

SciPost Phys. Lect. Notes 11 (2019)

# An introduction to spontaneous symmetry breaking

**Aron J. Beekman[1], Louk Rademaker[2,3] and Jasper van Wezel[4]⋆**

**1** Department of Physics, and Research and Education Center for Natural Sciences,
Keio University, 3-14-1 Hiyoshi, Kohoku-ku, Yokohama 223-8522, Japan
**2** Department of Theoretical Physics, University of Geneva, 1211 Geneva, Switzerland
**3** Perimeter Institute for Theoretical Physics, Waterloo, Ontario N2L 2Y5, Canada
**4** Institute for Theoretical Physics Amsterdam, University of Amsterdam,
Science Park 904, 1098 XH Amsterdam, The Netherlands

⋆ J.vanwezel@uva.nl

## Abstract

Perhaps the most important aspect of symmetry in physics is the idea that a state does not need to have the same symmetries as the theory that describes it. This phenomenon is known as *spontaneous symmetry breaking*. In these lecture notes, starting from a careful definition of symmetry in physics, we introduce symmetry breaking and its consequences. Emphasis is placed on the physics of singular limits, showing the reality of symmetry breaking even in small-sized systems. Topics covered include Nambu-Goldstone modes, quantum corrections, phase transitions, topological defects and gauge fields. We provide many examples from both high energy and condensed matter physics. These notes are suitable for graduate students.

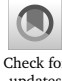

# Preface

Symmetry is one of the great unifying themes in physics. From cosmology to nuclear physics and from soft matter to quantum materials, symmetries determine which shapes, interactions, and evolutions occur in nature. Perhaps the most important aspect of symmetry in theories of physics, is the idea that the states of a system do not need to have the same symmetries as the theory that describes them. Such *spontaneous breakdown* of symmetries governs the dynamics

of phase transitions, the emergence of new particles and excitations, the rigidity of collective states of matter, and is one of the main ways classical physics emerges in a quantum world.

The basic idea of spontaneous symmetry breaking is well known, and repeated in different ways throughout all fields of physics. More specific aspects of spontaneous symmetry breaking, and their physical ramifications, however, are dispersed among the specialised literature of multiple subfields, and not widely known or readily transferred to other areas. These lecture notes grew out of a dissatisfaction with the resulting lack of a single general and comprehensive introduction to spontaneous symmetry breaking in physics. Quantum field theory textbooks often focus on the mathematical structure, while only leisurely borrowing and discussing physical examples and nomenclature from condensed matter physics. Most condensed matter oriented texts on the other hand, treat spontaneous symmetry breaking on a case-by-case basis, without building a more fundamental understanding of the deeper concepts. These lecture notes aim to provide a sound physical understanding of spontaneous symmetry breaking while at the same time being sufficiently mathematically rigorous. We use mostly examples from condensed matter theory, because they are intuitive and because they naturally highlight the relation between the abstract notion of the thermodynamic limit and reality. These notes also incorporate some more modern developments that shed light on previously poorly understood aspects of spontaneous symmetry breaking, such as the counting and dispersion relations of Nambu–Goldstone modes (Section 3.2) and the role and importance of the Anderson tower of states (Section 2.6).

A relatively new perspective taken in these notes revolves around the observation that the limit of infinite system size, often called the thermodynamic limit, is *not* a necessary condition for spontaneous symmetry breaking to occur in practice. Stronger, for almost all realistic applications of the theory of symmetry breaking, it is a rather useless limit, in the sense that it is never strictly realised. Even in situations where the object of interest can be considered large, the coherence length of ordered phases is generically small, and a single domain cannot in good faith be considered to approximate any sort of infinite size. Fortunately, even relatively small objects can spontaneously break symmetries, and almost all of the generic physical consequences already appear in some form at small scales. The central message of spontaneous symmetry breaking then, is that objects of all sizes reside in thermodynamically stable states, rather than energy eigenstates.

We will often switch between different types of symmetry-breaking systems. For example, most of these lecture notes concern the breakdown of continuous symmetries, but the most notable differences with discrete symmetries are briefly discussed whenever they are arise. Even within the realm of continuous symmetries, many flavours exist, each with different physical consequences emerging from their breakdown. We emphasise such differences throughout the lecture notes, but without attempting to be encyclopaedic. The main aim of these notes is to foster physical understanding, and variations on any theme will be presented primarily through the discussion of concrete examples.

Similarly, we will regularly switch between the Hamiltonian and Lagrangian paradigms. Many aspects of symmetry breaking can be formulated in either formalism, but sometimes one provides a clearer understanding than the other. Furthermore, it is often insightful to compare the two approaches, and we do this in several places in these lecture notes. For most of the discussion, we adopt the viewpoint that any physicist may be assumed to have a working knowledge of both formalisms, and we freely jump between them, choosing whichever approach is most suitable for the aspect under consideration.

These notes are based in part on lectures taught by J.v.W. at the Delta Institute for Theoretical Physics as a course for advanced MSc and beginning PhD students. They are suitable for a one- or even half-semester course for graduate students, while undergraduate students may find large parts fun and interesting to read. Knowledge of basic quantum mechanics

and Hamiltonian classical mechanics is essential, while familiarity with (classical) field theory, relativity, group theory, as well as basic condensed matter or solid state physics will be useful. Chapters 1 and 2 constitute the core of these lecture notes, explaining the physics of spontaneous symmetry breaking. The remaining sections contain various consequences of symmetries being broken, and could be taught more or less independently from one another. It should be noted, though, that Chapter 3 on Nambu–Goldstone modes is prerequisite for many other parts. Exercises are dispersed throughout the notes and provide an opportunity to deepen the understanding of concepts introduced in the main text. They are not intended to be a challenge. Answers to selected exercises are included in Appendix C, as is a bibliography that can by used as the starting point for further study. Finally, an overview of more advanced topics is presented in Appendix A.

We hope these lecture notes bring to the fore the ubiquitous effects of symmetry breaking throughout all realms of physics. They explain how to predict both the number and the nature of collective excitations arising in any physical system with a broken symmetry. They show that an Anderson–Higgs mechanism may arise from the breakdown of a global symmetry whenever it is accompanied by a gauge freedom, regardless of whether this occurs in general relativity, elementary particle physics or superconductors. They allow you to appreciate why sitting on a chair is essentially the same thing as levitating a piece of superconducting material in a magnetic field. And they prepare you for further exploring the many wonderful connections between fields of physics brought together by the unifying themes of symmetry and symmetry breaking.

**Notation**   We adopt relativistic notation for vector fields where $A^\nu = (A^t, A^n)$. Greek indices run over time and space and Roman indices run over space only. The metric is mostly minus $\eta_{\mu\nu} = \text{diag}(1, -1, -1, -1)$. The shorthand notation for derivatives is $\partial_\nu = \partial/\partial x^\nu$. The number of spatial dimensions is denoted as $D$, so that the spacetime dimension is $d = D + 1$. For arguments of fields we use $(x)$ to denote $(t, \mathbf{x})$, where bold-face vectors always refer to spatial $D$-dimensional vectors. Fields (like $\phi_a(x)$) can be viewed both as classical fields and as quantum field operators, depending on the context.

**Acknowledgements**   We thank Tomas Brauner and Hal Tasaki for discussions concerning Section 3.4, and Naoki Yamamoto for helpful comments on the manuscript. In general we have benefited from discussions with Haruki Watanabe and Yoshimasa Hidaka over the years. We thank our mentor Jan Zaanen for instilling upon us the idea that all condensates including superconductors and superfluids are to be seen as classical matter.

A. J. B. and J. v. W. are thankful for the gracious hospitality of Perimeter Institute, where the first draft for this manuscript was written. This research was supported in part by Perimeter Institute for Theoretical Physics. Research at Perimeter Institute is supported by the Government of Canada through the Department of Innovation, Science and Economic Development and by the Province of Ontario through the Ministry of Research, Innovation and Science.

A. J. B. is supported by the MEXT-Supported Program for the Strategic Research Foundation at Private Universities "Topological Science" (Grant No. S1511006) and by JSPS Grant-in-Aid for Early-Career Scientists (Grant No. 18K13502). L. R. is supported by the Swiss National Science Fund through an Ambizione grant.

# 1  Symmetry

## 1.1  Definition

Before talking about the breaking of symmetry, we need to define and understand what is meant by symmetry itself. Here, we need to distinguish between, on the one hand, the symmetries of the laws of nature, equations of motion, and the action or Hamiltonian, and on the other hand, the symmetries of states, objects, and solutions to the equations of motion. This distinction lies at the core of spontaneous symmetry breaking, which is said to occur whenever a physical state or object has less symmetry than the laws of nature that govern it.

### 1.1.1  Symmetries of states

Intuitively, we would say an object possess a symmetry, if it looks identical from different viewpoints. For example, a sphere looks identical from any angle, and is therefore concluded to be rotationally symmetric. To put it more formally, the continuous rotational symmetry in this case implies that the physical description of the sphere does not depend on its orientation in space. In the same way, an equilateral triangle possesses three-fold rotational symmetry, since it is unaffected by rotations over any multiple of 120°.

Within quantum mechanics, the formal definition of symmetric states closely mimics the intuitive one. Taking a 'different viewpoint' towards a given state is mathematically represented in the quantum formalism by applying a unitary transformation $U$ to it[1]. As a consequence, the symmetry of quantum states can be defined as follows:

**Definition 1.1** *A state $|\psi\rangle$ is said to be symmetric under a unitary transformation $U$ if the transformed state is identical to the original state, up to a phase factor:*

**symmetry – of states**

$$U|\psi\rangle = e^{i\varphi}|\psi\rangle. \tag{1.1}$$

The possible appearance of a phase factor in this definition is due to the usual axiom in quantum mechanics that the total phase of a quantum state is unmeasurable, and physical states therefore correspond to rays, rather than vectors, in the Hilbert space (see any textbook on quantum mechanics, such as [1, 2]). For the moment, the phase $\varphi$ may be ignored, and a symmetry may be understood to imply $U|\psi\rangle = |\psi\rangle$.

This simple definition for the symmetry of a state in quantum mechanics brings to light a seemingly trivial, but actually consequential, issue. Acting on the symmetric state $|\psi\rangle$ with $U$ does nothing. So how does $U$ differ from the identity operator? In terms of our earlier example, $U$ might represent a rotation of the sphere. Such a rotation can only exist (or only makes sense) when it is applied with respect to something else. If the observer holding the sphere rotates her hand to look at it from a different angle, and thus applies the operation $U$, the sphere is rotated with respect to the observer, who remains stationary. The fact that a state $|\psi\rangle$ is symmetric under $U$ can thus only be observed, and is only relevant, when there exist other states that are not symmetric under the same transformation. Phrased more generally, we notice that:

> A state, object, or system can only be defined to be symmetric with respect to a different, non-symmetric state, object, or system.

As we will see in Section 1.5, this seemingly obvious observation is paramount to understanding the difference between a symmetry and a gauge freedom.

---

[1]Classical physics can also be formulated in terms of operators acting on a Hilbert space, and then the definition of the symmetries of a classical state is similar to the quantum case discussed here.

### 1.1.2 Symmetries of Hamiltonians

Like states, also the laws of nature, or equivalently the equations of motion, may be said to possess symmetries. Just as for states, defining this second type of symmetry is most naturally done within quantum mechanics, where it again appears in a deceptively simple form. Since Schrödinger's equation and the dynamics of quantum states follow directly from the Hamiltonian, symmetries of the equation of motion correspond to symmetries of the Hamiltonian operator. In general, any quantum operator $A$ is called *invariant* under the unitary transformation $U$ if $U^{\dagger}AU = A$, or equivalently, if $[U,A] = 0$. If the Hamiltonian $H$ is invariant under some unitary transformation $U$, then $U$ is called a symmetry of the Hamiltonian. The rationale behind this definition is that the expectation value of a symmetric Hamiltonian $H$ in any state $|\psi\rangle$, symmetric or not, is invariant under application of the transformation $U$ to the state[2]:

$$\left(\langle\psi|U^{\dagger}\right)H\left(U|\psi\rangle\right) = \langle\psi|\left(U^{\dagger}HU\right)|\psi\rangle = \langle\psi|H|\psi\rangle. \tag{1.2}$$

The symmetry of Hamiltonians, and therefore quantum mechanical equations of motion, can thus be defined by stating that:

**Definition 1.2** *A unitary transformation $U$ is a symmetry of the Hamiltonian $H$ if their commutator vanishes:* $[U,H] = 0$.

**symmetry – of the Hamiltonian**

Notice that for a symmetric Hamiltonian, $U^{\dagger}HU = H$, and therefore eigenstates of $H$ are also eigenstates of $U^{\dagger}HU$. Both the eigenstates of $H$ and those of $U^{\dagger}HU$ may then be used as a complete basis of energy eigenstates in describing the dynamics of the system. Similarly, if a symmetric Hamiltonian $H$ has an eigenstate $|\psi\rangle$ with eigenvalue $E_{\psi}$, then the transformed state $U|\psi\rangle$ is also an eigenstate of $H$, with the same eigenvalue:

$$H(U|\psi\rangle) = UH|\psi\rangle = UE_{\psi}|\psi\rangle = E_{\psi}(U|\psi\rangle). \tag{1.3}$$

In case the state $|\psi\rangle$ is itself symmetric under $U$, the fact that its energy is invariant under the symmetry operation may seem obvious. The non-trivial implication of equation (1.3) is that for every energy eigenstate that is not itself symmetric, there exists a degenerate state, which can be reached by applying the symmetry transformation. Applying the operator $U$ a second time will lead to a next degenerate state, and so on.

Of course, symmetries can be defined within the Lagrangian formulation of physics just as well as within the Hamiltonian formalism. This definition will appear naturally in the next section, when we discuss Noether's theorem.

## 1.2 Noether's theorem

The dynamics of many physical systems may be described almost entirely in terms of conservation laws, like the conservation of energy, momentum, angular momentum, and so on. One of the most profound insights in all of physics is the fact that the existence of such conserved quantities is always rooted in symmetries of the applicable laws of nature. Translational invariance, for instance, which means that the equations of motion look the same at any spatial location, implies conservation of momentum. This holds true regardless of what the equations of motion happen to be.

---

[2]There are some unitary operators that are symmetries, with associated Noether currents, but which do not simply commute with the Hamiltonian. This applies in particular to Galilean and Lorentz boosts. A good treatment can be found in Ref. [2]. The implications of the fact that Galilean boosts are broken by any form of matter, have been understood only quite recently [3], and is beyond the scope of these notes, which is why we will restrict ourselves to symmetries that commute with the Hamiltonian.

### 1.2.1 Conserved quantities

The relation between symmetries and conserved quantities can be readily understood by realising that time evolution is determined by the Hamiltonian. Explicitly, the time evolution operator for time-independent Hamiltonians is shown by Schrödinger's equation to be the exponentiation of the Hamiltonian: $\mathcal{U}(t) = e^{iHt}$. As we saw in the previous section, a transformation $U$ defines a symmetry of the Hamiltonian if it commutes with $H$. It then follows that symmetry transformations also commute with the time evolution operator $\mathcal{U}(t)$. Because any symmetry transformation is unitary, it can be written as the exponential $U = e^{iQ}$ of some Hermitian operator $Q$. The fact that $U$ commutes with the time evolution operator, then implies that also $[Q, \mathcal{U}(t)] = 0$. This in turn means that the expectation value of $Q$ in any state $|\psi\rangle$ is conserved in time:

$$\langle \psi(t)|Q|\psi(t)\rangle = \big(\langle\psi|\mathcal{U}^\dagger(t)\big)Q\big(\mathcal{U}(t)|\psi\rangle\big) = \langle\psi|\big(\mathcal{U}^\dagger(t)Q\mathcal{U}(t)\big)|\psi\rangle = \langle\psi|Q|\psi\rangle. \qquad (1.4)$$

Now recall that in quantum mechanics, Hermitian operators represent observables. We thus find that the existence of the symmetry transformation $U$ directly implies a conservation law for the observable $Q$, which is known as a conserved quantity or *constant of motion*. Following an argument similar to equation (1.3), it can furthermore be shown that an eigenstate of $Q$ can only evolve in time to eigenstates of $Q$ with the same eigenvalue.

So the fact that the Hamiltonian generates time translations leads immediately to the conceptually important result:

> Any unitary symmetry $U$ corresponds to an observable $Q$ such that $U = e^{iQ}$. The observable $Q$ is a conserved quantity.

**Exercise 1.1 (Exponential of an operator)** The exponential of an operator $A$ is defined by its power series $e^A = \sum_{n=0}^{\infty} \frac{1}{n!}A^n$. Show that any operator that commutes with the Hamiltonian also commutes with the time evolution operator. Also show that if $U = e^{iQ}$ commutes with the Hamiltonian, $Q$ must commute with the Hamiltonian as well.

### 1.2.2 Continuity equations

The existence of any symmetry transformation implies a conservation law. For *continuous* symmetry transformations parametrised by a continuous variable, such as translations over a continuous distance or rotations over a continuous angle, there is an even stronger result, known as *Noether's theorem*. This theorem states that each continuous symmetry is associated with a *current* $j^\nu(\mathbf{x}, t)$ obeying the *local* conservation law or continuity equation $\partial_\nu j^\nu = 0$. For non-continuous, or discrete, symmetries we have only a global constant of motion $Q$, but no local continuity equation.

Before formally proving Noether's theorem, let us give an intuitive interpretation of its main result. Firstly, continuous symmetries can be parameterised by a continuous (real-valued) parameter $\alpha$. In terms of the unitary symmetry transformations discussed before, this means the Hamiltonian commutes with a family of related transformations $U_\alpha = e^{i\alpha Q}$. Taking continuous translations, for example, $U_\alpha$ would be the operator that translates a state over distance $\alpha$. In general, the operator $Q$ is called the *symmetry generator*, because any symmetry transformation $U_\alpha$ for finite $\alpha$ can be obtained from the action of $Q$. For transformations with an infinitesimally small value of the parameter $\alpha$, the operator $U_\alpha$ may be expanded in a Taylor series, and terms beyond first order may be neglected:

$$U_\alpha = 1 + i\alpha Q + \mathcal{O}(\alpha^2). \qquad (1.5)$$

Noether's theorem now follows from the observation that the conserved quantity $Q$ can always be written as an integral over purely local operators, $Q = \int d^D x \, \rho(\mathbf{x}, t)$, where $\rho(\mathbf{x}, t)$ is defined to be the local 'density of $Q$'. Even though the global observable $Q$ is conserved, the local observables $\rho(\mathbf{x}, t)$ are generally not. If, say, the value $\rho(\mathbf{x}, t)$ at position $\mathbf{x}$ increases, the conservation of the global $Q$ implies that the value of $\rho(\mathbf{x}', t)$ at some other position $\mathbf{x}'$ must be simultaneously reduced. This means there is a 'current of $\rho$' flowing from $\mathbf{x}'$ to $\mathbf{x}$. Calling this current $j^n(\mathbf{x}, t)$, the conservation of the global $Q$ is seen to be equivalent to the density and current together satisfying a continuity equation:

**continuity equation**

$$\partial_\nu j^\nu(\mathbf{x}, t) = \partial_t \rho(\mathbf{x}, t) + \partial_n j^n(\mathbf{x}, t) = 0. \tag{1.6}$$

Here, the density $\rho(\mathbf{x}, t)$ is written as the time-component of the four-vector $j^\nu(\mathbf{x}, t)$.

Argued the other way around, if a continuity equation holds, there must be a global $Q$ that is conserved in time. This can be seen by integrating the continuity equation over all space:

$$0 = \int d^D x \, \partial_\nu j^\nu(x) = \int d^D x \, \partial_t \rho(\mathbf{x}, t) + \int d^D x \, \partial_n j^n(\mathbf{x}, t)$$

$$= \partial_t Q(t) + \oint dS_n \, j^n(\mathbf{x}, t). \tag{1.7}$$

Here we used Gauss' divergence theorem in going to the last line. The second term is a boundary term at spatial infinity and assuming that $j^n$ falls off sufficiently quickly, this term vanishes. Therefore we find $\partial_t Q = 0$, or in other words, $Q$ is a conserved quantity. The direct relation between the presence of a symmetry and a corresponding continuity equation constitutes Noether's theorem.

### 1.2.3 Proving Noether's theorem

Noether's theorem applies to any theory of physics that has a continuous symmetry, and which can be described in terms of a Lagrangian, or minimum-action principle. It is equally valid in quantum physics, classical mechanics, gravity, and even as-yet unknown realms of physics. Here, we will present a proof for a general field theory containing several fields $\phi_a(x) = \phi_a(\mathbf{x}, t)$. The fields $\phi_a(x)$ need not take the form of a vector. The index $a$ could also for example label the two real components of a complex scalar field, or even different types of fields. The Lagrangian (density) is a functional of both the fields and their derivatives, $\mathcal{L} = \mathcal{L}[\phi_a(x), \partial_\nu \phi_a(x)]$, while the action, $S = \int dt d^D x \, \mathcal{L}$, is the integral of the Lagrangian over space and time.

**Lagrangian**

**action**

To prove Noether's theorem, we will compare the effects of infinitesimal symmetry transformations of the fields on the Lagrangian *before* and *after* imposing the equations of motion. In general, a transformation of the fields can be written as:

$$\phi_a(x) \rightarrow \phi_a'(x) = \phi_a(x) + \delta_s \phi_a(x). \tag{1.8}$$

This expression may be interpreted as the definition of the variation $\delta_s \phi_a = \phi_a' - \phi_a$. We will consider $\phi_a'(x)$ to be the field resulting from the action of a symmetry transformation on $\phi_a(x)$, and the variation is assumed to be infinitesimally small. We can then use variational calculus to write the effect of the transformation on the Lagrangian in terms of the transformations of the fields and their derivatives:

$$\mathcal{L} \rightarrow \mathcal{L}' = \mathcal{L} + \delta_s \mathcal{L}$$

$$\delta_s \mathcal{L} = \frac{\partial \mathcal{L}}{\partial \phi_a} \delta_s \phi_a + \frac{\partial \mathcal{L}}{\partial (\partial_\nu \phi_a)} \delta_s(\partial_\nu \phi_a). \tag{1.9}$$

Here $\partial\mathcal{L}/\partial\phi$ means taking a derivative of $\mathcal{L}$ as if it were an ordinary function of a variable $\phi$, and sums over $a$ and $\nu$ are implied by the Einstein summation convention. The expansion of $\delta_s\mathcal{L}$ in terms of infinitesimal variations of the fields is what makes Noether's theorem valid only for continuous, and not discrete, symmetries.

Equation (1.9) holds true for any infinitesimal variation $\delta_s\phi_a$ of the fields. It can be seen simply as the 'variational chain rule'. If the variation $\delta_s\mathcal{L}$ happens to be zero, the transformation that caused it can be called a symmetry of the action. In fact, even if $\delta_s\mathcal{L}$ is a total derivative $\partial_\nu K^\nu$ of some function $K^\nu$, we will call the transformation a symmetry. In that case, $\delta_s\mathcal{L}$ will only add a boundary term to the action, which does not affect the equations of motion. For most symmetry transformations the variation of the Lagrangian will simply be zero, but an important example of a non-vanishing boundary term is that of spacetime translations, which will be discussed in Section 1.3.3.

> A transformation of the fields, $\phi_a(x) \to \phi_a(x) + \delta_s\phi_a(x)$, is a symmetry of the action if the corresponding change in the Lagrangian $\mathcal{L}[\phi_a(x), \partial_\nu\phi_a(x)] \to \mathcal{L} + \delta_s\mathcal{L}$ is at most a total derivative: $\delta_s\mathcal{L} = \partial_\nu K^\nu$.

**symmetry – of the action**

Notice that so far, we have not specified whether or not the fields $\phi_a$ satisfy any equations of motion. The condition $\delta_s\mathcal{L} = \partial_\nu K^\nu$ defines what it means for a transformation to be a symmetry of the action, regardless of the type of field it acts on.

We now return to Eq. (1.9), and recall that it holds for any infinitesimal transformation of the fields, whether they constitute a symmetry or not. Using an elementary result from variational calculus, $\delta_s(\partial_\nu\phi_a) = \partial_\nu(\delta_s\phi_a)$, and performing an integration by parts allows us to write it in the form:

$$\delta_s\mathcal{L} = \partial_\nu\left(\frac{\partial\mathcal{L}}{\partial(\partial_\nu\phi_a)}\delta_s\phi_a\right) + \left[\frac{\partial\mathcal{L}}{\partial\phi_a} - \partial_\nu\left(\frac{\partial\mathcal{L}}{\partial(\partial_\nu\phi_a)}\right)\right]\delta_s\phi_a. \tag{1.10}$$

The part in the square brackets should look familiar: this is precisely what the Euler-Lagrange equations of motion prescribe to be zero. If we thus restrict attention to field configurations $\phi_a$ that satisfy these equations of motion, we are left with only the first term on the right-hand side, which is a total derivative. Notice that this condition for $\delta_s\mathcal{L}$ being a total derivative holds for specific field configurations $\phi_a$, without putting any requirements on the transformation $\delta_s\phi_a$ we consider. Conversely, the previous result of $\delta_s\mathcal{L} = \partial_\nu K^\nu$ constituting a symmetry of the action, was a requirement on the transformation $\delta_s\phi_a$, for arbitrary $\phi_a$ [4].

Noether's theorem is now obtained by considering fields that obey the Euler-Lagrange equation of motion, and transformations that are symmetries of the action. We can then subtract the two conditions on $\delta_s\mathcal{L}$, and obtain to the continuity equation

$$\partial_\nu\left(\frac{\partial\mathcal{L}}{\partial(\partial_\nu\phi_a)}\delta_s\phi_a - K^\nu\right) \equiv \alpha\partial_\nu j^\nu = 0. \tag{1.11}$$

Here we introduced an infinitesimal parameter $\alpha$ for later convenience, and defined the *Noether current* $j^\nu$ related to the symmetry transformation $\phi_a(x) \to \phi_a(x) + \delta_s\phi_a(x)$, as:

**Noether current**

$$j^\nu = \frac{1}{\alpha}\left(\frac{\partial\mathcal{L}}{\partial(\partial_\nu\phi_a)}\delta_s\phi_a - K^\nu\right) = \frac{\partial\mathcal{L}}{\partial(\partial_\nu\phi_a)}\Delta_s\phi_a - \frac{1}{\alpha}K^\nu. \tag{1.12}$$

In the final expression, we introduced the notation $\Delta_s\phi = \delta_s\phi/\alpha$. The presence of a continuous symmetry implying the existence of a Noether current that is locally conserved, $\partial_\nu j^\nu = 0$, is the main result of Noether's theorem.

**Theorem 1.1** (*Noether's theorem*). *To any continuous symmetry of a local action corresponds a current $j^\nu(\mathbf{x}, t)$ that is locally conserved. That is, it satisfies the continuity equation $\partial_\nu j^\nu(\mathbf{x}, t) = 0$.*

**Noether's theorem**

### 1.2.4 Noether charge

Recall that Eq. (1.7) showed any local continuity equation to imply the existence of a globally conserved quantity. In the context of Noether's theorem, this is called the *Noether charge* (not to be confused with electric charge, which is in first instance not related) and may be defined as:

$$Q(t) = \int d^D x \; j^t(\mathbf{x}, t) = \int d^D x \left( \frac{\partial \mathcal{L}}{\partial(\partial_t \phi)} \Delta_s \phi - \frac{1}{\alpha} K^t \right). \tag{1.13}$$

Similarly $j^t = \rho$ is called the *Noether charge density*. Notice that because $\partial_\nu j^\nu = 0$, the Noether charge $Q(t)$ is in fact independent of time, or in other words, a constant of motion.

To see how the Noether charge is related to the symmetry transformation, consider the canonical momentum $\pi_a(x) = \partial \mathcal{L}/\partial(\partial_t \phi_a)$, conjugate to $\phi_a(x)$. In quantum mechanics, the field and conjugate momentum obey the commutation relations $[\pi_a(x), \phi_b(y)] = -i\delta_{ab}\delta(x-y)$. Focussing on the most commonly encountered case with $K^\nu = 0$ and $[\Delta_s \phi_b, \phi_a] = 0$, the commutator of the field and the Noether charge is found to be:

$$[i\alpha Q, \phi_a(x)] = i\alpha \int d^D y \; [\pi_b(y)\Delta_s \phi_b(y), \phi_a(x)]$$
$$= i\alpha \int d^D y \; [\pi_b(y), \phi_a(x)]\Delta_s \phi_b(y) = \alpha \Delta_s \phi_a(x) = \delta_s \phi_a(x). \tag{1.14}$$

Because the commutator of the Noether charge and the field equals the variation of the field, the Noether charge is also called the *generator* of the symmetry. A symmetry transformation of the fields can now be written as:

$$\phi_a(x) \to \phi_a'(x) = \phi_a(x) + i\alpha[Q, \phi_a(x)]$$
$$= e^{i\alpha Q} \phi_a(x) e^{-i\alpha Q} + \mathcal{O}(\alpha^2). \tag{1.15}$$

In the final line, we applied the Baker–Campbell–Hausdorff formula while assuming $\alpha$ to be infinitesimal. Since this expression for the symmetry transformation is of the same form as Eq. (1.5), we see that the Noether charge $Q$ indeed corresponds to the observable $Q$ obeying $[Q, H] = 0$ in the Hamiltonian formalism. This result also holds in the more general case with nonzero boundary terms $K^\nu$, but the derivation is lengthier.

It should be emphasised that the local form of Noether's theorem $\partial_\nu j^\nu = 0$ depends only on the symmetry of the action, and is valid for any state that satisfies the equations of motion. In quantum mechanics, it is an operator identity that does not refer to any particular state. However, if the physical state of a system happens not to share the symmetry of the action— that is, if the symmetry of the action is spontaneously broken—one should be careful about what the continuity equation implies physically. We will come back to this point in Chapter 3, when discussing so-called Nambu–Goldstone modes.

## 1.3 Examples of Noether currents and Noether charges

### 1.3.1 Schrödinger field

A complex scalar field $\psi(\mathbf{x}, t)$ is called a Schrödinger field when it has the action:

$$S[\psi, \psi^*] = \int dt d^D x \left( i\frac{\hbar}{2}\big(\psi^*(\partial_t \psi) - (\partial_t \psi^*)\psi\big) - \frac{\hbar^2}{2m}(\partial_n \psi^*)(\partial_n \psi) - V(\mathbf{x})\psi^*\psi \right). \tag{1.16}$$

Here the potential $V(\mathbf{x})$ is an ordinary function of space. The reason $\psi$ is called a Schrödinger field is that the Euler–Lagrange equation obtained by varying with respect to $\psi^*$ looks like the

Schrödinger equation:

$$0 = \partial_t\left(\frac{\partial\mathcal{L}}{\partial(\partial_t\psi^*)}\right) + \partial_n\left(\frac{\partial\mathcal{L}}{\partial(\partial_n\psi^*)}\right) - \frac{\partial\mathcal{L}}{\partial\psi^*} = -i\hbar\partial_t\psi - \frac{\hbar^2}{2m}\partial_n^2\psi + V(\mathbf{x})\psi. \qquad (1.17)$$

One way to handle the two degrees of freedom contained in a complex scalar field is to treat $\psi$ and $\psi^*$ independently, with commutation relation $[\psi(x),\psi^*(y)] = \delta(x-y)$. The canonical momenta associated with the Schrödinger field and it complex conjugate can be identified as:

$$\pi = \partial\mathcal{L}/\partial(\partial_t\psi) = i\hbar\psi^*/2, \qquad\qquad \pi^* = \partial\mathcal{L}/\partial(\partial_t\psi^*) = -i\hbar\psi/2. \qquad (1.18)$$

The action in Eq. (1.16) has a continuous symmetry. It is invariant under phase rotations of the form

$$\psi(x) \rightarrow e^{-i\alpha}\psi(x), \qquad\qquad \psi^*(x) \rightarrow e^{i\alpha}\psi^*(x), \qquad (1.19)$$

**phase rotation**

with $\alpha$ a real and continuous parameter. Notice that $\alpha$ does not depend on $x$, so that the phase rotation is a 'global' transformation, affecting all points in the system in the same way. Taking $\alpha$ to be infinitesimal, the exponent can be expanded and the variations of the field under a phase rotation become $\Delta_s\psi(x) = -i\psi(x)$ and $\Delta_s\psi^*(x) = i\psi^*(x)$. The Noether current and conserved Noether charge can now be identified:

$$j^t = \pi\Delta_s\psi + \pi^*\Delta_s\psi^* = \hbar\psi^*\psi, \qquad\qquad Q = \int d^D x\, \hbar\psi^*\psi, \qquad (1.20)$$

$$j^n = i\frac{\hbar^2}{2m}\left((\partial_n\psi^*)\psi - \psi^*(\partial_n\psi)\right). \qquad (1.21)$$

Harking back to the correspondence to the Schrödinger equation, the quantity $\int \psi^*\psi = \int |\psi|^2$ is of course just the total amplitude of the wave function, which is indeed should retain its normalisation in any well-defined quantum theory. In the Lagrangian treatment, the conservation of the norm can be interpreted as a consequence of the invariance of Eq. (1.16) under global phase rotations. Similarly, the local amplitude of the wave function, $|\psi(x)|^2$, can only change when it flows elsewhere in the form of a probability current $j^n$. Noether's theorem can then be interpreted as a continuity equation for the probability current.

### 1.3.2 Relativistic complex scalar field

Rather than starting from Schrödinger's equation, we can also consider a different form for the action of a complex scalar field:

$$S = \int dt\, d^D x\left(\frac{1}{c^2}(\partial_t\psi^*)(\partial_t\psi) - (\partial_n\psi^*)(\partial_n\psi) - V(\mathbf{x})\psi^*\psi\right). \qquad (1.22)$$

This action is invariant under Lorentz transformations and therefore said to be "relativistic". Its equation of motion is known as the Klein–Gordon equation. Besides Lorentz invariance, the action also has the same global phase rotation symmetry as Eq. (1.16). However, the canonical momentum is now $\pi = \partial\mathcal{L}/\partial(\partial_t\psi) = \partial_t\psi^*/c^2$. The spatial part of the Noether current is the same as that in Eq. (1.21), up to a factor $\hbar^2/2m$, but the relativistic Noether charge becomes:

$$Q = \int d^D x\, (\pi\Delta_s\psi + \pi^*\Delta_s\psi^*) = \int d^D x\, i\frac{1}{c^2}\left(\psi^*(\partial_t\psi) - (\partial_t\psi^*)\psi\right). \qquad (1.23)$$

This conserved charge plays the role of the conserved field normalisation in the relativistic Klein-Gordon theory.

### 1.3.3  Spacetime translations

One of the few examples of a symmetry transformation on the fields changing the Lagrangian by a total derivative, is that of spacetime translations. Defining spacetime coordinates as usual, $x^\nu = (t, \mathbf{x})$, a global spacetime translation may be written as:

**spacetime translations**

$$
\begin{aligned}
\phi(x^\nu) \to \phi'(x^\nu) &= \phi(x^\nu + \alpha^\nu) \\
&= \phi(x^\nu) + \alpha^\mu \partial_\mu \phi(x^\nu) + \mathcal{O}(\alpha^2).
\end{aligned}
\tag{1.24}
$$

Here $\alpha^\nu$ is a constant spacetime vector. Since there are $D+1$ independent symmetry transformations, translating the field in $D$ spatial and one temporal directions, the variation $\Delta_s^\nu \phi(x) = \partial_\nu \phi$ is also a spacetime vector. Without referring to any specific action, and thus without using the equations of motion, we can see that the variation of the action under this symmetry transformation gives rise to a boundary term:

$$
\Delta_s^\nu \mathcal{L} = \frac{\partial \mathcal{L}}{\partial \phi} \Delta_s^\nu \phi + \frac{\partial \mathcal{L}}{\partial(\partial_\mu \phi)} \Delta_s^\nu (\partial_\mu \phi_a) = \frac{\partial \mathcal{L}}{\partial \phi} \partial_\nu \phi + \frac{\partial \mathcal{L}}{\partial(\partial_\mu \phi)} \partial_\nu \partial_\mu \phi = \partial_\nu \mathcal{L}.
\tag{1.25}
$$

In other words, the variation of the Lagrangian can be written as:

$$
\mathcal{L}' - \mathcal{L} = \alpha^\nu \partial_\nu \mathcal{L} = \partial_\nu (\alpha^\nu \mathcal{L}) \equiv \partial_\nu K^\nu.
\tag{1.26}
$$

Here, we used the fact that $\alpha^\nu$ is constant in space and time to take it inside the partial derivative.

Because there are $D+1$ independent continuous symmetry transformations, we expect to find $D+1$ conserved charges. Defining the 'relativistic canonical momenta', $\pi^\mu = \partial \mathcal{L}/\partial(\partial_\mu \phi)$, we can directly identify the $D+1$ Noether currents labelled by $\nu$:

$$
j_\nu^\mu = \pi^\mu \Delta_s^\nu \phi - \delta^\mu{}_\nu \mathcal{L} = \pi^\mu \partial_\nu \phi - \delta^\mu{}_\nu \mathcal{L}.
\tag{1.27}
$$

The tensor $j_\nu^\mu$ contains $D+1$ Noether currents labelled by $\nu$, each of which has $D+1$ spacetime components indexed by $\mu$. The tensor $\delta^\mu{}_\nu$ is the Kronecker delta. Notice that in writing this form of $j_\nu^\mu$, we still did not need to refer to any particular form of the action.

The conserved, global Noether charges associated with the Noether currents are:

$$
Q_t = \int \mathrm{d}^D x \, j_t^t = \int \mathrm{d}^D x \, \pi^t \partial_t \phi - \mathcal{L} = \int \mathrm{d}^D x \, \mathcal{H} = H,
\tag{1.28}
$$

$$
Q_n = \int \mathrm{d}^D x \, j_n^t = \int \mathrm{d}^D x \, \pi^t \partial_n \phi.
\tag{1.29}
$$

The conserved charge associated with time translation symmetry is seen to be the Hamiltonian, which is the energy operator. Similarly, $Q_n$ is the total momentum associated with the field $\phi$. The tensor of Noether currents describes the local flow of energy and momentum, and is usually referred to as the canonical energy-momentum tensor or stress-energy tensor. The conservation of the Noether charges now shows that energy is conserved because of the time translation symmetry of the action, and that the spatial translation symmetry of the action ensures conservation of momentum. Moreover, the entire derivation could be done without specifying a particular form of the action, and we therefore find that energy and momentum are conserved in any physical theory described by an action that is invariant under spacetime translations.

**symmetry – time translation**

**Exercise 1.2 ($SO(2)$ and $U(1)$ symmetry)** Consider an action of two real fields $A(x)$ and $B(x)$:

$$\mathcal{S} = \int \mathrm{d}t\,\mathrm{d}^D x\; \frac{1}{c^2}(\partial_t A)^2 + \frac{1}{c^2}(\partial_t B)^2 - (\partial_n A)^2 - (\partial_n B)^2 - V(\mathbf{x})(A^2 + B^2). \tag{1.30}$$

Show that the action is invariant under so-called $SO(2)$ rotations of the fields:

$$\begin{pmatrix} A \\ B \end{pmatrix} \rightarrow \begin{pmatrix} A' \\ B' \end{pmatrix} = \begin{pmatrix} \cos\alpha & -\sin\alpha \\ \sin\alpha & \cos\alpha \end{pmatrix} \begin{pmatrix} A \\ B \end{pmatrix}. \tag{1.31}$$

Determine the Noether current and Noether charge associated with this symmetry transformation. Compare your result with that of Section 1.3.2 when writing the complex field there as $\psi = A - iB$. In group theory this correspondence between phase rotations of a complex scalar variable and rotations within a vector of two real components, is known as the isomorphism between the groups $U(1)$ and $SO(2)$.

---

**Exercise 1.3 (Noether's trick)** There is a technique, sometimes called *Noether's trick*, that can be used to obtain the Noether current more directly. It considers a transformation of the action based on the global symmetry, but in which the parameter of the symmetry transformation is instead taken to depend on space and time: $\alpha \rightarrow \alpha(x)$. The action is then no longer invariant under the transformation. However, the term that is first order in $\partial_\mu \alpha$ will be of the form $\delta S|_{\mathcal{O}(\partial\alpha)} = \int \mathrm{d}x\, j^\mu \partial_\mu \alpha$, where $j^\mu$ turns out to be precisely the Noether current. (The reason is that the left-hand side still vanishes, $\delta S = 0$, for solutions of the equations of motion, so by partial integration, $\int \mathrm{d}x\, \alpha(\partial_\mu j^\mu) = 0$, which is true for arbitrary $\alpha$ if the Noether current in conserved.)

Derive the Noether current in this way for the Schrödinger field and the relativistic complex scalar field and compare your results with the calculations above.

**Noether's trick**

## 1.4 Types of symmetry transformations

In our definitions of symmetries and our treatment of their relation to conservation laws, we did not yet need to be very precise about the different types of symmetry transformations that may be encountered in various physical situations. However, not all symmetry transformations are susceptible to the spontaneous symmetry breaking that is the subject of the remainder of these lecture notes. Moreover, classifications of symmetry transformations have been introduced for as long as symmetry and symmetry breaking have been studied, and it is not obvious how some of these historical concepts fit into the modern framework presented here. In this and the following section we therefore give a brief account of the different types of transformations that may be encountered in the literature, focussing on the physical relevance of each distinction to the phenomenon of spontaneous symmetry breaking.

### 1.4.1 Discrete versus continuous symmetries

Contrary to a continuous symmetry, a discrete symmetry cannot be parametrised by a continuous, real variable. Simply put, you either do the discrete transformation or you do not. You cannot do it just a little bit. For some types of discrete symmetries, such as reflections, performing an arbitrary fraction of the symmetry transformation is simply not possible. In other cases, such fractional transformations are possible, but they are not symmetries. For example, a triangle is symmetric under rotations of 120°, but not under rotations over any

**discrete symmetry**

smaller angle. This is unlike the continuous rotational symmetry of a circle, which looks the same after rotations over any arbitrary angle.

While there is no mathematical difference between continuous and discrete symmetries beyond their parametrisation, there is an important physical difference: Noether's theorem applies only to continuous symmetries. The proof of Noether's theorem, and thus of the existence of a locally conserved current, requires the invocation of infinitesimal symmetry transformations. This is not possible for discrete symmetries[3]. Other concepts related to symmetry breaking, including the emergence of Nambu–Goldstone modes (Chapter 3) and the Mermin–Wagner theorem (Section 4.2), likewise only apply to broken continuous symmetries.

In short, there is more richness in the breaking of continuous symmetries than in discrete ones. Some examples of discrete symmetry breaking are certainly interesting and instructive — for instance, it is worth considering the Ising model when discussing the stability of broken symmetry states in Section 2.7 — but in these lecture notes we refer to them only in passing.

### 1.4.2 Anti-unitary symmetries

In Section 1.1.1, the symmetry of a state was defined as a unitary transformation that leaves the state unaffected, up to a total phase. A different point of view for why symmetry transformations are required to be unitary, is that such transformations conserve the inner product between any two states, $\left(\langle\psi|U^{\dagger}\right)\left(U|\psi'\rangle\right) = \langle\psi|\psi'\rangle$. This stringent condition can be relaxed somewhat, and we could instead consider transformations that only leave inner products unaffected up to a phase factor. This then allows for so-called *anti-unitary symmetries*, whose transformations turn out to satisfy $\left(\langle\psi|U^{\dagger}\right)\left(U|\psi'\rangle\right) = \langle\psi|\psi'\rangle^{*}$.

One very important anti-unitary symmetry is *time reversal symmetry*, which reverses the flow of time. Like other symmetries, time-reversal symmetry can be spontaneously broken. Examples of systems with broken time-reversal symmetries are ferromagnets and the *A*-phase of superfluid helium-3 [6]. However, we will largely ignore time-reversal and other anti-unitary symmetries in these lecture notes, since they are necessarily discrete.

**time reversal symmetry**

### 1.4.3 Global symmetries versus local symmetries

Spontaneous symmetry breaking occurs in physical systems with many microscopic degrees of freedom, such as a large collection of atoms, electrons, spins, or a continuous field spread out over all space and time. A symmetry of the full system is then defined by how it acts on the individual constituents (i.e. the atoms, the field amplitude at each location, and so on). A *global symmetry* is a symmetry that acts in the same way on each individual constituent. All of the examples in Section 1.3 concerned global symmetries, from the global rotation of the phase of the wave function in Section 1.3.1, to the global shift of spacetime coordinates in Section 1.3.3.

**global symmetry**

As will become clear in the next chapter, only global symmetries can be spontaneously broken. There also exist, however, many kinds of *local symmetries*, defined as a physical symmetry transformations that act differently on different local degrees of freedom. These are not to be confused with gauge freedoms, which are purely mathematical manipulations leaving a system's description invariant, but which do not correspond to any physical transformation of the actual system. Such gauge freedoms may have important consequences, and will be discussed in detail in Section 1.5, but for now, we will focus on actual local symmetries.

**local symmetry**

The easiest example of a local symmetry appears in a classical ideal gas of $N$ particles of mass $m$ with positions $\mathbf{X}_i(t)$ and momenta $\mathbf{P}_i(t)$. The index $i$ labels the particles and runs from

---

[3]Notice that in some cases, a remnant of the conserved Noether charges may survive even in systems with only a discrete symmetry. The discrete translation symmetry of a crystalline lattice, for example, is responsible for the fact that lattice momentum (or crystal momentum) is conserved modulo reciprocal lattice vectors [5].

1 to $N$. Since ideal particles do not interact, the Hamiltonian contains only the kinetic energy of the individual particles, and can be written as $H = \frac{1}{2m} \sum_i \mathbf{P}_i^2$. Because the Hamiltonian does not depend on the position of any particle, it is not affected by translations of each particle *individually*:

$$\mathbf{X}_i(t) \to \mathbf{X}_i(t) + \mathbf{a}_i. \tag{1.32}$$

translation symmetry

The local displacements $\mathbf{a}_i$ may be different for each $i$. We could even consider an extreme case in which the displacement is zero for all particles except one, making it obvious that the symmetry is local, rather than global.

You may argue that the example of the ideal gas is somewhat artificial, since the particles are really independent, and each come with their own symmetry. This example can be easily extended, however, to that of a free field theory [7]. Consider, for example, the relativistic complex field of Eq. (1.22) with no external potential, $V(\mathbf{x}) = 0$. Local spacetime displacements of the fields are then described by the transformation:

$$\psi(x) \to \psi(x) + \alpha(x). \tag{1.33}$$

Here, $\alpha(x)$ is complex-valued and depends on the spacetime coordinate $x$. If $\alpha(x)$ is completely general, the local shift is not a symmetry of the action. However, if $\alpha$ satisfies the equations of motion $\partial^2 \alpha = 0$ where $\partial^2 = \partial^\nu \partial_\nu$, then using $\partial^\nu \psi^* \partial_\nu \alpha + \partial^\nu \psi \partial_\nu \alpha^* = \partial^\nu(\psi^* \partial_\nu \alpha + \psi \partial_\nu \alpha^*) - \psi^* \partial^2 \alpha - \psi \partial^2 \alpha^*$ we see that the transformation only adds a boundary term to the action:

$$\delta_s S = \int \mathrm{d}t \mathrm{d}^D x \; \partial_\nu(\psi \partial^\nu \alpha^* + \psi^* \partial^\nu \alpha). \tag{1.34}$$

The equations of motion were imposed here for $\alpha$, but not for $\psi$. Although local symmetries cannot be spontaneously broken, as we will discuss in Exercise 2.3, they do give rise to conserved charges. For the free complex scalar field, we could derive Noether currents related to $\delta_s \psi = \alpha(x)$ at each $x$ individually. One interpretation of the corresponding Noether charges, is that each of the components in the Fourier transform of $\psi(x)$ is individually conserved [7].

The examples of the ideal gas and free field both concern non-interacting systems, but local symmetries may exist in interacting systems as well. For example, interacting particles in a disordered potential in some cases undergo *many-body localisation* (MBL), and the MBL-phase is characterised precisely by having a large number of emergent local symmetries (that are difficult to write down explicitly). Even in such interacting systems, spontaneous symmetry breaking does not occur for local symmetries.

many-body localisation

### 1.4.4 Active versus passive, and internal versus external symmetries

In addition to the physically relevant distinctions between global versus local and continuous versus discrete symmetries, various other classifications of symmetry transformations may be encountered in the literature. These are, in our opinion, not useful for the clarification of any physical effects, and often lead to confusion. For the sake of completeness, we will briefly comment on some of these alternative notions here, but they have no role anywhere else in these lecture notes.

The first distinction drawn in the literature, by mainly mathematically inspired authors, is between so-called active and passive transformations. An active transformation is said to be "an actual transformation of the coordinates and fields" as if to physically manipulate the system, whereas a passive transformation would be a coordinate transformation, or a "relabelling of the numerical values assigned to coordinates and fields". Taking the perspective of a physicist, we should note that the effect of both transformations on the description of the

system is the same, and hence, that there is only a philosophical distinction between active and passive transformations. For an opposing viewpoint, see for instance Ref. [8].

Another distinction made by some authors, is that between internal and external symmetry transformations. An external symmetry, which is also sometimes called a spacetime symmetry, is said to involve a transformation of spacetime coordinates, while internal symmetries concern properties of the fields other than its spacetime coordinate. For example, the phase-rotation symmetry of Section 1.3.1 would be an internal symmetry, whereas the translations in Section 1.3.3 are an example of external or spacetime symmetries. Notice that external symmetries encompass not just translations and rotations, but also for example dilatations and boosts. From a practical physical point of view, there is no fundamental difference between breaking a global spacetime symmetry or a global internal symmetry: in either case, observable effects result from the transformation properties of the physical fields [4]. This observation not withstanding, some specific physical effects may of course be special to certain types of symmetry. For instance, there are cases in which the dispersion relation of Nambu–Goldstone modes is fractional (that is, $\omega \propto q^{\gamma}$ with $\gamma$ non-integer), and these only occur for specific broken spatial symmetries [9].

## 1.5  Gauge freedom

Some transformations that can be applied to our model descriptions may leave the Lagrangian or Hamiltonian invariant, and yet have no physical consequence whatsoever. They do not give rise to Noether currents or charges, cannot be spontaneously broken, and are not associated with any sort of Nambu–Goldstone modes. Instead, these transformations appear purely as a mathematical property of the models with which we choose to describe nature. Any such mathematical transformation that leaves the description of nature invariant but does not correspond to a physical effect, may be called a *gauge freedom*. These freedoms can take many forms, and often complicate the interpretation of how best to represent a physical system.

**gauge freedom**

### 1.5.1  Relabelling your measuring rod

A straightforward example of a gauge freedom is the fact that in any description of nature, the origin of the coordinate system may be freely chosen. Surely, the physics of any system, object, field theory, or anything else cannot depend upon this choice. More generally, we are free to choose any type of coordinate system that we like, be it Cartesian, spherical or something more exotic. We are even free to switch from using one type of coordinate system to another at any point in time. Nothing physical ever changes as a result of this choice. The equations we use may look different in different coordinate systems, but they represent the same physical object.

The freedom of choosing axes does not apply only to the coordinates of space and time. As soon as we set out to measure any physical quantity whatsoever, we must choose a scale, which must have a zero and a set distance between units. Consider temperature, for example. Nothing physical changes in the system whose temperature we take, when we replace the thermometer's Celcius scale with one using Fahrenheit. Both scales correspond to an arbitrary choice of zero, and an arbitrary distance between units, in the sense that significance is arbitrarily assigned to the freezing point of water or the average temperature of a human body. The zero of the Kelvin scale is less arbitrary, but still, no physical process will change the moment you express its temperature in Kelvins, rather than Celcius or Fahrenheit. In fact, the term "gauge invariance" was coined by Hermann Weyl in 1919 in German as *eichinvarianz*, where *eich* (gauge) refers to the scale or standard dimension used by a measurement device [10]. It is the 'choice of tick marks', and we are free to choose the ticks on our measurement devices, without ever affecting the physical properties of the objects we measure.

Right now, this emphasis on the arbitrariness of the choice of coordinate system may sound a bit pedantic, but exactly the same arbitrariness underlies more sophisticated forms of gauge freedom. It is therefore worthwhile keeping this simple example in mind whenever gauge freedoms appear in any model of physics.

### 1.5.2 Superfluous degrees of freedom

A perhaps more profound type of gauge freedom encountered in many theories of physics, appears when new degrees of freedom are introduced to simplify our mathematical description of nature. They are often dynamic fields, which have equations of motion of their own, but which do not correspond to any observable, physical quantities.

The most familiar example of this type of gauge freedom can be found in Maxwell's equations of electromagnetism. Here, the physical observables are the electric and magnetic fields **E** and **B**. These fields do not take arbitrary forms, but are *constrained* by the Faraday–Maxwell equation and the requirement that the magnetic field is solenoidal:

**Maxwell electromagnetism**

$$\nabla \times \mathbf{E} + \partial_t \mathbf{B} = 0, \qquad\qquad \nabla \cdot \mathbf{B} = 0. \tag{1.35}$$

These equations fix three of the six components that together make up **E** and **B**. The constraints can be explicitly enforced by writing **E** and **B** in terms of the scalar and vector potentials $V$ and **A**:

$$\mathbf{E} = -\nabla V - \partial_t \mathbf{A}, \qquad\qquad \mathbf{B} = \nabla \times \mathbf{A}. \tag{1.36}$$

Written in this way, it is clear that the divergence of **B** vanishes and the Faraday–Maxwell equation is satisfied, regardless of what form the fields $V$ and **A** take.

Using the scalar and vector potentials, Maxwell's equations too take on a more convenient form. Introducing the four-potential $A_\mu = (\frac{1}{c}V, \mathbf{A})$ and the relativistic gradient $\bar{\partial}_\mu = (\frac{1}{c}\partial_t, \partial_m)$, the Maxwell action can be written as:

$$S_{\text{Maxw}} = \frac{1}{2\mu_0} \int \mathrm{d}t\mathrm{d}^3x \left( \frac{1}{c^2}\mathbf{E}^2 - \mathbf{B}^2 \right) = \frac{1}{4\mu_0} \int \mathrm{d}t\mathrm{d}^3x \, (\bar{\partial}_\mu A_\nu - \bar{\partial}_\nu A_\mu)^2. \tag{1.37}$$

Here, $c$ is the speed of light and $\mu_0$ is the vacuum permeability or magnetic constant. The equations of motion associated with this action are precisely the other two Maxwell's equations. It can be easily checked that the action is left invariant when the relativistic gradient of an arbitrary scalar field $\alpha(x)$ is added to the vector potential:

**gauge transformation**

$$A_\nu(x) \to A_\nu(x) + \bar{\partial}_\nu \alpha(x). \tag{1.38}$$

This transformation does not affect the physical fields **E** and **B**. It is therefore not a (local) symmetry, but rather a *gauge transformation*. The four-potential $A_\nu(x)$ is often called a *gauge field*, or *gauge potential* to reflect this role.

**gauge field**

One way to understand why a gauge freedom emerges from expressing the Maxwell equations in terms of the four-potential, is to note that $A_\mu$ has four components, while only three components are needed to completely determine **E** and **B** after implementing the constraints of Eq. (1.35). The fourth component of $A_\mu$ is redundant, and there is some freedom in choosing its value. Since the Maxwell equations take on a simple form in terms of $A_\nu$, it is often convenient to do calculations in terms of the four-potential, rather than **E** and **B**. At the end of any calculation, however, the final results should not depend on the superfluous degree of freedom which was introduced purely for mathematical convenience. That is, performing a gauge transformation of the type of Eq. (1.38) can never affect any predictions for physical observables.

It should be noted at this point, that gauge freedom is referred to as "gauge symmetry" throughout much of the physics literature. Although the action is left invariant by gauge transformations, however, it is misleading to call them symmetries, because they do not correspond to the measurable properties of any physical degree of freedom. There are no conserved currents or charges associated with gauge transformations. Gauge freedoms can also never be broken. Not only because they are local transformations, which cannot be spontaneously broken anyway (see Exercise 2.3), but also, and more fundamentally, because gauge freedoms do not correspond to any physical manipulation and therefore there cannot exist any measurable quantity that could conceivably be observed to vary under a gauge transformation.

> Gauge transformations are not symmetries.

To make matters even more confusing, global transformations of internal symmetries, and in particular global phase rotations like Eq. (1.19), are sometimes called "global gauge transformations" [11–13]. This terminology is simply outdated. A transformation is either a symmetry or a gauge freedom, and the two notions should not be mixed.

Despite the fact that gauge freedoms are concerned with superfluous degrees of freedom, they are not just obnoxious complications arising from our ineptness in finding better mathematical representations of the laws of nature. In fact, our understanding of elementary particle physics relies heavily on the structure of gauge transformations, with the gauge fields (connected to the W, the Z, the photon and the gluons) appearing as force fields mediating interactions between fermions. No matter how useful, however, gauge freedom is never a kind of symmetry, and much confusion can be avoided by taking that fact to heart.

The gauge freedom in Maxwell electromagnetism actually appears in conjunction with a physical symmetry. In the presence of electrically charged matter, the part of the action describing the interaction between charges and the electromagnetic fields can be written as:

$$S_{\text{int}} = \int \mathrm{d}t\mathrm{d}^3x \; eA_\nu j^\nu. \tag{1.39}$$

Here, $j^\nu$ is the Noether four-current of a global $U(1)$ symmetry, and $e$ is the coupling constant, which in this case is the elementary electron charge. This form of the action, tying gauge fields to matter fields, is known as *minimal coupling* and appears more generally in theories **minimal** that combine symmetries and gauge freedom. Now, if we perform a gauge transformation of **coupling** the type of Eq. (1.38), we obtain an additional term in the action:

$$S_{\text{int}} \to S_{\text{int}} + \int \mathrm{d}t\mathrm{d}^3x \; e(\bar{\partial}_\nu\alpha)j^\nu = S_{\text{int}} - \int \mathrm{d}t\mathrm{d}^3x \; e\alpha(\bar{\partial}_\nu j^\nu). \tag{1.40}$$

This expression clearly shows that the gauge fields can only ever be minimally coupled to a conserved Noether current, satisfying $\bar{\partial}_\nu j^\nu = 0$, because only then the action will be invariant under gauge transformations. The local $U(1)$ gauge freedom in the Maxwell action is therefore tightly connected to the simultaneously present global $U(1)$ symmetry.

There is another way to understand the link between global symmetries and gauge freedom, which uses Noether's *second* theorem. Briefly stated, it says that if the transformations **Noether's** that leave the Lagrangian invariant (including both symmetries and gauge freedoms) depend **second** on some parameters $\alpha_n(x)$ and their derivatives $\partial_\mu\alpha_n(x)$, then there are constraints on the **theorem** possible field configurations, regardless of whether or not the equations of motion are satisfied. We will not attempt to prove Noether's second theorem here, or even to state it in general form. Instead, we will illustrate its implications by considering the example of a gauge field $A_\mu$ that is coupled to a complex scalar field $\psi$. As we will discuss at length in Chapter 7, this is the main ingredient of the Ginzburg–Landau theory for superconductivity. Foretelling the

theory of superconductivity, assume that the theory is invariant under a local transformation that acts on the fields as:

$$\delta_s \psi = -\mathrm{i}\alpha(x)\psi, \qquad \delta_s \psi^* = \mathrm{i}\alpha(x)\psi^*, \qquad \delta_s A_\mu = -\frac{\hbar}{e^*}\partial_\mu \alpha(x). \tag{1.41}$$

Here, $e^*$ is the electric charge of the field $\psi$.

We can now write the Euler–Lagrange equation of motion obtained by varying $\mathcal{L}$ with respect to $\psi$ as $E_\psi = 0$, and similarly for variations with respect to $\psi^*$ and $A_\mu$. The expression $E_\psi$ is the term between square brackets in Eq. (1.10). Noether's second theorem then states that:

$$E_\psi(-\mathrm{i}\psi) + E_{\psi^*}(\mathrm{i}\psi^*) = (-\frac{\hbar}{e^*})\partial_\mu E_{A_\mu}. \tag{1.42}$$

Again, this identity holds whether or not the equations of motion are satisfied. In classical physics, only field configurations that obey the equations of motion are usually of any importance, and for these Noether's second theorem reduces to a trivial equation. Within quantum field theory, however, the calculation of quantum corrections involves contributions from so called *off-shell* field configurations that do not satisfy the equations of motion. Noether's second theorem, and the closely related Ward–Takahasi identities, then impose important and influential constraints on the field configurations that need to be considered.

**Ward–Takahashi identity**

Noether's second theorem also gives an alternative way of understanding of the fact that gauge fields can only couple to conserved currents. To see this, consider a situation in which the equations of motion for $A_\mu$ are satisfied, so that $E_{A_\mu} = 0$, and the right hand side of Eq. (1.42) vanishes. Furthermore, the left-hand side can be rewritten as $E_\psi \Delta_s \psi + E_{\psi^*}\Delta_s \psi^*$. Since we also know that for complex scalar fields $\frac{\partial \mathcal{L}}{\partial \psi}\Delta_s \psi + \frac{\partial \mathcal{L}}{\partial \psi^*}\Delta_s \psi^* = 0$, the left-hand side is then precisely of the form $\partial_\mu j^\mu$, with $j^\mu$ the Noether current associated the global part of the symmetry transformation on the field $\psi$. Equating the left and right hand sides gives Noether's (first) theorem, $\partial_\mu j^\mu = 0$, which we can now interpret as saying that as long as the gauge field satisfies its equation of motion, the current it couples to must be a conserved one. This is then true regardless of whether or not the field configuration of $\psi$ itself obeys its equations of motion. A more thorough account of this viewpoint is given in Ref. [14].

A final point: even when the *global $U(1)$* symmetry is spontaneously broken, the gauge freedom and corresponding gauge invariance persists. However, the fact that there is a coupling with gauge fields as in Eq. 1.39 leads to the so-called Anderson–Higgs effect, which will be discussed in detail in Section 7.3.

### 1.5.3 Distinguishing gauge freedom from symmetry

Looking only at the action, there does not seem to be much difference between the $\alpha(x)$ describing local spacetime translations in Eq. (1.33), and the $\alpha(x)$ describing local gauge transformations in Eq. (1.38). The former, however, is a symmetry transformation that yields conserved Noether currents, whereas the latter is a gauge transformation signifying the presence of a superfluous degree of freedom. It would thus be convenient if there were a way of telling these two physically distinct types of local transformation apart, even though both appear as local transformations leaving the action invariant. Fortunately, there is a method for identifying constraints which lead to local gauge freedoms, such as the Faraday–Maxwell equation and the requirement of the magnetic field being solenoidal in electromagnetism. The method may be referred to as the Dirac treatment of Hamiltonian constraints, and starts from the expression of the Hamiltonian in terms of canonical fields and their conjugate momenta. Any relations between these, such that linear combinations of the fields and momenta vanish, then constitute constraints. This in turn implies there are associated redundant degrees of freedom. A detailed discussion can be found in Refs. [15, 16].

The distinction between symmetries and gauge freedom becomes even more subtle for global transformations, which are not easily described in terms of constraints on the Hamiltonian. To illustrate this, consider a many-body spin system whose action is invariant under the global rotation of all spins simultaneously. Surely, the global spin rotation is a symmetry, which may be spontaneously broken into a ferromagnetic arrangement. Taking a different point of view however, one could also argue that the global transformation which seemingly rotates the direction of all spins, really only describes the rotation of the coordinate system that we use to measure the spin direction with. Such a relabelling of coordinates is the archetype of a global gauge transformation, which should not have any physical implications whatsoever, and which cannot be spontaneously broken.

The way out of this paradox lies in the observation we made at the beginning of this chapter, that symmetry can only be defined with respect to a reference. As long as we exclude from the universe any objects that can measure the magnetisation of our material, it is impossible to tell whether the spins have aligned to a certain direction. Global rotations of all spins are then unobservable by construction, even for the ferromagnet. The magnetisation becomes measurable only if we allow some interaction to exist between the magnet and an external reference. For example, even if the spin-rotation symmetry is not spontaneously broken, the paramagnet can be subjected to an externally applied magnetic field which forces the spins to align in a given direction according to the coupling:

$$\mathcal{L}_{\text{coupling}} = -\mathbf{h} \cdot \mathbf{S}(x). \tag{1.43}$$

Here, the spins are described by $\mathbf{S}(x)$, while $\mathbf{h}$ is the uniform applied field. Since the external field is applied in some given direction, the total action including the coupling to the field is no longer invariant under global rotations of the spins, and the ground state may be magnetised. Incidentally, notice that this is an example of explicit symmetry breaking. In contrast, spontaneous symmetry breaking occurs when the ferromagnetic state survives in the limit of the field strength $|\mathbf{h}|$ going to zero, as will be discussed in detail in the next chapter.

**symmetry breaking – explicit**

The crucial observation is now, that in the presence of the external field, the global spin-rotational symmetry and the global rotation of the coordinate system are different. Rotating the coordinate system that we use to define directions is space, implies a simultaneous transformation of *both* the spins $\mathbf{S}(x)$, and the applied field $\mathbf{h}$. After all, the directions of both are described within our arbitrarily chosen coordinate system. It is easily checked that the coupling of Eq. (1.43) is invariant under this global gauge transformation, which therefore applies equally to the ferromagnetic and paramagnetic states. On the other hand, the term $\mathcal{L}_{\text{coupling}}$ is not invariant under the global rotation of only the spins $\mathbf{S}$, keeping the reference field $\mathbf{h}$ fixed. This physical, global symmetry is broken in the ferromagnetic state, whether it be spontaneously or explicitly.

Notice that even though the coupling between the applied field and the magnet vanishes, the fact that an external field may exist in principle is crucial. In other words, the global symmetry that may be broken in a magnet is not simply the rotation of all its spins, but rather the global spin-rotation relative to some reference. To put this in a more mathematically precise formulation, we can consider the applied magnetic field to be generated by a second, external ferromagnet, which itself is also invariant under global spin rotations. Each magnet in complete isolation then has a global symmetry group denoted by $SU(2)$ (see below). As long as interactions between the two magnets are strictly forbidden, the total system of two magnets has a combined $SU(2)_S \times SU(2)_h$ symmetry, where the indices indicate the magnet with which each symmetry is associated. When interactions between the two magnets are allowed, the symmetry of the combined system is reduced to the so-called diagonal subgroup which contains only simultaneous rotations of $\mathbf{S}$ and $\mathbf{h}$. The symmetry breaking that occurs in a ferromagnet is thus the reduction of $SU(2)_S \times SU(2)_h$ to its diagonal subgroup, rather than

**diagonal subgroup**

the breakdown of just $SU(2)_S$ that we might naively expect. As we will see in Section 2.2, this is the case even with spontaneously broken symmetries, for which the coupling to an external field may be infinitely weak, but must necessarily be allowed to exist.

## 1.6 Symmetry groups and Lie algebras

In the final section of this chapter, we give a brief overview of some of the mathematical structure underlying symmetry transformations. This is not intended to be complete treatment of any of the topics discussed but as a reminder or as an entry point towards further study, for which many excellent books may be consulted [17, 18].

### 1.6.1 Symmetry groups

Symmetry transformations correspond to manipulations of a physical system that leave a particular state or action invariant. This definition alone has three important implications that together determine how symmetry transformations are represented mathematically:

1. The combined effect of two consecutive symmetry transformations is also a symmetry transformation. This is obvious because if the first transformation does not affect the state, then the second transformation simply acts on the initial state. Likewise, if both transformations leave the action invariant regardless of what state they act on, then acting consecutively will also have no effect on the action. The consecutive action of two symmetry transformations may be considered an (ordered) *product* of transformations.

2. There always exists a unity, or identity, transformation. This is the transformation that does nothing to *any* state or action it acts on, even non-symmetric ones. It is a symmetry transformation, albeit a trivial one. We can call this trivial symmetry the identity transformation $\mathbb{I}$.

3. For each symmetry transformation $U$ there is an inverse transformation $U^{-1}$ such that $UU^{-1} = U^{-1}U = \mathbb{I}$. The existence of a symmetry transformation implies that for each state, we can identify the state it transforms into under the action of $U$. The inverse transformation can then be defined as the operation that takes the transformed states back to the original ones. It is itself a symmetry transformation, because by the definition of $U$, the transformed and original states are either equal or possess the same action.

These properties are obeyed by the set of symmetry transformations $\{U\}$ for any given system, and endow them with the mathematical structure of a *group*. The description of groups and **group** their various properties and manipulations is a large and vibrant area of mathematics. If this is your first encounter with group theory, it is probably helpful to consult a more complete text, such as Refs. [17,18]. Here, we will only remind you of some of the most important ingredients of group theory, with a focus on the parts relevant to the discussion of spontaneous symmetry breaking.

Let us start with some definitions. First, if a set of symmetry transformations can be parametrised by one or more continuous variables, the group they form is called *continuous*. If it cannot, the group is said to be *discrete*. Furthermore, if the number $N$ of transformations within a discrete group is finite, it is called a finite group of order $N$, otherwise it is an infinite discrete group. Second, if all symmetry transformations in a group commute, that is **Abelian** $U_1 U_2 = U_2 U_1$ for all possible pairs of transformations $U_1, U_2$ in the group, the group is called **group** *Abelian* or *commutative*. If not all elements commute, the group is called *non-Abelian* or *non-commutative*. **non-Abelian**

Groups appear in many places throughout all of physics. As a reminder of just how common **group** they are, consider the following examples:

- The trivial group consists of one element, the identity $e$ or $\mathbb{I}$.

- The cyclic group $\mathbb{Z}_n$ or $C_n$ is a finite and Abelian group of order $n$, which describes the rotation symmetries of a regular, $n$-sided polygon (triangle, square, etc.)

- The dihedral group $D_n$ is a finite group of order $2n$ describing the rotations and reflections of a regular, $n$-sided polygon. It is non-Abelian if $n$ is larger than two.

- The discrete translation group $\mathbb{Z}^D$ describes translations on a regular $D$-dimensional lattice. It is discrete, infinite and Abelian.

- The translation group $\mathbb{R}^D$ describes translations of $D$-dimensional empty space. It is continuous and Abelian.

- The rotations of a circle are given by the group $SO(2)$ of real, orthogonal $2 \times 2$-matrices with determinant 1. It is continuous and Abelian.

- The rotations and reflections of a circle are given by the group $O(2)$ of real, orthogonal $2 \times 2$-matrices. It is continuous and non-Abelian.

- The rotations of a sphere are given by the group $SO(3)$ of real, orthogonal $3 \times 3$-matrices with determinant 1. It is continuous and non-Abelian.

- The rotations and reflections of a sphere are given by the group $O(3)$ of real, orthogonal $3 \times 3$-matrices. It is continuous and non-Abelian.

- The set of complex unitary $2 \times 2$-matrices with determinant 1 form a continuous and non-Abelian group called $SU(2)$. It describes spin rotations and the weak force.

- The continuous, non-Abelian group hosting complex unitary $3 \times 3$-matrices with determinant 1 is $SU(3)$. It underlies the strong nuclear force.

---

**Exercise 1.4 (Dihedral groups)** Consider dihedral groups $D_n$. Denote the identity by $e$, the rotations over multiples of $2\pi/n$ by $r, r^2, \ldots, r^{n-1}$, and the reflections by $s, sr, sr^2, \ldots, sr^{n-1}$, where $s^2 = e$.
**a.** Make a table of all the multiplication rules ($g_1$ in columns, $g_2$ in rows and their product $g_1 g_2$ as the entries) within the group $D_4$.
**b.** Check explicitly that every element in $D_4$ has an inverse.
**c.** Give the multiplications for general $n$ of i) two rotations $r^k$ and $r^l$; ii) a rotation $r^k$ and the reflection $s$; iii) two reflections $sr^k$ and $sr^l$.

---

A subset $H$ of the elements making up a group $G$ may by itself also form a group, if it possesses all of the three properties that define a group. In particular, the multiplication of elements in the subset must be closed, so that $h_1 h_2 \in H$ for all $h_1$ and $h_2$ in $H$. If the subset is a group by itself, it is called a *subgroup* of $G$. The set of all rotations within the dihedral **subgroup** group $D_n$, for example, form a subgroup, known as the cyclic group $C_n$. Similarly, translations in the $x$-direction are a subgroup of all translations in three-dimensional space. Subgroups appear in the discussion of symmetry breaking, because all the transformations that still leave the system invariant after a system has gone through a symmetry-breaking transition, form a subgroup of the original symmetry group.

For a given group $G$ and subgroup $H$, we may identify a (left) *coset* for each element $g \in G$, denoted by $gH$. It is defined to be the set of all elements $gh$ with $h \in H$.[4] The collection of all **coset** cosets associated with different elements of $g$ together, form a set that is denoted by $G/H$, and called the *quotient set*. Neither the individual cosets, nor the quotient set are typically groups **quotient set**

---

[4]The right coset is defined as $\{hg \mid h \in H\}$. Left and right cosets are not generally identical, but there is a bijection between them. In these lecture notes we use only left cosets, and refer to them as just cosets.

by themselves. As an example, consider the group of rotations of a regular hexagon, $C_6$. It has six elements, which may be written as $r^k$, with $k = 0, \ldots, 5$. Here, $r$ is a rotation over $2\pi/6$, and $r^0 = e$. This group has a subgroup $C_2$, consisting of the rotations of a line (a two-sided polygon). The subgroup as elements $e$ and $r^3$. The unique cosets that can be generated from $C_6$ and its subgroup $C_2$ are $\{e, r^3\}$, $\{r, r^4\}$ and $\{r^2, r^5\}$. The quotient set $C_6/C_2$ is a set with three elements, each of which may be represented by a single element from its corresponding coset, here for instance $\{e, r, r^2\}$. Cosets play an important role in the classification of broken-symmetry states, as we will see in Section 2.5.1.

---

**Exercise 1.5 (Equivalence and Quotient sets)** Two elements in the same coset may be said to be equivalent. That is, $g_1 \sim g_2$ if $g_1 = g_2 h$ for some $h \in H$.
Show that this is indeed an equivalence relation in the mathematical sense, by showing that it satisfies the three properties: i) $g \sim g$; ii) if $g_1 \sim g_2$ then $g_2 \sim g_1$; iii) if $g_1 \sim g_2$ and $g_2 \sim g_3$ then $g_1 \sim g_3$, for all $g, g_1, g_2, g_3 \in G$.
   The set of cosets, that is, the quotient set $G/H$, can alternatively be defined as the set of equivalence classes under this definition of equivalence.

---

**Exercise 1.6 (Subgroups of Dihedral groups)** In the dihedral group $D_n$, the set $\{e, s\}$ forms a subgroup.
**a.** Find all cosets with respect to this subgroup.
If $n$ is even, the element $\{r^{n/2}\}$ commutes with all other elements. It is therefore called a *central* element, and the subgroup $\{e, r^{n/2}\}$ is a *normal subgroup*. Normal subgroups have the property that $ghg^{-1} \in H$ for all $g \in G$ and all $h \in H$.
**b.** Find all cosets with respect to the normal subgroup $\{e, r^{n/2}\}$.
**c.** The set of cosets of a normal subgroup has a group structure itself. What is the group of the cosets in this case?

*central element*

*normal subgroup*

---

### 1.6.2 Lie groups and algebras

If the parameters of continuous symmetry transformations in a group are smooth, the group is called a *Lie group*. To be precise, a Lie group is a continuous group with the structure of a differentiable manifold, but for our purposes, having a description in terms of smooth functions suffices. All continuous groups encountered in these lecture notes, including all examples we have already seen, are Lie groups. The number of variables needed to parametrise the full set of continuous transformations, is called the *dimension* of the Lie group.

*Lie group*

   Unlike discrete groups, Lie groups always contain transformations that are infinitely close to the identity. The fact that the transformations can be written in terms of differentiable functions, then allows for them to be expanded around the identity. For example, consider the continuous transformation $U_\alpha = \exp(i\alpha_a Q_a)$, with $\alpha_a$ a set of small parameters and $Q_a$ a set of Hermitian operators, both labeled by the index $a = 1, \ldots, N$ (because symmetry transformations must be unitary, they can always be written as the exponent of a Hermitian operator). The identity is the symmetry transformation with all $\alpha_a$ equal to zero, and a general continuous transformation can be expanded around the identity as $U_\alpha = 1 + i\alpha_a Q_a + \mathcal{O}(\alpha^2)$. Because continuous symmetry transformations close to the identity are determined entirely by the action of the operators $Q_a$, these are called the *generators* of the Lie group. We actually already used an expansion around the identity in our proof of Noether's theorem, when we considered infinitesimal symmetry transformations in Eq. (1.5), and found the generators $Q_a$ of the symmetry to be conserved Noether charges.

   The sum of two symmetry generators is itself a generator, which implies that the set of symmetry generators has the mathematical structure of a vector space. Generators can also

be multiplied, but their product is generally not itself a generator. Instead their commutator (or more generally the Lie bracket) is a linear combination of other generators:

$$[Q_a, Q_b] = i \sum_c f_{abc} Q_c . \qquad (1.44)$$

Here the *structure constants* $f_{abc}$ are real numbers. The vector space of symmetry generators together with their commutation relations is called a *Lie algebra*. The Lie algebra is determined entirely by its structure constants. If a group is Abelian, then all structure constants of the Lie algebra are zero.

**structure constants**

**Lie algebra**

    While each Lie group possesses a single Lie algebra, a Lie algebra does not uniquely define the Lie group. The reason is, that the Lie algebra describes the structure of the symmetry generators, rather than the symmetry transformations themselves. The two are equivalent for continuous transformations close to the identity, but in addition to those, the complete group may also contain some discrete transformations, that cannot be expanded around the identity. For example, the groups $SO(3)$ and $O(3)$ have the same Lie algebra, describing continuous rotations in three dimensions, but the discrete reflections present in $O(3)$ are not captured by the Lie algebra. The Lie algebra is important for the structure of Nambu–Goldstone modes, discussed in Section 3.2.

---

**Exercise 1.7 ($SU(2)$ Lie Algebra)** The generators for the Lie algebra of the group $SU(2)$ are the Pauli matrices:

$$\sigma_x = \frac{1}{2}\begin{pmatrix} 0 & 1 \\ 1 & 0 \end{pmatrix}, \qquad \sigma_y = \frac{1}{2}\begin{pmatrix} 0 & -i \\ i & 0 \end{pmatrix}, \qquad \sigma_z = \frac{1}{2}\begin{pmatrix} 1 & 0 \\ 0 & 1 \end{pmatrix}, \qquad (1.45)$$

**a.** Show that the structure constants are given by the Levi-Civita symbol $\varepsilon_{abc}$, which is defined by the element $\varepsilon_{xyz}$ being one, and by being completely antisymmetric in $a$, $b$, and $c$.

**b.** Using the property of the Pauli matrices that $(\sigma_a)^2 = \mathbb{I}$ for $a = x$, $y$, or $z$, and the operator expansion $e^A = \sum_{n=0}^{\infty} \frac{1}{n!} A^n$, find an explicit expression for the continuous transformations (or Lie group elements) $U(\alpha_a) = e^{i\alpha_a \sigma_a}, a = x, y, z$. Do *not* assume $\alpha_a$ to be small.

**c.** Explicitly write out the multiplication of $U(\alpha_x)$ and $U(\alpha_y)$.

---

### 1.6.3 Representation theory

Groups and algebras are abstract mathematical concepts. They consist of abstract elements that are defined only by the way they can be multiplied or added together. To tie these elements to operators carrying out symmetry transformations, we need *representations* of the abstract group structures.

**representation**

    In practice, we almost always consider a representation to be a set of matrices which have the same structure, in terms of rules for multiplication and addition, as the group or algebra they represent. As an example, consider the cyclic group $C_n$. On an abstract level, this group is defined to be a set of $n$ elements written as $r^k$ with $k$ between zero and $n$, together with the multiplication rule $r^k r^l = r^{k+l \mod n}$. If we consider a regular $n$-sided polygon centred at the origin of two-dimensional space, however, we might want to identify the abstract operation $r$ with a rotation of the polygon over an angle of $2\pi/n$ around its centre. The representation of $r$ is then a $2 \times 2$ matrix acting on vectors $(x, y)$ which describe coordinates in the two-dimensional space. Explicitly, it would be given by:

$$r^k \mapsto \begin{pmatrix} \cos 2\pi k/n & \sin 2\pi k/n \\ -\sin 2\pi k/n & \cos 2\pi k/n \end{pmatrix}. \qquad (1.46)$$

This is called a real and two-dimensional representation, because $r^k$ is given as a $2 \times 2$ matrix with real elements.

We can also describe the polygon in a different way. Instead of using Cartesian coordinates of the form $(x, y)$, we can draw the polygon at the centre of the complex plane, in which points are denoted by $x + iy$. The same rotations of the same polygon would in that case be represented by $r^k \mapsto \mathrm{e}^{\mathrm{i}2\pi k/n}$. This is called a complex, one-dimensional representation, because $r^k$ is written as a $1 \times 1$ matrix with complex elements. We thus see that the abstract notion of rotating a polygon can be represented in different ways, depending on how we choose to describe the polygon. The important point to notice, is that regardless of which representation we choose, its elements satisfy the same properties, or rules of multiplication and addition, as the abstract group elements.

In most cases, which representation of a group is being used, will be obvious from the context in which it appears. Consider, for example, a theory with two complex fields $\psi_1$ and $\psi_2$, and the Lagrangian:

$$\mathcal{L} = \frac{1}{2}|\partial_\nu \psi_1|^2 + \frac{1}{2}|\partial_\nu \psi_2|^2 - V(|\psi_1|^2 + |\psi_2|^2). \tag{1.47}$$

Here, $V$ is some function that depends only on the combination of fields $|\psi_1|^2 + |\psi_2|^2$. You will probably notice that this Lagrangian in invariant under transformations of the fields $\psi_1$ and $\psi_2$ that keep the value of $|\psi_1|^2 + |\psi_2|^2$ fixed. This immediately suggest a natural representation for the combination of the two fields as a complex-valued vector $\psi = (\psi_1, \psi_2)$, whose length is held fixed by the symmetry transformations. The group of operations that act on complex two-component vectors and keeps their length fixed, is $SU(2)$, and its natural representation in this case is in terms of two-dimensional matrices:

$$\psi \rightarrow \mathrm{e}^{-\mathrm{i}\alpha_a \sigma_a}\psi, \qquad\qquad \psi^* \rightarrow \mathrm{e}^{\mathrm{i}\alpha_a \sigma_a}\psi^*. \tag{1.48}$$

Here the $\sigma_a$ are Pauli matrices, which generate the Lie algebra of $SU(2)$, see Exercise 1.7. Writing out the components of the vectors and matrices explicitly, the symmetry transformations are:

$$\psi_m(x) \rightarrow \psi'_m(x) = \sum_{a,n} \mathrm{e}^{-\mathrm{i}\alpha_a (\sigma_a)_{mn}}\psi_n(x). \tag{1.49}$$

Notice that here, $m, n = 1, 2$ are indices of the 2-vectors $\psi$, while $a = x, y, z$ denotes the index of the Pauli matrices.

Within the Lagrangian formalism, it is important to keep in mind the distinction between the representations of a symmetry group, and operations on Hilbert space. In the example of the $SU(2)$-invariant 2-vector field, for instance, the matrices $\sigma_a$ act on the fields $\psi$, which themselves are (eigenvalues of) operators acting on Hilbert space. In contrast to what we saw in the Hamiltonian formalism in section 1.2.2, the representations of the Lie algebra generators, $\sigma_a$, in this case do not correspond directly to the symmetry generators, $Q_a$, describing conserved Noether charges. Like any other observable in the Lagrangian formulation, the Noether charges can always be expressed in terms of the fields $\psi_n$ themselves. In this specific case, they are:

$$
\begin{aligned}
Q_a &= \pi(-\mathrm{i}\sigma_a \psi) + (\mathrm{i}\psi^\dagger \sigma_a)\pi^\dagger \\
&= -\mathrm{i}(\partial_t \psi^\dagger)\sigma_a \psi + \mathrm{i}\psi^\dagger \sigma_a(\partial_t \psi) \\
&= -\mathrm{i}(\partial_t \psi_m^*)(\sigma_a)_{mn}\psi_n + \mathrm{i}\psi_m^*(\sigma_a)_{mn}(\partial_t \psi_n).
\end{aligned} \tag{1.50}
$$

These conserved charges $Q_a$ are clearly related to the representations $\sigma_a$ of the Lie group generators, but the two are not the same thing.

> **Exercise 1.8 (Representations of $U(1)$ and $\mathbb{Z}$)**
> **a.** The one-dimensional complex representation of the group $U(1)$ of phase rotations over an angle $\alpha$ are given by $e^{in\alpha}$, for $n \in \mathbb{Z}$.
> Show that this representation obeys the group properties of $U(1)$.
> **b.** The one-dimensional complex representation of the group $\mathbb{Z}$ of lattice translations over a multiple $n$ of the elementary lattice vector are given by $e^{i\alpha n}$ with $\alpha \in [0, 2\pi)$.
> Show that this representation obeys the group properties of $\mathbb{Z}$.
> **c.** The groups $U(1)$ and $\mathbb{Z}$ also have zero-dimensional representations. Show that the *trivial representation* with only the identity element obeys the group properties of both groups (and indeed of any group).
> Representations where distinct group elements are represented by distinct matrices are called *faithful*. This is an example of a non-faithful representation.

## 2 Symmetry breaking

A symmetric system, described by a Hamiltonian, Lagrangian, or action that is left invariant under a unitary transformation, typically has a symmetric equilibrium configuration. In quantum physics, one can even prove that any symmetric system either has a unique and symmetric ground state, or a degenerate set of ground states related by the symmetry transformation. Looking around in our everyday world, however, we rarely see any truly symmetric objects. How things manage to be in stable configurations that seemingly evade the symmetries of the laws of nature, is explained by the theory of spontaneous symmetry breaking.

> Spontaneous symmetry breaking (SSB) is the phenomenon in which a stable state of a system (for example the ground state or a thermal equilibrium state) is not symmetric under a symmetry of its Hamiltonian, Lagrangian, or action.

**symmetry breaking – spontaneous**

Before delving into a formal discussion, we will first give a short description of the consequences of spontaneous symmetry breaking. This allows us to introduce some of the central concepts associated with this topic: the *order parameter*, the *tower of states*, effectively *restricted configuration space* and *singular limits*. We will then give three detailed examples of spontaneous symmetry breaking: in classical physics, the harmonic solid and the antiferromagnet. The final part of this chapter is devoted to three recurring themes in symmetry breaking: the thermodynamic limit, the tower of states, and stability.

### 2.1 Basic notions of SSB

When the state $|\psi\rangle$ of a system is not left invariant by a symmetry transformation $U$ of the Hamiltonian that describes the system, the state is said to have spontaneously broken the symmetry. This single observation immediately implies that for every symmetry-broken state there exist a multitude of related states, which all share the same energy. After all, for a given transformation $U$, the fact that the state $|\psi\rangle$ breaks the symmetry implies that it is different from $U|\psi\rangle$. At the same time, the Hamiltonian being symmetric implies that it commutes with the symmetry transformation. The two inequivalent states $|\psi\rangle$ and $U|\psi\rangle$ must therefore have the same energies. Continuing this way, we can define a whole set of distinct symmetry-broken states, which all have the same energy, by performing all possible symmetry transformations $U$ on a given initial symmetry-broken state $|\psi\rangle$. For example, a rock or other solid piece of material is typically localised in a single position in space, while the Hamiltonian for objects in homogeneous space is translationally invariant. Moving the rock to another position yields a

distinct state, but with the same energy. The set of all these inequivalent but degenerate states consists of the rock being localised in all possible positions.

The set of symmetry-related states allows us to define the *order parameter operator* $\mathcal{O}$ as an operator whose eigenstates are the inequivalent states in the set, and whose eigenvalues are different and non-zero for each. Additionally, the order parameter operator should have zero expectation value for symmetric states. This order parameter operator, in general, does not commute with the Hamiltonian (barring a few important exceptions that will be discussed in Section 3.2). In the example of a solid object, its symmetry-broken states are eigenstates of the position operator $X$, which serves as an order parameter operator. The Hamiltonian is translationally invariant, implying that it commutes with the momentum operator $P$, and therefore not with the order parameter operator.

**order parameter**

The fact that symmetry-broken states are eigenstates of an operator that does not commute with the Hamiltonian, raises a conundrum: these states cannot be eigenstates of the Hamiltonian. Somehow, all the symmetry-broken states that you see around you every day, including the table in front of you right now, are not energy eigenstates. In fact they cannot even be (thermal) mixtures of energy eigenstates, and they are therefore not in thermal equilibrium!

That these states can nonetheless exist and be stable, is owing to a large degree to the *singularity* of the *thermodynamic limit*. The thermodynamic limit for a system of $N$ particles and volume $V$, is defined by taking both the limit $N \to \infty$ and $V \to \infty$, while keeping the ratio $N/V$ fixed. This means that intensive quantities like density and temperature do not change as the limit is taken, while extensive quantities like $N$ and entropy grow to infinity. It turns out that, even though the order parameter operator and the Hamiltonian in general do not commute, the commutator expectation value vanishes in the thermodynamic limit. Furthermore, the symmetry-broken states become orthogonal to one another in that limit, and they become degenerate with the symmetric exact eigenstates of the Hamiltonian. Precisely in the limit then, the symmetry-broken states actually are eigenstates of $H$, and may occur in thermal equilibrium. The fact that the situation for truly infinite $N$ and $V$ is qualitatively different from that of any finite volume, no matter how large, makes the limit *singular*.

**singular limit**

**thermodynamic limit**

A main theme that will be emphasised throughout these lecture notes, is that the thermodynamic limit is a mathematical idealisation that serves as a guide to what happens in the real world, but does not actually describe it. A real system may have $N$ and $V$ very large, but not infinite. So the question of how real, finite-sized objects may be observed in symmetry-broken configurations, still remains.

In addressing this question, the first observation to be made is that the spectra of symmetric Hamiltonians have some common properties. Using a Fourier transformation, the Hamiltonian can always be separated into a centre-of-mass, or collective part at zero wave number, and a part for finite wave numbers containing all possible information about the internal degrees of freedom. Moreover, these two parts commute. The essential observation is now that, in order to describe the breaking of a global symmetry, we only need to consider the collective part of the Hamiltonian. For example, a solid object localised in space has a collective Hamiltonian that describes its centre-of-mass position and the collective motion of all of its $N$ atoms of mass $m$ being displaced in unison. In free space, the collective and symmetric Hamiltonian is just that of a free particle of mass $mN$, and its lowest energy levels are spaced by an amount of the order of $\Delta E \sim 1/N$. These low-energy eigenstates of the collective Hamiltonian make up what is called the *tower of states*. The states in this tower are highly collective and non-local. That is, they cannot be written as *product states* of the form $|\psi\rangle = \otimes_j |\psi_j\rangle$, with $|\psi_j\rangle$ a single-particle state. For the solid object, the tower of states consists of eigenstates of total momentum, which are collectively delocalised over all of space.

**tower of states**

**product state**

Because they are non-local, the energy eigenstates in the tower are increasingly *unstable* towards local interactions as the system size is increased. This instability of exact energy

eigenstates prevents them from being realised in our everyday world, because even the weakest asymmetric interaction suffices to destabilise them completely. Stable states, on the other hand, are local, may be written as tensor products of single-particle states, and are not very sensitive to local perturbations. For the rock, they are the localised eigenstates of the position operator. They are superpositions of states in the tower, and are not generally energy eigenstates. However, since the energy spacing within the tower scales as $1/N$, the energy uncertainty of the stable states is very small for large systems. Similarly, the stable states are not orthogonal, but their overlap drops as $e^{-N}$, so that once the system ends up in a stable state, the probability of tunnelling to another state is exponentially suppressed. Most importantly, these stable states are not symmetric, and they are degenerate in the sense that they have the same energy expectation value, which is very close to the energy of the exact ground state of the Hamiltonian.

Spontaneous symmetry breaking for finite-sized objects is the phenomenon that a system may exist in a stable state that is not an exact eigenstate of its symmetric collective Hamiltonian. But how does a system single out just one of the many distinct, stable, symmetry-broken states? Clearly the symmetric Hamiltonian cannot account for this, and one is forced to consider an external perturbation which explicitly breaks the symmetry and favours one of the stable states over all others. A large symmetric system is exceedingly sensitive to such symmetry-breaking disturbances, and a perturbation with an energy scale as small as $\sim 1/N$ suffices to single out a particular stable state. This is the *spontaneous* part of the symmetry breaking: no matter how small the perturbation, it will entirely determine the fate of sufficiently large systems [5]

Considering the symmetric collective Hamiltonian together with an arbitrarily small symmetry-breaking perturbation, the combined ground state of the full system is a stable symmetry-broken state. In the thermodynamic limit, it extrapolates to an eigenstate of the order parameter operator. On the other hand, in the strict absence of any perturbations, the symmetric ground state is the true ground state for systems of any size. The fact that the state encountered in the thermodynamic limit changes qualitatively if even an infinitesimally weak external perturbation is added or removed, is a clear manifestation of its singular nature.

Because the overlap between distinct symmetry-broken states is exponentially suppressed for large system sizes, we can treat a system with a spontaneously broken symmetry as if, for all practical purposes, it has a single, symmetry-broken ground state. All other symmetry-broken states are inaccessible to the system, and its entire dynamics takes place in a restricted part of Hilbert space that contains the symmetry-broken state and its excitations. In other words, **configuration space** (or phase space) is effectively restricted to a small subspace. As long as we are interested in the physics of the symmetry-broken phase (and not for instance in phase transitions), we need to consider only an effective Hamiltonian describing physics within the small symmetry-broken subspace, and we may safely disregard the rest. This an important instance of *ergodicity breaking* where part of the phase space is not accessible on reasonable timescales, and global thermal equilibrium can never be reached [20, 21]. It also occurs in disordered systems such as glasses, which we discuss briefly in Section A.1.

*configuration space – restricted*

*ergodicity breaking*

## 2.2 Singular limits

The possibility of spontaneously breaking a symmetry is closely related to the singular nature of the thermodynamic limit. Although singular limits occur throughout all parts of physics, and indeed daily life, they are not commonly encountered in standard physics curricula. To develop some feeling for them, consider the example of a classical perfect cylinder. If we

---

[5]The term *spontaneous symmetry breaking*, while accurate in this sense, is not ideal when trying to explain symmetry breaking to non-specialists. It was introduced by Baker and Glashow [19], and so far nobody has come up with a workable alternative.

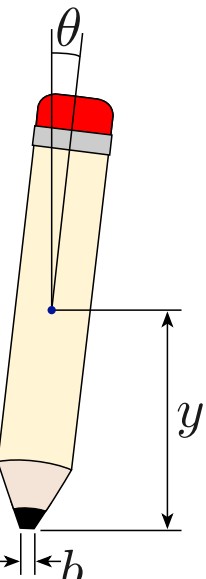

Figure 2.1: Balancing a blunted pencil becomes increasingly difficult if we sharpen it. The limits $b \to 0$ and $\theta \to 0$ do not commute, and provide an example of singular limits.

sharpen such a cylinder so that it ends in a tip on one end, we get a pencil-shaped object, which we could try and balance on a table, as shown schematically in Figure 2.1 below. In this picture, $y$ is the distance between the centre of mass of the pencil and the surface of the table, $\theta$ is the angle between the central axis of the pencil and a line normal to the table surface, and $b$ is the diameter of the base area of the pencil's tip, which we assume to be flat and circular.

If we manage to perfectly balance the pencil, so that $\theta$ is strictly zero, it will be symmetric under rotations around the axis normal to the table surface. The equations of motion describing the pencil, which include the effects of inertia as well as gravity acting perpendicular to the table surface, are invariant under the same rotations. If you ever tried to balance a sharp pencil on its tip in real life, however, you probably discovered this is a very hard thing to do. In fact, if the pencil is sharp enough, it becomes practically impossible to balance it, and the pencil will always fall over when released. As the pencil drops flat onto the table, it does so in a single direction, and will thus no longer be symmetric under rotations around the axis normal to the table. That is, by tipping over in a specific but uncontrollable direction, the pencil has spontaneously broken its symmetry.

More precisely, the fact that it seems impossible to balance a sharp pencil in a symmetric state, can be described mathematically in terms of two limits. Trying to balance the pencil perfectly corresponds to taking the limit $\theta \to 0$, while sharpening the tip of the pencil corresponds to taking $b \to 0$. The spontaneous breakdown of rotational symmetry can then be understood to be a consequence of the fact that these two limits do not commute: non-commuting limits

$$\lim_{b \to 0} \lim_{\theta \to 0} y > 0,$$

$$\lim_{\theta \to 0} \lim_{b \to 0} y = 0. \tag{2.1}$$

That is, if we manage to really, perfectly balance a pencil (taking $\theta$ to zero first), it will stay upright in a symmetric state no matter how sharp the pencil happens to be (even if $b$ approaches zero). On the other hand, if we really take an infinitely sharp pencil (taking $b$ to zero first),

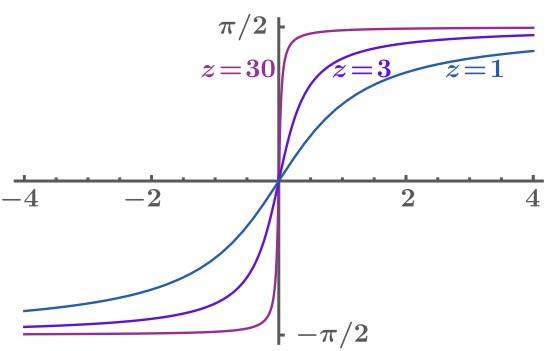

Figure 2.2: The function $y = \arctan(zx)$ has non-commuting limits $x \to 0$ and $z \to \infty$.

any *infinitesimal* deviation from being perfectly upright ($\theta$ approaching zero) suffices to tip it over and break the symmetry. The fact that in the limit $b \to 0$ the pencil becomes infinitely sensitive to the perturbation $\theta$, makes the limit of $b$ going to zero a *singular limit*. In general, the failure of two limits to commute is both a necessary and a distinctive signal of the presence of a singular limit.[6]

Mathematically, the fact that limits do not commute implies the development of a non-analytic feature in some function. This can be illustrated by plotting the function $y = \arctan(zx)$. For any given value of $z$, the function $y(x)$ is a smooth curve, going through the origin at $x = 0$, as shown in the figure below. As long as $y(x)$ remains a smooth function, its value at $x$ approaching zero does not depend on whether $x$ goes to zero from above or below. As $z$ increases however, the function $y(x)$ becomes steeper and steeper around the origin, until in the limit $z \to \infty$ it becomes a step function. At that point, the value of $y(x)$ with $x$ approaching zero is no longer zero, and does depend on how you approach it:

$$\lim_{z\to\infty}\lim_{x\to 0} y = 0, \qquad \lim_{x\downarrow 0}\lim_{z\to\infty} y = 1, \qquad \lim_{x\uparrow 0}\lim_{z\to\infty} y = -1. \qquad (2.2)$$

In the limit of $z \to \infty$, the value of the function $y(x)$ can change qualitatively under even infinitesimally small changes of $x$ around zero. Or, in other words, if the value $y(0)$ is something you could measure, the result of your measurement depends infinitely sensitively on how well you control the value of $x$.

Notice that in the physical example of a pencil balancing on its tip, neither of the limits $b \to 0$ and $\theta \to 0$ can in practice be realised. The tip will always be a little blunt, and no matter how steady your hand is, the balancing will never be absolutely perfect. The meaning of the singular limits then, is to say that for a sufficiently sharp pencil, it becomes arbitrarily difficult to balance it, and thus a sharp pencil in practice always tips over. The other ordering of limits implies that a sufficiently well-balanced cylinder may safely be sharpened (keeping the angle $\theta$ constant) without tipping it over. Of course the practical value of the latter order or limits is limited in real life.

There are a few particularities to be noticed about the pencil spontaneously breaking rotational symmetry. The first concerns the question of precisely which symmetry is broken. As we said before, the balanced pencil has a global rotational symmetry, meaning that we rotate the pencil in its entirety. This transformation does not involve the table. The symmetry is therefore global in the sense of affecting every single piece of the pencil, but it occurs relative to the table, which is held fixed. The rotation is thus not the global gauge freedom of rotating the universe as a whole. In this simple classical setting pointing out the difference between

---

[6]A nice introduction of the role of singular limits in different fields of physics is given in Ref. [22].

global transformations within a fixed reference frame and a truly global gauge freedom may seem a bit esoteric. But when you consider that both the table and the pencil are built up out of quantum mechanical atoms and molecules, which in turn consist of indistinguishable elementary particles, it is not at all obvious that a global transformation which acts only on the particles inside the pencil but not those inside the table is a natural thing to consider. This is of course the same issue as the one we discussed in Section 1.5.3, and again the conclusion is that one has to be careful in identifying the symmetry that is spontaneously broken.

Secondly, it should be noted that the symmetric state is unstable while the broken-symmetry states are stable, in accordance with the discussion of Section 2.1. In classical physics the stable, symmetry-broken states are also ground states even for finite-sized systems, while the symmetric state has higher energy.

Finally, the Hamiltonian describing the pencil cannot account for the direction in which the cylinder will fall. For this, an external perturbation favouring a certain $\theta$ is necessary. The spontaneous aspect of the symmetry breaking lies in the fact that this perturbation may be arbitrarily small for a sufficiently sharp pencil.

---

**Exercise 2.1 (Classical magnet)** The fact that all symmetry-broken states of for example a pencil lying flat on the table are degenerate may seem like an innocent statement, but taken at face value it actually represents a serious conundrum, even within the confines of classical physics. To see this clearly, consider the example of a classical magnet, consisting of many microscopic bar magnets coupled together by nearest-neighbour interactions, so that its internal energy is:

$$E = \sum_{\mathbf{x},\boldsymbol{\delta}} -|J|\mathbf{S}_{\mathbf{x}} \cdot \mathbf{S}_{\mathbf{x}+\boldsymbol{\delta}} \,. \tag{2.3}$$

Here $\mathbf{x}$ labels the position of the bar magnet with magnetisation $\mathbf{S}_{\mathbf{x}}$, and $\boldsymbol{\delta}$ connects nearest neighbours. All bar magnets have the same size $|\mathbf{S}_{\mathbf{x}}| \equiv s$. Clearly, the energy of the full magnet can be minimised by having all microscopic bar magnets point in the same direction. It is invariant however, under the simultaneous rotation of all bar magnets around their individual centres. That is, there is a global symmetry which dictates that all states of maximum magnetisation are degenerate, regardless of the direction of total magnetisation.

**a.** Remember that thermal expectation values may be computed as

$$\langle M \rangle_T = \frac{\sum_{\text{states}} e^{-E(\text{state})/k_B T} M(\text{state})}{\sum_{\text{states}} e^{-E(\text{state})/k_B T}} \,. \tag{2.4}$$

Show that the expectation value of the total magnetisation is zero for any temperature, even $T = 0$.

Notice what this implies: classical magnets in thermal equilibrium cannot have a well-defined north or south pole, at any temperature. Clearly this rigorous result is at odds with everyday experience, in which magnets do have a nonzero and permanent magnetisation. The resolution of the paradox of course lies in the spontaneous breakdown of symmetry, which we can describe mathematically by adding a small symmetry-breaking magnetic field to the system:

$$E' = \sum_{\mathbf{x},\boldsymbol{\delta}} -|J|\mathbf{S}_{\mathbf{x}} \cdot \mathbf{S}_{\mathbf{x}+\boldsymbol{\delta}} - h\hat{n} \cdot \mathbf{S}_{\mathbf{x}} \,. \tag{2.5}$$

Here $h$ is the strength of the applied magnetic field, which we will send to zero at the end of the calculation, and the unit vector $\hat{n}$ is its direction.

**b.** Argue that in this case the expectation value for the magnetisation at zero temperature will be $\langle M \rangle_T = N s \hat{n}$, where $N$ is the number of microscopic bar magnets.

**c.** We now again find a set of non-commuting limits:

$$\lim_{N \to \infty} \lim_{h \to 0} \langle M \rangle_T / N = 0$$
$$\lim_{h \to 0} \lim_{N \to \infty} \langle M \rangle_T / N = s \hat{n} . \tag{2.6}$$

Explain in words what these limits mean for magnets in our everyday world.

Even after breaking their symmetry, real magnets do not thermalise in the way described by Eq. (2.4). The reason for this is that states with different magnetisation, while formally connected via thermal fluctuations, are actually not accessible on ordinary time scales. All magnets being simultaneously rotated over the same angle in a single thermal fluctuation is exceedingly unlikely to occur for large magnets. This is a clear example of a large part of configuration space being effectively inaccessible in a system with a spontaneously broken symmetry.

## 2.3 The harmonic crystal

To see how spontaneous symmetry breaking is realised in the quantum realm, we consider the example of a particularly simple model for a quantum crystal. The Hamiltonian for a collection of atoms with mass $m$ in which neighbouring atoms are held together by harmonic forces with characteristic frequency $\omega_0$, is given by:

$$H = \sum_{\mathbf{x}, \delta} \frac{\mathbf{P}^2(\mathbf{x})}{2m} + \frac{1}{2} m \omega_0^2 (\mathbf{X}(\mathbf{x}) - \mathbf{X}(\mathbf{x} + \delta))^2 . \tag{2.7}$$

Here, $\mathbf{P}(\mathbf{x})$ and $\mathbf{X}(\mathbf{x})$ are the momentum and position operator of the atom at equilibrium position $\mathbf{x}$, with commutation relation $[X_a(\mathbf{x}), P_b(\mathbf{x}')] = i\hbar \delta_{ab} \delta(\mathbf{x} - \mathbf{x}')$. The atomic position $\mathbf{x}$ may be in one-, two-, or three-dimensional space, and the connections $\delta$ between interacting atoms may cover nearest neighbours, next-nearest neighbours, or any other interatomic distance. In fact, for the following arguments, even the quadratic form of the potential is not essential, and we could straightforwardly include anharmonic interactions as well. We are thus not just considering an oversimplified model for a hypothetical piece of material, but also the family of Hamiltonians which in principle describe all solids, including the very chair on which you sit.

The crucial point to notice about this Hamiltonian, is that it commutes with the operator for total (or centre-of-mass) momentum:

$$\mathbf{P}_{\text{tot}} \equiv \sum_{\mathbf{x}} \mathbf{P}(\mathbf{x}), \qquad\qquad [H, \mathbf{P}_{\text{tot}}] = 0. \tag{2.8}$$

The fact that the Hamiltonian for a crystal commutes with the total-momentum operator implies that all eigenstates of the crystal are total-momentum eigenstates. Because of Heisenberg's uncertainty principle, states in which the value of the total momentum can be known with certainty must be states in which the centre-of-mass position is entirely unpredictable. In other words, the total momentum eigenstates, and hence the eigenstates of the crystal Hamiltonian, are wave functions that are spread out over all of space. Clearly this is not the ground state you would expect for a piece of matter that you can hold in your hand. Much less that of an object on which you can safely sit.

To see why objects in our everyday world, with its translationally symmetric energy eigenstates, can in fact be localised, we first write the Hamiltonian in momentum space with wave numbers $\mathbf{k}$, and separate it into two independent parts:

$$H = H_{\text{CoM}} + \sum_{\mathbf{k} \neq 0} H_{\text{int}}(\mathbf{k}). \tag{2.9}$$

The part of the Hamiltonian for non-zero values of $\mathbf{k}$ describes the internal dynamics of the crystal, in terms of all of its phonon (quantised sound) excitations, and their interactions. The **phonon** centre-of-mass part at $\mathbf{k} = 0$ on the other hand, describes the collective dynamics of the crystal as a whole. It can be straightforwardly shown that in the case of the crystal, the collective part of the Hamiltonian is given by:

$$H_{\text{CoM}} = \frac{\mathbf{P}_{\text{tot}}^2}{2mN}. \tag{2.10}$$

Here, as always, $N$ is the number of particles. For now, we will only consider the properties of the collective part of the Hamiltonian, and completely ignore the internal, phonon-related part. This is possible because the two parts of the Hamiltonian commute, which implies that good quantum numbers of one part are also good quantum numbers for the other. Moreover, we will discover shortly that the energies in the spectrum of the collective part of the Hamiltonian are far smaller than even the lowest possible excitation energy of a single phonon. At extremely low temperatures, the collective part of the Hamiltonian is therefore the only part that matters.

From now on, for the sake of simplicity we will focus on a one-dimensional system, so we do not need to keep track of any spherical harmonics and other complications. Notice that this does not affect the generality of any of our conclusions. The collective Hamiltonian in Eq. (2.10) is that of a free particle of mass $mN$, and its eigenstates are total-momentum states, with energies $E_{\mathbf{P}} = \mathbf{P}^2/2mN$. The resulting spectrum of collective excitations, the tower of states, is sketched in the figure below.

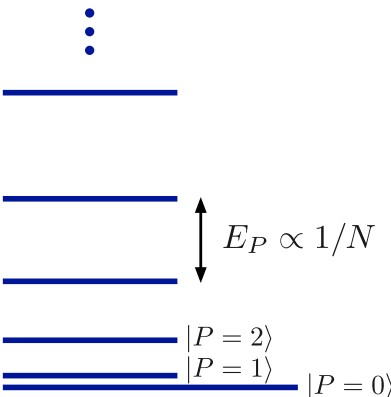

Figure 2.3: Tower of states

The ground state of the crystal is the state with total momentum $\mathbf{P} = 0$. Its wave function is completely and evenly spread out over all of space, with equal amplitude and even equal phase at every possible position. The energy separating the ground state from the collective excitations with non-zero total momentum, is inversely proportional to the total mass $mN$ of the whole crystal. This means that as we consider a larger and larger crystal, it becomes easier and easier to make excitations of the crystal. In fact, in the limit $N \to \infty$ of an infinitely large crystal, it would cost no energy at all to make collective excitations, and all states in the collective part of the spectrum become degenerate with the ground state. In that limit then,

a wave packet of total momentum states with a well-defined centre-of-mass position would have the same energy expectation value as the zero-momentum state.

Of course real crystals are not infinitely large, and superpositions of momentum states do cost energy to create. But the pieces of matter we are interested in contain a very large, albeit finite, number of atoms. It is therefore reasonable to wonder just how difficult it would be to make a superposition of crystal eigenstates which has a well-defined centre-of-mass postion. In terms of the collective Hamiltonian, this amounts to adding a perturbation which tends to localise the crystal:

$$H'_{\text{CoM}} = \frac{\mathbf{P}^2_{\text{tot}}}{2mN} + \mu \mathbf{X}^2_{\text{CoM}}. \tag{2.11}$$

Here, $\mathbf{X}_{\text{CoM}} = 1/N \sum_{\mathbf{x}} \mathbf{X}(\mathbf{x})$ is the centre of mass position of the crystal, while $\mu$ is the strength of a potential that tends to localise the crystal at the origin of our coordinate system. We will consider $\mu$ to be a very small perturbation, and take the limit $\mu \to 0$ at the end of our calculation.

> **Exercise 2.2 (Commutator of $\mathbf{P}_{\text{tot}}$ and $\mathbf{X}_{\text{CoM}}$)** Show that $\mathbf{P}_{\text{tot}}$ and $\mathbf{X}_{\text{CoM}}$ obey canonical commutation relations.

Since $\mathbf{P}_{\text{tot}}$ and $\mathbf{X}_{\text{CoM}}$ obey canonical commutation relations, the perturbed Hamiltonian $H'_{\text{CoM}}$ is that of a harmonic oscillator, with the well-known energies and ground state:

$$E_n = \hbar \omega (n + 1/2)$$

$$\psi_0(\mathbf{x}) = \langle \mathbf{x} | n = 0 \rangle = \left( \frac{2mN\mu}{\pi^2 \hbar^2} \right)^{3/8} e^{-\sqrt{\frac{mN\mu}{2\hbar^2}} |\mathbf{x}|^2}, \tag{2.12}$$

with $\omega = \sqrt{\frac{2\mu}{mN}}$. The ground state of the harmonic crystal in the presence of a perturbation is a Gaussian wave packet. Its width in position space, $\sigma^2 = \hbar / \sqrt{2mN\mu}$, decreases as the number of particles in the crystal grows. The occurrence of spontaneous symmetry breaking is again signalled by two non-commuting limits:

$$\lim_{N \to \infty} \lim_{\mu \to 0} |\psi_0(\mathbf{x})|^2 = \text{constant},$$

$$\lim_{\mu \to 0} \lim_{N \to \infty} |\psi_0(\mathbf{x})|^2 = \delta(\mathbf{x}). \tag{2.13}$$

As before, this is an indication that the thermodynamic limit $N \to \infty$ is singular. For a system of any finite size, no matter how large, the perturbation can be made small enough for the ground state wave function to be essentially spread out over the entire universe. But for an infinitely large system, *any* perturbation, no matter how weak, is enough to completely localise the wave function in a single position. In the thermodynamic limit, the localisation happens even in the presence of only an infinitesimal potential, which in effect means that the wave function localises spontaneously. Notice that the energy of the state does not depend on the order of limits. In both cases $E_0 = \hbar\omega/2 = 0$. This implies that indeed in the thermodynamic limit the symmetry-broken, localised state of the crystal has the same energy as the exact, plane-wave ground state.

Real materials are not infinitely large, and thus neither of the limits in equation (2.13) strictly speaking applies to real pieces of matter. This is not just a practical consideration, but a point of view that is increasingly advocated even in formal mathematical approaches to spontaneous symmetry breaking [23, 24]. The importance of the non-commuting limits, is to signal that even if $N$ is not yet truly infinite, the approach towards the thermodynamic limit is singular. This implies in particular that as you consider larger and larger pieces of matter,

a weaker and weaker perturbation suffices to make its ground state a localised wave packet. Imagine for example a typical 'finite size' iron crystal, with a volume of one cubic centimeter. Iron has an atomic mass of $55.8u = 9.27 \times 10^{-26}$ kg, and a lattice constant of $a = 2.856 \times 10^{-10}$ m. This means the iron crystal contains approximately $N = 4 \times 10^{22}$ atoms. So how strong does the symmetry breaking field need to be in order to reasonably localise such a crystal? The units of $\mu$ are kg s$^{-2}$ or N m$^{-1}$ (Newton per meter). The weakest possible forces that can currently be measured are of the order of zeptonewtons ($10^{-21}$ N). The weakest possible symmetry breaking field for our piece of iron that would still be measurable, is thus something like a zeptonewton per centimeter, or $\mu \sim 10^{-19}$ N m$^{-1}$. Even with such a weak perturbation, the width of the crystal's ground state wave function is constrained to be about $2 \times 10^{-12}$ m. That is, two orders of magnitude smaller than the unit cell of the iron crystal itself. Clearly, it is practically impossible to find any sort of everyday-sized object in a momentum eigenstate.

---

**Exercise 2.3 (Elitzur's theorem for a free gas)** We stated in Section 1.4.3 that, in contrast to global symmetries, *local* symmetries cannot be spontaneously broken. To understand why this is the case, consider an ideal gas of $N$ non-interacting particles with positions $\mathbf{X}_j$ and momenta $\mathbf{P}_j$, described by:

$$H_0 = \sum_j \frac{\mathbf{P}_j^2}{2m}. \tag{2.14}$$

**a.** The ideal gas has a local translation symmetry. Write down the operator $U$ which describes local translations, and show that the Hamiltonian is invariant under it.

We can try and break the local symmetry by adding a perturbation which acts on one particle only:

$$H' = \sum_j \left( \frac{\mathbf{P}_j^2}{2m} \right) + \frac{1}{2}\kappa \mathbf{X}_{j=2}^2. \tag{2.15}$$

**b.** What is the ground state $|\psi\rangle$ of the perturbed Hamiltonian?
**c.** Calculate the expectation value in the symmetry-broken state of both the unperturbed energy and the uncertainty in position of the perturbed particle:

$$\bar{E} = \langle\psi| H_0 |\psi\rangle,$$
$$\Delta \mathbf{X}_{j=2}^2 = \langle\psi| \mathbf{X}_{j=2}^2 |\psi\rangle - \left( \langle\psi| \mathbf{X}_{j=2} |\psi\rangle \right)^2. \tag{2.16}$$

**d.** What are the values of $\bar{E}$ and $\Delta \mathbf{X}_{j=2}^2$ in the limit $\kappa \to 0$?
**e.** Is there any limit that does not commute with the limit $\kappa \to 0$? In particular, does $\kappa \to 0$ commute with the thermodynamic limit $N \to \infty$? What do your answers imply for the local symmetry of $H_0$?

---

The exercise above is suggestive of the general fact that continuous local symmetries cannot be spontaneously broken. This result is known as *Elitzur's theorem* [25], and it has the status of a mathematical theorem. Its only assumption is that the physics of the system is described by a minimum action principle, which is the case for all presently known theories of physics. The central ingredient needed to prove the theorem is the realisation that for locally symmetric systems, there cannot exist a singular limit of the kind we encountered for global symmetries. Systems with a local symmetry therefore lack the instability that comes with global symmetry and they are always robust against small perturbations. This is not a result that can be negotiated with by "smart engineering" of a model or system. Spontaneously

**Elitzur's theorem**

breaking a continuous local symmetry is just impossible.

## 2.4 The Heisenberg antiferromagnet

A second instructive example of spontaneous symmetry breaking in quantum physics, is that of the Heisenberg antiferromagnet. Consider a cubic lattice with a spin-half degree of freedom on every site, and isotropic interactions between neighbouring spins:

**Heisenberg model**

$$H = J \sum_{\langle i,j \rangle} \mathbf{S}_i \cdot \mathbf{S}_j. \tag{2.17}$$

Here $\mathbf{S}_i = \hbar \boldsymbol{\sigma}_i$ is the spin operator on site $i$, with $\boldsymbol{\sigma}_i$ the Pauli matrices as introduced in Exercise 1.7, and the usual commutation relations $[S_i^a, S_j^b] = i\hbar \delta_{ij} \epsilon_{abc} S_i^c$. The indices $i$ and $j$ label all sites of the cubic lattice, and $\langle i, j \rangle$ indicates that we sum over nearest-neighbour pairs of spins only. Expanding the product of spin vectors, the Hamiltonian becomes:

$$\begin{aligned} H &= J \sum_{\langle i,j \rangle} S_i^z S_j^z + S_i^y S_j^y + S_i^x S_j^x \\ &= J \sum_{\langle i,j \rangle} S_i^z S_j^z + \frac{1}{2} \left( S_i^+ S_j^- + S_i^- S_j^+ \right). \end{aligned} \tag{2.18}$$

The operators $S_i^{\pm} = S_i^x \pm i S_i^y$ are the spin raising and lowering operators. The strength of the interaction between neighbouring spins is denoted by the coupling constant $J$. If it is negative, the energy is minimised whenever neighbouring spins point in the same direction. The ground state is then a ferromagnet, with all spins in the entire material pointing in the same direction, which is spontaneously chosen. We will come back to the very special case of spontaneous symmetry breaking in ferromagnets in Exercise 2.7.

For positive $J$, the energy is lowered by neighbouring spins being anti-parallel. If we divide the cubic lattice into two sublattices, as shown in the Fig. 2.4, then all nearest-neighbour pairs of spins can be made anti-parallel by choosing all spins on the $A$-sublattice to point up, and all spins on the $B$-sublattice to point down. Such a perfect *antiferromagnetic* arrangement of spins is known as a *Néel state*. Again the axis along which the spins point either up or down is spontaneously chosen. Global rotations of the spins around this axis are still symmetry transformations in the Néel state (they are "unbroken"), while spin-rotations around all other axes are spontaneously broken.

**antiferro-magnet**

---

**Exercise 2.4 (Noether current of the Heisenberg magnet)**

The Heisenberg Hamiltonian of Eq. (2.17) is not in canonical form. That is, it is not expressed in term of operators and their conjugate momenta. We then cannot perform a Legendre transformation to obtain a Lagrangian and action, and we cannot use the method of Section 1.2.3 to obtain the Noether current. However, we already have the symmetry generators $Q^a = S_{\text{tot}}^a$, expressed as volume 'integrals' over what must be the Noether charge density $j_a^t(i) = S_i^a$. We can then obtain the conservation law by calculating the time derivative explicitly.

**a.** Calculate the Heisenberg equation of motion $\partial_t S_i^a = \frac{i}{\hbar} [H, S_i^a]$, using the spin commutation relations.

The vector connecting a point $i$ on a lattice with a neighbouring point $i + \delta$ may be written as $v_{i,i+\delta}$. In this notation, the (lattice) divergence of a vector at $i$ is given by $\sum_{\delta} v_{i,i+\delta}$.

**b.** Show that the right-hand side of the equation of motion has the form of the lattice

---

divergence of a vector. That vector corresponds to the spatial Noether current $j_a^m$, and the equation of motion is the lattice version of $\partial_t j_a^t + \partial_m j_a^m = 0$.

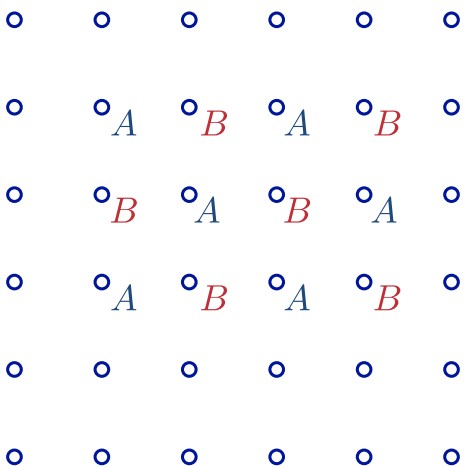

Figure 2.4: A square lattice can be divided into two inequivalent *sublattices A* and *B*. In the case of a perfect Néel antiferromagnet, the spins on the two sublattices point in opposite direction.

If the spins were classical bar magnets, the Néel state would have been the ground state of the antiferromagnet. The quantum mechanical Hamiltonian however, contains terms like $S_i^+ S_j^-$, of which the Néel state is not an eigenstate. The Néel state can therefore not be the ground state of the Heisenberg Hamiltonian. In fact, for a three dimensional lattice, it is still an open question what the precise ground state of the antiferromagnetic Heisenberg Hamiltonian is.

The Hamiltonian of equation (2.17) is invariant under global, simultaneous rotations of all spins around any axis. You can check that the Hamiltonian indeed commutes with the generators $S^a = S_{\text{tot}}^a = \sum_i S_i^a$ of spin-rotations around all axes $a \in x, y, z$. It follows that the Heisenberg Hamiltonian also commutes with the operator for total spin $S^2 = \mathbf{S} \cdot \mathbf{S}$:

$$\left[H, S^2\right] = 0. \tag{2.19}$$

Since the Hamiltonian is invariant under global rotations of all spins, you might expect that like in the case of a crystal, it will be sufficient to consider the collective part of the Hamiltonian in order to see if its symmetry can be spontaneously broken. This is true, but in the case of the antiferromagnet we do need to be careful about what this collective part is precisely. Because the $A$ and $B$ sublattice are different, there is a collective mode which rotates all spins on the $A$ sublattice, while the ones on the $B$ sublattice are left invariant, and the other way around. These two modes correspond to two components in the Fourier transfrom of the spin rotation operators, rather than only the one component that defined the collective motion of the crystal. Together, the two Fourier components make up the collective part of the Heisenberg Hamiltonian in equation (2.17):

$$H = H_{\text{coll}} + \sum_{\mathbf{k} \neq 0, \pi/a} H(\mathbf{k}), \qquad\qquad H_{\text{coll}} = \frac{J}{N} \mathbf{S}_A \cdot \mathbf{S}_B. \tag{2.20}$$

This collective Hamiltonian is known as the Lieb-Mattis model. Here, the collective operator for the total spin on the $A$ sublattice is defined as $\mathbf{S}_A = \sum_{i \in A} \mathbf{S}_i$, and similarly for $\mathbf{S}_B$. The non-collective, internal modes which make up $H(\mathbf{k})$, are the spin-wave analogue to the phonons in a crystal, and are known as *magnons*.

**Lieb-Mattis model**

**magnon**

SciPost Phys. Lect. Notes 11 (2019)

The eigenstates of the collective Hamiltonian can be easily found by re-writing it in terms of the total spin operator $\mathbf{S} = \mathbf{S}_A + \mathbf{S}_B$:

$$H_{\text{coll}} = \frac{J}{2N} \left( S^2 - S_A^2 - S_B^2 \right). \tag{2.21}$$

Because all operators in this expression commute, we can immediately write the eigenstates and the corresponding energies in terms of the quantum numbers for total spin and total spin on the sublattices:

$$H_{\text{coll}} |S_A, S_B, S, S^z\rangle = E(S, S_A, S_B) |S_A, S_B, S, S^z\rangle,$$

$$E(S, S_A, S_B) = \frac{J\hbar^2}{2N} \left( S(S+1) - S_A(S_A+1) - S_B(S_B+1) \right). \tag{2.22}$$

The spectrum of low-energy eigenstates is shown schematically in Figure 2.5 below. The ground state is the state with $S_A$ and $S_B$ equal to their maximal value of $N/4$, and the total spin $S$ equal to zero. Because $S = 0$ implies that $\langle S^x \rangle = \langle S^y \rangle = \langle S^z \rangle = 0$, the ground state does not break spin-rotation symmetry. It is a unique, non-degenerate state in which the spins on each sublattice are all exactly aligned with each other, and anti-aligned with neighbours on a different sublattice, but in which no direction in space is different from any other. You can think of the total spin singlet ground state as a superposition of infinitely many Néel states, pointing in all possible directions. Clearly, this is a highly non-local state.

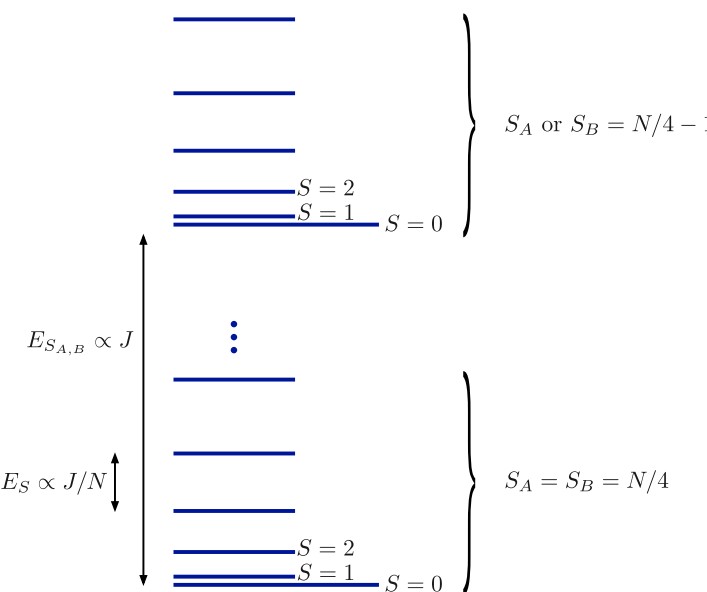

Figure 2.5: The states with lowest energy in the Lieb-Mattis Hamiltonian.

Flipping one of the individual spins on for example the $A$ sublattice would decrease $S_A$ by one. Such an excitation costs an energy of the order of $J$, and if the temperature is low enough, these excitations will not be present. It is also possible, however, to make collective excitations which increase the value of the total spin $S$ by one. Such excitations cost an energy that is only proportional to $J/N$, and for a large enough system these excitations can never be neglected. In fact, in the limit of an infinitely large system, with $N \to \infty$, there is a again a tower of states labeled by different values of $S$ which all become degenerate with the ground state[7]. This suggests that again, in that limit even an infinitesimally small perturbation might have a qualitative effect and spontaneously break the spin-rotational symmetry.

---

[7]In this context it is also referred to as the *Anderson tower of states* after P.W. Anderson, who argued its existence already in 1952 [26]. These entire lecture notes in fact are very much in the spirit of the first chapter of Anderson's seminal textbook [11].

The perturbation or symmetry breaking field to consider in this case, is a field that prefers the spins to be arranged into a Néel state. As before, the reasoning is that in principle we should consider all possible perturbations and find that none of them have any effect in the limit where they are infinitesimally small, except for the one that happens to stabilise the classically expected ground state. Since we already know what a classical antiferromagnet looks like, we will straight away consider the only relevant perturbation:

$$H'_{\text{coll}} = \frac{J}{N} \mathbf{S}_A \cdot \mathbf{S}_B - \mu \left( S_A^z - S_B^z \right). \tag{2.23}$$

Since the perturbation breaks spin-rotational symmetry, it does not commute with the original Hamiltonian. The states $|S, S_A, S_B, S^z\rangle$ are therefore no longer eigenstates of the perturbed Hamiltonian. In order to find the new eigenstates, we need to calculate the matrix elements of the perturbed Hamiltonian in the basis of the old eigenstates. Because the excitations that change $S_A$ and $S_B$ cost far more energy than an infinitesimal perturbation is expected to provide, we can safely assume that $S_A = S_B = N/4$ throughout. And because the quantum number $S^z = S_A^z + S_B^z$ is a good quantum number even for the perturbed Hamiltonian, we can additionally fix $S^z = 0$. The only remaining relevant quantum number is then the total spin $S$, and we can abbreviate the eigenstates of the original Hamiltonian by writing $|S, S_A = S_B = N/4, S^z = 0\rangle \equiv |S\rangle$. Calculating the matrix elements of the symmetry-breaking field in this basis involves some rather tedious exercises in the addition of angular momentum, but the result is known to be:

$$\langle S' | S_A^z - S_B^z | S \rangle = \delta_{S',S-1} \, S \sqrt{\frac{(N/2+1)^2 - S^2}{4S^2 - 1}} + \delta_{S',S+1} \, S' \sqrt{\frac{(N/2+1)^2 - S'^2}{4S'^2 - 1}}$$

$$\approx \left( \delta_{S',S-1} + \delta_{S',S+1} \right) \frac{N}{4}. \tag{2.24}$$

In the final line we simplified the matrix element by using the fact that only the states with $1 \ll S \ll N$ will contribute to the perturbed ground state. That this is indeed the case will become clear shortly.

The Hamiltonian in the $|S\rangle$ basis can now be explicitly written as:

$$H'_{\text{coll}} \approx \sum_S \frac{J\hbar^2}{2N} S^2 |S\rangle \langle S| - \mu \frac{N}{4} |S\rangle \langle S+1| - \mu \frac{N}{4} |S+1\rangle \langle S|, \tag{2.25}$$

where again we used the assumption $1 \ll S \ll N$ to simplify the diagonal term, and we left out the constant contribution to the energy from the $S_A$ and $S_B$ quantum numbers. We can simplify the expression further by writing the eigenstates of the perturbed Hamiltonian as $|n\rangle = \sum_S \psi_n(S) |S\rangle$, and by approximating $S$ to be a continuous variable and taking the continuum limit in the Schrödinger equation:

$$\langle S | H | n \rangle = \langle S | E_n | n \rangle,$$

$$\frac{J\hbar^2}{2N} S^2 \psi_n(S) - \frac{\mu N}{4} \left( \psi_n(S+1) + \psi_n(S-1) \right) = E_n \psi_n(S),$$

$$\Rightarrow \quad \left( \frac{J\hbar^2}{2N} S^2 - \frac{\mu N}{2} \right) \psi_n(S) - \frac{\mu N}{4} \frac{\partial^2 \psi_n(S)}{\partial S^2} = E_n \psi_n(S)$$

$$-\frac{1}{2} \frac{\partial^2 \psi_n(S)}{\partial S^2} + \frac{1}{2} \omega^2 S^2 \psi_n(S) = \epsilon_n \psi_n(S). \tag{2.26}$$

In the third line we used the discrete version of the second derivative to take the continuum limit, while in the fourth line we defined the variables $\omega^2 = 2J\hbar^2/(\mu N^2)$ and $\epsilon_n = (2E_n + \mu N)/(\mu N)$.

The final expression is just the Schrödinger equation for a harmonic oscillator with $m = \hbar = 1$, whose eigenstates and energies are well known. We should keep in mind the subtlety that in the present problem, the quantum number $S$ can only be positive. We should therefore only accept solutions of equation (2.26) which have a node at the origin. That is, the eigenstates of the perturbed antiferromagnet are the (positive $S$ part of the) harmonic oscillator eigenstates with odd values for the quantum number $n$, and the eigenvalue corresponding to the lowest allowed state is $\epsilon_1 = (1 + 1/2)\omega$. The ground state of the perturbed antiferromagnet is therefore the state $|n = 1\rangle$, with ground state energy:

$$E_1 = \frac{\mu N}{2}\left(\frac{3}{2}\omega - 1\right) = \frac{3}{4}\sqrt{2\mu J \hbar^2} - \frac{\mu N}{2}. \tag{2.27}$$

From the ground state energy we can again understand what happens in the thermodynamic limit. If we first take the limit $\mu \to 0$, then the energy of the perturbed system is just equal to the energy of the unperturbed system, and the symmetric ground state is unaffected. However, if we first take the limit $N \to \infty$ while keeping $\mu$ non-zero (but infinitesimally small), then the first term in the energy can be neglected, and we find $E_1 = -\mu N/2$, with $N \to \infty$. But this is precisely the maximum amount of energy that can only possibly by gained by having all spins on the $A$ sublattice point upwards, and all spins on the $B$ sublattice point downwards. The ground state in this limit thus must be the Néel state, and the spin-rotational symmetry is spontaneously broken.

An alternative way of seeing the broken symmetry, is to calculate the expectation value of the difference in magnetisation between the two sublattices in the thermodynamic limit:

$$\lim_{N\to\infty}\lim_{\mu\to 0}\langle n = 1|S_A^z - S_B^z|n = 1\rangle = 0,$$

$$\lim_{\mu\to 0}\lim_{N\to\infty}\langle n = 1|S_A^z - S_B^z|n = 1\rangle = \frac{N}{2}. \tag{2.28}$$

Again, the final line indicates that for a large enough piece of antiferromagnetic material, even an infinitesimally small symmetry breaking field suffices to completely align all of the spins. The spin-rotational symmetry of the Heisenberg Hamiltonian is thus spontaneously broken in the thermodynamic limit.

## 2.5 Symmetry breaking in the thermodynamic limit

The spontaneous breakdown of symmetry in both the harmonic crystal and the antiferromagnet is signalled by the fact that the thermodynamic limit in those systems is singular. As emphasised repeatedly however, the practical implication of this singularity is that symmetric states are unstable and symmetry-breaking states may be stable already in large, but finite, systems. The thermodynamic limit thus foreshadows the behaviour of finite-sized objects. Because the stable and unstable states become degenerate in the thermodynamic limit, many aspects of symmetry breaking are easier to describe there, even though they actually apply more generally. We will make use of this fact here, and follow the standard approach of classifying broken-symmetry states entirely within the thermodynamic limit.

### 2.5.1 Classification of broken-symmetry states

For mathematical convenience, we will from here on consider only Hamiltonians and broken-symmetry states that have some degree of translational invariance. It is sufficient for the invariance to be discrete. That is, we consider states with a periodic arrangement of unit cells on a regular lattice. The examples of the crystal and the antiferromagnetic Néel state clearly fall in this class, but it also applies more generally to almost any symmetry-breaking system

of interest, at least on a course-grained level. Translational invariance allows for a Fourier transformation with well-defined wave numbers, which is a prerequisite for the introduction of Nambu–Goldstone modes in Chapter 3. It also ascertains that we can write the broken-symmetry state as a product state of the form $|\psi\rangle = \otimes_i |\psi_i\rangle$, with $|\psi_i\rangle$ the same local wave function for every unit cell $i$. If the translational symmetry happens to be continuous, rather than discrete, the index $i$ may be replaced by a continuous parameter $\mathbf{x}$. In the following, we will use $i$ and $\mathbf{x}$ interchangeably.

Just the assumption of translational invariance is sufficient to guarantee that distinct symmetry-breaking states are orthogonal in the thermodynamic limit. To see this, consider the overlap between two normalised broken-symmetry states $|\psi\rangle$ and $|\psi'\rangle$:

$$
\begin{aligned}
\langle \psi' | \psi \rangle &= \left( \otimes_{\mathbf{x}'} \langle \psi'(\mathbf{x}') | \right) \left( \otimes_{\mathbf{x}} | \psi(\mathbf{x}) \rangle \right) \\
&= \prod_{\mathbf{x}} \langle \psi'(\mathbf{x}) | \psi(\mathbf{x}) \rangle = \langle \psi'(\mathbf{x}) | \psi(\mathbf{x}) \rangle^N .
\end{aligned}
\tag{2.29}
$$

Here we used the vanishing overlap between local states $|\psi(\mathbf{x})\rangle$ and $|\psi(\mathbf{x}')\rangle$ for $\mathbf{x} \neq \mathbf{x}'$ to write the overlap of product states as a product of local overlaps. In the final step, we also used translational invariance by taking $|\psi(\mathbf{x})\rangle$ to be independent of $\mathbf{x}$. The number of unit cells is denoted $N$. If the two symmetry-breaking states $|\psi\rangle$ and $|\psi'\rangle$ are distinct, then the local overlap $\langle \psi'(\mathbf{x}) | \psi(\mathbf{x}) \rangle$ must be smaller than one, and the full inner product $\langle \psi' | \psi \rangle$ vanishes as $N$ is taken to infinity. On the other hand, if the states are not distinct, and differ only by a phase $|\psi'(\mathbf{x})\rangle = e^{i\varphi} |\psi(\mathbf{x})\rangle$, the inner product is one, up to a phase factor. We thus find that any two symmetry-breaking states in the thermodynamic limit must be either equivalent or orthogonal.

At this point, we can apply the group theory of Section 1.6.1 to begin the classification of all possible distinct symmetry-breaking states. Recall that symmetry transformations make up a group $G$, with elements $g$. As we saw before, a specific symmetry transformation $g$ is represented in quantum physics by a unitary operator acting on Hilbert space. With a slight abuse of notation, and to avoid introducing too many symbols, we will write $g$ both for the element in the group of symmetry transformations, and for the operator it represents. We can then consider a state $|\psi\rangle$ that breaks some of the symmetries in the group $G$, but is invariant under others. For symmetry transformations $g$ that are broken, $|\psi\rangle$ and $g|\psi\rangle$ are distinct, inequivalent states. Assuming some degree of translational invariance, they must then be orthogonal in the thermodynamic limit. For unbroken symmetry transformations $g$, on the other hand, we know from the definition of a symmetric state in equation (1.1), that $g|\psi\rangle = e^{i\varphi}|\psi\rangle$. The set of all such unbroken transformations together forms a subgroup $H \subset G$ called the *residual symmetry group*.

**Exercise 2.5 (Subgroup of unbroken transformations)** Recall the definition of a subgroup from Section 1.6.1 and show that the set of transformations that leave $|\psi\rangle$ invariant indeed form a subgroup.

residual symmetry group

Now consider two transformations, $g_1$ and $g_2$, which happen to satisfy the relation $g_1 = g_2 h$ for some element $h$ of the residual symmetry group. We then see that $g_1|\psi\rangle = e^{i\varphi} g_2|\psi\rangle$, which implies that $g_1|\psi\rangle$ is equivalent to $g_2|\psi\rangle$. Conversely, if $g_1$ and $g_2$ do not satisfy $g_1 = g_2 h$ for any $h \in H$, it follows that $g_1|\psi\rangle$ and $g_2|\psi\rangle$ are distinct and orthogonal broken-symmetry states. Since the operators corresponding to a broken symmetry transform a given symmetry-breaking state into an inequivalent one, they are often said to correspond to abstract 'rotations' within the space of broken-symmetry states. Starting from the initial state $|\psi\rangle$, we can label each symmetry-breaking state $|\psi'\rangle$ that is distinct from it by some $g$ for which $|\psi'\rangle = g|\psi\rangle$. The resulting list of labels for inequivalent broken-symmetry states consists of a subset of the

elements $g$ of the symmetry group $G$, in which no two elements $g$ and $g'$ satisfy the relation $g = g'h$ for any $h \in H$. Looking back at Section 1.6.1, this coincides precisely with the definition of the quotient set $G/H$.

> Inequivalent symmetry-broken states are classified by the cosets $gH$ as elements of the quotient set $G/H$, where $G$ is the group of all symmetry transformations and $H \subset G$ is the subgroup of unbroken transformations.

Notice that this classification of symmetry-breaking states in terms of cosets only applies to the thermodynamic limit. In a system of finite size, distinct broken-symmetry states need not be precisely orthogonal. This is easy to see in the case of a finite object breaking for example continuous rotational symmetry. The Hilbert space in that case is finite-dimensional, but there are infinitely many symmetry-broken states labelled by all possible directions in space. These correspond to the infinitely many cosets of the rotational symmetry group.

To make the classification more concrete, consider for example a Hamiltonian with a global phase-rotational symmetry, described by the continuous group of symmetry transformations $G = U(1)$, see Eq. 1.19. For a stable state that breaks the phase rotations, so that there is no symmetry left, the residual symmetry group is just the trivial group $H = e$, containing only the identity operator. The quotient set is then $G/H = U(1)$ and distinct or inequivalent broken-symmetry states may be labeled by their phase factors $e^{i\varphi}$. This global phase rotation actually is the broken symmetry characterising a superfluid, and superfluids with different phase values may be distinguished by observing a Josephson current between them, as we will see in Section 2.5.5. More generally, whenever all symmetry transformations in a group are simultaneously broken, the symmetry-breaking states are simply labelled by the elements of the full symmetry group.

As a second example, suppose that the spin-rotation symmetry in the Heisenberg antiferromagnet of Section 2.4 is broken down to only rotations around a single axis. This is the case for example in the classical Néel state. Then the group describing the spin-rotational symmetry of the Hamiltonian is $G = SU(2)$, while the residual symmetry group describing the leftover rotations around a single axis is $H = U(1)$. It can be shown that the quotient set $G/H$ equals $SU(2)/U(1) \simeq S^2$, which corresponds to the set of points on the surface of a sphere. These points indicate the possible directions of the residual rotation axis, or equivalently, the direction of the sublattice magnetisation. Note that while $S^2$ classifies all possible symmetry-broken states, it does not have a group structure itself.

For continuous groups, the classification of broken symmetry states can also be expressed in terms of the generators of the continuous symmetry transformations, defined in Eq. (1.5). In this context, generators $Q$ of which the broken-symmetry state under consideration is an eigenstate, are called *unbroken generators* or *unbroken Noether charges*, and any finite transformation generated by $Q$ is also unbroken. Conversely, generators that do not leave the state invariant are called *broken*. The continuous symmetry group may be broken down to either a continuous or a discrete subgroup, and even to the trivial group. The dimension of the quotient set $G/H$ for continuous groups $G$ is said to equal the number of broken generators. The algebraic relations between broken and unbroken generators will play an important role in the classification of Nambu–Goldstone modes in Chapter 3.

### 2.5.2 The order parameter

Having classified all possible inequivalent symmetry-breaking states, it would be useful to have an operator whose expectation value can be used to distinguish between them. Ideally, such an operator would have expectation value zero in any symmetric state, and a unique non-zero expectation value for each of the sets of equivalent broken-symmetry states. As it turns

out, such an operator can be defined, and is called the *order parameter operator*. Actually the word *order parameter* is often used to denote any quantity whose expectation value is non-zero in the broken-symmetry phase, and zero in the symmetric state, without having the additional benefit of distinguishing between inequivalent symmetry-breaking configurations. A well-known example can be found in the theory of superconductivity, where the amplitude of the gap function (see Section 7.2) is often called the superconducting order parameter. In these lecture notes, we will stick to the more narrow definition, which has the added advantage that it will be instrumental in deriving the Goldstone theorem in Chapter 3.

**order parameter operator**

To define a fool-proof recipe for identifying an order parameter operator, we first need to slightly update our definition of what constitutes a symmetric state.

**symmetry – of states**

**Definition 2.1** *Let $U = e^{i\alpha Q}$ be a symmetry transformation that commutes with the Hamiltonian, and which is parameterised by a discrete or continuous variable $\alpha$. A state $|\psi\rangle$ breaks this symmetry if there exists any operator $\Phi$ such that:*

$$\langle\psi|[Q, \Phi]|\psi\rangle \neq 0. \tag{2.30}$$

*If no such operator $\Phi$ exists, the state is symmetric under the transformation $U$.*

This definition is consistent with our earlier intuitive definition of symmetric states being eigenstates of the symmetry transformation, because if $U|\psi\rangle = e^{i\varphi}|\psi\rangle$, the left-hand side of Eq. (2.30) vanishes for any operator $\Phi$, and the state is said to be symmetric.

The operator $\Phi$ appearing in Eq. (2.30) will be a field $\Phi(x)$ acting locally in space in all cases we shall encounter in these lecture notes. It is called the *interpolating field*. With it, we can define the *order parameter operator* $\mathcal{O}(x)$ related to a broken symmetry $Q$, and its expectation value $O(x)$, which is known as the (local) *order parameter*:

**interpolating field**

**order parameter**

$$\mathcal{O}(x) = [Q, \Phi(x)], \qquad\qquad O(x) = \langle\psi|\mathcal{O}(x)|\psi\rangle. \tag{2.31}$$

Notice that if $|\psi\rangle$ is translationally invariant, then so is $O(x)$. Also, $\Phi(x)$ and $\mathcal{O}(x)$ are not necessarily Hermitian, but they can always be used to construct an observable quantity, such as $\mathcal{O} + \mathcal{O}^\dagger$ or $\mathcal{O}\mathcal{O}^\dagger$. The order parameter $O(x)$ is just the left-hand side of Eq. (2.30). Therefore, the order parameter is automatically zero if $|\psi\rangle$ is symmetric and non-zero if $|\psi\rangle$ breaks a symmetry under the new definition of the symmetry of states. It thus clearly satisfies the requirement of distinguishing symmetric from symmetry-breaking states.

To make sure that the order parameter also distinguishes beween inequivalent broken-symmetry states, we can require that any good order parameter operator $\mathcal{O}$ has eigenvalues that map in a one-to-one fashion onto the quotient space $G/H$ introduced in Section 2.5.1. This way, the order parameter $O(x)$ will not only be different for distinct broken-symmetry states and equal for states related by residual symmetry transformations, but it will also inherit the structure of the quotient space. In particular, this means that states which are close to each other, according to the topological structure of the Lie group $G$, will have only a small difference in their corresponding order parameter values.

As it turns out, it is always possible to find an order parameter operator that satisfies these constraints, because Eq. (2.30) does not uniquely determine the order parameter and interpolating field. For example, multiplying $\Phi$ by a constant, or taking its Hermitian conjugate, yield alternative definitions of an interpolating field that still obey Eq. (2.30). In almost all cases, a convenient choice for the order parameter operator, which maps onto the quotient space $G/H$, is suggested by the physics of the symmetry-breaking system itself.

To make the formal definitions of the order parameter and interpolating field more concrete, consider the example of the Heisenberg antiferromagnet, which we discussed in Section 2.5.1. The Hamiltonian has $SU(2)$ spin-rotational symmetry, which is broken down in the

antiferromagnetic state to just $U(1)$ rotations around the axis of sublattice magnetisation. Inequivalent broken-symmetry states correspond to antiferromagnetic configurations with the sublattice magnetisation pointing in different directions. All possible directions constitute the set of points on the surface of a sphere, $S^2$, which indeed coincides with the quotient $SU(2)/U(1) \simeq S^2$. For a specific broken-symmetry state with the sublattice magnetisation along the $z$-axis, the symmetry generators $S^x$ and $S^y$ are spontaneously broken, while $S^z$, which generates rotations around the $z$-axis, is unbroken. We can then define the *staggered magnetisation* as $N_i^a = (\pm 1)^i S_i^a$, where $i$ is a position index that is even on the $A$-sublattice and odd on the $B$-sublattice, and $a$ denotes a direction in spin space. To describe the breaking of rotations generated by $S^x$, we can choose $N^y$ to be the interpolating field, which yields $\sum_{ij}[S_i^x, N_j^y] = i \sum_{ij} \delta_{ij} N_i^z = i \sum_i N_i^z$ as the order parameter operator. Similarly, for the breakdown of rotations around the $y$-axis, we can choose $N^x$ as the interpolating field, which also corresponds to $N^z$ as the order parameter operator. The staggered magnetisation identified as the order parameter in this way, is also the natural choice for an antiferromagnet, because the Néel state is precisely an eigenstate of the staggered magnetisation operator.

**staggered magnetisation**

As a second example, recall the Schrödinger field theory of Section 1.3.1. This model has a global $U(1)$ symmetry describing uniform rotations of the phase of a complex scalar field $\psi(x)$. To find out what it means for this symmetry to be broken, we can for example consider the field $\psi(x)$ itself to be an interpolating field. The associated order parameter is then:

$$[Q, \psi(x)] = \int d^d x' \, \hbar[\psi^*(x')\psi(x'), \psi(x)] = -\hbar\psi(x). \tag{2.32}$$

Here we used the commutation relation $[\psi^*(x'), \psi(x)] = -\delta(x - x')$. The order parameter operator thus turns out to be given by the field $\psi(x)$ itself. Because an eigenstate of the order parameter is a symmetry-breaking state, the states that break the global phase rotation symmetry must be eigenstates of the operator $\psi(x)$. But that operator is just the annihilation operator for quanta of the $\psi$-field. Within Fock space, it is possible to construct eigenstates of the annihilation operator, and these are called *coherent states*. You can check that the state $e^{\int d^D x \, \phi(x)\psi^*(x)}|\text{vac}\rangle$ is an eigenstate of the annihilation operator $\psi(x)$, with complex eigenvalue $\phi(x)$. Because this coherent state is an eigenstate of the field operator, $|\phi(x)|^2$ is the expectation value of the number of field quanta, or particles, at position **x**. Notice that if we assume translational invariance as before, $\phi$ is independent of **x**. Expanding the exponential in the definition of the coherent state, it is seen to be a superposition of infinitely many states with different numbers of excited field quanta. Conversely, an eigenstate of the number or density operator $\psi^*(x)\psi(x)$ can be written as a superposition of infinitely many coherent states which all have the same absolute value of $\phi(x)$, but which differ in phase. The phase and modulus of $\phi$ can in fact be shown to be conjugate variables, with canonical commutation relations.

**coherent state**

The symmetry-breaking state of the Schrödinger field is a coherent state, which has an indefinite number of particles or quanta. This is in direct accordance with the fact that the symmetry generated by $Q \propto \int \psi^*\psi$ is associated with the conservation of the number of field quanta. Distinct symmetry-breaking states are characterised by the phase of $\phi$. Performing rotations with $\exp(i\alpha Q)$ will lead to other order parameter operators $e^{i\alpha}\psi(x)$ which indeed correspond to broken-symmetry states with different phase values that can be labelled by elements of the coset $U(1)/1 \simeq U(1)$. The formation of a state with indeterminate particle number and precise phase-value is a good interpretation of what happens in Bose–Einstein condensates such as superfluids and superconductors. In these symmetry-broken states of matter, it costs zero energy to add or remove a particle in the condensate.

### 2.5.3 The classical state

Given the definition of the order parameter operator $\mathcal{O}$ in Eq. (2.31), you might be tempted to believe that broken-symmetry states are simply the eigenstates of $\mathcal{O}$, with eigenvalues $O$. This would certainly justify the definition of the order parameter as the expectation value of the order parameter operator, and in most cases it also agrees very well with our expectation for a perfectly ordered state. This is because in translationally invariant systems, the eigenstates of the local operator $\mathcal{O}(x)$ are tensor products of local eigenstates $|\psi\rangle = \otimes_{\mathbf{x}} |\psi(\mathbf{x})\rangle$, and they correspond directly to the states of a classical Hamiltonian in which all operators are replaced by their expectation values. We will call the eigenstates of the order parameter operator *classical states*.

**classical state**

The symmetry-breaking states encountered in real quantum systems are typically not classical states. This is easy to understand, because although a symmetry-breaking perturbation may dominate the shape of the ground state for sufficiently large systems, it is not the only contribution to the Hamiltonian. The remaining, symmetric part of the Hamiltonian contributes to the ground state and takes it away from the classical ideal. We have already seen this for the antiferromagnet in Section 2.4. The true, quantum, broken-symmetry states typically have order parameter expectation values close to those of the classical state. The quantum states can therefore be thought of as arising in a perturbation theory around the classical states. The differences between the classical state and the true quantum broken-symmetry state, are then known as *quantum corrections* to the classical state. These corrections consist of a part at zero wave number, related to the tower of states that will be discussed in more detail in Section 2.6, and a part at nonzero momentum, which is the topic of Chapter 4.

**quantum corrections**

### 2.5.4 Long-range order

For any given uniform ground state, the order parameter identifies whether it has broken or unbroken symmetries, and distinguishes between inequivalent symmetry-breaking states. In practical calculations however, one often does not know the exact ground state of a given Hamiltonian, and many systems exist in conditions that are not perfectly uniform. In those cases, a particularly useful alternative way of quantifying the occurrence of spontaneous symmetry breaking, is through the behaviour of the so-called *two-point correlation function*:

$$C(\mathbf{x},\mathbf{x}') = \langle \psi | \mathcal{O}^{\dagger}(\mathbf{x})\mathcal{O}(\mathbf{x}') | \psi \rangle . \tag{2.33}$$

Here $\mathcal{O}$ is the local order parameter operator. Clearly, if $|\psi\rangle$ is uniform in space and an eigenstate of the order parameter, the correlation function is the same for any choice of the coordinates $\mathbf{x}$ and $\mathbf{x}'$, and equal to the square of the uniform order parameter. The advantage of the two-point function, however, is that it can be used also in less clear-cut cases. The behaviour of the two-point function can be divided roughly into two distinct classes, depending on its functional form as the distance $|\mathbf{x}-\mathbf{x}'|$ is taken to infinity:

$$C(\mathbf{x},\mathbf{x}') \propto \begin{cases} \text{constant} & \textit{long-range ordered} \\ e^{-|\mathbf{x}-\mathbf{x}'|/l} & \textit{disordered} \end{cases} \qquad |\mathbf{x}-\mathbf{x}'| \to \infty . \tag{2.34}$$

**long-range order**

Here $l$ is a length scale called the *correlation length*. For *long-range ordered* systems, the spatial average of the local order parameter will be non-zero, and the correlation length $l$ diverges. The presence of long-range order is therefore associated with the breaking of a symmetry. In fact, the two-point correlation function signals the propensity to break symmetry already for finite-size systems that really have a symmetric ground state. Even though the order parameter expectation value is exactly zero, the two-point function shows correlations for long separations. This can be easily understood by considering the extreme example of the two-spin

**correlation length**

singlet state $|\uparrow\downarrow\rangle - |\downarrow\uparrow\rangle$, which is the ground state of the two-site Heisenberg antiferromagnet. Clearly, the singlet state has no preferred direction of staggered magnetisation, but the two spins are definitely anti-parallel. Especially in numerical investigations, the two-point function is typically easy to 'measure' even with only limited information about the spectrum, and is widely used in establishing the presence of order and spontaneous symmetry breaking. We will come back to this in more detail when we discuss stability in Section 2.7.

Notice that the separation of states into long-range ordered and disordered is not exhaustive. A special case may occur in low dimensions when the two-point function is proportional $|\mathbf{x} - \mathbf{x}'|^c$, for some exponent $c$. This is called *algebraic long-range order* or *quasi long-range order*. We will discuss this type of order in Section 6.3.

> **algebraic long-range order**

You may have also encountered the term "off-diagonal long-range order" or "ODLRO" in the literature. This term was introduced by Oliver Penrose and Lars Onsager in the 1950s [27, 28], in the context of superfluidity in helium, as a way of contrasting the superfluid order with ordering in solids. The concept is largely historical, and the distinction between ODLRO and other types of order has become obsolete with the modern definition of the order parameter in Eq. (2.31). To see this, we will first consider the usual definition of ODLRO in terms of the $N$-particle wave function $\Psi(\mathbf{x}_1, \ldots, \mathbf{x}_N)$. We can then define:

> **long-range order – off-diagonal**

$$\rho(\mathbf{x}_1, \ldots, \mathbf{x}_N, \mathbf{y}_1, \ldots, \mathbf{y}_N) = \Psi^*(\mathbf{x}_1, \ldots, \mathbf{x}_N)\Psi(\mathbf{y}_1, \ldots, \mathbf{y}_N), \tag{2.35}$$

for two sets of coordinates $\mathbf{x}_i$ and $\mathbf{y}_i$. For coinciding coordinates $\mathbf{y}_i = \mathbf{x}_i$, the matrix $\rho(\mathbf{x}_i) \equiv \rho(\mathbf{x}_i, \mathbf{x}_i)$ is just the usual *density matrix*, giving the probability for finding the $N$ particles at positions $\mathbf{x}_i$. The two-particle reduced density matrix can be found by integrating over all but two of the coordinates:

> **density matrix**

$$\rho_{\mathrm{D},2}(\mathbf{x}_1, \mathbf{x}_2) = \int d\mathbf{x}_3 \cdots d\mathbf{x}_N \ \Psi^*(\mathbf{x}_1, \mathbf{x}_2, \mathbf{x}_3, \ldots, \mathbf{x}_N)\Psi(\mathbf{x}_1, \mathbf{x}_2, \mathbf{x}_3, \ldots, \mathbf{x}_N). \tag{2.36}$$

If space is uniform, the reduced density matrix is invariant under global translations, and can only depend on the difference of the two coordinates, so that $\rho_{\mathrm{D},2}(\mathbf{x}_1, \mathbf{x}_2) = \rho_{\mathrm{D},2}(\mathbf{x}_1 - \mathbf{x}_2)$. *Diagonal long-range order* is said to occur if $\rho_2(\mathbf{x}_1 - \mathbf{x}_2)$ is periodic in $\mathbf{x}_1 - \mathbf{x}_2$. This is the usual ordering we find for solids such as the harmonic crystal introduced in Section 2.3. It is called diagonal because we only need to consider diagonal elements of $\rho(\mathbf{x}_i, \mathbf{y}_i)$, with $\mathbf{y}_i = \mathbf{x}_i$.

> **long-range order – diagonal**

Alternatively, we can consider another type of two-point function:

$$\rho_{\mathrm{O},2}(\mathbf{x}, \mathbf{y}) = \int d\mathbf{x}_2 \cdots d\mathbf{x}_N \ \Psi^*(\mathbf{x}, \mathbf{x}_2, \ldots, \mathbf{x}_N)\Psi(\mathbf{y}, \mathbf{x}_2, \ldots, \mathbf{x}_N). \tag{2.37}$$

That is, we choose all wave function coordinates except the first to coincide, and integrate over them. Note that this entails one more integration than the definition of the reduced density matrix in Eq. (2.36). In fact, $\rho_{\mathrm{O},2}$ and $\rho_{\mathrm{D},2}$ represent two very different physical quantities that have little to do with one another. For superfluids in particular, $\rho_{\mathrm{O},2}$ is almost identical to the two-point correlation function of Eq. (2.33), if we choose the order parameter operator $\mathcal{O}(x)$ to be the field operator $\Psi(x)$. *Off-diagonal long-range order* (ODLRO) is said to occur if $\rho_{\mathrm{O},2}(\mathbf{x}, \mathbf{y})$ does not vanish as $\mathbf{x} - \mathbf{y}$ is taken to infinity. It was called off-diagonal because it involves off-diagonal elements of $\rho(\mathbf{x}_i, \mathbf{y}_i)$.

> **long-range order – off-diagonal**

Although we need to be careful when applying Eq. (2.34) to crystals, because they retain discrete translational symmetry and $C(\mathbf{x}, \mathbf{x}')$ approaches a periodic function rather than a true constant at large separations, the two-point function does capture the breakdown of symmetry in both crystals and superfluids in essentially the same way. In fact, both diagonal and off-diagonal long-range order are part of a much larger family of possible types of ordered states that are all classified by the behaviour of the two-point correlation function at large separation. There is nothing special about either the order occurring in crystals (DLRO), or that

in superfluids (ODLRO). When off-diagonal long-range order was originally introduced, the concept of symmetry breaking for internal degrees of freedom and its embedding within the larger theory of symmetry breaking in general were not yet developed, and ODLRO was a way to capture the long-range ordering of the internal $U(1)$-phase degree of freedom contained in the $N$-body wave function.

### 2.5.5 The Josephson effect

When we first discussed symmetric states in Section 1.1, we pointed out that symmetry must always be defined with respect to some reference. In the example of a crystal breaking translational symmetry, the broken translations are really defined with respect to an outside observer, who can for example measure the distance between the crystal and herself. More generally, in order to observe the breakdown of a symmetry in a given state, an observer needs some reference frame with respect to which the broken symmetry can be measured. For the reference frame to be able to distinguish between inequivalent symmetry-breaking states, it must itself be in broken-symmetry state. That is, you cannot measure the position of a crystal with respect to a uniform fluid permeating all of space, and it is not possible to measure a direction of magnetisation with a piece of plastic that cannot itself be magnetised. Although this might seem obvious for crystals and magnets, one could wonder what it implies for materials hosting less intuitive forms of broken symmetry, like the $U(1)$ phase-rotation symmetry associated with conserved particle number, which we argued in Section 2.5.2 to be broken in superfluids? This question found a literal manifestation in the spontaneous tunnelling current that was predicted by Josephson in 1962 to occur between two separated pieces of superconducting material [29].

**Josephson effect**

Since the origin of the Josephson effect lies in the broken $U(1)$ symmetry, we will discuss it here for neutral superfluids rather than superconductors. As we saw before, a superfluid is a state with broken $U(1)$ phase-rotation symmetry and a complex order parameter $\psi = |\psi|e^{i\varphi}$. In terms of observable properties, the superfluid is characterised by its ability to host *supercurrents* that flow without viscosity. One way to understand this, is by noticing that the supercurrent is identical to the conserved Noether current associated with the broken $U(1)$ symmetry:

$$j^n = i((\partial_n \psi^*)\psi - \psi^*(\partial_n \psi)). \tag{2.38}$$

The order parameter field $\psi$ is the expectation value of the field operator, but is often referred to in the more popular literature as a "macroscopic wave function". Although somewhat misleading, this terminology does emphasise the fact that like a quantum wave function, the field $\psi(x)$ satisfies equations of motion with spatial derivatives, which force it to be continuous. As a consequence, the field does not abruptly vanish at the boundary of a sample, but rather falls off exponentially into the vacuum. For two samples of superfluid separated by a small gap, the order parameter fields extending into the gap from both sides can overlap. Just like for quantum mechanical wave functions, this implies the possibility for field quanta to tunnel from one sample to the other, which is the essence of the Josephson effect.

For a junction of width $w$ between superfluids with constant order parameters $\psi_1$ and $\psi_2$, the order parameter field inside the junction can be written as:

$$\psi(x) = Ae^{-x/\xi} + Be^{x/\xi}. \tag{2.39}$$

Here, $A$ and $B$ are complex constants, and the decay length of the order parameter field in the vacuum is $\xi$. The samples are assumed to extend indefinitely in the $y$ and $z$ direction while being semi-infinite in the $x$ direction, with one sample having an edge at $x = -w/2$ and the other at $x = w/2$. You can think of the field in the junction as a superposition of the decaying

order parameter fields from either side. The boundary conditions are given by the field values in the samples, $\psi(-w/2) = \psi_1$ and $\psi(w/2) = \psi_2$, so that we find:

$$A = \frac{e^{w/2\xi}\psi_1 - e^{-w/2\xi}\psi_2}{2\sinh(w/\xi)}, \qquad\qquad B = \frac{e^{w/2\xi}\psi_2 - e^{-w/2\xi}\psi_1}{2\sinh(w/\xi)}. \qquad (2.40)$$

The current density in the junction is given by Eq. (2.38), and equals $j^x = 2i(B^*A - A^*B)/\xi$, independent of the position $x$. Substituting the values of $A$ and $B$ yields the expression:

$$j^x = i\frac{\psi_2^*\psi_1 - \psi_1^*\psi_2}{\xi\sinh(w/\xi)} = \frac{2|\psi_1||\psi_2|}{\xi\sinh(w/\xi)}\sin(\varphi_2 - \varphi_1). \qquad (2.41)$$

We thus find a current per unit area $j_x$ flowing through the junction, which is proportional to the sine of the phase difference between the two superfluid order parameters.

The flow of supercurrent without a chemical potential difference between the superfluid samples is an interesting physical observation in and of itself. In the context of symmetry breaking however, it also gains a more fundamental interpretation. The phase of the order parameter for one superconducting sample can be determined with respect to the phase of a second sample by measuring the Josephson current flowing between them. This is precisely analogous to the way that the position of a crystal can be determined only with respect to the position of some other object with broken translational symmetry. Historically, the discovery of the Josephson effect was therefore the deciding factor in settling the debate of whether or not a symmetry was spontaneously broken in superconductors (see for instance Ref. [30]). More generally, the calculation of the Josephson effect should actually be applicable in some form to two pieces of material with any type of spontaneously broken symmetry [31]. There are very few known examples of generalised Josephson effects outside of superconductivity and superfluidity, but at least conceptually, the Josephson effect offers an unambiguous general way of measuring the order parameter of any sample with respect to a reference broken-symmetry state.

---

**Exercise 2.6 (Josephson effect)** The Josephson effect equations for describing the generalised Josephson current between any two symmetry-breaking objects can be derived from a very simple model due to Feynman [32]. Here you write the global order parameter operators of the two systems at the left (L) and right (R) as $\Psi_L(t)$ and $\Psi_R(t)$ (no **x**-dependence). The Hamiltonian is taken to be of the form:

$$H = H_L + H_R + H_K, \qquad (2.42)$$

where the left and right systems are described by local Hamiltonians $H_L$ and $H_R$. The coupling between the order parameters across the junction is described by $H_K$.

**a.** For superconductors, the order parameter operators correspond to field operators $\psi_i$, with commutation relation $[\psi_i, \psi_{i'}^*] = \delta_{i,i'}$. The coupling is given by:

$$H_K = K(\psi_R^*\psi_L + \psi_L^*\psi_R), \qquad (2.43)$$

where $K$ is a coupling constant with units of energy. Using the Heisenberg equations of motions $-i\hbar\partial_t A = [H, A]$, derive an expression for the Josephson current $I_J = \partial_t(\psi_L^*\psi_L)$. Compare your result with Eq. (2.41).

**b.** Now consider two ferromagnets, with magnetisation vectors $\mathbf{M}_i$, and commutation relations $[M_i^a, M_{i'}^b] = i\epsilon_{abc}\delta_{i,i'}M_i^c$. They are coupled to each other via the interaction

described by:

$$H_K = K\mathbf{M}_L \cdot \mathbf{M}_R. \tag{2.44}$$

Derive an expression for the "spin Josephson current" $\partial_t \mathbf{M}_L$. Your result will agree with the much more sophisticated calculation based on a microscopic description of the tunnelling of electrons between two ferromagnets [33].

## 2.6 The tower of states

As emphasised in several places already, the true ground state of a finite-sized quantum system without any symmetry-breaking field present, is typically symmetric. Except for symmetry-breaking states associated with conserved order parameters that will be introduced in Exercise 2.7 and discussed in more detail in Section 3.4, the true ground state of a symmetric Hamiltonian is unique, and is an eigenstate of the symmetry generators. Since the symmetry transformations are global, the symmetric ground state also has a global structure. In particular, it typically contains *long-range entanglement* between distant parts of the system, which therefore all strongly depend on each other (as discussed in more detail in Section A.2). Furthermore, as we will see in the next section, the symmetric state is unstable. In terms of the spectrum of eigenstates of the symmetric Hamiltonian, however, the long-range entangled nature of the ground state is in no way exceptional. There is a whole set of eigenstates that can be seen as low-energy, global excitations on top of the ground state, which all share the same feature.

Crudely speaking, if the order parameter can be defined in terms of some canonical observable, the symmetric Hamiltonian must contain a kinetic energy proportional to the total canonical momentum squared. This is easy to see, because the eigenstates of momentum are symmetric combinations of all possible canonical positions. Within a finite volume $V$, the Fourier transform of the canonical momentum operator is given by $\mathbf{p}(\mathbf{x}) = \sum_{\mathbf{k}} e^{i\mathbf{k}\cdot\mathbf{x}}\mathbf{p}_{\mathbf{k}}$. The total momentum is proportional to the $\mathbf{k} = 0$ component of this decomposition:

$$\begin{aligned}
\mathbf{p}_{\text{tot}} &= \int d\mathbf{x}\, \mathbf{p}(\mathbf{x}) = \sum_{\mathbf{k}} \int d\mathbf{x}\, e^{i\mathbf{k}\cdot\mathbf{x}}\mathbf{p}_{\mathbf{k}} \\
&= V\sum_{\mathbf{k}} \delta_{\mathbf{k}}\mathbf{p}_{\mathbf{k}} = V\mathbf{p}_{\mathbf{k}=0}.
\end{aligned} \tag{2.45}$$

Here we used the representation of the Kronecker delta function given by $\frac{1}{V}\int d\mathbf{x}\, e^{i\mathbf{k}\cdot\mathbf{x}} = \delta_{\mathbf{k}}$[8]. The term in the Hamiltonian proportional to the total momentum comes from the $\mathbf{k} = 0$ part of the usual kinetic energy operator:

$$\begin{aligned}
H_{\text{kin}} &\propto \int d\mathbf{x}\, \mathbf{p}^2(\mathbf{x}) = V\sum_{\mathbf{k}} \mathbf{p}_{\mathbf{k}} \cdot \mathbf{p}_{-\mathbf{k}} \\
&= \frac{1}{V}\mathbf{p}_{\text{tot}}^2 + V\sum_{\mathbf{k}\neq 0} \mathbf{p}_{\mathbf{k}} \cdot \mathbf{p}_{-\mathbf{k}}.
\end{aligned} \tag{2.46}$$

The second term in the final line combines with the potential energy to describe internal excitations like phonons, magnons, supercurrents, and so on. The first term on the other hand, is just the (canonical) kinetic energy of the object as a whole.

Since $V \propto N$, the modes corresponding to the total momentum will be quantised in any confining potential with spacing $1/N$, as in Figs. 2.3 and 2.5. This set of global eigenstates of canonical total momentum is referred to as the *tower of states*, or *Anderson tower of states*, or

---

[8]Note that $\frac{1}{V}\sum_{\mathbf{k}} e^{-i\mathbf{k}\cdot\mathbf{x}} = \delta(\mathbf{x})$ yields the Dirac delta function.

occasionally as the *thin spectrum*. In the thermodynamic limit, all states in the tower become exactly degenerate.

As we will see in Section 3, systems with a spontaneously broken symmetry have gapless, propagating excitations, called Nambu–Goldstone modes. These modes exist at non-zero wave number and for a finite system of linear size $L$, the lowest possible energy they can take is proportional to $1/L$. In spatial dimension $D > 1$, the states in the $\mathbf{k} = 0$ tower of states have energies proportional to $1/V \propto 1/L^D$ and these are therefore much lower in energy than even the $\mathbf{k} > 0$ states with the lowest possible energies. Since all states in the tower are eigenstates of the total canonical momentum at zero wave number, they all have a global structure, and they are all symmetric. The classical symmetry-broken states are superpositions of the states in the tower, with the special property that they can be written as local product states of the form $\otimes_{\mathbf{x}} |\psi(\mathbf{x})\rangle$. They are not energy eigenstates, but because the energy spacing between the states in the tower is so small, the classical states are very narrow wavepackets in energy space, which take on a single, well-defined energy expectation value in the thermodynamic limit.

Because the existence of a tower of global excitations is so intrinsically linked to spontaneous symmetry breaking, it is reasonable to ask whether these states influence any other measurable properties of a symmetry-broken object. To see this, consider the free energy of a quantum system with symmetric Hamiltonian $H$, which at temperature $T$ can be calculated using:

$$F = -k_{\mathrm{B}} T \ln Z,$$
$$Z = \sum_{|\psi\rangle} \langle \psi | e^{-\frac{H}{k_{\mathrm{B}} T}} | \psi \rangle. \tag{2.47}$$

The sum in the partition function $Z$ runs over all energy eigenstates $|\psi\rangle$ of the Hamiltonian. The free energy associated with the collective part of the Hamiltonian for an object consisting of $N$ interacting particles, is generically proportional to $\ln N$. Again crudely, this can be seen by defining the collective part of the Hamiltonian to be the kinetic energy associated with some total canonical momentum, so that:

$$Z_{\mathrm{coll}} = \sum_{\mathbf{p}_{\mathrm{tot}}} \langle \mathbf{p}_{\mathrm{tot}} | e^{-\frac{H_{\mathrm{coll}}}{k_{\mathrm{B}} T}} | \mathbf{p}_{\mathrm{tot}} \rangle \sim \int d\mathbf{p}_{\mathrm{tot}} \, e^{-\frac{\mathbf{p}_{\mathrm{tot}}^2}{k_{\mathrm{B}} T V}}$$
$$\propto \sqrt{k_{\mathrm{B}} T N}. \tag{2.48}$$

Since $Z_{\mathrm{coll}}$ is proportional to $\sqrt{N}$, the corresponding contribution of the tower of collective states to the free energy $F_{\mathrm{coll}}$ is proportional to $\ln N$. The free energy $F$ associated with the full Hamiltonian must always be proportional to $N$, because it is an extensive quantity. The relative contribution of the collective states to the total free energy, $F_{\mathrm{coll}}/F$, is then proportional to $\ln N / N$, and disappears in the limit of large system size. In other words, even though they are the only states with energies as low as $1/N$, there are so few states in the tower that they do not contribute to the free energy at any non-zero temperature, no matter how low. This part of the spectrum in fact is so 'thin' that it cannot be observed in any thermodynamic properties of the material, such as specific heat or conductivity, which are all determined by the free energy.

This observation is again fully general for collective states governing the spontaneous breakdown of any continuous symmetry. They always form an exceedingly thin part of the spectrum that is practically undetectable for any realistically sized system in our everyday world. Paradoxically, one of the few ways in which the presence of this part of the spectrum does have an influence on measurable quantities, is due precisely to its undetectable nature. If a material with a broken continuous symmetry is used to store quantum information, for example using the presence or absence of a magnon in an antiferromagnet as the zero and one states of a hypothetical qubit, then the presence of many states beyond any experimental

control acts as a sort of environment to the qubit. Even if one could entirely isolate such a system from any external influences, the qubit will decohere, because the information about the magnon state becomes entangled with the unmeasurable, thin part of the spectrum [34].

## 2.7  Stability of states

In spontaneous symmetry breaking, the exact ground state of a system is infinitely sensitive to perturbations, which therefore always yield a broken-symmetry state in the thermodynamic limit. However, such symmetric states are also generically unstable all by themselves. To illustrate this, consider a magnetic system with some local magnetisation $\sigma^z(\mathbf{x})$ defined at each position. If a measurement of the magnetisation at $\mathbf{x}$ can influence a subsequent measurement at a far-away positions $\mathbf{y}$, the system is *unstable against local measurements*. In other words, stability requires the expectation value $\langle \sigma^z(\mathbf{y}) \rangle$ at position $\mathbf{y}$ to be independent of the measurement of magnetisation $\sigma^z(\mathbf{x})$ at a position $\mathbf{x}$ far away from $\mathbf{y}$.

**instability – against local measurement**

A simple example of an *unstable* state is the ground state of the transverse field Ising model,

**Ising model**

$$H = -J \sum_{\langle ij \rangle} \sigma_i^z \sigma_j^z - \mu \sum_i \sigma_i^x. \tag{2.49}$$

This Hamiltonian is defined on any lattice of spin-$\frac{1}{2}$ states, with $\langle ij \rangle$ denoting nearest neighbours, and $\sigma_i^a$ Pauli matrices on site $i$. The coupling $J$ is positive and the transverse field is represented by $\mu$. This model has a discrete, global $\mathbb{Z}_2$ symmetry of simultaneously flipping all spins in the $z$-direction, so that $\sigma_i^z \to -\sigma_i^z$. If the transverse field is small, $0 < \mu \ll J$, the ground state of this model is approximately a superposition of all spins up and all spins down,

$$|\psi_0\rangle \approx |\uparrow\uparrow\uparrow \cdots\rangle - |\downarrow\downarrow\downarrow \cdots\rangle. \tag{2.50}$$

Adopting the continuum limit, in which $\sigma_i^z$ becomes the function $\sigma^z(\mathbf{x})$ of the continuous variable $\mathbf{x}$, notice that the expectation value of $\sigma^z(\mathbf{x})$ at any position equals zero, $\langle \sigma^z(\mathbf{x}) \rangle = 0$, as expected for a system with spin-flip symmetry. Measuring the $z$-component of the spin at position $\mathbf{x}$ will collapse the superposed ground state onto the component corresponding to the observed value of $\sigma^z(\mathbf{x})$. For example, if we happen to measure an up spin at $\mathbf{x}$, the entire state after the measurement has collapsed to $|\uparrow\uparrow\uparrow \cdots\rangle$, and subsequently measuring the $z$-component of spin at any position $\mathbf{y}$ will *always* yield up. The expectation value of $\sigma^z(\mathbf{y})$ has thus qualitatively changed because $\sigma^z(\mathbf{x})$ was measured, and the state of Eq. (2.50) is concluded to be unstable against local measurements.

Conversely, the broken-symmetry state $|\uparrow\uparrow\uparrow \cdots\rangle$ itself *is* stable against local measurements, since no measurement of $\sigma^z(\mathbf{x})$ for any $\mathbf{x}$ will influence the result of subsequent measurements at any other positions. In fact, this pattern is general, and the stability of the symmetry-breaking state is a direct consequence of its long-range order, discussed in Section 2.5.4. Local measurements will generically rapidly collapse an unstable symmetric state onto one of the possible broken-symmetry states. The definition of stability with respect to local measurements is especially relevant when considering the embedding of any given system in its local environment. Even the weakest interactions with an environment can easily amount to an effective measurement of local observables like the magnetisation $\sigma^z(\mathbf{x})$, and thus prevent symmetric states from being observed in any realistic setting.

The central ingredient in the definition of stability against local measurements, is the requirement for an unstable state, a *single* measurement influences the outcome of *many* subsequent measurements. To quantify the meaning of 'many', the concept of *cluster decomposition* can be used. A state is said to satisfy the cluster decomposition property if and only if for all local observables $a(\mathbf{x})$ and $b(\mathbf{y})$ we have

**cluster decomposition**

$$C_{ab}(\mathbf{x}, \mathbf{y}) = \langle a(\mathbf{x})b(\mathbf{y}) \rangle - \langle a(\mathbf{x}) \rangle \langle b(\mathbf{y}) \rangle \to 0 \text{ when } |\mathbf{x} - \mathbf{y}| \to \infty. \tag{2.51}$$

This means that measurements of any $a(\mathbf{x})$ and $b(\mathbf{y})$, provided $\mathbf{x}$ and $\mathbf{y}$ are far apart, will be independent. Cluster decomposition can therefore be considered a requirement for macroscopic stability.

It is easy to check that the ground state of the transverse field Ising model in Eq. (2.50) does *not* satisfy the cluster decomposition property. States like these are sometimes called *cat states*, in reference to Schrödinger's cat. The exact ground states of Hamiltonians suscepti- **cat states** ble to spontaneous symmetry breaking are almost always cat states. Conversely, the broken-symmetry states that may be stabilised in the thermodynamic limit always *do* satisfy the cluster decomposition property.

The concept of cluster decomposition is itself closely related to a thermodynamic restriction on fluctuations of extensive observables. In thermodynamics, the extensive observables of two subsystems can be added to find the corresponding extensive observable associated with the system as a whole. In other words, extensive observables can be written as a sum of local observables, $A = \sum_{\mathbf{x}} a(\mathbf{x})$. The expectation value of $A$ must therefore scale with the volume of the system, $\langle A \rangle \sim \mathcal{O}(V)$. In general, the *variance* of an observable scales as **variance** $\mathrm{Var}(A) = \langle A^2 \rangle - \langle A \rangle^2 \propto V^\alpha$. If the exponent $\alpha$ is two or greater, $\sqrt{\mathrm{Var}(A)}/\langle A \rangle$ does not vanish in the thermodynamic limit, and the fluctuations in $A$ are as large as its expectation value. The state is then said to be *thermodynamically unstable*. On the other hand, for states with **instability –** $\alpha = 1$, the fluctuations vanish in comparison to the expectation value, and these states are **thermody-** thermodynamically stable. Product states are always of this type. Some special states may **namic** have $1 < \alpha < 2$. These are thermodynamically stable, but they may be fragile in other senses. For instance, the critical systems that we will discuss in Section 5.5 fall in this class.

Unsurprisingly, a system that violates the cluster decomposition property is a superposition of macroscopically distinct states, and thus possesses macroscopic fluctuations of an extensive variable. This can be easily seen by writing the variance in terms of the two-point correlation function $C(\mathbf{x}, \mathbf{y})$,

$$
\begin{aligned}
\mathrm{Var}(A) &= \langle A^2 \rangle - \langle A \rangle^2 & (2.52) \\
&= \sum_{\mathbf{x},\mathbf{y}} \langle a(\mathbf{x}) a(\mathbf{y}) \rangle - \langle a(\mathbf{x}) \rangle \langle a(\mathbf{y}) \rangle & (2.53) \\
&= \sum_{\mathbf{x},\mathbf{y}} C_{aa}(\mathbf{x}, \mathbf{y}). & (2.54)
\end{aligned}
$$

In a product state, such as the ordered, symmetry-breaking states, the two-point function becomes equal to the product of local expectation values for $\mathbf{x}$ and $\mathbf{y}$ far apart. The variance is then dominated by contributions with $\mathbf{x} \sim \mathbf{y}$, and therefore scales as $\mathrm{Var}(A) \sim \mathcal{O}(V)$. On the other hand, if $C(\mathbf{x}, \mathbf{y})$ does not equal the uncorrelated product of expectation values for large $\mathbf{x} - \mathbf{y}$, the variance has contributions from all terms in the double sum and scales as $\mathrm{Var}(A) \sim \mathcal{O}(V^2)$. That is, fluctuations in a measurement of $A$ are as large as the observed average $A$ itself, indicating a highly unstable situation.

We can thus define stability in three equivalent ways: using the stability against local measurements, examining the cluster decomposition property, and considering the variance of extensive variables. The symmetric ground state of models that exhibit SSB are generically unstable, while classical broken-symmetry states are always stable under any of these three definitions of stability. The applicability of the rule that symmetric ground states are inherently unstable is further reaching than many other results presented in these lecture notes. For example, Noether's theorem only applies to continuous symmetries, and the tower of states is only relevant to systems in which the order parameter does not commute with the Hamiltonian (see Exercise 2.7). However, the instability of symmetric states is a property of *all* systems that exhibit spontaneous symmetry breaking.

**Exercise 2.7 (Heisenberg Ferromagnet)** We end this chapter by highlighting a special case within the realm of symmetry breaking. In daily parlance, the ferromagnet is often used as the simplest example of symmetry breaking. Unfortunately, as you will see, the properties of a ferromagnet make it exceptional, and unlike most other forms of symmetry broken states, such as antiferromagnets, superfluids, or even the symmetry breaking in the Standard Model of particle physics.

Consider the Heisenberg Hamiltonian Eq. (2.17), but now with *negative* coupling $J < 0$, so that pairs of spins prefer to be aligned, rather than anti-aligned. This can easily be accommodated by having all spins aligned, say in the $z$-direction.
**a.** Show that $S^x$ and $S^y$ are spontaneously broken according to Eq. (2.30), by finding appropriate interpolating fields.
   The order parameter operator for the state with all spins aligned in the $z$-direction is $S^z$, which is itself one of the symmetry generators of the symmetric Heisenberg Hamiltonian. The order parameter operator thus commutes with the Hamiltonian. The ferromagnet is truly exceptional, however, due to the following property:
**b.** Show that the state with all spins aligned (in the $z$-direction) is an eigenstate of the Hamiltonian.
*Hint:* use the second line of Eq. (2.18).
   This result implies that the fully magnetised, classical state in the sense of Section 2.5.3 is an exact ground state of the symmetric quantum mechanical Hamiltonian, for any system size. In fact, any fully magnetised state, with all spins simultaneously pointing in any direction, is a ground state. The ground state is thus far from unique, even for finite-sized ferromagnets. There is no tower of states and there are no quantum corrections. The system merely chooses a state that is stable against local perturbations from the degenerate set of ground states.
   We will examine the ferromagnet in more detail in Section 3.4. For now, the moral of this exercise is that you should mistrust any text that uses the ferromagnet as an archetype of spontaneous symmetry breaking. It truly is an exceptional case.

ferromagnet

# 3   Nambu–Goldstone modes

Every symmetry of the Hamiltonian or Lagrangian corresponds to a conserved quantity, regardless of whether or not the state of the system respects the symmetry. In homogeneous space for example, both a classical ball with spontaneously broken translational symmetry, and an electron in a symmetric, plane-wave state, will have a conserved total momentum. The intimate relation between the conserved global quantity and the possibility of spontaneously breaking a symmetry, was elucidated in Section 2.6, where we discussed the tower of states. This collective, $k = 0$, part of the spectrum consists of eigenstates of the conserved global quantity, which in the thermodynamic limit can be combined into a coherent-state superposition. Both the individual eigenstates and the symmetry-breaking superposition conserve the (expectation value of the) global quantity.

As we saw in Section 1.2.2, however, Noether's theorem has implications far beyond the global aspects of the system. For every continuous global symmetry, it guarantees the existence of a *locally* conserved current, obeying a local continuity equation. Again, this form of Noether's theorem holds regardless of whether or not the state of the system respects the symmetry. Moreover, the local conservation law is intimately tied to a generic property of the spectrum of systems with a spontaneously broken continuous symmetry. Rather than affecting

the collective states, however, the local continuity equation impacts the excitations at non-zero wave number, and guarantees the appearance of gapless modes known as *Nambu–Goldstone (NG) modes*. In particle physics and relativistic quantum field theory, these modes are referred to as (Nambu–)Goldstone bosons, and they are said to be massless instead of gapless. The difference is purely a matter of nomenclature.

**Nambu–Goldstone mode**

To understand the nature of the NG modes, consider the temporal component of the Noether current operator $j^t(x)$ related to the symmetry generator $Q = \int \mathrm{d}^D x\, j^t$ that is spontaneously broken. The NG mode $|\pi(\mathbf{k})\rangle$ can then be viewed as a plane-wave superposition of local excitations created by acting with the Noether current operator on the broken-symmetry state $|\psi\rangle$:

$$|\pi(\mathbf{k}, t)\rangle \propto \int \mathrm{d}^D x\, \mathrm{e}^{\mathrm{i}\mathbf{k}\cdot\mathbf{x}} j^t(\mathbf{x}, t)|\psi\rangle. \tag{3.1}$$

Goldstone's theorem, which we will introduce below, shows these states to be gapless. That is, their energy goes to zero as $\mathbf{k} \to 0$. Because low-energy excitations of a symmetry-broken state necessarily correspond to creating local Noether charge density, Noether's continuity equation guarantees that they will be dispersed over time. In other words, low-energy disturbances in the order parameter will be carried away like waves in a puddle carry away the local excitation of a raindrop, and systems with a spontaneously broken symmetry are thus endowed with a form of *rigidity* [11].

**rigidity**

## 3.1 Goldstone's theorem

Before delving into the proof for Goldstone's theorem and discussing some of its implications and more modern aspects, let us simply state the theorem and define its realm of applicability:

**Theorem 3.1** *(Goldstone's theorem). If a global, continuous symmetry is spontaneously broken in the absence of long-ranged interactions, and leaving some (discrete) translational symmetry intact, then there exists a mode in the spectrum whose energy vanishes as its wave number approaches zero.*

The theorem includes many assumptions, and in cases where these do not hold, the NG mode either ceases to exist or to be gapless. If a symmetry is explicitly broken by an external field $\mu$, for example, the NG mode will exist, but with a gap of size $\mu$ at $k \to 0$. If the broken symmetry is discrete, rather than continuous, there is no NG mode at all. And if the symmetry appears in conjunction with a gauge freedom encoding a long-ranged interaction, the NG mode may couple to the gauge field and develop a gap (this is called the Anderson–Higgs mechanism and will be addressed in Section 7.3). The original theorem also required Lorentz invariance, but non-relativistic versions have been derived later, which we shall address in Section 3.2.

The requirement that some translational invariance remains in the broken-symmetry state is the same as the one we needed to prove Noether's theorem in Section 1.2.3. We again need translational invariance only on a coarse-grained level, so that momentum is a good quantum number, and modes will have a definite value of momentum. We can then define a complete set of eigenstates of the Hamiltonian, $|n, \mathbf{k}\rangle$, labelled by their momentum $\mathbf{k}$ and energy $E_n(\mathbf{k})$, with $n$ encoding all relevant quantum numbers other than momentum. These states are orthogonal, $\langle n', \mathbf{k}'|n, \mathbf{k}\rangle = (2\pi)^D \delta_{nn'} \delta(\mathbf{k} - \mathbf{k}')$, and can be used to write a resolution of the identity:

$$\mathbb{I} = \sum_n \int \frac{\mathrm{d}^D k}{(2\pi)^D} \, |n, \mathbf{k}\rangle \langle n, \mathbf{k}|. \tag{3.2}$$

We can insert this into the definition of a broken-symmetry state of Eq. (2.30), in terms of the

interpolating field:

$$\langle\psi|[Q,\Phi]|\psi\rangle = \sum_n \int \frac{d^D k}{(2\pi)^D}\Big(\langle\psi|Q(t)|n,\mathbf{k}\rangle\langle n,\mathbf{k}|\Phi|\psi\rangle - \text{c.c.}\Big)$$
$$= \int_\Omega d^D x \sum_n \int \frac{d^D k}{(2\pi)^D}\Big(\langle\psi|j^t(x,t)|n,\mathbf{k}\rangle\langle n,\mathbf{k}|\Phi|\psi\rangle - \text{c.c.}\Big) \neq 0. \quad (3.3)$$

Here, c.c. indicates the complex conjugate, and in the second line the global conserved charge $Q = \int dx\, j^t$ is written as an integral over the Noether charge density. Goldstone's theorem addresses the modes as $\mathbf{k}$ approaches zero, but it is not concerned with the tower of states at precisely $\mathbf{k} = 0$. It is thus related to the behaviour of the Noether charge density integrated over a large, but finite part of space[9]. The primary assumption in the derivation of Goldstone's theorem then, is that we can take the integration volume $\Omega$ in the expression above to be large but finite. Because the interpolating field $\Phi$ is local, any contributions to the expectation value from outside the volume $\Omega$ are guaranteed to vanish in relativistic theories by causality. In non-relativistic, or effective, theories, it vanishes as long as the theory does not contain any long-ranged interactions. That is, all interactions should decay sufficiently quickly with distance.

With this caveat in mind, we can again use translational invariance, and write:

$$\langle[Q,\Phi]\rangle = \int_\Omega d^D x \sum_n \int \frac{d^D k}{(2\pi)^D}\Big(\langle\psi|e^{-\frac{i}{\hbar}(Ht-\mathbf{P}\cdot\mathbf{x})}j^t(0,0)e^{\frac{i}{\hbar}(Ht-\mathbf{P}\cdot\mathbf{x})}|n,\mathbf{k}\rangle\langle n,\mathbf{k}|\Phi|\psi\rangle - \text{c.c.}\Big)$$
$$= \int_\Omega d^D x \sum_n \int \frac{d^D k}{(2\pi)^D}\Big(e^{\frac{i}{\hbar}(E_n t-\mathbf{k}\cdot\mathbf{x})}\langle\psi|j^t(0,0)|n,\mathbf{k}\rangle\langle n,\mathbf{k}|\Phi|\psi\rangle - \text{c.c}\Big)$$
$$= \sum_n \int d^D k\, \delta_\Omega(\mathbf{k})\Big(e^{\frac{i}{\hbar}E_n t}\langle\psi|j^t(0,0)|n,\mathbf{k}\rangle\langle n,\mathbf{k}|\Phi|\psi\rangle - \text{c.c.}\Big) \neq 0. \quad (3.4)$$

In the first line, the local Noether charge $j^t(\mathbf{x},t)$ was translated in time and space using the shift operators $e^{-iHt/\hbar}$ and $e^{-i\mathbf{P}\cdot\mathbf{x}/\hbar}$. In the second line we set $E_n$ to be the energy of the state $|n,\mathbf{k}\rangle$ relative to that of the state $|\psi\rangle$, and we invoked translational invariance to see that $|\psi\rangle$ is a zero-momentum state. In the final line we defined $(2\pi)^D\delta_\Omega(\mathbf{k}) \equiv \int_\Omega d^D \mathbf{x}\exp(i\mathbf{k}\cdot\mathbf{x})$ to be a strongly peaked function tending towards a Dirac delta function in the limit of large integration volume. Because $|\psi\rangle$ is assumed to be a broken-symmetry state, the order parameter cannot be zero. This implies there should be at least one state $|n,\mathbf{k}\rangle$ such that the integrand in the final line also does not vanish, even for large $\Omega$, when only contributions with momentum $\mathbf{k}$ tending to zero can contribute. This is the first part of the theorem: there must exist some state near zero momentum that is excited from the broken-symmetry state by both the local Noether charge $j^t(0,0)$ and the interpolating field $\Phi$.

Noether's theorem guarantees the global charge $Q$ to be time-independent. If $\Phi$ also does not depend on time, then in the thermodynamic limit where the broken state $|\psi\rangle$ is an energy eigenstate, the entire order parameter $\langle[Q,\Phi]\rangle$ is time-independent. For the right-hand side of Eq. (3.4) we then find:

$$\partial_t\langle[Q,\Phi]\rangle = \partial_t \sum_n \int \frac{d^D k}{(2\pi)^D}\delta_\Omega(\mathbf{k})\Big(e^{iE_n t}\langle\psi|j^t(0,0)|n,\mathbf{k}\rangle\langle n,\mathbf{k}|\Phi|\psi\rangle - \text{c.c.}\Big)$$
$$= \sum_n \int \frac{d^D k}{(2\pi)^D}\delta_\Omega(\mathbf{k})iE_n\Big(e^{iE_n t}\langle\psi|j^t(0,0)|n,\mathbf{k}\rangle\langle n,\mathbf{k}|\Phi|\psi\rangle - \text{c.c.}\Big) = 0. \quad (3.5)$$

---

[9]Formally, we should consider both the limit of the volume $V$ of our system tending to infinity, and that of the integration volume $\Omega$ tending to $V$. Taking $V \to \infty$ before taking $\Omega \to V$ then guarantees that the point $k = 0$ is excluded from any momentum integrals appearing in this section. This singular limit is discussed in detail in Ref. [35]

We already found that there must be at least one state, the NG mode, for which the term between brackets does not vanish. The final line then implies that the NG mode must have vanishing energy, $E_n(\mathbf{k}) \to 0$ as $\mathbf{k} \to 0$. This completes the proof of Goldstone's theorem: a system with a spontaneously broken symmetry has at least one excitation whose energy vanishes as its wave number approaches zero.

Notice that Goldstone's theorem is *constructive*, in the sense that it not only tells us that gapless modes exist whenever a symmetry is spontaneously broken, but also indicates how to find these modes. They can be excited from the symmetry-broken state by acting on it with either the local Noether charge operator, or the interpolating field.

## 3.2 Counting of NG modes

The derivation of Goldstone's theorem may at first sight seem to suggest that there is always one NG mode for each broken symmetry generator. This cannot be the case, however, since the Heisenberg ferromagnet is known to have only a single NG mode, while two spin-rotation symmetries are broken. Similarly, one may be tempted the assume that the energy of NG modes always vanishes linearly in momentum, $E_n \propto k$. For relativistic systems, this certainly is the case, since Lorentz symmetry dictates that time and space derivatives appear on equal footing in the action. However, in non-relativistic systems the Heisenberg ferromagnet again provides a counterexample to the general rule. Its single NG mode is quadratic in momentum, rather than linear.

Goldstone's theorem as we derived it above, actually just states there is at least one NG mode whenever any symmetry is broken, and it does not specify its dispersion relation other than that it is gapless, so there is no real contradiction with the observed properties of ferromagnets. How many NG modes should really be expected in any given system, and what replaces the seemingly intuitive rule of one mode per broken symmetry, was cleared up only recently. It cannot yet be found in any of the standard text books, but is readily accessible through either the original literature in Refs [36–43], or in the short review of Ref. [44].

In the derivation of Goldstone's theorem, we found that NG modes can be excited from the broken-symmetry state by either the generator of a broken symmetry, or the interpolating field. A special case then arises if the interpolating field $\Phi$ is itself also a generator of a broken symmetry. A clear example is again the Heisenberg ferromagnet, in which one of the spin-rotation operators, say $S^z$, obtains a non-zero expectation value. The commutator of the broken generators $S^x$ and $S^y$ is proportional to $S^z$, and can thus be used as an order parameter. The broken generators in this case act as interpolating fields for each other, and Eq. (3.4) shows that they must excite the *same* NG mode [38].

More generally, take any two symmetry generators $Q_{a,b} = \int_{\mathbf{x}} j_{a,b}^t(\mathbf{x})$, and consider the commutator expectation value

$$
\begin{aligned}
\langle [Q_a, j_b^t(\mathbf{x})] \rangle &= \int d^D y \, \langle [j_a^t(\mathbf{y}), j_b^t(\mathbf{x})] \rangle = \int d^D y \, \delta(\mathbf{x} - \mathbf{y}) \sum_c i f_{abc} \langle j_c^t(\mathbf{y}) \rangle \\
&= \sum_c i f_{abc} \langle j_c^t(\mathbf{x}) \rangle = \langle [j_a^t(\mathbf{x}), Q_b] \rangle .
\end{aligned}
\tag{3.6}
$$

If this commutator has non-zero expectation value in the broken-symmetry state, they are again seen to excite the same NG mode. After Watanabe and Murayama we call such NG modes *type-B*, while 'ordinary' NG modes are said to be *type-A* [41]. **Nambu–Goldstone mode – type-B**

From Eq. (3.6), it is clear that type-B modes cannot arise for Abelian symmetry groups, in which all generators commute with one another. To systematically count the number of NG modes of either type, we should construct the Watanabe–Brauner matrix [40] **Watanabe–Brauner matrix**

$$
M_{ab} = -i \langle \psi | [Q_a, j_b^t(\mathbf{x})] | \psi \rangle .
\tag{3.7}
$$

Here $a$ and $b$ label all broken symmetry generators, and $|\psi\rangle$ is the SSB state. The total number of broken generators is equal to the dimension of the quotient space $G/H$. Notice that since the broken-symmetry state is assumed to be translationally invariant, the matrix elements do not depend on the position $\mathbf{x}$. The numbers $n_A$ and $n_B$ of type-A and type-B Nambu–Goldstone modes are now given by:

$$n_A = \dim G/H - \operatorname{rank} M, \qquad\qquad n_B = \frac{1}{2}\operatorname{rank} M. \qquad (3.8)$$

The two independent proofs [41, 42] also show that (in almost all cases) type-A NG modes have linear dispersion while type-B modes have quadratic dispersion. This can be understood using the low-energy effective Lagrangian method [39, 41, 43, 45]. A detailed derivation is beyond the scope of these notes, but in short, one writes down the most general Lagrangian allowed by the symmetry of the problem, in terms of fields $\pi_a(x)$ which take values in the space of broken symmetry generators, the quotient space $G/H$. The number of these fields then equals the number of broken symmetry generators, but the fields are not necessarily independent. The gapless modes in the spectrum of this low-energy effective Lagrangian will correspond to the NG modes. The lowest order terms are

**Lagrangian – effective**

$$\mathcal{L}_{\text{eff}} = m_{ab}(\pi_a \partial_t \pi_b - \pi_b \partial_t \pi_a) + \bar{g}_{ab}\partial_t \pi_a \partial_t \pi_b - g_{ab}\nabla\pi_a \cdot \nabla\pi_b. \qquad (3.9)$$

Here $m_{ab}$, $\bar{g}_{ab}$ and $g_{ab}$ are coefficients, some of which are constrained by symmetry. For example, there are no terms linear in gradients, since we assume space to be isotropic. For the same reason, it is clear that the first term in $\mathcal{L}_{\text{eff}}$ breaks Lorentz invariance, and can only be non-zero in non-relativistic systems. Watanabe and Murayama have shown that the coefficients $m_{ab}$ are given precisely by the corresponding elements of $M_{ab}$ in Eq. (3.7) above [41].

---

**Exercise 3.1 (Number of type-B NG modes)** Part of the proof of Eq. (3.8) in Ref. [41] is the following. The matrix $M_{ab}$ is real and antisymmetric. Then there exists an orthogonal transformation $O$ such that $\tilde{M} = OMO^{\mathrm{T}}$ takes the form

$$\tilde{M} = \begin{pmatrix} M_1 & & & & & \\ & \ddots & & & & \\ & & M_m & & & \\ & & & 0 & & \\ & & & & \ddots & \\ & & & & & 0 \end{pmatrix}, \qquad M_i = \begin{pmatrix} 0 & \lambda_i \\ -\lambda_i & 0 \end{pmatrix}, \qquad (3.10)$$

where $m = \frac{1}{2}\operatorname{rank} M$ and the $\lambda_1, \ldots, \lambda_m$ are real and non-zero. To prove this:

**a.** Show that the eigenvalues of $M$ are purely imaginary. This means that the matrix $E$ with the imaginary eigenvalues on the diagonal can be obtained by some unitary transformation $E = UMU^\dagger$.

**b.** Show that the non-zero eigenvalues come in conjugate pairs $i\lambda_i, -i\lambda_i$. Since there are rank $M$ non-zero eigenvalues, this implies rank $M$ is even, so $n_B$ in Eq. (3.8) is integer.

**c.** For each $2 \times 2$-submatrix $e_i$ with a conjugate pair on the diagonal elements, find a unitary matrix $w_i$ such that $w_i e_i w_i^\dagger = M_i$.

You have found that $M$ is unitarily equivalent to $\tilde{M}$ by the unitary transformation $WU$ where $W$ has the submatrices $w_i$ on the top-left diagonal and other entries 0. Since $M$

> and $\tilde{M}$ are both real they are then also orthogonally equivalent, and the orthogonal matrix $O$ can be constructed from $WU$. See Problem 160 in Ref. [46].

From the effective Lagrangian, we can find the equations of motion for the fields and their dispersion relations. In systems where $m_{ab}$ is zero, including all relativistic systems, the effective Lagrangian describes modes with linear dispersions. More precisely, Fourier transforming the Lagrangian will introduce a frequency $\omega$ for every time derivative and a momentum $k$ for every gradient, so the dispersion will obey $\omega^2 \propto k^2$. If the coefficients $m_{ab}$ are not zero, their contribution to the Lagrangian will always dominate the second order derivatives at sufficiently low energies, and the dispersion will be quadratic, $\omega \propto k^2$. Notice that terms like $\pi_a \partial_t \pi_a$ (no summation over $a$) are total derivatives in the Lagrangian, and vanish in the action. Therefore, $m_{ab}$ must be antisymmetric, and the first term in Eq. (3.9) can only be non-zero in systems where two fields are coupled. The reduction in the number of gapless NG modes because two generators excite the same mode, and the fact that type-B modes have quadratic dispersion, are thus seen to go hand-in-hand.

Finally, notice that in the effective Lagrangian, it is possible for the coefficients $g_{ab}$ to be zero. In that case, higher-order terms must be taken into account, and it is therefore possible that type-A modes have quadratic dispersion, or rather $\omega^2 \propto k^4$. This is the case, for example, for so-called Tkachenko modes in vortex lattices in rotating superfluids [43].

### 3.3 Examples of NG modes

To see how the formal considerations of Sections 3.1 and 3.2 impact the observable properties of real materials, we will present a short selection of practical examples. This list is far from exhaustive, but should give you a feeling for the extent to which the Goldstone's theorem shapes the physics of all systems subject to spontaneous symmetry breaking.

**Superfluid** The superfluid was argued in Section 2.5.2 to be described by a complex scalar field theory, in which the field operator itself acts as the order parameter. The action is invariant under rotations of the phase of the field, and Noether's theorem shows this symmetry to be associated with the conservation of particle number. In the superfluid phase, the $U(1)$ phase-symmetry is spontaneously broken and the number of particles in the superfluid condensate is indeterminate. There is one broken symmetry generator and one NG mode, which may be excited by finite-wave-number rotations of the phase variable in the complex scalar field. The NG mode is type-A and its dispersion is linear in momentum. The supercurrent (a particle current that flows without viscosity) is a direct manifestation of this NG mode.

**Crystal** Crystals in $D$ spatial dimensions break the symmetries of space, $D$ translations and $\frac{1}{2}D(D-1)$ rotations. The translation group is Abelian, so the associated NG modes are all type-A, with linear dispersions. They are called phonons, or sound waves, and there is one for each direction of space. The rigidity due to breaking of translational symmetry is shear rigidity, whose non-zero value is the traditionally used quantity for distinguishing solids from liquids.

The broken rotational symmetries in the crystal do not lead to any additional NG modes. As was shown only recently [47–49], rotations and translations are not independent symmetry operations. Consequently, the NG fields excited by broken translations and broken rotations are also not independent, and contain redundant degrees of freedom. The broken rotations do therefore not lead to independent NG modes. Intuitively, this means that if you try to excite a rotational NG mode by applying torque stress to a crystal, you instead end up exciting transverse sound modes. In fact, Lorentz boosts are also spontaneously broken in the crystal (or any other medium), but like rotations, they do not lead to independent NG modes [50].

**Nambu–Goldstone mode – redundant**

**Antiferromagnet** The Heisenberg antiferromagnet of Section 2.4 breaks two out of three spin-rotational symmetries, say $S^x$ and $S^y$. The commutator in the off-diagonal elements of the Watanabe-Brauner matrix is the magnetisation $S^z$, which vanishes. The NG modes excited by the broken symmetry generator are thus independent, so there are two type-A NG modes with linear dispersions, called spin waves. These can be viewed as plane waves of precessions for the spins on each sublattice.

**Ferromagnet** The Heisenberg ferromagnet of Exercise 2.7 breaks the same spin-rotation symmetries as the antiferromagnet. This time, however, the magnetisation $S^z$ is an order parameter whose expectation value does not vanish in the broken-symmetry state. The Watanabe-Brauner matrix is therefore non-zero and the modes excited by the two broken symmetry generators are not independent. There is then one type-B NG mode with a quadratic dispersion.

**Canted antiferromagnet** Adding a term that favours orthogonal alignment of neighbouring spins to the antiferromagnetic Heisenberg Hamiltonian will lead to a symmetry-broken state in which all spins uniformly cant away from the preferred direction in the Néel state. The result is a state with a total uniform magnetisation as well as a staggered or sublattice magnetisation in a perpendicular direction. This state breaks all three spin-rotation symmetries. In this case, the broken generator of rotations around the axis of uniform magnetisation will excite one type-A NG mode, while the remaining two broken generators excite one type-B NG mode.

---

**Exercise 3.2 (Chiral symmetry breaking)** Consider a complex scalar doublet $\Phi = \begin{pmatrix} \phi_1 & \phi_2 \end{pmatrix}^{\mathrm{T}}$ where $\phi_1$ and $\phi_2$ are complex scalar fields, with Lagrangian:

$$\mathcal{L} = \frac{1}{2}(\partial_\mu \Phi^\dagger)(\partial^\mu \Phi) - \frac{1}{2}r\Phi^\dagger\Phi - \frac{1}{4}u(\Phi^\dagger\Phi)^2. \tag{3.11}$$

**a.** Show that this Lagrangian is invariant under $\Phi \to L\Phi$ where $L \in SU(2)$ is a unitary $2 \times 2$ matrix with determinant 1.

**b.** Show that this Lagrangian is furthermore invariant under

$$\phi_1 \to r_2\phi_1 + r_1\phi_2^*, \qquad\qquad \phi_2 \to -r_1\phi_1^* + r_2\phi^2, \tag{3.12}$$

with $r_1^*r_1 + r_2^*r_2 = 1$.

If we write $\Phi$ as a $U(2)$ matrix $\check{\Phi}$ and collect $r_1$, $r_2$ in an $SU(2)$-matrix $R$:

$$\check{\Phi} = \begin{pmatrix} \phi_2^* & \phi_1 \\ -\phi_1^* & \phi_2 \end{pmatrix}, \qquad\qquad R = \begin{pmatrix} r_2 & -r_1 \\ r_1^* & r_2^* \end{pmatrix}, \tag{3.13}$$

then $\Phi^\dagger\Phi = \frac{1}{2}\mathrm{Tr}\,\check{\Phi}^\dagger\check{\Phi}$, and the Lagrangian is invariant under:

$$\check{\Phi} \to L\check{\Phi}R^\dagger. \tag{3.14}$$

This is called *chiral symmetry*. The Lagrangian is also invariant under global U(1) phase rotations $\Phi \to e^{i\alpha}\Phi$, and the full symmetry group is $SU(2)_{\mathrm{L}} \times SU(2)_{\mathrm{R}} \times U(1)$, but we will disregard the $U(1)$ symmetry here.

The groups $SU(2)_{\mathrm{L}}$ and $SU(2)_{\mathrm{R}}$ are generated by $Q_a^{\mathrm{L}}$ and $Q_a^{\mathrm{R}}$, which satisfy the $SU(2)$-relations $[Q_a^E, Q_b^F] = i\epsilon_{abc}\delta_{EF}Q_c^E$, $E, F = \mathrm{L, R}$ (see Exercise 1.7). Define the *vector* and *axial charges* $Q_a^{\mathrm{V}} = Q_a^{\mathrm{L}} + Q_a^{\mathrm{R}}$ and $Q_a^{\mathrm{A}} = Q_a^{\mathrm{L}} - Q_a^{\mathrm{R}}$.

**chiral symmetry**

**c.** Show that these satisfy the algebra relations:

$$[Q_a^{\mathrm{V}}, Q_b^{\mathrm{V}}] = \mathrm{i}\epsilon_{abc} Q_c^{\mathrm{V}}, \tag{3.15}$$

$$[Q_a^{\mathrm{A}}, Q_b^{\mathrm{A}}] = \mathrm{i}\epsilon_{abc} Q_c^{\mathrm{V}}, \tag{3.16}$$

$$[Q_a^{\mathrm{V}}, Q_b^{\mathrm{A}}] = \mathrm{i}\epsilon_{abc} Q_c^{\mathrm{A}}. \tag{3.17}$$

This implies the $Q_a^{\mathrm{V}}$ generate a subgroup but the $Q_a^{\mathrm{A}}$ do not.

If $r < 0$, for $u > 0$, the potential has a minimum at $\langle \Phi \rangle = \begin{pmatrix} 0 & v \end{pmatrix}^{\mathrm{T}}$ with $v \in \mathbb{R}$, or $\langle \check{\Phi} \rangle = \mathrm{diag}(v, v)$ (see Section 5.2).

**d.** Show that the vector transformations ($L = R$ in Eq. (3.14)) leave the expectation value $\langle \check{\Phi} \rangle$ invariant, while the axial transformations ($L = R^\dagger$ in Eq. (3.14)) do not.

The symmetry is spontaneously broken by $\langle \check{\Phi} \rangle$ from $SU(2)_{\mathrm{L}} \times SU(2)_{\mathrm{R}}$ to the diagonal subgroup $SU(2)_{\mathrm{L+R}}$ generated by the vector charges $Q_a^{\mathrm{V}}$. Since there are three broken generators $Q_a^{\mathrm{A}}$ we expect three type-A NG modes. The Lagrangian expressed in terms of $\check{\Phi}$, with the symmetry of Eq. (3.14) describes the Higgs field in the Standard Model of elementary particles, before coupling to other fields. The reason that these NG bosons are not massless particles in the Standard Model, will be explained in Section 7.3.

    $SU(3) \times SU(3) \to SU(3)$ chiral symmetry breaking occurs in quantum chromodynamics (QCD), in the limit where quark masses go to zero.

### 3.3.1 NG-like excitations

There are several systems which harbour excitations that are clearly related to the physics of spontaneous symmetry breaking and NG modes, but that do not satisfy all of the assumptions underlying Goldstone's theorem. Again, we give a short selection of examples to give you a feeling for how NG-like excitations extend into systems that strictly speaking fall just outside the realm of spontaneous symmetry breaking.

**Gapped NG mode** If a system with a spontaneously broken symmetry is exposed to an external field that explicitly breaks the same symmetry, there is an NG-like excitation with an energy gap proportional to the external field. This has been dubbed a *gapped* or *massive* NG mode [51,52]. The typical example is that of spin waves in a ferromagnet exposed to a magnetic field parallel to the magnetisation. This situation may be interpreted as a model without explicit symmetry breaking, at the cost of having modified, time-dependent symmetry generators [53].

**Pseudo NG mode** If a symmetry is broken explicitly, due to a weak coupling to other fields (or other degrees of freedom) rather than by an external field, there is a bosonic particle with an energy gap, which otherwise has all the characteristics of a NG mode. This is now called a *pseudo NG mode* or *pseudo NG boson*. The most famous example are the lightest eight pseudoscalar mesons in the Standard Model, pseudoscalar particles whose mass is small because the approximate $SU(3) \times SU(3)$ chiral symmetry is broken spontaneously (see Exercise 3.2).

**Quasi NG mode** In some cases, the ground state of a system may have a larger symmetry group than the Hamiltonian itself, *and* that symmetry may be broken spontaneously. In that case, an NG-like excitation emerges, which is now called a *quasi NG mode* [54], although

confusingly these same modes used to be called a pseudo NG boson [55]. This occurs in particle physics, where the charged pions obtain part of their mass in this way, as well as in certain technicolour and supersymmetry models. In condensed matter physics quasi NG modes are found in helium-3 superfluids and spinor Bose-Einstein condensates.

**Goldstino** The broken generators $Q_a$ are creation operators for NG modes when acting on the broken-symmetry state. Since they generally obey some set of commutation relations, the NG modes are bosons. Usually they are scalar particles, although sometimes it makes sense to assign a vector or tensor structure to several NG modes. However, if the symmetry generators happen to satisfy *anti*-commutation relations, the NG modes are fermions. This is the case for *supersymmetry*, which is a possible extension of the Poincaré algebra of spacetime symmetries with fermionic supersymmetry generators $Q_a$. If supersymmetry is spontaneously broken, the associated fermionic NG modes are called Goldstinos.

**Goldstino**

**supersymmetry**

### 3.4 Gapped partner modes

Type-B Nambu–Goldstone modes may be excited from the broken-symmetry states by two distinct broken symmetry generators, whose commutator has a non-zero expectation value. In the effective Lagrangian description of Eq. (3.9), this was signalled by two fields $\pi_1$ and $\pi_2$ not being independent. Even though the modes are not independent, they do originate from two distinct symmetry transformations, and one might therefore wonder whether there should not be two degrees of freedom or modes associated with the two fields in the Lagrangian. In fact, there generally are two modes, but the second mode is gapped [56–59]. To see this, consider the effective Lagrangian for a system with two coupled broken symmetry generators, and nothing else [60]:

**gapped partner mode**

$$\mathcal{L}_{\text{eff}} = 2M(\pi_1\partial_t\pi_2 - \pi_2\partial_t\pi_1) + \frac{1}{c^2}(\partial_t\pi_1)^2 + \frac{1}{c^2}(\partial_t\pi_2)^2 - \frac{1}{2}(\nabla\pi_1)^2 - \frac{1}{2}(\nabla\pi_2)^2. \qquad (3.18)$$

The dispersion relations for the two modes described by this Lagrangian are:

$$\omega_{\pm} = \sqrt{c^2k^2 + M^2c^4} \pm Mc^2. \qquad (3.19)$$

That is, there is one gapless NG mode with dispersion $\omega_- = \frac{k^2}{2M} + \ldots$, and one gapped partner mode with dispersion $\omega_+ = 2Mc^2 + \frac{k^2}{2M} + \ldots$. The coefficients $2M$ and $c$ can be interpreted as an effective mass and velocity.

In the limit $M \to 0$, the two modes decouple and we obtain a degenerate, linear dispersion $\omega_{\pm} = ck$. This corresponds to having two type-A NG modes, which is indeed consistent with $M$ being the expectation value of the off-diagonal element in the Watanabe-Brauner matrix and going to zero in this limit. In the opposite limit of $c \to \infty$, the gap goes to infinity and we are effectively left with only a single, gapless mode. This corresponds to having only terms with single time derivatives in Eq. (3.18), which is the case for example for the Heisenberg ferromagnet, and indeed there is no gapped mode in the spectrum of the ferromagnet. Physically, the fact that the second mode disappears altogether for the ferromagnet can be understood by realising that its NG modes are always excited by a lowering of the maximally polarised spins. That is, in both $S^x = (S^+ + S^-)/2$ and $S^y = (S^+ - S^-)/2i$ only the $S^-$ part actually excites a mode. Acting with $S^+$ on a maximally polarised ferromagnet annihilates the state, and does not correspond to a physical excitation, recall Exercise 2.7. In this limit, the action of the two broken symmetry generators on the symmetry-breaking state are thus entirely equivalent, and there really is only a single excitation associated with them. More generally, however, non-zero values for both $M$ and $c$ always indicate the presence of both a type-B NG mode and an accompanying gapped partner mode.

Notice that at first sight, the existence of a gapped mode excited by a broken symmetry generator seems to invalidate the proof of Goldstone's theorem in Section 3.1, in which we argued that all modes excited by broken generator must be gapless to ensure that the order parameter would be time-independent. Although there is no general proof of which we are aware, it has been checked in several cases that this paradox is resolved by observing that the terms in Eq. (3.5) always contain a product of two matrix elements, of the form:

$$E_n \langle \psi | j^t(0,0) | n, \mathbf{k} \rangle \langle n, \mathbf{k} | \Phi | \psi \rangle . \tag{3.20}$$

In all verified cases where $j^t(0,0)$ excites a gapped partner with non-zero energy $E_n$, it turns out that $\langle \psi | \Phi | n, \mathbf{k} \rangle$ is proportional to the energy of the accompanying gapless mode, which vanishes for $\mathbf{k} \to 0$. The existence of the gapped partner mode therefore does not contradict the time-independence of the order parameter[10].

---

**Exercise 3.3 (Heisenberg Ferrimagnet)** A *ferrimagnet* can be described by the Heisenberg Hamiltonian $H = J \sum_{\langle i,j \rangle} \mathbf{S}_i \cdot \mathbf{S}_j$ on a square lattice with positive coupling $J$, but for a system in which the spins on the A- and B-sublattices have different sizes.      **ferrimagnet**

**a.** Let the spins on the *A*-sites have spin-$S_A$ and those on the *B*-sites have spin-$S_B$. Calculate the average magnetisation per unit cell $\langle S_{\text{tot}}^z \rangle / (N/2)$ for the Néel-like state in which all spins are in eigenstates of their $S_j^z$ operators with eigenvalues $m_A = S_A$ and $m_B = -S_B$.
**b.** This state breaks the spin-rotation symmetries generated by $S^x$ and $S^y$. Calculate the matrix elements of the Watanabe-Brauner matrix defined in Eq. (3.7) (take the average per unit cell).
**c.** The local magnetisation $S_i^z$ is an order parameter operator for this state, but it turns out that there is also a second order parameter: calculate the expectation value per unit cell for the staggered magnetisation operator $N_i^z = (-1)^i S_i^z$.

We thus find that the breaking of spin-rotational symmetry in the ferrimagnet may be described by (at least) two distinct order parameters, one of which commutes with the Hamiltonian. Clearly then, it is not sufficient to find an order parameter operator that does not commute with the Hamiltonian to claim that any symmetry-breaking system is of type-A. One should calculate all the elements of the Watanabe-Brauner matrix, as you did in this exercise.

With a bit more effort, you can see that the ferrimagnet has one quadratically dispersing NG mode and one gapped partner mode, whose gap scales with the difference in spin sizes, $\Delta \propto |S_A - S_B|$. If the spins on the two sublattices have the same size, the system is an ordinary antiferromagnet with two gapless, linearly dispersing modes (as described by Eq. (3.19) in the limit $M \to 0$). In the opposite limit, as $\Delta$ grows, exciting the gapped partner mode costs ever more energy, and the spectrum looks more and more like that of a ferromagnet with only a single gapless, quadratically dispersing, NG mode.

---

### 3.4.1 The tower of states for systems with type-B NG modes

NG modes may be seen as the $k > 0$ cousins of the collective excitations with zero wave number that make up the tower of states in systems with a spontaneously broken symmetry. The close relation between the internal and collective modes persists in the distinction between systems with type-A and those with type-B modes. We have seen that the Heisenberg antiferromagnet and the harmonic crystal, for example, have unique ground states and a tower of low-energy

---

[10]We thank Tomas Brauner for discussions concerning this point.

Table 3.1: Properties associated with different types of Nambu–Goldstone modes. The accurate forms of the dispersion relations can be found in Eqs. 3.19.

|  | NG mode dispersion | ground state degeneracy | tower of states |
|---|---|---|---|
| type-A | $\propto k$ | no | yes |
| type-B ferro | $\propto k^2$ | yes, $\mathcal{O}(N)$ | no |
| type-B ferri | $\propto k^2$, $M + k^2$ | yes, $\mathcal{O}(N)$ | no |

states, all of which are unstable. This is general for systems with type-A NG modes. We have also seen in Exercise 2.7 that the ferromagnet has a macroscopically degenerate ground state and no tower of states. The number of exact ground states is infinite in the thermodynamic limit, but of order $N$ for finite systems, with $N$ the number of particles.

As it turns out, in systems with a type-B NG mode and a gapped partner mode (which have both $M$ and $c$ non-zero in Eq. (3.18)) there is a ground state degeneracy, and still no tower of states. While we are not aware of a general proof, the relation to the collective modes can be seen in the explicit example of an antiferromagnet on the Lieb lattice[11]. The Hamiltonian is the same as that of the usual Heisenberg antiferromagnet, Eq. (2.17), but on the Lieb lattice there are twice as many A sites as there are B sites, as shown in Fig. 3.1. The classical ground state is a Néel-type state with spins pointing antiparallel on the two sublattices. It has non-zero magnetisation $\langle S^z \rangle = S(N_A - N_B) = \frac{1}{3}SN$, with $N_{A,B}$ denoting the number of sites on the $A$ and $B$ sublattices, and $N$ the combined total number of sites. Because the magnetisation is finite, the NG modes of the Lieb-lattice antiferromagnet will be type-B. Just like the usual square-lattice antiferromagnet discussed in Section 2.4, any exact ground states of the full model have finite overlap with those of a corresponding Lieb–Mattis model (the $k = 0$-part of the Hamiltonian, with infinte-range interactions). Using that fact, the ground states can be shown to have total spin $S_{\text{tot}} = S(N_A - N_B)$, and degeneracy $2S_{\text{tot}} + 1$, which is of order $N$. The energy required to excite any of the remaining $k = 0$ states is of order $\mathcal{O}(J)$, in contrast to the usual antiferromagnet with type-A NG modes, where $k = 0$ excitations have energy vanishing as $\mathcal{O}(J/N)$. In other words, there is no tower of states. The relation between types of NG modes and the spectrum of collective excitations is summarised in Table 3.1.

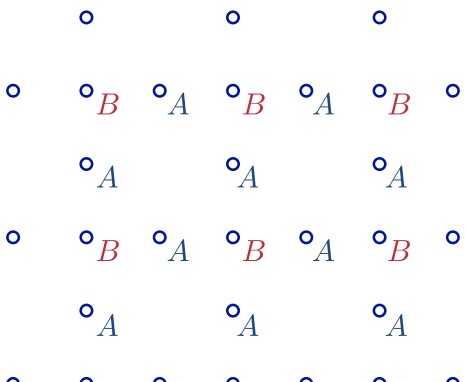

Figure 3.1: The Lieb lattice. Each A site has only B neighbours and vice versa, but there are twice as many A sites as there are B sites.

---

[11]We thank Hal Tasaki for pointing out this example, which is discussed also in his textbook [61]

# 4 Quantum corrections and thermal fluctuations

The eigenstates of the local order parameter operator, which are the classically expected symmetry-broken states, are not in general eigenstates of their corresponding symmetric Hamiltonian. Except for systems with conserved order parameters and type-B NG modes, such as ferromagnets and ferrimagnets, symmetric Hamiltonians have unique, symmetric ground states. That the 'classical states' nevertheless typically bear a large resemblance to the states actually realised in nature, is owing to their stability, in the sense of Section 2.7.

Despite their particular stability, however, the classical states are not exactly the states found in actual quantum materials, due to the somewhat subtle effect of excitations with finite wave number reducing the perfect local order preferred by the $k = 0$, collective part of the Hamiltonian in the thermodynamic limit. The differences between the classically expected and actually encountered state at zero temperature are known as *quantum corrections*. At non-zero temperature, these may be supplemented by *thermal fluctuations*, which further suppress the local order parameter.

**quantum corrections**

Notice the distinction made between thermal *fluctuations* and quantum *corrections*. The latter are often referred to as "quantum fluctuations" in the literature. The reasoning being that the effect of quantum corrections in reducing the order parameter amplitude is quite similar to that of the thermal fluctuations. Indeed, they may even lead to a *quantum phase transition* at zero temperature which is in some ways analogous to the usual, thermally induced, phase transition (see also the discussion in Section A.4). We believe this terminology is misleading. Thermal fluctuations describe actual small, random fluctuations within a thermal state due to, for example, Brownian motion. At zero temperature, on the other hand, nothing ever fluctuates. The ground state may be unique all the way from a maximally ordered phase to just before a quantum phase transition, despite the expectation value of the local order parameter strongly decreasing. For any given value of system parameters, nothing about the ground state evolves or fluctuates in time. We therefore prefer the term *quantum corrections* to denote the difference between the actual quantum ground state and the perfectly ordered, classical state.

Rather than describing the generic properties of quantum corrections to a general ordered state, we will study the specific example the Heisenberg antiferromagnet in Section 4.1. This allows us both to introduce some of the commonly encountered mathematical techniques in studying the actual ground states of ordered systems, and to show in detail a realistic description of a system in which quantum corrections are of significant magnitude. Indeed, in most ordered states that we encounter in daily life, quantum corrections to the classical state are "utterly negligible" [11]. For example, a chair or table, even at low temperatures, is extremely well-described by its classical state with infinitely well-defined position.

In stark contrast to these everyday objects, stand systems in low dimensions, for which quantum corrections and thermal fluctuations are generically so large that they altogether prevent the formation of any ordered state and the spontaneous breaking of continuous symmetries. A heuristic derivation of the Mermin–Wagner–Hohenberg–Coleman theorem which explains both the effect of dimensionality, and the link between quantum corrections and thermal fluctuations, is given in Section 4.2.

## 4.1 Linear spin-wave theory

There is no known exact expression for the broken-symmetry state realised by the Heisenberg antiferromagnet in the thermodynamic limit, in dimensions higher than one. One thing we can do to describe it, however, is to start from the classical state (the eigenstate of the Néel order parameter operator), and use the variational principle to look for deviations that lower the energy expectation value. This is done in a systematic wave in *spin-wave theory*, which is

based on a reformulation of the Heisenberg Hamiltonian in terms of boson operators acting on the classical state, so that each bosonic excitation lowers the order parameter expectation value. In *linear* spin-wave theory, the bosonic Hamiltonian is additionally linearised, allowing for a ground state to be identified by direct diagonalisation. We can thus find the exact ground state of an approximate Hamiltonian, and trust that it can serve as an approximate ground state to the exact Hamiltonian. As the name suggests, the linear spin-wave theory considers only the first non-trivial order in a systematic expansion of the order parameter. For Heisenberg spin-$S$ antiferromagnets on bipartite lattices with $z$ nearest neighbours, it turns out that the results obtained this way are good beyond expectation, even for $S = \frac{1}{2}$ and $z = 4$, where neither $1/S$ nor $1/z$ are expected to be very good expansion parameters.

The Hamiltonian for the spin-$S$ Heisenberg antiferromagnet on a bipartite lattice in $d$ dimensions, is given by Eq. 2.17:

$$H = J \sum_{\langle i,j \rangle} \left( S_i^x S_j^x + S_i^y S_j^y + S_i^z S_j^z \right) = \frac{J}{2} \sum_{j\delta} \left( S_j^x S_{j+\delta}^x + S_j^y S_{j+\delta}^y + S_j^z S_{j+\delta}^z \right). \tag{4.1}$$

Here, the coupling constant $J$ is positive, so that neighbouring spins prefer to anti-align. The lattice vectors $\delta$ run over the $z$ connections of any site to all of its nearest neighbours, and the factor $\frac{1}{2}$ is included to avoid double counting. Because the lattice is bipartite, it can be divided into $A$- and $B$-sublattices. For the square lattice, we saw this before in Figure 2.4. Anticipating antiferromagnetic Néel order, we expect the spins on the $A$-sublattice to have a positive magnetisation in the broken-symmetry state, and spins on the $B$-sublattice to have negative magnetisation. To avoid the inconvenience of having to keep track of the local magnetisation direction on each sublattice, we introduce rotated spin operators $\mathbf{N}_j$, defined as:

$$N_{j\in A}^a = S_j^a; \qquad N_{j\in B}^x = S_j^x, \qquad N_{j\in B}^y = -S_j^y, \qquad N_{j\in B}^z = -S_j^z. \tag{4.2}$$

That is, the coordinate system for spins on the $B$ sublattice is rotated over an angle of $\pi$ around the $x$-axis with respect to the coordinate system used for spins on the $A$-sublattice. The new spin operators $\mathbf{N}_j$ are still proper spin-$S$ operators, obeying the commutation relations associated with the $SU(2)$ algebra, $[N_i^a, N_j^b] = \delta_{ij} i\epsilon_{abc} N_j^c$.[12]

The Heisenberg Hamiltonian expressed in terms of the rotated spins is:

$$H = \frac{J}{2} \sum_{j\delta} \left( \tfrac{1}{2} N_j^+ N_{j+\delta}^+ + \tfrac{1}{2} N_j^- N_{j+\delta}^- - N_i^z N_{j+\delta}^z \right). \tag{4.3}$$

In writing this, we made use of the fact that on bipartite lattices, sites on the $A$-sublattice always have nearest neighbours on the $B$-sublattice, and the other way around. The operators $N_j^\pm = N_j^x \pm i N_j^y$ are the raising and lowering operators for transformed spins. The classical Néel state is an eigenstate of $N_{\text{tot}}^z = \sum_k N_j^z$, with maximal eigenvalue. This is clearly not an eigenstate of the Hamiltonian, due to the first two terms in Eq. (4.3).

Starting from the fully developed Néel state, local excitations are made by applying the spin-lowering operator $N_j^-$. These lower the value of the local staggered magnetisation by the same quantised amount each time they act, and are therefore similar in their effect to the ladder operators of a harmonic oscillator, or more generally, to boson creation operators. This similarity can be made exact by formally expressing the transformed spin operators in terms

---

[12]Note that in Section 2.5.2 and Exercise 3.3, we introduced the order parameter operators $N_i^a = (-1)^i S_i^a$. While that former definition is simpler to write, for spin wave theory it is important that the $N_i^a$ operators satisfy the standard $SU(2)$ commutation relations, and therefore we define them here according to Eq. (4.2).

of boson operators $a_j$ and $a_j^\dagger$:

$$N_j^+ = \sqrt{2S}\sqrt{1 - \frac{1}{2S}n_j}\, a_j, \tag{4.4}$$

$$N_j^- = \sqrt{2S}a_j^\dagger\sqrt{1 - \frac{1}{2S}n_j}, \tag{4.5}$$

$$N_j^z = S - a_j^\dagger a_j. \tag{4.6}$$

The boson operators obey canonical commutation relations $[a_i, a_j^\dagger] = \delta_{ij}$ and $n_j = a_j^\dagger a_j$. The square root in these expressions is defined by its power series expansion. It can be checked that the definition of $\mathbf{N}_j$ in terms of bosons still respects the $SU(2)$ algebra of the spin operators. The bosons introduced in this way of writing spin operators are known as *Holstein–Primakoff bosons*, and we should define the Néel state to correspond to the bosonic vacuum, $a_j|\text{Néel}\rangle = 0$, so that it has the correct eigenvalue of $S$ for all of the $N_j^z$ operators. States with non-zero numbers of bosons correspond to states with non-maximal values of the Néel order parameter.

**Holstein–Primakoff transformation**

The Holstein–Primakoff transformation is exact, but the square roots in its definition prevent a simple diagonalisation of the Heisenberg Hamiltonian written in terms of Holstein–Primakoff bosons. We therefore make a linear approximation of the square roots, keeping only the first terms in their power series expansion, so that none of the approximate expressions are more than bilinear in $a$ and $a^\dagger$. That is, we approximate the spin raising and lowering operators as $N_j^+ \approx \sqrt{2S}a_j$ and $N_j^- \approx \sqrt{2S}a_j^\dagger$, while the expression for $N_j^z$ remains unaltered. This can really only be expected to be a good approximation for large $S$ and low numbers of boson excitation $\langle a_j^\dagger a_j\rangle$, but the resulting approximate antiferromagnetic ground state and its excitations will turn out to give quite accurate results even for spin-$\frac{1}{2}$ systems.

The linear approximation for the expression of the transformed spin operators yields the approximate form of the Heisenberg Hamiltonian:

$$H \approx \frac{1}{2}JS\sum_{j\delta}\left(a_j a_{j+\delta} + a_j^\dagger a_{j+\delta}^\dagger\right) - \frac{1}{2}zNJS^2 + zJS\sum_j a_j^\dagger a_j. \tag{4.7}$$

Here $N$ is the total number of sites, and $z$ is again the number of nearest neighbours, or *co-ordination number*. To diagonalise the approximate Hamiltonian, we first use the fact that is invariant under lattice translations, by writing it in terms of Fourier transformed operators $a_j = \frac{1}{\sqrt{N}}\sum_k e^{ik\cdot j}a_k$. The momentum-space bosons still obey the canonical commutation relations $[a_k, a_{k'}^\dagger] = \delta_{kk'}$, and the terms appearing in the Hamiltonian become:

$$\sum_j a_j^\dagger a_j = \frac{1}{N}\sum_{jkk'} e^{i(-k\cdot j + k'\cdot j)}a_k^\dagger a_{k'} = \sum_k a_k^\dagger a_k, \tag{4.8}$$

$$\sum_{j\delta} a_j a_{j+\delta} = \frac{1}{N}\sum_{j\delta kk'} e^{i(k\cdot j + k'\cdot j + k'\cdot\delta)}a_k a_{k'} = \sum_{k\delta} e^{-ik\cdot\delta}a_k a_{-k}, \tag{4.9}$$

$$\sum_{j\delta} a_j^\dagger a_{j+\delta}^\dagger = \frac{1}{N}\sum_{j\delta kk'} e^{i(-k\cdot j - k'\cdot j - k'\cdot\delta)}a_k^\dagger a_{k'}^\dagger = \sum_{k\delta} e^{ik\cdot\delta}a_k^\dagger a_{-k}^\dagger. \tag{4.10}$$

Here we used the definition of the delta function $\frac{1}{N}\sum_j e^{ij\cdot(k-k')} = \delta_{k,k'}$. To simplify notation, we introduce $\gamma_k \equiv \frac{1}{z}\sum_\delta e^{ik\cdot\delta}$. Notice that for the square lattice, $\gamma_k$ is real and reduces to a sum over cosines. The Hamiltonian in momentum space now reads:

$$H = -\frac{1}{2}NJzS^2 + JzS\sum_k\left[a_k^\dagger a_k + \frac{1}{2}\gamma_k(a_k a_{-k} + a_k^\dagger a_{-k}^\dagger)\right]. \tag{4.11}$$

The products of two creation operators and two annihilation operators in this expression still hinder a simple identification of the ground state. The appearance of these terms once again indicates that the Néel state, or bosonic vacuum, is not an eigenstate of the Heisenberg Hamiltonian. To find the exact ground state of the linearised Hamiltonian, we perform a so-called *Bogoliubov transformation*, and introduce a second set of boson creation and annihilation operators $b_k^\dagger$ and $b_k$:

**Bogoliubov transformation**

$$a_k = \cosh u_k \; b_k + \sinh u_k \; b_{-k}^\dagger, \qquad a_k^\dagger = \cosh u_k \; b_k^\dagger + \sinh u_k \; b_{-k}. \qquad (4.12)$$

Here, $u_k$ is an unknown but real function of $k$, obeying $u_k = u_{-k}$. The new operators again satisfy the canonical boson commutation relations $[b_k, b_{k'}^\dagger] = \delta_{kk'}$. The approximate Hamiltonian can be written in terms of the new boson operators:

$$H = -\frac{1}{2}JNzS^2 + JzS\sum_k \Big[\sinh^2 u_k + \tfrac{1}{2}\gamma_k \sinh 2u_k + (\cosh 2u_k + \gamma_k \sinh 2u_k)b_k^\dagger b_k$$

$$+ \tfrac{1}{2}(\gamma_k \cosh 2u_k + \sinh 2u_k)(b_k^\dagger b_{-k}^\dagger + b_k b_{-k})\Big]. \qquad (4.13)$$

This expression for the Hamiltonian would be diagonal if the terms in the final line vanish. Since the Bogoliubov transformation was introduced in Eq. (4.12) in terms of an arbitrary function $u_k$, we are free to now consider the particular choice for $u_k$ that renders the off-diagonal terms in the Hamiltonian zero. That is, we choose $u_k$ such that $\gamma_k \cosh 2u_k + \sinh 2u_k$ equals zero. Using the general relation $\cosh^2 x - \sinh^2 x = 1$, this amounts to:

$$\sinh 2u_k = \frac{-\gamma_k}{\sqrt{1-\gamma_k^2}}, \qquad \cosh 2u_k = \frac{1}{\sqrt{1-\gamma_k^2}}. \qquad (4.14)$$

Notice that this expression for $u_k$ is ill-defined at zero wave number, where $\gamma_{k=0} = 1$. The collective, centre-of-mass part of the Hamiltonian at zero wave number corresponds to the tower of states, just as in Eqs. (2.9), (2.20) and (2.46). The diagonalisation of the Hamiltonian using a Bogoliubov transformation here forces us to treat these collective excitations separately from the internal, finite-wave-number excitations corresponding to quantum corrections and NG modes.

Writing the approximate Hamiltonian in terms of the Bogoliubov transformed excitations, diagonalising it by our choice of $u_k$, and omitting the $k = 0$ part, we finally find the *linear spin wave Hamiltonian*:

$$H = -\frac{1}{2}JNzS^2 - \frac{1}{2}JNzS + JzS\sum_k \sqrt{1-\gamma_k^2}(b_k^\dagger b_k + \frac{1}{2})$$

$$= \underbrace{-\frac{1}{2}JNzS^2}_{\text{classical}} \underbrace{-\tfrac{1}{2}JzS\sum_k\Big(1-\sqrt{1-\gamma_k^2}\Big)}_{\text{quantum corrections}} + \underbrace{JzS\sum_k\sqrt{1-\gamma_k^2}\,b_k^\dagger b_k}_{\text{NG modes}}. \qquad (4.15)$$

The Hamiltonian consists of three parts. The first two parts describe the energy expectation value in the ground state, while the final term contains all excitations that can propagate with non-zero wave number, and describes the NG modes. Their energy is positive, so they will be absent in the ground state, which is the vacuum for the $b$ bosons defined by $b_k|0\rangle = 0$. If excited, the NG modes can reduce the expectation value of the local order parameter even further. This happens for example at non-zero temperatures, where the occupation number of the bosonic NG modes follows the Bose–Einstein distribution function.

On a square lattice, the dispersion relation for the NG modes can be approximated at low wave numbers to be:

$$E_k = zJS\sqrt{1-\gamma_k^2} \approx 2\sqrt{D}JSk + \dots, \qquad (4.16)$$

Table 4.1: The ground state energy density for the Heisenberg antiferromagnet on square and cubic lattices with $z = 2D$. Indicated first for general spin, and then for the specific cases of $S = 1/2$ and $S = 1$, are the absolute energy per site as well as the relative energy gain with respect to the classical expectation value $E/JN = -\frac{1}{2}zS^2$.

| $E_0/JN$ | absolute | relative | $S = \frac{1}{2}$ | | $S = 1$ | |
|---|---|---|---|---|---|---|
| $D = 2$ | $-2S(S + 0.1579)$ | $15.8/S$ % | $-0.658$ | $31.6\%$ | $-2.32$ | $15.8\%$ |
| $D = 3$ | $-3S(S + 0.0972)$ | $9.72/S$ % | $-0.896$ | $19.4\%$ | $3.29$ | $9.72\%$ |

where we used $z = 2D$. As expected for type-A NG modes, the dispersion is linear in wave number. Because the antiferromagnet breaks two spin rotations, we should in fact expect to find two type-A NG modes, and it may seem like our linear spin-wave description is missing an entire branch of excitations. In fact, this is a consequence of introducing rotated spin operators in Eq. (4.2). In terms of the rotated spin operators, all spins in the lattice look equivalent, and the unit cell is thus half as large as it was for the original spins. In reciprocal space, this implies a doubling of the Brillouin zone. In the dispersion of Eq. (4.16), half the excitations are thus folded out to higher wave numbers. The dispersion going linearly to zero at $k_x = k_y = \ldots = \pi$, corresponds to the second branch of NG modes, which would be folded back to $k = 0$ if we return to using the same, non-rotated, coordinated system at every site.

Returning to the first two terms in the Hamiltonian of Eq. (4.15), we see that the first part is just the energy expectation value of the classical Néel state. The second part is negative and lowers the energy below that of the classical state. It represents the difference in ground state energy between the exact quantum ground state (of the approximate Hamiltonian) and the classical state, or in other words, it shows the quantum correction to the ground state energy. The ground state energy can be evaluated numerically in the continuum limit, by replacing sums with integrals. The results for square and cubic lattices are listed in Table 4.1. Particularly for spin-$\frac{1}{2}$ antiferromagnets, the quantum corrections to the ground state energy are seen to be substantial.

Like the energy, the expectation value of the order parameter is affected by quantum corrections. To see this, we can start from the expression in Eq. (4.6), of the staggered magnetisation in terms of the original Holstein-Primakoff bosons:

$$\langle 0| \frac{1}{N} \sum_j N_j^z |0\rangle = S - \frac{1}{N} \sum_j \langle 0| a_j^\dagger a_j |0\rangle = S - \frac{1}{N} \sum_k \langle 0| a_k^\dagger a_k |0\rangle. \tag{4.17}$$

The ground state $|0\rangle$ appearing in this equation is the vacuum of the Bogoliobov transformed particles $b_k$, rather than the original Holstein-Primakoff bosons $a_k$. Using the Bogoliubov transformation of Eq. (4.12), and the fact that the ground state does not contain any $b$-excitations, the staggered magnetisation may be written as:

$$\langle 0| \frac{1}{N} \sum_j N_j^z |0\rangle = S - \frac{1}{N} \sum_k \sinh^2 u_k$$

$$\approx S + \frac{1}{2} - \frac{1}{(2\pi)^D} \int_{-\pi}^{\pi} d^D k \frac{1}{2} \frac{1}{\sqrt{1 - \gamma_k^2}}. \tag{4.18}$$

Here we again took the continuum limit in the second line. The integral diverges in one dimension, indicating that the quantum corrections to the order parameter in that case are strong enough to suppress the order altogether. This in fact turns out to be a general phenomenon, to which we return in Section 4.2.

Table 4.2: The order parameter density in the Heisenberg antiferromagnet on square and cubic lattices with $z = 2D$. Indicated first for general spin, and then for the specific cases of $S = 1/2$ and $S = 1$, are the absolute value of the order parameter expectation value, and its relative suppression compared to the classical expectation value $\langle N^z \rangle / N = S$.

| $\langle N^z \rangle / N$ | absolute | relative | $S = \frac{1}{2}$ | | $S = 1$ | |
|---|---|---|---|---|---|---|
| $D = 2$ | $S - 0.1966$ | $19.7/S$ % | 0.303 | 39.3% | 0.803 | 19.7% |
| $D = 3$ | $S - 0.0784$ | $7.8/S$ % | 0.422 | 15.6% | 0.922 | 7.8% |

The integral in the final line of Eq. (4.18) may be evaluated numerically. The results are displayed in Table 4.2, and they show that the quantum corrections to the order parameter take it substantially away from its value in the classical Néel state, especially for low-spin antiferromagnets. The sizes of these quantum corrections in the Heisenberg antiferromagnet are exceptional. In most ordered systems in three dimensions, quantum corrections are tiny.

The linear spin-wave approximation we used here to estimate the effect of quantum corrections turns out to give unexpectedly good results compared to more precise, variational methods. The best estimates for the ground state energy, for example, are within a few percent of the results found here [62].

*XY*-model

**Exercise 4.1 (*XY*-model quantum corrections)** The *XY*-model describes interactions between rotors in a plane, which have a global $U(1)$ rotational symmetry, and which can be written in terms of spin operators as:

$$H_{XY} = J \sum_{\langle ij \rangle} \left( S_i^x S_j^x + S_i^y S_j^y \right) \tag{4.19}$$

We will calculate the quantum corrections for the *XY*-model in $d$ dimensions using linear spin-wave theory [63]. For simplicity, assume $J < 0$.
**a.** We start with a trick: taking a different reference frame, the Hamiltonian can be expressed as $\tilde{H}_{XY} = J \sum_{\langle ij \rangle} \left( S_i^x S_j^x + S_i^z S_j^z \right)$ with respect to a rotated coordinate frame. In $\tilde{H}_{XY}$, write $S_j^x$ in terms of the raising and lowering operators $S_j^\pm$.
**b.** Write the Hamiltonian in terms of Holstein–Primakoff bosons, using:

$$S_j^+ = \sqrt{2S} \sqrt{1 - \frac{1}{2S} n_j}\, a_j \approx \sqrt{2S}\, a_j,$$

$$S_j^- = \sqrt{2S}\, a_j^\dagger \sqrt{1 - \frac{1}{2S} n_j} \approx \sqrt{2S}\, a_j^\dagger,$$

$$S_j^z = S - n_j. \tag{4.20}$$

**c.** First perform a Fourier transformation on the Hamiltonian, and then the Bogoliubov transformation of Eq. (4.12).
**d.** Choose the function $u_k$ such that the Hamiltonian becomes diagonal in the Bogoliubov-transformed operators.
**e.** Numerically evaluate the ground state energy density $E/N$ and the order parameter density $\langle S^z \rangle / N$ in two and three dimensions for general $S$.

The results for the order parameter should be $S - 0.0609$ in two dimensions, and $S - 0.0225$

in three dimensions. The quantum corrections are considerable, but not as large as for the antiferromagnet.

## 4.2 Mermin–Wagner–Hohenberg–Coleman theorem

The lowering of the order parameter expectation value in the broken-symmetry state of the Heisenberg antiferromagnet, as compared to the classical Néel state, is a general property of systems undergoing spontaneous symmetry breaking. In low dimensions ($D = 1$ for the antiferromagnet) quantum corrections may even preclude the existence of a non-zero order parameter altogether. A similar thing happens at elevated temperatures, where thermal fluctuations may prevent spontaneous symmetry breaking in dimension two or lower. This thermal limit to ordering is known as the *Mermin–Wagner–Hohenberg theorem*, while the zero-temperature absence of spontaneous symmetry breaking in one spatial dimension is known in quantum field theory as the *Coleman theorem*.

**Mermin–Wagner–Hohenberg–Coleman theorem**

The calculation showing the divergence of quantum corrections in the Heisenberg antiferromagnet cannot be neatly generalised to apply to all systems with spontaneous symmetry breaking. We will therefore consider the effective Lagrangian of Eq. (3.18), which may be considered a course-grained description of a symmetry-breaking system with NG modes, but no other low-energy excitations. We will find that considering systems with either type-A or type-B NG modes turns out to have significant implications for the way in which quantum corrections affect the broken symmetry. The highest spatial dimension in which quantum corrections or thermal excitations prevent the onset of long-range order, is called the *lower critical dimension*. In the remainder of this section, we will work out the lower critical dimensions for systems with various types of NG modes, both at zero and non-zero temperature. The analysis will require the use of imaginary-time path integrals, and may skipped by readers not interested in the technical analysis. The results are summarised in Table 4.3.

**lower critical dimension**

Table 4.3: The effect of different types of Nambu–Goldstone modes on the emergence of long-range order. The accurate forms of the dispersion relations can be found in Eqs. 3.19. Indicated for each case are the presence or absence of ground state degeneracy and of a tower of states at $k = 0$, whether or not the ground state is affected by quantum corrections, and the lower critical dimensions for both zero and non-zero temperature.

| | NG mode dispersion | ground state degeneracy | tower of states | quantum corrections | lower critical dimension | |
| --- | --- | --- | --- | --- | --- | --- |
| | | | | | $T = 0$ | $T > 0$ |
| type-A | $\propto k$ | no | yes | yes | 1 | 2 |
| type-B ferro | $\propto k^2$ | yes, $\mathcal{O}(N)$ | no | no | 0 | 2 |
| type-B ferri | $\propto k^2, M + k^2$ | yes, $\mathcal{O}(N)$ | no | yes | 0 | 2 |

### 4.2.1 The variance of the order parameter

The spontaneous breakdown of symmetry, and the associated emergence of long-range order, is described in general by some local order parameter operator $\mathcal{O}$, obtaining a non-zero expectation value. For the order to survive the effect of quantum corrections, and that of thermal fluctuations, the variance of the order parameter should be smaller than its expectation value in the symmetry-breaking ground state, or in thermal equilibrium. We thus consider the

variance of the local order parameter, defined as:

$$
\begin{aligned}
\langle \mathcal{O}(\mathbf{x},t)^2 \rangle - \langle \mathcal{O}(\mathbf{x},t) \rangle^2 &= \lim_{\mathbf{x}'\to\mathbf{x},t\to t'} \langle \mathcal{O}(\mathbf{x},t)\mathcal{O}(\mathbf{x}',t') \rangle - \langle \mathcal{O}(\mathbf{x},t) \rangle \langle \mathcal{O}(\mathbf{x}',t') \rangle \\
&= \lim_{\mathbf{x}'\to\mathbf{x},t'\to t} \langle \delta\mathcal{O}(\mathbf{x},t)\delta\mathcal{O}(\mathbf{x}',t') \rangle.
\end{aligned}
\tag{4.21}
$$

In the second line, we expanded the order parameter around its expectation value as $\mathcal{O}(\mathbf{x},t) = \langle \mathcal{O}(\mathbf{x},t) \rangle + \delta\mathcal{O}(\mathbf{x},t)$, and used the fact that the average of the Gaussian fluctuations vanishes, so that $\langle \delta\mathcal{O}(\mathbf{x},t) \rangle = 0$. At very low temperatures or energies, the gapless NG modes $\pi_a(\mathbf{x},t)$ dominate the fluctuations of the order parameter:

$$
\lim_{\mathbf{x}'\to\mathbf{x},t'\to t} \langle \delta\mathcal{O}(\mathbf{x},t)\delta\mathcal{O}(\mathbf{x}',t') \rangle = \lim_{\mathbf{x}'\to\mathbf{x},t'\to t} \sum_a \langle \pi_a(\mathbf{x},t)\pi_a(\mathbf{x}',t') \rangle + \dots
\tag{4.22}
$$

A more precise expression of the variance in terms of NG modes can be found in [43], but this approximate form suffices to understand their role in establishing the lower critical dimensions. Notice that $\langle \pi_a(\mathbf{x},t)\pi_a(\mathbf{x}',t') \rangle$ precisely coincides with the definition for the real-space *propagator* of the NG mode.

Because we are interested in long-ranged ordered symmetry-breaking systems, with some form of translational invariance, it will be most convenient to calculate the propagator in momentum space. Taking a Fourier transformation, it becomes:

$$
\begin{aligned}
G(0) &\equiv \lim_{\mathbf{x}'\to\mathbf{x},t'\to t} \sum_a \langle \pi_a(\mathbf{x},t)\pi_a(\mathbf{x}',t') \rangle \\
&= \lim_{\mathbf{x}'\to\mathbf{x},t'\to t} \int \frac{\mathrm{d}\omega}{2\pi} \int \frac{\mathrm{d}^D k}{(2\pi)^D} \, e^{i\mathbf{k}\cdot(\mathbf{x}-\mathbf{x}')-i\omega(t-t')} \sum_a \langle \pi_a(-\mathbf{k},-\omega)\pi_a(\mathbf{k},\omega) \rangle \\
&= \int \frac{\mathrm{d}\omega}{2\pi} \int \frac{\mathrm{d}^D k}{(2\pi)^D} \sum_a \langle \pi_a(-\mathbf{k},-\omega)\pi_a(\mathbf{k},\omega) \rangle.
\end{aligned}
\tag{4.23}
$$

To obtain an expression for the thermal average from this ground state expectation value, we can apply the common trick of analytic continuation to imaginary time ($t \to -i\tau$), and introduce bosonic Matsubara frequencies $\omega \to i\omega_n$ [64–66]. Here $\omega_n$ may take the values $\frac{2\pi}{\hbar\beta}n$, with $n$ integer and $\beta = 1/k_\mathrm{B}T$. This leads to an expression for the NG mode propagator of the form:

$$
G(0) = \frac{1}{\beta} \sum_n \int \frac{\mathrm{d}^D k}{(2\pi)^D} \sum_a \langle \pi_a(-\mathbf{k},-i\omega_n)\pi_a(\mathbf{k},i\omega_n) \rangle.
\tag{4.24}
$$

**imaginary time**

**Matsubara frequency**

This expression now applies to non-zero temperatures, while the zero-temperature result is recovered in the limit $\beta \to \infty$. The Matsubara frequencies in the arguments of the fields are written explicitly as $i\omega_n$, to emphasise that they are purely imaginary. We will explicitly carry out the sum over Matsubara frequencies [65]. The dependence of the remaining integral over momentum $\mathbf{k}$ on the number of spatial dimensions $D$ will then determine the lower critical dimension.

### 4.2.2 Matsubara summation

Here, we give a brief and incomplete introduction to the general technique of Matsubara summation, before applying it to the specific calculation of how type-A and type-B NG modes affect the variance of a local order parameter. The objective will be to analytically evaluate the summation in Eq. 4.24, which is of the form $G(0) = \frac{1}{\beta}\sum_n g(i\omega_n)$ both for systems with type-A NG modes, and for those with type-B modes.

Using a more-or-less standard trick [65, 67], the sum over Matsubara frequencies can be changed to a sum over the poles of $g$. The first step is to replace the function $g(i\omega_n)$ by the

function $g(z)$, which has the exact same functional form, but in which the purely imaginary Matsubara frequencies are replaced by a general complex variable $z$. Having done this *analytic continuation*, we can use the *Cauchy residue theorem* to replace the sum by a contour integral. In general, this theorem relates a contour integral in the complex plane by a sum over the poles enclosed within the contour:

$$\oint_{\mathcal{C}} f(z) = 2\pi i \sum_n \text{Res } f(z_n). \tag{4.25}$$

Here, $z_j$ are the complex coordinates of poles of the function $f(z)$, and Res indicates the residue at those poles. For simple poles, of order one, the residue is Res $f(z_n) = \lim_{z \to z_n} ((z - z_n) f(z))$. The trick for evaluating Matsubara summations is now to notice that the function $F(z) = (e^{\hbar\beta z} - 1)^{-1}$ happens to have poles of order one, precisely at the coordinates $z_n = 2i\pi n/\hbar\beta$, which coincide with the values $i\omega_n$ of the Matsubara frequencies in our sum. This means that we can write the entire set of terms $g(i\omega_n)$ that we need to sum over as the residues of a suitably chosen function in the complex plane:

$$\text{Res}_{z=z_n} (g(z)F(z)) = \lim_{z \to z_n} \left( (z - z_n) \frac{g(z)}{e^{\hbar\beta z} - 1} \right) = \frac{1}{\hbar\beta} g(z_n). \tag{4.26}$$

Putting together Eqs. (4.25) and (4.26), we can now write the Matsubara summation as a contour integral:

$$\frac{1}{\hbar\beta} \sum_n g(i\omega_n) = \frac{1}{\hbar\beta} \sum_n g(z_n) \qquad \text{analytic continuation}$$

$$= \sum_n \text{Res}_{z=z_n} \frac{g(z)}{e^{\hbar\beta z} - 1} \qquad \text{the trick}$$

$$= \sum_n \frac{1}{2\pi i} \oint_{\mathcal{C}_n} \frac{g(z)}{e^{\hbar\beta z} - 1}. \qquad \text{residue theorem} \tag{4.27}$$

The contours $\mathcal{C}_n$ in the final line are small loops in the complex plane, tightly enclosing the poles at $z_n$ of the function $F(z)$ (see Fig. 4.1, middle figure). The function $g(z)$ however, may also have poles of itself. For sake of clarity, consider an example in which $g(z)$ has two poles on the real axis, at $z_j = \pm 1$. We can then use the fact that contour integrals in the complex plane may be freely reshaped as long as no poles are crossed by the deforming contour. This, and the fact that any circular contour integral with infinite radius must vanish (such as the one in Fig. 4.1, left figure), allows us to equate the contour integral over poles on the imaginary axis to another contour integral over poles on the real axis (Fig. 4.1, right figure).

Using the shift of the integration contour, the Matsubara summation may now be written as:

$$\frac{1}{\hbar\beta} \sum_n g(i\omega_n) = \sum_n \frac{1}{2\pi i} \oint_{\mathcal{C}_n} \frac{g(z)}{e^{\hbar\beta z} - 1}$$

$$= -\sum_j \frac{1}{2\pi i} \oint_{\mathcal{C}_j} \frac{g(z)}{e^{\hbar\beta z} - 1} \qquad \text{shift contour}$$

$$= -\sum_j \text{Res}_{z=z_j} \frac{g(z)}{e^{\hbar\beta z} - 1} \qquad \text{residue theorem}$$

$$= -\sum_j \frac{1}{e^{\hbar\beta z_j} - 1} \lim_{z \to z_j} (z - z_j) g(z). \tag{4.28}$$

The equation in the final line contains a sum over just a small number of poles $z_j$ of the function $g(z)$, and is much easier to evaluate than the original expression involving a sum over infinitely many Matsubara frequencies.

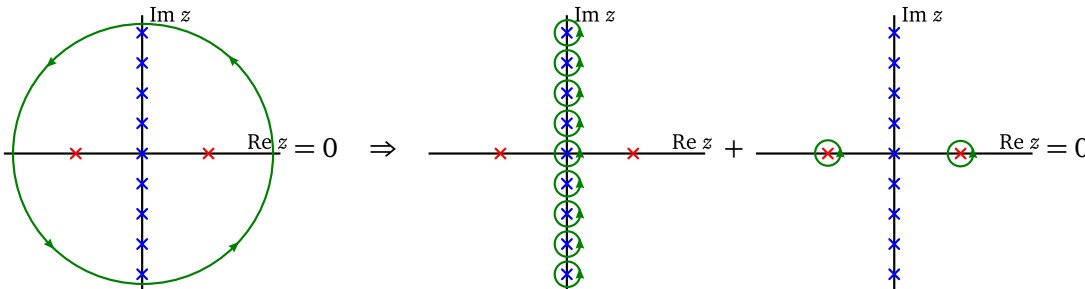

Figure 4.1: The contour integrals used in evaluating the sum over Matsubara frequencies, for a propagator with two poles on the real axis. The integral over the contour at infinity on the left hand side vanishes because normalisability requires all fields to decay sufficiently fast as they approach infinity. The contour may be deformed, however, into a sum over small contours surrounding all of the poles. This implies the equality on the right hand side.

### 4.2.3 General NG mode propagator

The general effective Lagrangian Eq. (3.18) with NG dispersions $\omega_\pm = \sqrt{c^2 k^2 + M^2 c^4} \pm M c^2$ (Eq. (3.19)) has propagator

$$
\begin{aligned}
G(0) &= \int \frac{d\omega}{2\pi} \int \frac{d^D k}{(2\pi)^D} \Big[ \frac{-c^2}{(\omega + \omega_+)(\omega - \omega_-)} + \frac{-c^2}{(\omega - \omega_+)(\omega + \omega_-)} \Big] \\
&\to -c^2 \frac{1}{\hbar\beta} \sum_n \int \frac{d^D k}{(2\pi)^D} \Big[ \frac{1}{(z_n + \omega_+)(z_n - \omega_-)} + \frac{1}{(z_n - \omega_+)(z_n + \omega_-)} \Big] \\
&= c^2 \int \frac{d^D k}{(2\pi)^D} \sum_j \lim_{z \to z_j} \frac{z - z_j}{e^{\hbar\beta z_j} - 1} \Big[ \frac{1}{(z + \omega_+)(z - \omega_-)} + \frac{1}{(z - \omega_+)(z + \omega_-)} \Big]. \quad (4.29)
\end{aligned}
$$

We see that the first term has poles at $z_j = \omega_+$ and $z_j = -\omega_-$ and the second term has poles at $z_j = -\omega_+$ and $z_j = \omega_-$. Evaluating Eq. (4.28) we find

$$
\begin{aligned}
G(0) &= c^2 \int \frac{d^D k}{(2\pi)^D} \frac{1}{\omega_+ + \omega_-} \Big[ \frac{1}{e^{\hbar\beta\omega_+} - 1} - \frac{1}{e^{-\hbar\beta\omega_+} - 1} + \frac{1}{e^{\hbar\beta\omega_-} - 1} - \frac{1}{e^{-\hbar\beta\omega_-} - 1} \Big], \\
&= \int \frac{d^D k}{(2\pi)^D} \frac{c}{\sqrt{k^2 + M^2 c^2}} \Big[ n_B(\hbar\omega_+) + n_B(\hbar\omega_-) + 1 \Big], \quad (4.30)
\end{aligned}
$$

where we have defined the Bose–Einstein distribution $n_B(\varepsilon) = (e^{\beta\varepsilon} - 1)^{-1}$.

### 4.2.4 Type-A NG modes

Applying the general technique of Matsubara summation to the particular problem of evaluating the propagator of NG modes, we start from the effective Lagrangian of Eq. 3.9, which for the case of a single type-A mode can be simplified to:

$$
\mathcal{L}_{\text{eff}} = \frac{1}{c^2} \partial_t \pi \partial_t \pi - \nabla \pi \cdot \nabla \pi. \quad (4.31)
$$

Using Fourier transforms to write the Lagrangian as a function of momentum and Matsubara frequencies, it becomes:

$$
\mathcal{L}_{\text{eff}} = \pi(-\mathbf{k}, -i\omega_n) \Big( \frac{1}{c^2} (i\omega_n)^2 - k^2 \Big) \pi(\mathbf{k}, i\omega_n). \quad (4.32)
$$

As usual, the propagator of the NG modes, which according to Eq. (4.23) corresponds to the variance of the order parameter, is found by inverting the quadratic part of the Lagrangian:

$$\langle \pi(-\mathbf{k}, -i\omega_n)\pi(\mathbf{k}, i\omega_n)\rangle = \frac{c^2}{(i\omega_n)^2 - c^2 k^2}. \tag{4.33}$$

This defines the function $g(i\omega_n)$, which needs to be summed over the bosonic Matsubara frequencies. We can then apply the procedure of Eqs. (4.27) and (4.28), and write the Matsubara summation as a sum over the poles of $g(z)$, which in the present case lie at $z_\pm = \pm ck$:

$$\begin{aligned}
G(0) &= \int d^D k \sum_{j=\pm} \text{Res}_{z=z_j} \frac{1}{e^{\hbar\beta z} - 1} \frac{c^2}{(z - z_+)(z - z_-)} \\
&= \int d^D k \left( \frac{1}{e^{\hbar\beta z_+} - 1} \frac{c^2}{z_+ - z_-} + \frac{1}{e^{\hbar\beta z_-} - 1} \frac{c^2}{z_- - z_+} \right) \\
&= \int d^D k \frac{c}{2k} \left( \frac{1}{e^{\hbar\beta c k} - 1} - \frac{1}{e^{-\hbar\beta c k} - 1} \right) \\
&= \left( \int d\Omega \right) \int dk \, k^{D-2} c \left( n_B(\hbar c k) + \frac{1}{2} \right).
\end{aligned} \tag{4.34}$$

In the final line, $n_B(\varepsilon) = (e^{\beta\varepsilon} - 1)^{-1}$ is the Bose–Einstein distribution , and we introduced spherical coordinates for the momentum integral. The angular part, $\int d\Omega$, evaluates to the surface area of a $D$-dimensional hypersphere, since the integrand only depends on $k = |\mathbf{k}|$.

At zero temperature, or $\beta$ approaching infinity, the contribution from the Bose factor vanishes. We may also avoid any divergence of the integral for high momenta by introducing an upper limit in the integral, corresponding to the inverse lattice spacing or some other inverse length scale set by the microscopic lattice that was ignored in the effective, coarse grained Lagrangian that we started with. Even so, the momentum integral still diverges in dimensions lower than or equal to one, due to the factor $k^{D-2}$. Such a low-momentum divergence is often called an *infrared divergence*. This divergence indicates that the variance of the order parameter is unbounded in dimensions one or lower, and therefore certainly larger than its expectation value. In other words, the quantum corrections due to the presence of type-A NG modes preclude spontaneous symmetry breaking and the formation of long-range order in dimensions one or lower, even at zero temperature. This is known in the quantum field theory literature as the Coleman theorem.

**infrared divergence**

For non-zero temperatures, we can expand the Bose-Einstein distribution for small values of $\hbar\beta c k$, since the dominant contribution to the integral will come from low $k$ values, for any non-zero value of $\beta$. We thus use the expansion:

$$\frac{1}{e^x - 1} = \frac{1}{x} + \frac{1}{2} + \mathcal{O}(x). \tag{4.35}$$

Substituting this into the expression for the variance of the order parameter in Eq. (4.34), we find that the momentum integral is now over a function proportional to $T k^{D-3}$. The presence of thermal fluctuations thus has an even larger effect on the formation of order than the quantum corrections at zero temperature, and the variance of the order parameter diverges in a system with only type-A NG modes in spatial dimensions two or lower for *any* non-zero temperature. This is known as the Mermin–Wagner–Hohenberg theorem.

### 4.2.5 Type-B NG modes

Interestingly, the result for the lower critical dimensions for systems with only type-B NG modes is different from that for systems with only type-A modes. Recall that type-B modes arise

whenever there are terms with a single time derivative in the effective Lagrangian. Type-B systems with and without gapped partner modes behave in the same way as far as the lower critical dimensions are concerned. For definiteness, we consider here the simplest type-B system, without gapped partner modes, which is described by:

$$\mathcal{L}_{\text{eff}} = 2m\left(\pi_1 \partial_t \pi_2 - \pi_2 \partial_t \pi_1\right) - \nabla \pi_a \cdot \nabla \pi_a. \tag{4.36}$$

Notice that there are necessarily two NG fields, $\pi_1$ and $\pi_2$, coupled by the terms with time derivatives. Using Fourier transforms we can again write the Lagrangian as a function of momentum and Matsubara frequencies, and express it in matrix form:

$$\mathcal{L}_{\text{eff}} = \begin{pmatrix} \pi_1(-\mathbf{k}, -i\omega_n) & \pi_2(-\mathbf{k}, -i\omega_n) \end{pmatrix} \begin{pmatrix} -k^2 & -2im(i\omega_n) \\ 2im(i\omega_n) & -k^2 \end{pmatrix} \begin{pmatrix} \pi_1(\mathbf{k}, i\omega_n) \\ \pi_2(\mathbf{k}, i\omega_n) \end{pmatrix}. \tag{4.37}$$

In this case, the sum of propagators for the two NG fields is found by inverting the quadratic part of the Lagrangian and taking the trace over the resulting matrix:

$$g(i\omega_n) = \frac{2k^2}{(2mi\omega_n)^2 - k^4} = \frac{1}{2m}\left(\frac{1}{i\omega_n - \frac{k^2}{2m}} - \frac{1}{i\omega_n + \frac{k^2}{2m}}\right). \tag{4.38}$$

Since both fields contribute to variance of the order parameter, we can directly use this combined expression of $g(i\omega_n)$ in the summation over Matsubara frequencies in Eqs. (4.27) and (4.28). In this case the poles of the analytically continued function $g(z)$ lie at $z_\pm = \pm k^2/2m$:

$$\begin{aligned} G(0) &= \int d^D k \sum_{i=\pm} \text{Res}_{z=z_i} \frac{1}{e^{\hbar\beta z} - 1} \frac{1}{2m}\left(\frac{1}{z - \frac{k^2}{2m}} - \frac{1}{z + \frac{k^2}{2m}}\right) \\ &= \int d^D k \frac{1}{2m}\left(\frac{1}{e^{\hbar\beta \frac{k^2}{2m}} - 1} - \frac{1}{e^{-\hbar\beta \frac{k^2}{2m}} - 1}\right) \\ &= \int d^D k \frac{1}{2m} \coth\left(\tfrac{1}{2}\hbar\beta \frac{k^2}{2m}\right). \end{aligned} \tag{4.39}$$

To evaluate the variance at zero temperature, we need to take the limit $\beta \to \infty$ in the hyperbolic cotangent in the final line. Notice that the momentum $k$ of the NG modes may be arbitrarily small, but not zero, because the exact $k = 0$ modes form the tower of collective states. Also taking the mass to be non-zero, the hyperbolic cotangent then necessarily evaluates to one in the zero-temperature limit. The momentum integral is then over a constant, and has no infrared divergence. In fact, if we again introduce an upper (ultraviolet) cutoff on the integral representing the discreteness of the atomic lattice, the variance is finite in any spatial dimension. Spontaneous symmetry breaking and long-range order may thus occur at zero temperature in any dimension for systems with only type-B NG modes.

At non-zero temperatures, the dominant contribution to the momentum integral will again come from low momenta. Expanding the integrand for small $k$ in this case leads to $\coth(x) = 1/x + \dots$. The thermal population of type-B NG modes then induces a variance of the order parameter according to:

$$\begin{aligned} G(0) &= \int (2\pi)^D k \frac{2}{\hbar\beta k^2} + \dots \\ &= (\int d\Omega) \int dk \, k^{D-3} \frac{2T}{\hbar k_{\text{B}}}. \end{aligned} \tag{4.40}$$

This integral diverges due to the low-$k$ contributions in dimensions two or lower. That is, even though there are no quantum corrections to the order at zero temperature, thermal population of type-B NG modes at non-zero temperatures yields the same lower critical dimension as thermal population of type-A NG modes. Starting from Eq. (4.30) the same result is found for type-B systems with gapped partner modes.

The final classification of lower critical dimensions in systems with only type-A or only type-B NG modes is summarised in Table 4.3 on page 71. As an interesting aside, notice that the same method used above can also be applied to calculate the lower critical dimension when the dispersion relation of type-A NG modes is not linear. For instance, for the Tkachenko modes mentioned in Section 3.2, which are type-A modes with $\omega \propto k^2$, the lower critical dimension will be two at zero temperature and four at any non-zero temperature. Systems with such modes therefore cannot order at any non-zero temperature even in three dimensions [68].

# 5 Phase transitions

So far, we discussed properties of equilibrium phases of matter in which a symmetry of the action or Hamiltonian is spontaneously broken. We have not discussed how such phases are created, or how a symmetry can be broken in practice. That this is a relevant question, is clear from the fact that at infinite temperature, the thermal density matrix for any physical system must be the identity matrix, which is left invariant by all possible symmetry transformation. To find a long-range ordered state at low temperatures, some symmetries must then be broken at a specific *critical temperature*, $T_c$, during the cooling process. That is, there must be a *phase transition* from the symmetric high-temperature state to the symmetry-breaking low-temperature state. The study of phase transitions is a major field in and of itself, and is introduced in detail in several excellent textbooks [20, 69, 70]. Here, we give a brief and limited overview of only some of the central theoretical concepts related to phase transitions, which are most relevant to spontaneous symmetry breaking.

**phase transition**

## 5.1 Classification of phase transitions

Every ordered, symmetry-broken phase has a non-zero expectation value of the order parameter operator. In contrast, the order parameter will be zero in the corresponding symmetric, or disordered, phase. We can therefore distinguish between order and disorder by considering the value of the order parameter. Although the distinction is sharp, in the sense that the order parameter is either zero or not, we can still distinguish between different types of phase transitions by considering the way in which the order parameter goes to zero at the phase transition.

**phase transition – discontinuous**

- In **discontinuous** or **first-order** phase transitions, the order parameter jumps discontinuously from zero to a non-zero value at the critical temperature. Similarly, there is a sudden change in entropy, and the going through the transition requires the release of latent heat. First-order transitions often show hysteresis in the thermal evolution of the order parameter as the system is cycled across the phase transition. Phase transitions between states characterised by the same broken symmetries, such as the gas-to-liquid transition, are almost always first-order. But transitions that do involve the breakdown of a symmetry can also be discontinuous, with the liquid-to-solid transition a famous example.

**phase transition – continuous**

- In **continuous** or **second-order** phase transitions the order parameter increases continuously from zero as the critical temperature is traversed. The entropy also changes continuously. On the other hand, the correlation length and related energy scales diverge at the critical temperature. In fact, at the critical temperature of a second-order phase transition, systems become scale-invariant, in the sense that physical properties no longer depend on the length (or energy) scale at which they are probed. Many symmetry-breaking phase transitions are second-order, with the onsets of superfluidity, (anti)ferromagnetism and many phases of liquid crystals as famous examples.

**scale invariance**

The terminology of first and second order phase transitions stems from the now largely obsolete Ehrenfest classification of phase transitions, in which a transition is said to be of $n$-th order if the $n$-th derivative of the free energy with respect to temperature is discontinuous at the critical temperature. While not yet experimentally accessible at Ehrenfest's time, it has now become clear that for instance the heat capacity in many systems is not just discontinuous, but in fact diverges upon approaching the critical temperature [20], complicating a direct application of Ehrenfest's rules. We therefore prefer a classification based only on the behaviour of the order parameter. This has the additional advantage that the continuous or discontinuous evolution of the order parameter can be directly generalised to *quantum phase transitions* which are phase transitions occurring at zero temperature as function of some other parameter, such as pressure, density or magnetic field.

**Ehrenfest classification**

**quantum phase transition**

To summarise the behaviour of the order parameter as a function of multiple parameters affecting the order (including temperature), it is often useful to draw a *phase diagram*. This is a plot with the externally controllable parameters such as temperature and pressure on each of the axes. Within this parameter space, the different symmetric and symmetry-breaking phases are indicated, while phase transitions are denoted by lines separating the different phases. A line of second-order phase transitions is sometimes called a critical line. A line of first-order transitions can end in a point. If the transition exactly at this point is continuous, then that point is called a critical point. As an example, the phase diagram of helium-4, which contains all these features, is shown in Fig. 5.1.

**phase diagram**

**critical point**

## 5.2 Landau theory

The modern view of phase transitions, also known as *Landau theory*, can be said to have originated with the *equation of state* introduced by Van der Waals to explain the transition from gas to liquid due to attractive interparticle interactions. It posits that below a critical temperature, $T_c$, the pressure of a collection of interacting particles will *decrease* when volume is decreased, in contrast to what we expect for any gas. In this phase transition, no symmetry is broken, as both the ideal gas and the ideal liquid are symmetric under spatial transformations. Nevertheless, the liquid has a preferred density that does not depend on the volume of the container it is in, and we can use this to define an 'order parameter' $o$, which is only non-zero in the liquid phase. It is not an order parameter in the sense of Section 2.5.2, because no symmetries are broken and this parameter does not and cannot take values in any broken-symmetry space $G/H$. The role of the preferred density in the description of the gas-liquid phase transition, however, is the same as that of symmetry-based order parameters in other transitions.

**Landau theory**

### 5.2.1 The Landau functional

In general, equations of state may be expressed in terms of thermodynamic quantities such as pressure, entropy, and so on. These thermodynamic quantities can always be written as derivatives of the free energy. The central idea of Landau theory is that the free energy of any

**free energy**

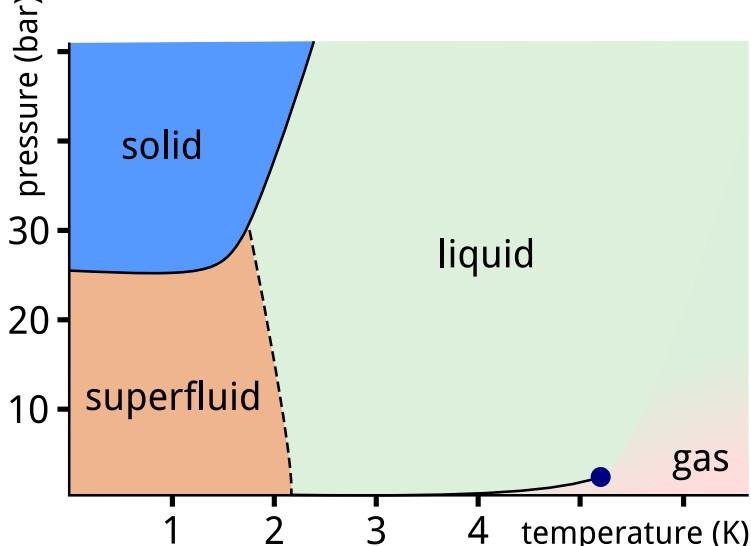

Figure 5.1: Phase diagram of helium-4. The different equilibrium phases as a function of pressure and temperature are marked by different colours. Solid lines represent the first-order phase transitions between liquid and a solid (involving the breaking of a symmetry), and between liquid and gas (without breaking any symmetry). The liquid–gas transition line ends in a critical point (dark blue dot). The scale invariance at this point is signalled for instance by its critical opalescence. Beyond this line it is no longer possible to clearly distinguish between a liquid and a gas, and there is no abrupt transition, but a smooth crossover between the two phases. The dashed line denotes the second-order phase transition from a liquid to a superfluid.

system undergoing a phase transition can be written as a functional of the order parameter, which itself depends on temperature. In a continuous phase transition, the order parameter $o(T)$ is guaranteed to be small close to the critical temperature, and one can carry out a Taylor expansion for small $o(T)$ in that region. Which types of terms appear in such an expansion depend strongly on the nature of the order parameter and the symmetries of the system. In the simplest case of a real-valued scalar field $o$, the Taylor expansion of the Landau functional reads

$$\mathcal{F}_{\mathrm{L}}[o, T] = \mathcal{F}[0, T] + \frac{1}{2}r(T)o^2 + \frac{1}{4}u(T)o^4 + \ldots \tag{5.1}$$

Here we assumed the free energy to be invariant under $o \to -o$, which guarantees that that terms with odd powers of $o$ are absent in the expansion. This symmetry with respect to the order parameter is a commonly occurring property of the free energy, but certainly not a general requirement. Given the free energy expansion, the actual value of the order parameter $o(T)$ realised at any given temperature, can be found by minimizing the free energy (see also Fig. 5.2). Because the expectation value of the observable local order parameter should always be finite, the free energy functional $\mathcal{F}_{\mathrm{L}}[o, T]$ should always be bounded from below. This is guaranteed if the highest-order term in the expansion of Eq. (5.1) has a positive prefactor. In this case, that means we should have $u(T) > 0$. If we somehow determine or measure the value of $u(T)$ and find it to be negative, we should carry out the expansion of the free energy to higher order, continuing until we arrive at a term with positive prefactor.

Assuming for now that $u(T)$ is indeed positive, then if $r(T)$ happens to also be positive, the lowest free energy can be found in the symmetric state, with $o(T) = 0$. However, if $r(T)$

is negative, minimisation of the free energy yields a nonzero order parameter:

$$\frac{\partial \mathcal{F}_{\mathrm{L}}[o,T]}{\partial o} = -|r(T)|o + u(T)o^3 = 0 \;\Rightarrow\; o = \pm\sqrt{\frac{|r(T)|}{u(T)}}. \tag{5.2}$$

A phase transition can now be described as a process in which a variation of temperature leaves $u(T)$ positive, but causes $r(T)$ to change sign. The order parameter then obtains a non-zero value at the temperature for which $r(T)$ goes through zero, which defines the critical temperature. In Fig. 5.2 the free energy is plotted for different values of $r$. The order parameter value where the free energy is minimal is seen to smoothly change from zero to non-zero values as $r$ changes from positive to negative values, and this evolution thus describes a continuous phase transition.

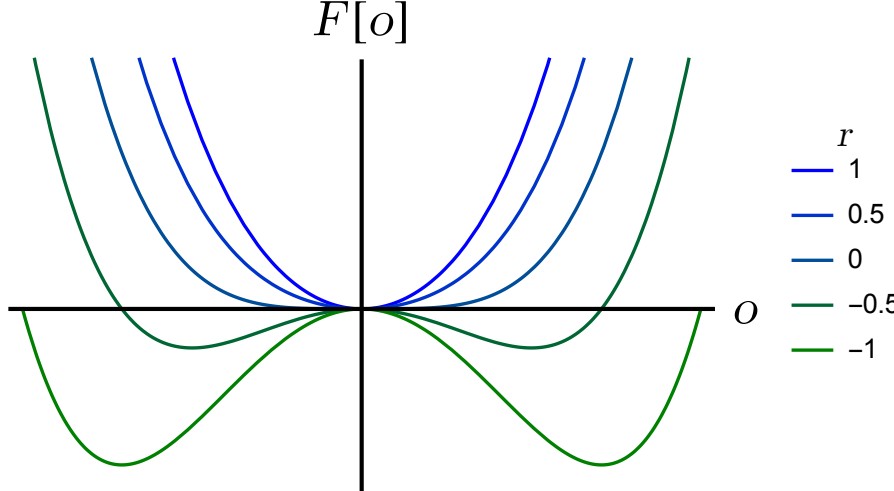

Figure 5.2: The Landau free energy as a functional of the order parameter $o$ during a continuous phase transition described by Eq. (5.2), with $u > 0$ for several values of $r$. The minimum of $\mathcal{F}_{\mathrm{L}}$ moves continuously from zero tot non-zero values as $r$ decreases from positive to negative values.

Close to the critical temperature $T_{\mathrm{c}}$, the function $r(T)$ can be Taylor expanded to first order in $T - T_{\mathrm{c}}$, and will be of the form:

$$r(T) \approx r_0 \frac{T - T_{\mathrm{c}}}{T_{\mathrm{c}}} \equiv r_0 t. \tag{5.3}$$

Here, the *reduced temperature* $t = (T - T_{\mathrm{c}})/T_{\mathrm{c}}$ is a dimensionless quantity that measures the distance from the critical temperature. Furthermore, because $u$ does not change sign, we can Taylor expand it to zeroth order, and assume it to be constant in a small enough region of temperature around $t = 0$. The minimisation of the free energy in Eq. (5.2) then gives an explicit prediction of *how* the order parameter goes to zero close to the critical temperature:

$$o(T) \propto (T - T_{\mathrm{c}})^{\frac{1}{2}}. \tag{5.4}$$

As it turns out, the scale invariance of the system at a continuous phase transition guarantees the behavior of the order parameter close to the critical temperature to always be of the form $o(T) \sim (T - T_{\mathrm{c}})^{\beta}$, where $\beta$ is called a *critical exponent* (not to be confused with the inverse temperature $\beta = 1/k_{\mathrm{B}}T$). Similar critical exponents are associated with the behaviour of the specific heat at the transition, $C_V \sim |T - T_{\mathrm{c}}|^{\alpha}$, the susceptibility, $\chi \sim |T - T_{\mathrm{c}}|^{\gamma}$, and so on. We will come back to these exponents in Section 5.5, where the values of the critical exponents are found to depend solely on the form of the Landau free energy, highlighting the importance of symmetry in the analysis of phase transitions.

### 5.2.2 First-order phase transitions

Contrary to continuous phase transitions, there is no guarantee that an expansion of the free energy in powers of the order parameter makes sense close to a discontinuous phase transition. Nevertheless, Landau theory turns out to describe these types of transitions as well. Consider a situation in which the prefactor $u$ of the quartic term is negative, and the expansion of the free energy is carried out to sixth order:

$$\mathcal{F}_{\mathrm{L}}[o, T] = \mathcal{F}[0, T] + \frac{1}{2}r(T)o^2 + \frac{1}{4}u(T)o^4 + \frac{1}{6}w(T)o^6 + \dots \tag{5.5}$$

Here, we assume $w(T)$ is positive for all temperatures of interest, to ensure that the free energy is bounded from below. For simplicity, consider $r(T) = w(T) = 1$. Then there is a discontinuous jump in the location of the minimum of the free energy from zero to a non-zero value, as $u$ is decreases continuously from $u > -2$ to $u < -2$. This is shown pictorially in Fig. 5.3. Because the order parameter jumps discontinuously from zero to a non-zero value, the phase transition described by these parameters is first order. At discontinuous phase transitions, there is no scale invariance, so that thermodynamic quantities are not expected to have an algebraic form, and there are no critical exponents.

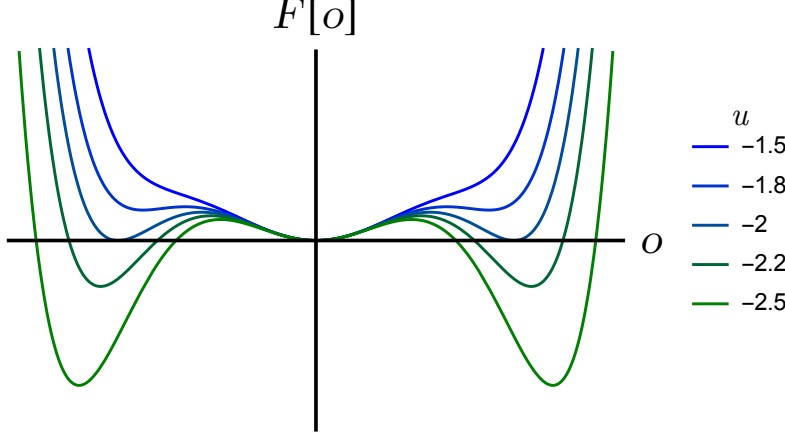

Figure 5.3: The Landau free energy as a functional of the order parameter $o$ during a discontinuous phase transition described by Eq. (5.5), with $r = w = 1$, for several values of $u$. The location of the minimum of $\mathcal{F}_{\mathrm{L}}$ jumps from zero to non-zero values as $u$ crosses the value $-2$.

## 5.3 Symmetry breaking in Landau theory

Landau theory can describe both continuous and discontinuous phase transitions in terms of an 'order parameter', whose value changes from zero to non-zero at the critical temperature. To see how this description of phase transitions is related to spontaneous symmetry breaking, consider the shape of the free energies in Figs. 5.2 and 5.3. At the lowest temperatures, there are two minima with equal energies, related by the transformation $o \rightarrow -o$. That is, the free energy has a discrete $\mathbb{Z}_2$ symmetry, and when the system realises a specific ground state, with either a positive or a negative value for the order parameter, the symmetry is spontaneously broken.

### 5.3.1 The Mexican hat potential

As an example of how a continuous symmetry breaking is manifested within the framework of Landau theory, consider a complex scalar field $\psi(x)$ with a continuous $U(1)$ phase-rotation

symmetry. The ordering transition in which the phase rotation symmetry is broken may be described by the Landau free energy functional:

$$\mathcal{F}_{\mathrm{L}}[\psi, T] = \mathcal{F}[0, T] + \frac{1}{2}r(T)\psi^*\psi + \frac{1}{4}u(T)(\psi^*\psi)^2. \tag{5.6}$$

This expression is invariant under phase rotations of the field $\psi$, since it only depends on powers of its squared amplitude. In analogy to the theory for a real scalar field in Eq. (5.1), we can assume $u$ to be constant close to the critical temperature, and $r(T)$ to be linear function changing sign at $t = 0$. At high temperatures, for $r > 0$, the free energy has a single minimum at $\psi = 0$, while at low temperatures, for $r < 0$ the minimum will be obtained for configurations with non-zero amplitude $|\psi|$. The amplitude of the field is therefore the relevant Landau order parameter. Note that this is similar to our discussion in Section 2.5.2, where we showed that the field operator $\psi$ in a complex scalar field theory obtains an expectation value $\langle\psi\rangle \neq 0$ in the ordered state.

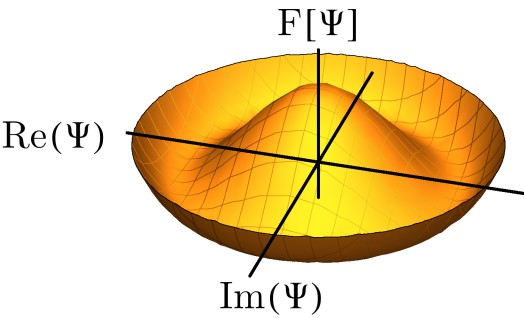

Figure 5.4: Plot of the "Mexican hat" free energy in Eq. (5.6) for $r < 0$.

The typical form of the low-temperature free energy is then depicted in Figure 5.4. This shape of the free energy is colloquially known as the "Mexican hat potential" because of its shape. It is the prototype example in discussions of spontaneous symmetry breaking in the context of Landau theory. There is a continuum of states with amplitude $|\psi| = \sqrt{|r|/u}$ and arbitrary value for the phase, that all minimise the free energy. The free energy is invariant under $U(1)$-rotations, but only single points on the circle of minima are stable. This circle is isomorphic to the quotient space $U(1)/1 \simeq U(1)$ classifying all possible broken-symmetry states. Upon traversing the phase transition, one of the states in the circle, and hence a value for the phase, will be spontaneously chosen.

Having realised a minimum-energy state with a spontaneously chosen order parameter, the free energy in Fig. 5.4 shows that excitations which change the phase of the order parameter but not its amplitude, do not cost any potential energy. The potential, or free energy, along the circle is *flat*, and this is therefore known as a *flat direction* in field space [71]. The Nambu–Goldstone mode associated with $U(1)$-symmetry breaking is precisely a very long wavelength modulation of the phase of the field $\psi$. That is, it is an excitation for which the order parameter oscillates along the flat direction in order parameter space. This result is readily generalised to more complicated instances of symmetry breaking. Given the low-temperature free energy, excitations that oscillate in flat directions correspond to Nambu–Goldstone modes.

**flat direction**

### 5.3.2 From Hamiltonian to Landau functional

The Landau functional $\mathcal{F}_{\mathrm{L}}$ is constructed purely as an expansion in terms of the order parameter, with symmetry dictating both the form of the order parameter and whether or not any given terms are allowed to be non-zero. The values of the coefficients in the expansion are typically determined by fitting predictions for thermodynamic quantities to experimental data, and it may therefore seem like the "macroscopic" Landau theory is entirely phenomenological, and disconnected from the detailed "microscopic" theory defined by a Lagrangian or Hamiltonian. In fact, this is not the case. It is often possible to find both the terms which appear in the Landau functional, and the values of the corresponding coefficients, starting from a microscopic description.

Rather than giving a general and abstract procedure, we will illustrate the connection by considering the specific example of the Heisenberg antiferromagnet on a $D$-dimensional hypercube, defined by the Hamiltonian of Eq. (2.17):

$$H = J \sum_{i,\delta} \mathbf{S}_i \cdot \mathbf{S}_{i+\delta}. \tag{5.7}$$

Here, $i$ runs over all $N$ sites and $\delta$ over the $z = 2d$ connections to nearest neighbours. A good choice for the antiferromagnetic order parameter operator is the staggered magnetisation, $\mathcal{O}_i = (-1)^i \mathbf{S}_i$. We consider a vector of order parameters here to emphasise that there is no preferred axis in the symmetric system. The expectation value of the order parameter operator, $\mathbf{m} = \langle (-1)^i \mathbf{S}_i \rangle$, will be independent of position $i$ in both the disordered and the translationally invariant antiferromagnetic state.

The first step towards formulating a macroscopic description of the antiferromagnet is to rewrite all operators in terms of an average value, called the *mean-field value*, plus deviations. The spin operators $\mathbf{S}_i$, for example, can be written as a sum of the order parameter $\mathbf{m}$ plus deviations $\delta \mathbf{S}_i$:

$$\mathbf{S}_i = (-1)^i \mathbf{m} + \delta \mathbf{S}_i. \tag{5.8}$$

Using this notation, the Heisenberg Hamiltonian becomes:

$$
\begin{aligned}
H &= J \sum_{i\delta} \left( (-1)^i \mathbf{m} + \delta \mathbf{S}_i \right) \cdot \left( (-1)^{i+1} \mathbf{m} + \delta \mathbf{S}_{i+\delta} \right) \tag{5.9} \\
&= -JNz|\mathbf{m}|^2 - 2Jz \sum_i (-1)^i \mathbf{m} \cdot \delta \mathbf{S}_i + J \sum_{i\delta} \delta \mathbf{S}_i \cdot \delta \mathbf{S}_{i+\delta}. \tag{5.10}
\end{aligned}
$$

This form of the Hamiltonian now allows us to make a rigorous approximation. If the system is ordered, the expectation values of the deviations $\delta \mathbf{S}_i$ will be small, which suggests that we may neglect terms in the Hamiltonian that are quadratic or higher order in the deviations. The Hamiltonian resulting from setting these fluctuation terms to zero, is called the *mean-field Hamiltonian*:

**mean-field theory**

$$
\begin{aligned}
H[m] &= -JNz|\mathbf{m}|^2 - 2Jz \sum_i (-1)^i \mathbf{m} \cdot \delta \mathbf{S}_i \\
&= +JNz|\mathbf{m}|^2 - 2Jz \sum_i (-1)^i \mathbf{m} \cdot \mathbf{S}_i. \tag{5.11}
\end{aligned}
$$

In the second line, we reintroduced the original spin operators, using $\delta \mathbf{S}_i = \mathbf{S}_i - (-1)^i \mathbf{m}$. Notice that this results in a sign change in front of the first term.

The mean-field Hamiltonian depends on the order parameter $|\mathbf{m}|$, whose value we do not know. Nevertheless, it is only linear in spin operators and can therefore be solved exactly.

In the present context this means that we can compute the partition function $Z = \text{Tr} e^{-\beta H}$, and from there the thermodynamic free energy $\mathcal{F} = -\frac{1}{\beta} \log Z$. We can find both of these as functions of the unspecified order parameter $\mathbf{m}$.

Since the Hamiltonian is proportional to $\mathbf{m} \cdot \mathbf{S}_j$, the energy eigenstates can be chosen to coincide with those of $S_j^z$, by defining the $z$-axis to be parallel to $\mathbf{m}$. The partition function can then be computed by explicitly summing over all eigenvalues of the operators $S_j^z$. For the case of spin-$\frac{1}{2}$, this yields:

$$
\begin{aligned}
Z &= \sum_{S_1^z = \pm 1/2} \sum_{S_2^z = \pm 1/2} \cdots \sum_{S_N^z = \pm 1/2} e^{-\beta J N z |\mathbf{m}|^2} e^{2Jz\beta|\mathbf{m}|S_1^z} e^{2Jz\beta|\mathbf{m}|S_2^z} \cdots e^{2Jz\beta|\mathbf{m}|S_N^z} \\
&= e^{-\beta J N z |\mathbf{m}|^2} \left(2\cosh J z \beta |\mathbf{m}|\right)^N .
\end{aligned}
\tag{5.12}
$$

Taking the logarithm, the free energy associated with this partition function is:

$$
\mathcal{F} = J N z |\mathbf{m}|^2 - \frac{N}{\beta} \log\left(2\cosh J z \beta |\mathbf{m}|\right) .
\tag{5.13}
$$

As expected, we find an expression for the free energy in terms of the mean-field order parameter. To be able to compare our expression to a Landau functional, we perform a Taylor expansion of $\mathcal{F}$ for small $\mathbf{m}$:

$$
\mathcal{F}/N = -\frac{\log 2}{\beta} + J z (1 - J z \beta/2)|\mathbf{m}|^2 + \frac{1}{12}(Jz)^4 \beta^3 |\mathbf{m}|^4 + \ldots
\tag{5.14}
$$

This has exactly the shape of the Landau free energy of Eq. (5.1).

At high temperatures, or low $\beta$, the coefficients of both the quadratic and quartic terms are positive, and the free energy is minimised for $|\mathbf{m}| = 0$. That is, at high temperatures we do not expect to find an antiferromagnetically ordered state, and the spin-rotational symmetry is unbroken. As the temperature is lowered, the coefficient of the quadratic term decreases, until it goes through zero at $J z \beta = 2$, and becomes negative. At that point, the free energy is minimised by a value for the order parameter that is not zero, indicating that the symmetry is broken, and long-range antiferromagnetic order established.

Although the expansion of the free energy yields precisely the description of the of the antiferromagnetic phase transition we expected, you may object that the analysis is not internally consistent. We started out be defining $\mathbf{m}$ as the ground state expectation value of the order parameter operator, but we ended up claiming to find the value of $\mathbf{m}$ from the minimisation of the free energy, which does not explicitly involve taking any expectation value. As it turns out, however, minimizing the mean-field free energy is *exactly* the same as computing the mean-field expectation value of the order parameter operator.

---

**Exercise 5.1 (Mean-field order parameter)** The expectation value of the order parameter is found self-consistently in any mean-field theory. The consistency condition allows us to derive the mean-field equations in two different ways.

**a.** Show that in a system described by the mean-field Hamiltonian Eq. (5.11), the thermal expectation value of the magnetisation $|\mathbf{m}| = |\langle (-1)^i \mathbf{S}_i \rangle|$ is given by $\frac{1}{2} \tanh J z \beta |\mathbf{m}|$.

**b.** Use the exact expression for the free energy in Eq. (5.13) to compute the magnetisation $\mathbf{m}$ by minimizing $\mathcal{F}$. (Hint: Compute $\frac{\partial \mathcal{F}/N}{\partial |\mathbf{m}|} = 0$.)

The expectation value of the order parameter operator with respect to the mean-field Hamiltonian is thus indeed the same as the equilibrium value with respect to the Landau free energy.

---

One of the great triumphs of theoretical physics is the connection made by Gor'kov, in a similar fashion to what we did above for the Heisenberg antiferromagnet, between the microsopic Bardeen-Cooper-Schrieffer theory of superconductivity and the phenomenological Ginzburg–Landau theory. An accessible derivation can be found in the book by De Gennes [72].

## 5.4 Spatial fluctuations

In all expansions of the free energy that we considered so far, we assumed the local order parameter to always have the same value everywhere in space. This assumption is certainly appropriate when looking for the equilibrium state of a translationally invariant system, be it symmetric or symmetry-breaking. Considering the approach of Landau theory more generally, however, there is no reason not to consider configurations of a system that are described by a spatially varying order parameter. As you might expect, the free energy of such perturbed states may be used to study the role of fluctuations near a phase transition.

 As before, the free energy is expanded in powers of the order parameter, which is assumed to be small. This time however, we simultaneously do an expansion in powers of the spatial derivatives of the order parameter, which are also assumed to be small. That is, we consider a system near a symmetry-breaking phase transition, with smooth or long-wavelength fluctuations. The study of Landau functionals that include spatial derivatives is often called *Ginzburg–Landau theory*, although this term is also used by some to refer exclusively to a theory of superconductivity, including gauge fields, that we will encounter in Chapter 7). **Ginzburg–Landau theory**

 The minimal extension of the Landau theory for second-order phase transitions in Eq. (5.1), including only the lowest possible power of spatial derivatives, is given by:

$$\mathcal{F}_{\mathrm{GL}}[o(\mathbf{x}), T] = \mathcal{F}[0, T] + \int \mathrm{d}^D x \left\{ \frac{c^2}{2} [\nabla o(\mathbf{x})]^2 + \frac{r(T)}{2} [o(\mathbf{x})]^2 + \frac{u(T)}{4} [o(\mathbf{x})]^4 + \dots \right\}. \quad (5.15)$$

This Ginzburg–Landau functional depends on the order parameter *field*, $o(\mathbf{x})$, which may have different values at different positions. Notice that if $o(\mathbf{x})$ is dimensionless, the constant $c^2$ must have units of energy density times length squared.

 One thing Ginzburg–Landau theory can tell us, is what the typical size of fluctuations in the order parameter field will be. To do this, we use the trick of adding a small local perturbation to the potential [73], and seeing what configuration of the order parameter field is established in response. In general, an externally applied potential $\mu(\mathbf{x})$ can be introduced as:

$$\mathcal{F}_\mu[o, T] \;=\; \mathcal{F}_{\mathrm{GL}}[o, T] - \int \mathrm{d}^D x' \mu(\mathbf{x}') o(\mathbf{x}'). \quad (5.16)$$

For now, consider a perturbation that only affects the system at a single location $\mathbf{x}$, so that the potential is a delta function, $\mu(\mathbf{x}') = \mu_0 \delta(\mathbf{x}' - \mathbf{x})$. As usual, the equilibrium configuration of the order parameter field is the one that minimises the free energy. Since the order parameter is now itself a position-dependent function, however, the minimum of the free energy is found by setting the *functional derivative* $\frac{\delta \mathcal{F}}{\delta o(\mathbf{x})} = 0$ to zero[13]. In our example, this results in:

$$-c^2 \nabla^2 o(\mathbf{x}) + r o(\mathbf{x}) + u[o(\mathbf{x})]^3 = \mu_0 \delta(\mathbf{x}). \quad (5.17)$$

Because the perturbation is small, we may assume that the deviations of the order parameter field from its uniform average value $\bar{o} = \langle o \rangle$ are small $o(\mathbf{x}) = \bar{o} + \delta o(\mathbf{x})$. Discarding terms of order $\mathcal{O}\big((\delta o)^2\big)$, then yields:

$$-c^2 \nabla^2 \delta o(\mathbf{x}) + r \bar{o} + r \delta o(\mathbf{x}) + u \bar{o}^3 + 3u \bar{o}^2 \delta o(\mathbf{x}) = \mu_0 \delta(\mathbf{x}). \quad (5.18)$$

The small and local perturbation will not affect the average value of the order parameter, $\bar{o}$, which is therefore equal to the value we found in the uniform Landau theory, Eq. (5.2). For temperatures above the critical temperature, the average order parameter should be zero,

---

[13]A functional derivative acts just like a normal derivative, but with respect to a function instead of a variable. The core relation is $\frac{\delta f(x)}{\delta f(y)} = \delta(x-y)$. This means, for example, that $\frac{\delta}{\delta f(y)} \int \mathrm{d}^d x \, a(x) f(x) = \int \mathrm{d}^d x \, a(x) \delta(x-y) = a(y)$.

while for low temperatures we expect to find $\bar{o} = \sqrt{|r|/u}$. Substituting these averages, the equation for the equilibrium configuration becomes:

$$-c^2\nabla^2\delta o(\mathbf{x}) + r\delta o(\mathbf{x}) = \mu_0\delta(\mathbf{x}) \qquad \text{for } T > T_c,$$
$$-c^2\nabla^2\delta o(\mathbf{x}) - 2r\delta o(\mathbf{x}) = \mu_0\delta(\mathbf{x}) \qquad \text{for } T < T_c. \qquad (5.19)$$

These are ordinary differential equations for the response $\delta o(\mathbf{x})$ to a local perturbation $\mu_0\delta(\mathbf{x})$, which can be straightforwardly solved. In three spatial dimensions, the solution is given by:

$$\delta o(\mathbf{x}) = \frac{\mu_0}{4\pi c^2}\frac{e^{-|\mathbf{x}|/\xi}}{|\mathbf{x}|}, \qquad \begin{cases} \xi = \sqrt{\frac{c^2}{r}} & T > T_c, \\ \xi = \sqrt{\frac{c^2}{-2r}} & T < T_c. \end{cases} \qquad (5.20)$$

The deviation $\delta o(\mathbf{x})$ of the order parameter field from its average value thus falls off exponentially in all directions. It does so with a characteristic length scale, $\xi$, which is called the *coherence length* [14]. This is the length scale over which fluctuations of the order parameter, or in other words, deviations of the magnitude of $o(\mathbf{x})$, persist. It is sometimes referred to as the *healing length*, because the order parameter field returns to its average value $\bar{o}$ within this length scale from an external perturbation. But it is also the typical size of spontaneously generated, thermal, fluctuations. At length scales larger than the coherence length, perturbations and fluctuations have little effect, and the order parameter field is well approximated by it average value. The order and broken symmetry completely determine the way the system looks at those scales. On the other hand, at scales shorter than the coherence length, the local configuration of the order parameter field is dominated by perturbations, and the average order parameter will be hard to distinguish among the microscopic fluctuations. In a way, the long-range order *emerges* from the underlying local physics on length scales larger than the coherence length.

**coherence length**

Notice that coherence length is not the same as the correlation length associated with the two-point correlation function of Eq. (2.33). The former indicates the size of a single fluctuation, while the latter corresponds to the likelihood of two distant regions behaving the same way. In a long-ranged ordered system, the correlation length may be infinitely long, while the coherence length remains finite.

The parameter $r$ in the definition of the coherence length in Eq. (5.20) depends on temperature. In fact, it is the parameter that goes from positive to negative values in the Landau description of a second-order phase transition. As the transition temperature is approached, $r$ must therefore go to zero, and the coherence length will *diverge*. Such divergences turn out to be a general feature of second-order phase transitions, in which not only the coherence length, but all relevant length (and energy) scales diverge as the system advances towards the phase transition. This will be discussed in more detail in Section 5.5.

The divergence of fluctuations as the phase transition is approached poses a problem for the expansion of the free energy in Ginzburg–Landau theory, because it was based on the assumption that variations in the order parameter field are small. As temperature is tuned towards its critical point, there must therefore be a value $t_G \neq 0$ at which the Ginzburg–Landau theory is no longer applicable. Remarkably, this so-called *Ginzburg temperature* can be determined from within Ginzburg–Landau theory itself. To do this, consider the correlation function $\langle o(\mathbf{x})o(0)\rangle$. If the system is long-range ordered, the order parameter should not vary much as a function of position, and the value of the correlation function is close to $\langle o(\mathbf{x})\langle o(0)\rangle$. In other words, the variance of the order parameter should be small, as compared to the value of the order parameter itself:

$$\int d^3x \, \langle o(\mathbf{x})o(0)\rangle - \langle o(\mathbf{x})\rangle\langle o(0)\rangle \ll \int d^3x \, \langle o(\mathbf{x})\rangle^2. \qquad (5.21)$$

---

[14]Eq. (5.19) differs by a factor of $1/\sqrt{2}$ from another definition commonly used in superconductivity [74]

Ginzburg
criterion

The *Ginzburg criterion* now states that whenever this inequality is violated, Ginzburg–Landau theory breaks down. Because we know the size of a single fluctuation is just the coherence length of Eq. (5.20), the integrals in the inequality should be taken from zero to the coherence length. The Ginzburg criterion thus really determines whether a local fluctuation is sufficient to destroy the local order, rather than finding out whether many fluctuations together destroy the global long-ranged order. The latter criterion would instead give the critical temperature.

Notice that the right-hand side of the inequality is easily evaluated, because it just integrates over the constant average value of the order parameter $\langle o \rangle^2 = |r|/u$. To evaluate the left-hand side, we can use an incarnation of the fluctuation-dissipation theorem, which relates the thermal average of fluctuations to the derivatives of the free energy in the presence of perturbations [20, 73]. In the Ginzburg–Landau theory, this relation can be seen simply as a property of the free energy of Eq. (5.16), combined with the definition of the thermal average, $\langle A \rangle = \text{Tr}\, A\, e^{-\beta \mathcal{F}}/Z$, where the trace runs over all possible states of the system:

$$\begin{aligned}
\frac{1}{\beta}\frac{\delta}{\delta\mu(0)}\langle \delta o(\mathbf{x}) \rangle &= \frac{1}{\beta}\frac{\delta}{\delta\mu(0)}\langle o(\mathbf{x}) - \bar{o} \rangle \\
&= \frac{1}{\beta}\frac{\delta}{\delta\mu(0)}\text{Tr}\left[(o(\mathbf{x}) - \bar{o})\, e^{-\beta\mathcal{F}_\mu}/Z\right] \\
&= \text{Tr}\left[(o(\mathbf{x}) - \bar{o})\, o(0)\, e^{-\beta\mathcal{F}_\mu}/Z\right] \\
&= \langle o(\mathbf{x})o(0) \rangle - \langle o(\mathbf{x}) \rangle\langle o(0) \rangle.
\end{aligned} \tag{5.22}$$

Here we used $\langle o(\mathbf{x}) \rangle = \bar{o}$ in going to the last line. To evaluate this expression, notice that we already found $\delta o(\mathbf{x})$ in Eq. (5.20). Taking the functional derivative then yields:

$$\langle o(\mathbf{x})o(0) \rangle - \langle o(\mathbf{x}) \rangle\langle o(0) \rangle = \frac{k_B T}{4\pi c^2}\frac{e^{-|\mathbf{x}|/\xi}}{|\mathbf{x}|}. \tag{5.23}$$

We can insert this into the Ginzburg criterion of Eq. (5.21), and in three dimensions use $\int d^3x\, e^{-x/\xi}/x = \int d\Omega \int dx\, x e^{-x/\xi}$ and $\partial_\xi e^{-a/\xi} = (a/\xi^2)e^{-a/\xi}$ to evaluate the integral (from $x = 0$ to $x = \xi$):

$$\begin{aligned}
\frac{k_B T}{c^2}(1 - 2/e)\xi^2 &\ll \frac{4}{3}\pi\xi^3\bar{o}^2 \\
\frac{3 - 6/e}{4\pi}\frac{k_B T}{c^2\xi\bar{o}^2} &\ll 1.
\end{aligned} \tag{5.24}$$

Close to the phase transition, the temperature in the numerator is approximately $T_c$. In the denominator, we can use our results from the uniform Landau theory to substitute $\xi = c/\sqrt{2|r|}$ and $\bar{o}^2 = |r|/u$. We also know that at the phase transition, $r$ changes sign, so that we can also write $r \approx r_0 t$ close to transition. Putting everything together, the entire fraction on the left hand side is then seen to diverge as $1/\sqrt{t}$ as the critical temperature is approached. There must therefore be a region of temperatures around the critical temperature for which the Ginzburg criterion is violated, and Ginzburg–Landau theory breaks down. The *Ginzburg temperature* indicating the approximate size of this region can be defined as the reduced temperature $t_G$ at which the fraction in Eq. (5.24) equals one. This results in:

Ginzburg
temperature

$$\sqrt{t_G} = \sqrt{2}\frac{3 - 6/e}{4\pi}\frac{k_B T_c u}{\sqrt{r_0}c^3}. \tag{5.25}$$

The Ginzburg temperature is not an exact, quantitative bound up to which Ginzburg–Landau theory can be trusted. Rather, it indicates the order of magnitude for the reduced temperature at which thermal fluctuations become important. For reduced temperatures of

the order of $t_G$ and below, the approximations on which Ginzburg–Landau theory is based are not justified, and more sophisticated methods should be used to describe the system. The applicability of Ginzburg–Landau theory to any realistic situation may seem precarious, since the theory is based on the assumption that the average order parameter is not too large, but also breaks down when it becomes too small, close to the transition. In practice, it turns out that the regime over which the theory is reliable is actually very large for most symmetry-breaking systems. To give you a feeling, the Ginzburg temperature in superconductors may vary from $t_G \sim 10^{-16}$ for strongly type-I superconductors to $t_G \sim 10^{-4}$ for strongly type-II superconductors.

During the analysis of the coherence length and Ginzburg temperature we discarded in Eq. (5.18) all terms with powers of the fluctuation higher than one. This approximation is similar to the one we made in our discussion of the Heisenberg antiferromagnet in Section 5.3.2, and as in that case, it implies that the Ginzburg–Landau theory is an example of a mean-field theory. The expression for the Ginzburg temperature in Eq. (5.24) is the result for three spatial dimensions. In general, the left hand side is proportional to $t^{\frac{D-4}{2}}$ [20]. In dimensions $D \geq 4$, the suppression of local order due to fluctuations thus no longer diverges, and the results of mean-field theory are robust all the way up to the critical temperature. The spatial dimension below which thermal fluctuations qualitatively affect the phase transition and invalidate a mean-field description, is called the *upper critical dimension*. For the Ginzburg–Landau theory of this section, the upper critical dimension is four. This is to be contrasted with the lower critical dimension introduced in Section 4.2, at and below which fluctuations are so violent they prevent the establishment of long-range order altogether. For the Ginzburg–Landau theory of Eq. 5.15 the lower critical dimension is two. At non-zero temperatures we thus find that only in three dimensions there can exist a phase transition, in which local fluctuations destroy a long-range ordered phase.

**upper critical dimension**

## 5.5 Universality

The Landau theory of phase transitions is based on an expansion of the free energy in powers of the local order parameter. Both the nature of the order parameter and the allowed terms in the expansion are entirely determined by the symmetries of the phases on either side of the transition, which therefore also determine many observable properties of the phase transition. The temperature dependence of the order parameter near the phase transition, for example, was shown in the mean-field theory of Eq. (5.4) to be a power law with exponent $\beta = \frac{1}{2}$. This value of the critical exponent depends only on the fact that we considered a second-order phase transition involving a real and scalar order parameter, both of which follow directly from the symmetry being broken in the phase transition.

Notice the profound implication of this observation: knowing only the symmetries on either side of the phase transition, and nothing whatsoever about the microscopic Hamiltonian, we can already deduce real, observable properties of the system near its phase transition. This is an example of *universality* in physics, because it implies that observable properties near a phase transition characterised by a certain symmetry may be universal, and shared among even completely different physical systems. Models that have the same symmetry properties, leading to the same universal behaviour near phase transitions, can then be collected into *universality classes*. For instance, the Ising model is in the same universality class as the liquid–gas transition, and the superfluid transition in helium-4 is in the same universality class as the $XY$-model of Eq. (4.19). More practically, universality guarantees that the experimental measurement of for example the temperature dependence of specific heat near a phase transition does not depend on any microscopic details like impurities in the sample, stray magnetic fields, or the fact that the material being measured may not be completely described any simple theoretical model. Universal quantities can be measured and compared with theoretical

**universality**

**universality class**

predictions in spite of any such practical difficulties.

Precisely at a second-order phase transition or critical point the universality becomes even stronger. The correlation function of Eq. (5.23) depends on the coherence length $\xi$. At the phase transition, $\xi$ diverges, and the correlation function becomes proportional to $1/|\mathbf{x}|$. This function is an example of a *scale invariant* function, which does not define a typical length, **scale invariance** and looks the same at every scale. It should be contrasted with functions like $\cos(x/x_0)$, or $x^2(x^2 - x_0^2)$, which depend on a parameter $x_0$ that defines a characteristic length scale. The direct physical consequence of observables being described by scale invariant functions, is that they will look the same regardless of the scale at which they are measured. A famous example is the *critical opalescence* at the critical point of the liquid–gas transition (see Figure 5.1). The normally transparent water suddenly becomes opaque there, and it does so for light at all possible wavelengths. The reason is the presence of scale-invariant fluctuations, which cover all length scales and therefore scatter light at all wavelengths, including scales far beyond any related to atomic or molecular properties.

The correlation function at the phase transition, $c(|\mathbf{x}|) \propto 1/|\mathbf{x}|$, is in fact a so-called *homogeneous function*, satisfying

$$C(|\mathbf{x}|) = b^{\kappa} C(b|\mathbf{x}|). \tag{5.26}$$

Here $b$ may be any real number, and $\kappa$ is a characteristic exponent, which in this case equals one. The fact that the correlation function is homogeneous is the key ingredient in the theory of *renormalisability* and the *renormalisation group* description of phase transitions. These **renormalisation** approaches are a way of describing the temperature region immediately around the critical temperature, where the coherence length diverges and Ginzburg–Landau theory breaks down. Crudely, the idea is to first identify some small length $a$ in the microscopic model, which could for example be the lattice constant. We can then coarse-grain or average over any physical excitations or fluctuations that occur at length scales between $a$ and $ba$, where $b > 1$. Like the original Hamiltonian, the coarse grained description can be used to formulate an effective description in terms of a Landau free energy, but this time using the rescaled coordinate $\mathbf{x}' = b\mathbf{x}$. Because none of the symmetries of the model are affected by the coarse-graining, the Landau free energy will look the same as for the original model, but with different values for its parameters. Using Eq. (5.26), the coherence length in the coarse-grained model can then be expressed as a function of the original one, $\xi' = f(\xi)$. If the coarse-grained coherence length happens to be smaller than the coherence length of the original model, the effect of fluctuations is smaller and the critical temperature may be approached more closely before the Ginzburg–Landau theory becomes invalid. Repeating the coarse-graining many times, we can even hope to approach the critical temperature arbitrarily closely.

This procedure is known as the real-space renormalisation group. Several similar procedures exist, including renormalistion in momentum space, or even in terms of the order parameter fields themselves. These approaches capture the effects of thermal fluctuations near phase transitions in many realistic settings. The study of the renormalisation group is consequently a major field of study on its own, for which several excellent textbooks are available [20, 67, 70, 75].

# 6 Topological defects

We have seen that fluctuations of the local order parameter from its average value play an important role in establishing the stability of the broken-symmetry state. Their proliferation at sufficiently high temperatures can cause the long-range order to melt and induce a phase transition, while the Mermin-Wagner-Hohenberg theorem shows that in sufficiently low dimensions, fluctuations can even prevent the occurrence of long-range order altogether. The

fluctuations we considered in these analyses were invariably of the form of Nambu-Goldstone modes. That is, they were small, wave-like modulations of the order parameter. One may wonder if there exist any other types of fluctuations that influence the order of the broken-symmetry state. In fact, a whole other class of such alternative excitations exist, known as *topological excitations* or *topological defects*.

**topological defect**

## 6.1 Meaning of *topological* and *defect*

The most intuitive example of a topological defect is that in a state of $U(1)$ order. Consider for example the $XY$-model of Eq. (4.19), whose degrees of freedom can be visualised by unit vectors confined to a two-dimensional plane. In the ordered state, all vectors point in the same, spontaneously chosen, direction. Nambu–Goldstone modes correspond to plane wave excitations, in which the direction of the vectors oscillates as the system is traversed (see Fig. 6.1a). For long-wavelength excitations, neighbouring vectors are never far from parallel, and the energy cost of creating NG-modes may be arbitrarily low. It is also possible, however, to imagine modulations of the direction when going around a circle, rather than propagating in a straight line, as shown in Fig. 6.1b. Even though in this configuration most neighbouring spins are also close to parallel, this *vortex* is fundamentally different from the plane wave. To see this, consider a closed contour like the red line in Fig. 6.1b. Because the vectors are locally parallel, they must rotate over an integer multiple of $2\pi$ as we go around the contour. This integer is called the *winding number*. The value of this integer does not depend on where precisely the contour is drawn, as long as it encircles the centre of the vortex configuration. It is therefore a property of the vortex itself, called the *topological charge* or *topological invariant*. The charge may be understood to be *topological* because the contour used to determine it can be freely deformed. Furthermore, any smooth change of the vector-field configuration does not alter the winding number. A more precise discussion will be given in Section 6.4.

**vortex**

**winding number**

**topological charge**

**topological invariant**

Somewhere within the red contour, there must be a *singularity* in the order parameter field, because the winding of the vectors is independent of the contour size. Shrinking the contour as much as possible, it must then end up in a single point at which the direction of the order parameter vector is undefined: there is a *defect* in the order parameter field. In actual physical systems, the singularity is avoided, either because the order occurs in a discrete crystal lattice, or because the amplitude of the order parameter can go to zero at the singular point, like in a superfluid. The *core* of the vortex around the singularity has radial size of the order of the coherence length $\xi$ defined in Eq. (5.20).

**topological defect**

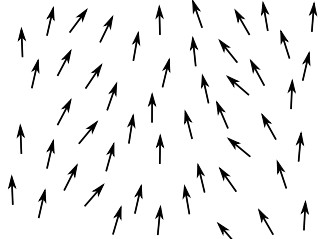
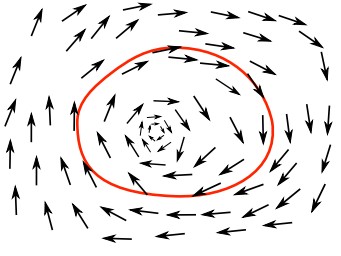
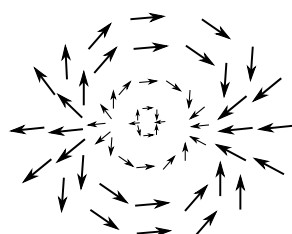

(a) Nambu–Goldstone mode     (b) $N = 1$ vortex     (c) $N = 2$ vortex

Figure 6.1: Left: $XY$-model SSB state perturbed by NG modes. Centre: a single vortex with winding number one. Following the red contour, spins wind by $2\pi$, independent of the position and shape of the contour, as long as it encloses the vortex core. At the vortex core, the phase in not defined: there is a singularity. If the size of the spins can vary, it will shrink to zero at the core. Right: a single vortex with winding number two.

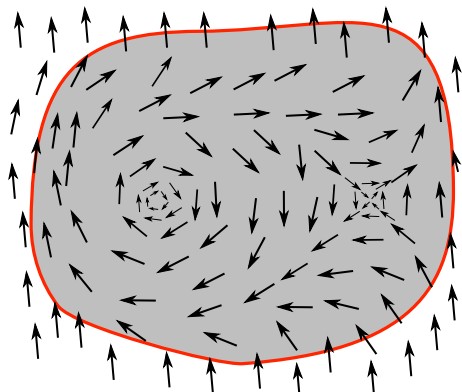

Figure 6.2: Two vortices with opposite topological charge. The total configuration is topologically neutral: following the red contour which encloses both vortex cores, the vectors do not wind at all. Furthermore, far away from the core, the vectors all point in the same spontaneously chosen direction.

Topological excitations like the vortex exist for ordered states in any dimension. A one-dimensional chain of vectors that prefer to be aligned ferromagnetically, for example, may have all vectors to the left of the origin pointing up, and all vectors to the right pointing down. There is then a zero-dimensional topological defect, called a domain wall, at the origin. Likewise, a two-dimensional system may host a vortex, as we have seen already, and vectors in a three-dimensional volume may be arranged to all point outwards from the centre, like the needles on a hedgehog, in what is known as a monopole configuration. More generally, for a $D$-dimensional system, defects of dimension $D-1$ are collectively called domain walls, of $D-2$ vortices, and of $D-3$ monopoles, although the nomenclature may vary for different subfields in physics. Vortices, such as those in Fig 6.1 have cores that are pointlike objects in 2D and linelike in 3D.

Starting from a perfectly ordered configuration, creating a single topological defect involves changing the orientation of the order parameter almost everywhere in the system. Such a defect therefore typically costs a lot of energy, often even scaling with the logarithm of the volume of the system or faster. It also makes it extremely unlikely that such defects are introduced spontaneously by thermal or other fluctuations. Systems with isolated topological defects can be created, however, when defects are forced in from the outside. Consider for example a superfluid in a container. The superfluid order makes its flow dissipationless, but it also causes the fluid to be irrotational in its ground state, so that its order parameter field has vanishing vorticity. If we now start to spin the container, trying to apply an external torque on the superfluid, at first nothing will happen. The superfluid will remain perfectly still inside container, seemingly ignoring its rotation. Once the externally applied torque exceeds the energy cost of forming a single vortex, however, a topological defect will move into the system from the side, and cause the phase of the order parameter to wind throughout the entire superfluid. This way, *quantised* amounts of angular momentum, proportional to the winding number of the total vortex configuration, may be imposed on the superfluid.

While a single topological defect may not be easily created, it is very stable once formed, since you need to make an extensive amount of change to the system to remove the defect. Furthermore local disturbances to the order parameter cannot alter the topological charge. For this reason, topological defects are under investigation for use in for instance quantum computation and information storage.

In stark contrast to the effort required for creating isolated topological defects, it is common for them to occur in *topologically neutral* combinations. For vortices, the total topological

charge of multiple defects can be found by drawing a contour like in that Figure 6.1, enclosing all vortex cores. If the phase of the order parameter does not wind along this contour, the defects together form a neutral configuration, such as the one depicted in Figure 6.2. Such neutral combinations affect the orientation of the order parameter within only an isolated part of the system, and their energetic cost grows with the separation beween cores, rather than the system size. They can therefore be created as thermal excitations.

## 6.2 Topological melting: the $D = 1$ Ising model

To see the importance of topological defects in the study of long-range order and spontaneously broken symmetries, consider a one-dimensional chain with classical Ising spins. Unlike the usual spin, Ising spins are classical objects that always point either up or down. A ferromagnet made of Ising spins therefore has only two possible broken-symmetry states, either with all spins up, or all down. This is an example of discrete symmetry breaking, rather than the continuous symmetries considered in most of these lecture notes. The Hamiltonian for the Ising ferromagnet is given by (recall Eq. (2.49)):

$$H = -J \sum_i \sigma_i^z \sigma_{i+1}^z. \tag{6.1}$$

The ground state is two-fold degenerate and has energy $E_0 = -JN$, where $N$ is the number of spins in the chain. Starting from a spontaneously chosen ground state with all spins up, the simplest excitation to create is a single spin flip, as shown in Fig. 6.3b. This costs an energy $E_f = 4J$, because the flipped spin is now aligned antiferromagnetically with two neighbours, each causing a change in energy on the bond from $-J$ to $+J$. The single spin flip can be thought of as a localised version of the spin waves of the Heisenberg ferromagnet. Because Ising spins cannot be continuously rotated from up to down, a localised spin-flip is the best one can do, and what used to be a massless NG mode in the system with continuous symmetry is now a gapped excitation in the discrete case.

A single spin flip can be seen to reduce the total magnetisation by 2, and even a large but not-extensive number of spin flips cannot completely remove the magnetisation. The one-dimensional Ising ferromagnet thus seems to be stable at non-zero temperatures.

This conclusion, however, turns out the be wrong, because we neglected the *topological defects* of the ferromagnetic state. For the one-dimensional chain, a topological defect is a domain wall created by splitting the chain in two segments, and taking all the spins to be up in one segment, and down in the other, as in Fig. 6.3c. Since only a single pair of neighbouring spins is now aligned antiferromagnetically, the energetic cost of the domain wall is only $E_{dw} = 2J$. That is, in this special one-dimensional case, the cost of a domain wall is lower than that of a spin flip. Even more importantly, a single domain wall involves a reorientation of a macroscopic number of spins, and thus strongly affects the average magnetisation. Even if we include only states with a single domain wall in the low-energy effective model, the consequences are drastic. In a chain with $N$ spins and a domain wall at position $j$, the magnetisation is $M = 2j - N$.

There are about $N$ possible configurations to put a single domain wall so the entropy is $S = \ln N$. The free energy of a single wall is then $F_{\text{one wall}} = 2J - k_B T \ln N$. For large systems at finite temperature, the entropic gain outweighs the energetic cost to introduce domain walls into the system. The thermal expectation value of the magnetisation therefore vanishes completely in the thermodynamic limit, and ferromagnetic order cannot occur in the one-dimensional Ising chain at any non-zero temperature. This is the simplest example of **topological melting**, in which the local order is destroyed by the proliferation of topological defects rather than NG modes.

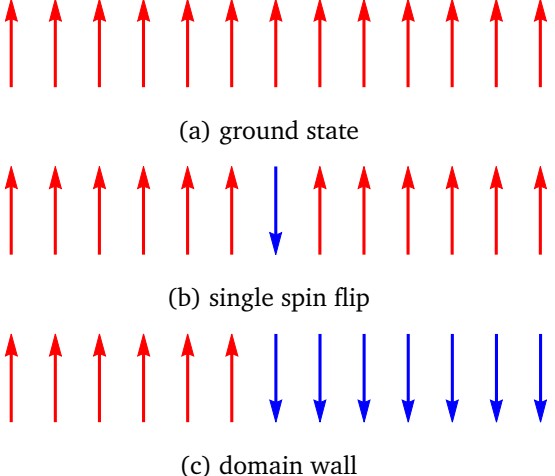

(a) ground state

(b) single spin flip

(c) domain wall

Figure 6.3: Top: the ferromagnetic ground state of the one-dimensional Ising model. Centre: a single spin flip, which only marginally affects the total magnetisation. Bottom: a single domain wall, which is a topological defect with macroscopic effect on the total magnetisation. In one dimension, the domain wall has lower energy than the spin flip.

## 6.3 Berezinskii–Kosterlitz–Thouless phase transition

The Mermin–Wagner–Hohenberg–Coleman theorem of Section 4.2 shows that thermal fluctuations will destroy long-range order at any non-zero temperature in two-dimensional systems that have type-A NG modes at zero temperature. Rather than the exponentially decaying correlation functions that characterise truly disordered states, however, it has been shown that some two-dimensional systems have low-temperature correlation function Eq. (2.33) that decay as power laws:

$$C(\mathbf{x}, \mathbf{x}') \propto |\mathbf{x} - \mathbf{x}'|^{-c}, \qquad |\mathbf{x} - \mathbf{x}'| \to \infty. \tag{6.2}$$

Here, $c$ is a system-dependent real exponent. These types of correlation functions are not long-ranged, but they also are qualitatively different from the short-ranged, exponentially decaying correlation functions always prevail at sufficiently high temperatures. States with power law correlation functions are therefore said to exhibit *algebraic long-range order*. Going from low to high temperatures, there must be a critical temperature at which algebraic long-range order gives way to true disorder. Because a power law cannot be analytically continued to an exponential function, the correlation function becomes non-analytic at the critical temperature, which thus corresponds to a true phase transition rather than a smooth crossover. This phase transition occurs despite the fact that no symmetry is truly broken in either the low or the high temperature phase.

As it turns out, the phase transition in this case is described by another form of topological melting: the *unbinding* of pairs of topological defects. Topologically neutral configurations of defect–antidefect pairs, such as that in Fig. 6.2 can occur as finite-energy excitations in an otherwise ordered background. This implies that a system starting out in an ordered state at zero temperature will develop a thermal population of such pairs at non-zero temperatures. The energy cost associated with a defect–antidefect pair scales with the separation between their cores, and at low temperatures no defect will have sufficient energy to wander far from its antidefect partner. The low temperature phase then, is characterised by a thermal population of *bound* defect pairs, in what turns out to correspond to a state of algebraic long-range order.

As temperature is raised, the pairs become more prolific, and defects within a pair become further separated from their partners. At some temperature, the average separation between

partners becomes as large as the separation between pairs. At that point, an individual defect can no longer be associated with any particular antidefect, and single excitations may freely roam the system. In other words, there is an *unbinding* of topological defects. This picture, put forward by Berezinskii [76] as well as Kosterlitz and Thouless [77, 78], is now known as the BKT phase transition. The name is applied in particular to systems with $U(1)$ or $XY$-symmetry, but the phenomenon is much more general. The only requirement for it to occur is that stable point-like topological defects may be formed in two-dimensional systems with a spontaneously broken symmetry at zero temperature.

**BKT phase transition**

The reason that defect pairs must unbind at sufficiently high temperatures can be conveyed using a heuristic argument due to Kosterlitz and Thouless [78], inspired by the argument of the 1D Ising model. They show that in a system of $XY$-spins, the energy of single, isolated vortex is proportional to $E_{\text{one defect}} \propto \ln L/\xi$, where $L$ is the linear system size and $\xi$ its coherence length. There are about $L^2/\xi^2$ ways to put an object of area $\xi^2$ into a system with area $L^2$. The entropy associated with a single defect is therefore $S_{\text{one defect}} \approx k_{\text{B}} \ln L^2/\xi^2 \approx k_{\text{B}} 2 \ln L/\xi$. Notice that the energy and entropy scale with system size in the same way. This means the free energy associated with a single, isolated defect is:

$$F_{\text{one defect}} = E - TS \approx (J - k_{\text{B}}T) \ln L/\xi, \tag{6.3}$$

where $J$ is an energy scale that depends on the microscopic model. At low temperatures, the energy cost of creating a defect is higher than the entropy gain, and isolated defects will not occur. At sufficiently high temperatures however, the entropic term outweighs the energetic one, and isolated defects proliferate throughout the system, destroying any type of order. Notice that although this argument nicely shows that a thermal phase transition is unavoidable, it is only part of the story. It neglects the physics of defect–antidefect pairs which screen the interactions between defects. This allows them to drift further apart and lowers the energy cost of creating additional defect pairs, eventually culminating in a proliferation of defects at the critical temperature.

The BKT phase transition cannot be described within the usual Landau paradigm of phase transitions discussed in Section 5. It is sometimes said to be an "infinite-order" phase transition, because the free energy and all of its derivatives remain continuous throughout the transition. It does have distinct critical exponents, which are used to experimentally identify BKT transitions, and which may be calculated using an appropriate version of the renormalisation group. Evidence for BKT transitions was first found in films of superfluid helium, and later in anisotropic magnets, ultracold atomic gases, colloidal discs, and even thin-film superconductors.

## 6.4 Classification of topological defects

Which topological defects may arise in a certain ordered state is determined entirely by the broken symmetries that define its order parameter. The details of this classification are beyond the scope of these lecture notes. An excellent review may be found in Ref. [79]. Here, we restrict ourselves to a superficial introduction and a presentation of some examples.

Recall the discussion in Section 2.5.1, in which we argued that the possible values or directions of the order parameter in broken-symmetry states correspond to the quotient space $G/H$. Here $G$ is the group of all symmetry transformations of the symmetric system, and $H$ is the subgroup of unbroken transformations in the ordered state. If the direction of the order parameter is allowed to vary, different points in real space may correspond to different points in the quotient space. In other words, the state of such a system is described by a mathematical map from real space to the quotient space $G/H$.

The mathematical structures that categorise topologically distinct ways of mapping from

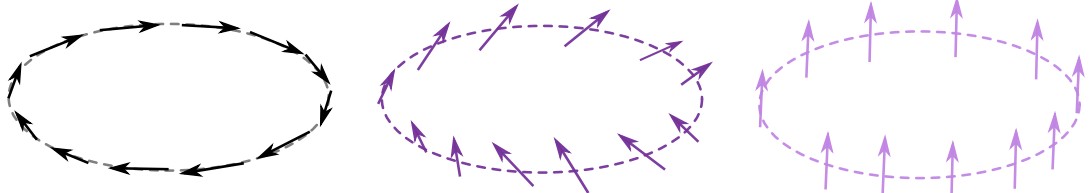

Figure 6.4: Removal of a vortex by rotating vectors out of the plane. This is sometimes called "escape in the third dimension", and shows the vortex is not a stable topological defect for order parameters that can be represented by a three-dimensional vector.

a certain space to another, are the so-called *homotopy groups*.[15] As an example, consider the winding of $XY$-vectors upon going around the vortex in Fig. 6.1. The system has a $U(1)$ symmetry, which is broken completely in the ordered state, so that the group of unbroken transformations is simply the trivial group $H = 1$. The quotient space is then $G/H = U(1)$, which is equivalent (isomorphic) to $S^1$, the set of points on a circle. The vectors in the ordered state can point in any direction in the two-dimensional plane, so that the possible orientations of the order parameter indeed trace out a circle. To each point in real space along the red contour of Fig. 6.1, a 1-loop, corresponds a point on the circle of possible order parameter orientations. As we go around the 1-loop in real space, we therefore also trace out a closed path on the circle. If the 1-loop in real space does not encircle any singularity, the path in the quotient space will cover only part of the circle. It can then be contracted to a point by smooth deformations. If a single vortex is enclosed within the 1-loop however, the quotient space will be traversed completely.

Notice that not any smooth deformation can transform the single covering of $S^1$ into a point, so the situation with and without a vortex give topologically distinct paths in the quotient space. Continuing in this way, the charge-two vortex in Fig. 6.1c corresponds to a path going around the quotient space twice, which again cannot be transformed into a either a point or a single covering by smooth deformations. The topological index quantifying the difference between all such paths in the quotient space, is the total number of times the circle is covered, with clockwise paths counting as positive and counterclockwise paths contributing negative terms. This is the winding number. Since the circle can be covered any integer number of times, the first homotopy group is $\pi_1(U(1)) \simeq \mathbb{Z}$ in this case.

In general, $p$-dimensional topological defects in a $D$-dimensional system are classified by the homotopy group $\pi_{D-p-1}(G/H)$. The contour that can be used to detect such defects is $(D - p - 1)$-dimensional. In the example of $XY$-vectors, a one-dimensional contour in a two-dimensional system is used to characterise a zero-dimensional, point-like defect.

Analysed this way, domain walls in the one-dimensional Ising model are characterised by $\pi_0(\mathbb{Z}_2) = \mathbb{Z}_2$ (the "zeroth" homotopy group $\pi_0$ just counts the disconnected components). That is, a link in the chain is either a domain wall, or it is not. Vortices in a $U(1)$ symmetry are point-like in 2D or line-like in 3D, characterised by $\pi_1(U(1)) = \mathbb{Z}$. The second homotopy group $\pi_2(U(1))$ turns out to be trivial, indicating there are no monopoles in 3D in such a state.

Ferromagnetic configurations of three-dimensional spins break $SU(2)$ symmetry down to $U(1)$, so that the order parameter takes values on the two-sphere $S^2 \simeq SU(2)/U(1)$, see Section 2.5.1. Any closed 1-loop on the surface of that sphere can be contracted to a point by smooth deformations. There are thus no topologically distinct paths, and the first homotopy group is trivial. This means there can be no stable vortices in Heisenberg ferromagnets. If a vortex is introduced in such a magnet, all spins can be smoothly rotated to a perpendicular

---

[15]A good introduction into the mathematics of homotopy groups is given by Ref. [80].

direction to remove the singularity, as indicated in Figure 6.4. It is possible, however, to have zero-dimensional defects in a three-dimensional Heisenberg ferromagnet, by arranging spins in a hedgehog or monopole configuration, pointing radially outward from the origin everywhere. Such monopoles are classified by the homotopy group $\pi_2(S^2) = \mathbb{Z}$, where in this case the integer index counts the number of times a two-dimensional surface (a 2-loop) covers the two-sphere.

Crystalline solids can have two types of $\pi_1$ topological defects (so points in 2D or lines in 3D), known as dislocations and disclinations, associated with translational and rotational symmetry breaking respectively. The dislocation has a vector-valued topological charge called the Burgers vector, and can be thought of as a row of misaligned atomic bonds within an otherwise regular lattice. The disclination is characterised by an angle, corresponding to a wedge of superfluous or deficient material. Famously, the idea that neutral pairs of dislocations in hexagonal lattices might proliferate led to the prediction of a new phase of matter called *hexatic liquid crystal*. It is liquid in the sense that it is translationally symmetric, but possesses 'hexatic' order as the rotational symmetry remains broken down to six-fold discrete rotations, $C_6$ [81–83]. The transition between the crystalline and hexatic liquid-crystalline phases is called *dislocation-mediated melting*, which is similar to the BKT phase transition. **dislocation-mediated melting**

Finally, one may also imagine *time-dependent* topological excitations. A defect that exists only at one point, an *event*, in space-time is called an *instanton*, and appears in the study **instanton** of Yang-Mills theories as well as in theories of nucleation at first-order phase transitions. In four-dimensional space-time they are enclosed by a three-dimensional contour, so they are characterised by the third homotopy group $\pi_3(G/H)$. For example in systems with spontaneously broken $SU(N)$ symmetry this homotopy group can be non-trivial, and instantons play an important role. The book by Shifman [84] is a good reference for this topic.

## 6.5 Topological defects at work

Topological defects play a role in many physical phenomena. We include a brief introduction to some of them, but do not attempt to be comprehensive in any sense.

### 6.5.1 Duality mapping

The traditional picture of a phase transition, following Landau, starts from the symmetric, disordered state and describes the emergence of a broken symmetry. As shown by the BKT transition, it may sometimes also be useful to take a complementary approach, and consider how the proliferation of topological defects leads towards a disordered state starting from the ordered, symmetry-breaking phase. In some cases, it may be possible to take this approach one step further and treat the topological defects as particles in their own right. The transition from the low- to the high-temperature phase can then be described as a Bose-Einstein condensation of defects, spontaneously breaking an associated symmetry. Seen this way, the low- and high-temperature phases are both ordered, but in very different ways. In fact, the order parameter for the defect condensate acts as a *disorder parameter* for the original particles, and vice versa.

The creation of a topological defect involves a reorientation of particles throughout the system. A creation operator for a topological defect is therefore extremely non-local in terms of the creation and annihilation operators of the underlying particles. Nevertheless, it sometimes so happens that writing all original creation and annihilation operators in a Hamiltonian in terms of defect operators results in a form that is as convenient as the original. You can then choose to describe the physics of the system either in terms of the original particles, or in terms of topological defects acting as particles. Both pictures give the same results, but one is often easier to apply in the low-temperature phase, and the other in the high-temperature phase.

The mathematical map between two descriptions of the same system is called a *duality mapping*. The first example of a duality in physics was established by Kramers and Wannier for the 2D Ising model [85], writing the model in terms of domain walls rather than original spins, which enabled Onsager to solve it exactly in 1944 [86]. The existence of such a duality mapping usually allows one to explore the properties of a critical point more easily or more thoroughly. This has met with considerable success in the description of $U(1)$-symmetry breaking in two and three dimensions, where it goes under the name of boson–vortex duality. Recently, the approach has been extended to systems involving multiple species of particles including fermions, in a so-called "web of dualities" [87, 88]. In all cases, the phase transition described by a duality mapping can be viewed as the unbinding or condensation of topological defects.

**duality mapping**

### 6.5.2 Kibble–Zurek mechanism

The dynamics of phase transitions generally falls outside the scope of these lecture notes. Nevertheless, it is worthwhile to mention here that one way in which topological defects come into existence in practice, is by going through a continuous phase transition 'too quickly'. As mentioned in Section 5.4, all length scales and energy scales diverge near a continuous phase transition. In fact, characteristic time scales diverge as well, and this includes the relaxation time, which is the time it takes for a system to dissipate any excitations. This effect of increasing time scales near a phase transition is called *critical slowing down*, and it plagues both numerical simulations of phase transitions, and their experimental study. When driving a system across a phase transition, the critical slowing down makes it impossible to retain equilibrium at all times. No matter how slowly and carefully you cool a system, close to the phase transition you will always exceed the relaxation rate. The implication is that all systems are necessarily in a highly excited state when entering the ordered phase. While relaxing towards equilibrium again, long-range order is gradually built up, but the topological stability of defects that were present in the excited state prevent them from being removed by local relaxation mechanisms. The result is an ordered state with a non-zero density of topological defects.

**critical slowing down**

This way of creating topological defects by crossing a continuous phase transition was first proposed by Kibble [89] to explain structure formation in the universe after the Big Bang. It was later refined by Zurek [90], who derived the expected density of topological defects associated with any given quench rate (the rate of temperature change) and universality class. It is now referred to as the Kibble–Zurek mechanism.

**Kibble–Zurek mechanism**

### 6.5.3 Topological solitons and skyrmions

In our discussion of topological defects so far, we neglected a special category of topological objects, called *topological solitons* (this name is sometimes applied only to systems in one spatial dimension). These objects are topological in the sense that they have a topological charge, which takes quantised values, and which cannot be altered by smooth deformations. In contrast to the usual topological defects, however, they do not require any singularity in the order parameter field, and they have only a finite energy, which does not scale with system size and which is strongly localised near the centre of the soliton.

**soliton**

To understand how such an object can be created, consider an XY ferromagnet on a one-dimensional line. The topological soliton will be localised near the centre of the line, but importantly the spins far away from the centre are as good as unaffected by its presence. That is, the order parameter is undisturbed and constant almost everywhere along on the line. To describe the soliton, we now do a mathematical transformation which maps the points at the boundary (or at infinity) onto a single point. This is allowed since the order parameter takes the same value at these points. For the one-dimensional line, this implies that many points

from both sides of the line will be taken to the same point, turning the line into a circle. This mathematical procedure is called *compactification*. The soliton spin configuration in real space corresponds to a vortex configuration on the compactified space. The vortex core lies in the centre of the circle, so the order parameter field along the circle is smooth and well-defined everywhere, as shown in Fig. 6.5.

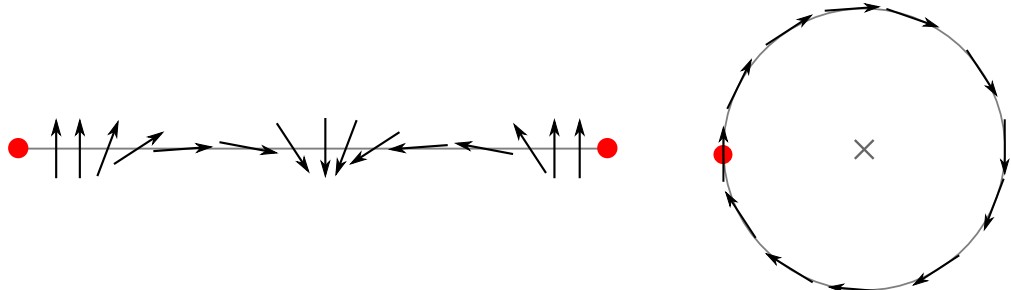

Figure 6.5: A topological soliton of charge one. Left: the configuration in real space, on a finite line. The spins at the boundaries (red dots) point in the same direction. As the line is traversed from left to right, the spins wind smoothly over a $2\pi$ angle. Right: the (red) points at the boundary of the line are mapped to the same point on a circle, illustrating the topological nature of the soliton. The cross indicates the position of the associated singularity, which lies outside of the one-dimensional space on which the order parameter field is defined.

Similar topological solitons also exist in higher dimensions $D$, categorised by $\pi_D(G/H)$. For example, the two-dimensional plane may be compactified into a 2-sphere, using the stereographic projection. An $S^2$-valued order parameter, like the magnetisation of a Heisenberg ferromagnet, can then be arranged in a hedgehog or monopole configuration on the sphere, with all spins pointing radially outward. Folding the sphere back out into a flat plane, the resulting spin configuration is called a skyrmion. This type of topological soliton appears in **skyrmion** quantum Hall systems and some magnetic materials. In nuclear physics the same configuration of spins is called a *baby skyrmion*, while the name *skyrmion* is reserved for its three-dimensional siblings, which were introduced by Skyrme [91] as a possible way of creating pointlike objects within a smooth three-dimensional vector field.

# 7 Gauge fields

We briefly discussed gauge freedom in Section 1.5.2, but the main focus of these lecture notes so far has been on systems with a global symmetry in the absence of gauge fields. Although gauge freedom can never be broken, its presence does affect the physical phenomenology of symmetry-breaking phases and phase transitions. Here, we will introduce some of these effects by considering the explicit example of the *superconducting* state, in which a global phase **supercon-** rotation symmetry is broken in the presence of a local $U(1)$ gauge freedom. This example **ductor** makes apparent much of the physics that also appears in more complicated, and even non-Abelian, types of gauge freedom, whose description is more involved mathematically. Non-Abelian gauge fields are briefly discussed in Exercise 7.4.

## 7.1 Ginzburg–Landau superconductors

Real-world superconducting materials are metals that are cooled to very low temperatures, where they go through a phase transition and become superconductors. This instability of the metallic state can be understood in terms of a microscopic model by the famous Bardeen–Cooper–Schrieffer (BCS) theory [92]. One of the main ingredients in this theory is the *Cooper instability* which explains that any attractive force between electrons in a Fermi liquid will lead to the formation of bound states of two electrons with opposite spin and momenta, called Cooper pairs. In the BCS theory, the attractive force between electrons arises from their interaction with phonons in the crystal lattice. The Cooper pair contains two fermions, which as a whole behaves like a boson. They can thus Bose-condense at sufficiently low temperatures, and the superconducting state can be viewed as a superfluid of Cooper pairs. Because the bosons in this case are electrically charged, the dissipationless flow of the superfluid is actually a resistance-free electric supercurrent.

**BCS theory**

**Cooper pair**

The symmetry-breaking transition in superconductors is the Bose-condensation of Cooper pairs. To discuss the broken symmetry, its relation to gauge freedom, and its observable consequences, we can largely ignore the fact that Cooper pairs really consist of two electrons bound by phonons. Instead, we start straight away from a (metallic) normal fluid of charged bosons. The Landau potential of Eq. (5.6) describes the free energy of a complex order parameter field $\psi(x)$, whose ordered state we argued corresponds to a neutral superfluid. The effect of having charged Cooper pairs rather than neutral bosons can be seen only if we allow for fluctuations in their density, since the average electronic (or Cooper pair) charge is balanced precisely by the positive charge of the ionic lattice. In the Landau potential, the squared amplitude of the field, $|\psi(x)|^2$, represents the density of bosons. Adding the lowest order term in an expansion of gradients of the density, as we did for the Ginzburg–Landau theory of Section 5.4, we would expect a contribution to the energy of the form $\frac{\hbar}{2m^*}|\nabla\psi|^2$, with $m^*$ is the mass of a single boson. Because fluctuations in the density of Cooper pairs are charged, they both create and are affected by electromagnetic fields. A convenient minimal way of introducing the coupling between such fields and local charge fluctuations was suggested by Peierls, and consists of simply making the substitution $\nabla\psi \to (\nabla - i\frac{e^*}{\hbar}\mathbf{A})\psi$, where $\mathbf{A}$ is the electromagnetic vector potential and $e^*$ is the electric charge of an isolated boson. In this case, each Cooper pair contains two electrons, so that $e^* = 2e$ with $e$ the electron charge. For convenience, we assume the electromagnetic scalar potential $V$ to be zero. The full free energy of the Ginzburg–Landau theory for superconductivity is then:

**Peierls substitution**

**Ginzburg–Landau theory**

$$\mathcal{F}_{\text{GL}} = \frac{\hbar^2}{2m^*}|(\nabla - i\frac{e^*}{\hbar}\mathbf{A})\psi|^2 + \frac{1}{2}r|\psi|^2 + \frac{1}{4}u|\psi|^4 + \frac{1}{2\mu_0}(\nabla\times\mathbf{A})^2. \tag{7.1}$$

Notice that this includes the potential energy of the electromagnetic field itself, with $\mu_0$ the magnetic constant, and that we consider only time-independent fields since we are interested in understanding equilibrium phases. The free energy $\mathcal{F}_{\text{GL}}$ can be used to explain a large part of the phenomenology of superconductivity, including its dissipationless current, the Meissner effect, vortex topological defects, and the Josephson effect. All of these are intimately related to symmetry breaking and will be discussed here. For other aspects, or a more detailed treatment, many excellent textbooks on superconductivity, such as Refs. [72, 74, 93, 94], may be consulted.

**Exercise 7.1 (Peierls substitution)** Verify that $\frac{e^*}{\hbar}\mathbf{A}$ has units of inverse length, to confirm that the Peierls substitution is dimensionally correct.
The reason that $\hbar$ appears in the substitution stems from the fact that electromagnetic

fields couple to charged matter via quantum electrodynamics.

The free energy $\mathcal{F}_{\text{GL}}$ is invariant under the *global $U(1)$ symmetry transformation*:

$$\psi(x) \to e^{-i\alpha}\psi(x). \tag{7.2}$$

As always, this global symmetry is associated with a conserved Noether current, which may be obtained either through the usual Noether procedure, or directly by applying the Peierls substitution to the Noether current of the neutral superfluid in Eq. (1.21). Either way, the resulting expression for the Noether current is:

$$\mathbf{j} = i\frac{\hbar^2}{2m^*}((\nabla\psi^*)\psi - \psi^*(\nabla\psi)) - \frac{e^*\hbar}{m^*}\psi^*\psi\mathbf{A} = \frac{\hbar^2}{m^*}|\psi|^2(\nabla\varphi - \frac{e^*}{\hbar}\mathbf{A}), \tag{7.3}$$

where we wrote $\psi = |\psi|e^{i\varphi}$. The conserved Noether charge transported by this current turns out to be the number of Cooper pairs. But because the Cooper pairs are electrically charged, a current of them also corresponds to an actual electric current. This is easily confirmed by considering the usual definition of the electric current $\mathbf{j}_{\text{e}} = -\delta\mathcal{F}/\delta\mathbf{A}$, which shows that it is indeed related to the Noether current by $\mathbf{j}_{\text{e}} = \frac{e^*}{\hbar}\mathbf{j}$. As always, the Noether current is manifested in the ordered phase by NG modes, whose lifetime goes to infinity in the long-wavelength limit. In this case, the infinitely long-lived current of Cooper pairs in the superconducting phase, is called a *supercurrent*. The coupling to the dynamic gauge field, however, suppresses finite-frequency modes, see Section 7.3.1. **supercurrent**

Besides the global symmetry, $\mathcal{F}_{\text{GL}}$ is also invariant under a local $U(1)$ *gauge transformation*:

**gauge transformation**

$$\psi(x) \to e^{-i\alpha(x)}\psi(x), \qquad\qquad \mathbf{A}(x) \to \mathbf{A}(x) - \frac{\hbar}{e^*}\nabla\alpha(x). \tag{7.4}$$

This gauge freedom is the result of having introduced superfluous degrees of freedom, namely the longitudinal component of the electromagnetic vector potential, which does not contribute to the observable electric and magnetic fields. Because of the minimal coupling between the order parameter field and electromagnetic vector potential in the free energy of Eq. (7.1), the phase of the field $\psi(x)$ also becomes subject to this gauge freedom. Stated differently, in $\mathcal{F}_{\text{GL}}$ any occurrence of the longitudinal component of $\mathbf{A}(x)$ can be traded for a suitable local rotation of the phase of the field $\psi(x)$. As emphasised in Section 1.5, gauge transformations are not symmetries. They are simply consistency requirements, and any physical observable derived from the free energy of Eq. (7.1) must be invariant under the transformations of Eq. (7.4). In particular, this gauge invariance can never be broken.

At this point, we should notice that for constant $\alpha(x) = \alpha$, the gauge transformation of Eq. (7.4) appears to coincide with the global symmetry transformation of Eq. (7.2), which we argued to be spontaneously broken in the superconducting phase. In fact, the situation is the same as the one we encountered in Section 1.5.3, where the spin-rotational symmetry that is broken in a ferromagnet seemed to coincide with the unbreakable global gauge freedom of choosing a coordinate system. Both for the ferromagnet and the superconductor, the distinction between gauge freedom and symmetry becomes clear once we use a more careful definition of the global symmetry with respect to an external reference. One way of doing this, is to consider a gauge-invariant definition for the order parameter.

## 7.2 Gauge-invariant order parameter

As pointed out before, ignoring any spatial variations in the Cooper pair density, the Landau potential for the superconductor is precisely the same as that for the neutral superfluid in

Eq. (5.6). In the superfluid case, we saw that for $r < 0$ the minimum of the potential is at $\langle \psi \rangle \neq 0$, and a global $U(1)$ symmetry is broken by choosing a particular phase for the minimum energy configuration $\langle \psi \rangle = e^{i\varphi} |\psi|$. Consequently, the field variable $\psi(x)$ itself could be used as the order parameter for the superfluid. Because the global symmetry being broken should not be affected by the local fluctuations that make a superconductor different from a superfluid, it is tempting to also introduce the field $\psi$ as an order parameter for the superconductor. However, the quantity $\psi(x)$ is *not* invariant under the gauge transformation of Eq. (7.4). Because any physical quantity must be gauge invariant, $\psi$ cannot be a good choice of order parameter.

The are three ways of dealing with this, each of them used in practice and throughout the literature.

1. Ignore the complication and simply use $\psi(x)$ as the order parameter. This is not as silly as is sounds. In many cases of interest, there is no external electromagnetic field, and induced fields are negligible. Then the vector potential is approximately zero, and $\mathcal{F}_{\mathrm{GL}}$ reduces to the free energy of a neutral superfluid. Simply ignoring any electromagnetic fields then suffices to get many physical predictions correct. In particular, the original publication of BCS theory used an order parameter of this form (although written differently) [92], and the Ginzburg–Landau [95] and Josephson [29] papers treated the order parameter in a similar way as well.

2. Choose a particular *gauge fix*. Just as in electromagnetism, we can impose additional, arbitrary, constraints on the vector potential and phases of $\psi$ to remove the freedom of doing gauge transformations. That is, given some configuration of $\psi$ and $\mathbf{A}$, we can choose to always apply the particular gauge transformation $\psi \to \psi'$, $\mathbf{A} \to \mathbf{A}'$ which makes the transformed fields $\psi'$ and $\mathbf{A}'$ satisfy the additional constraints. The constraints are often chosen to ensure a mathematically convenient or aesthetically pleasing form of the fields. For the superconducting theory, there are two very useful choices of constraints. One is the so-called *unitary gauge fix*, which demands the phase of the field $\psi$ to be zero everywhere, so that all degrees of freedom reside in the vector potential. The second is the *Coulomb gauge fix* (also called the *London gauge*), which imposes $\nabla \cdot \mathbf{A} = 0$ everywhere, so that the longitudinal degree of freedom is carried exclusively by the phase of the field $\psi$. Because the constraints can be implemented by a gauge transformation, they are guaranteed not to affect the values of any physically observable quantities (in this case, the electric and magnetic field and phase differences within the field $\psi$). Choosing a gauge fix is thus always an allowed thing to do at any step within a calculation, but it cannot affect any final physical predictions. For example, choosing to work within a unitary gauge fix may seem to make $\psi$ more acceptable as an order parameter. However, choosing the phase of $\psi$ to be zero in any calculation does not mean that we predict it to actually be so in any measurement. The physical outcome of any calculation must be gauge invariant, even if we choose to calculate it within a particular gauge fix.

**gauge fix**

3. Define a gauge-invariant but non-local order parameter. It is simply not possible to have a gauge-invariant local order parameter operator $\mathcal{O}(x)$ that includes only operators acting within a small neighbourhood of $x$. It is possible however, to define a gauge-invariant non-local order parameter operator, following a proposal by Dirac [96]:

$$\psi_{\mathrm{D}}(\mathbf{x}, t) = \psi(\mathbf{x}, t) \, e^{i \int d^3 y \, \mathbf{Z}(\mathbf{y}-\mathbf{x}) \cdot \mathbf{A}(\mathbf{y}, t)}. \tag{7.5}$$

Here $\mathbf{Z}(\mathbf{x})$ is defined to be a function satisfying $\nabla \cdot \mathbf{Z}(\mathbf{x}) = \frac{e^*}{\hbar} \delta(\mathbf{x})$. So $\mathbf{Z}$ is proportional to the electric field emanating from a point charge at $\mathbf{x} = 0$. The order parameter $\psi_{\mathrm{D}}(x)$ is non-local in the sense that you need to integrate over all of space to find its value at any

particular location. The Dirac order parameter $\psi_D(x)$ reduces to simply the field $\psi(x)$ in the Coulomb gauge fix $\nabla \cdot \mathbf{A} = 0$. Knowing that a gauge-invariant formulation of the order parameter exists, it is thus possible to impose a gauge fix and simply work with the field $\psi(x)$ as a local order parameter. Doing so, however, you should remember that a gauge fix was in fact imposed. The final predictions of your calculations should always be gauge invariant.

> **Exercise 7.2 (Dirac order parameter)** Verify that the Dirac order parameter is invariant under the gauge transformation of Eq. (7.4).

Using the Dirac order parameter, the difference between the global symmetry transformation of Eq. (7.2) and the uniform part of the local gauge freedom of Eq. (7.4) may be made clear. The symmetry that is broken upon entering the superconducting phase is the global $U(1)$ phase rotation symmetry of the field $\psi_D(x)$. Doing exercise 7.2, you may have noticed that $\psi_D(x)$ is invariant under any gauge transformation, *except* for the global transformation with constant $\alpha(x) = \alpha$. The reason for this, is that in the definition of the Dirac field in Eq. (7.5) the local phase of $\psi(x)$ is effectively measured, in a gauge invariant way, with respect to the phase of the field at infinity, where it is taken to be zero. The global transformation with $\alpha(x) = \alpha$, however, changes the phase of the field everywhere, including at infinity. This situation is precisely analogous to the way that rotating all spins in the universe will have no measurable effect on a ferromagnet that breaks spin-rotation symmetry. The orientation of the spins within one magnet can only be measured with respect to the direction of the magnetic field produced by a second. And likewise, the position of a crystal breaking translational symmetry is defined only with respect to a reference frame provided for example by the surrounding lab. The symmetry that can be spontaneously broken in a superconductor must therefore be a rotation of the phase of $\psi_D(x)$ which is constant throughout the piece of superconducting material, but which leaves the phase of an external reference superconductor fixed. The observable describing such relative phase differences within the Dirac order parameter, is the gauge-invariant equal-time correlation function:

$$C_D(\mathbf{x}, \mathbf{x}') = \langle \psi_D(\mathbf{x}, t) \psi_D^\dagger(\mathbf{x}', t) \rangle = \langle \psi(\mathbf{x}, t) \, e^{i \int d^3 y \, (\mathbf{Z}(\mathbf{y}-\mathbf{x}) - \mathbf{Z}(\mathbf{y}-\mathbf{x}')) \cdot \mathbf{A}(\mathbf{y}, t)} \psi^\dagger(\mathbf{x}', t) \rangle. \qquad (7.6)$$

In particular, when $\mathbf{x}$ and $\mathbf{x}'$ are taken to be points within two spatially separated superconductors, this correlation function is precisely proportional to the current measured in the *Josephson effect*, introduced in Section 2.5.5. The Josephson current thus provides a gauge-invariant global order parameter akin to the total magnetisation of a ferromagnet or the centre-of-mass position of a crystal. The local order parameter from which it is built, consists of the phase of $\psi_D(x)$, which is defined with respect to an external coordinate system, just like the local magnetisation within a magnet or the position of atoms within a crystal. The Josephson effect can even be used to measure the local value of the order parameter, by using a superconducting tip in a scanning-tunnelling experiment and registering the local value of the Josephson current.

> **Exercise 7.3 (Josephson junction array)** Consider a *Josephson junction array* consisting of a one-dimensional chain of superconducting islands. For simplicity, assume the electromagnetic field to be zero everywhere (this does not affect any of the results). Each island can be described by two coarse-grained observables: the average number of Cooper pairs on a site, $n_j$, and the average phase of the Dirac field $\psi_D$ on each site, $\theta_j$. These conjugate

variables obey $[\theta_j, n_{j'}] = i\delta_{j,j'}$. The Hamiltonian is given by:

$$H = \sum_j \frac{1}{2}Cn_j^2 - J\cos\left(\theta_j - \theta_{j+1} + \psi_j^{j+1}\right), \qquad \psi_j^{j+1} \equiv \frac{e^*}{\hbar}\int_j^{j+1} A^x(x')\,dx'$$

Here $C$ and $J$ are parameters known as the charging and Josephson energies, and $e^* = 2e$ is the charge of a Cooper pair. The phase $\psi_j^{j+1}$ comes from the Peierls substitution, with $A^x$ the $x$-component of the electromagnetic vector potential. Although each individual island is always superconducting, the chain as a whole has a superconducting transition temperature that depends on the ratio $J/C$.

**a.** Verify that the Hamiltonian is invariant under the gauge transformation of Eq. (7.4).

The Hamiltonian can be simplified by introducing new operators $\phi_j$:

$$\phi_j = \theta_j - \sum_{i=2}^j \psi_{i-1}^i, \qquad\qquad [\phi_j, n_{j'}] = i\delta_{j,j'}$$

Note that $\phi_j$ is non-local in terms of $\psi_j^{j+1}$.

**b.** Show that in terms of these, the Hamiltonian becomes approximately:

$$H \approx \sum_j \left[\frac{1}{2}Cn_j^2 + \frac{1}{2}J\left(\phi_j - \phi_{j+1}\right)^2\right]$$

**c.** Use Fourier and Bogoliubov transformations to diagonalise the Hamiltonian for $k \neq 0$. Show that its spectrum is given by $\hbar\omega(k) = \sqrt{4JC}\left|\sin\left(\frac{ka}{2}\right)\right|$

The modes in this spectrum are the Nambu–Goldstone modes associated with the chain as a whole being a superconductor. They appear here as gapless modes, because we neglected the dynamics of the electromagnetic field, see Section 7.3. The collective, $k = 0$ part not included in the NG-spectrum, is described by $H_{k=0} = Cn_{\text{tot}}^2/2N$, with $N$ the number of sites in the chain and $n_{\text{tot}} = \sum_j n_j$ the total number of particles in the entire chain. The average phase across the chain is given by $\phi_{\text{ave}} = 1/N\sum_j \phi_j$, with $[\phi_{\text{ave}}, n_{\text{tot}}] = i$.

**d.** Show that the collective Hamiltonian $H_{k=0}$ is invariant under the symmetry transformation $U = e^{i\alpha n_{\text{tot}}}$.

The symmetry of $H_{k=0}$ can be broken by introducing a symmetry breaking field:

$$H'_{k=0} = \frac{1}{2N}Cn_{\text{tot}}^2 + \frac{1}{2}J'\phi_{\text{ave}}^2$$

This could be interpreted as a coupling of the chain to an additional, external piece of superconductor with a fixed global phase. The operator $\phi_{\text{ave}}$ then measures the relative phase difference between the chain and the external superconductor.

**e.** Show that a sufficiently long chain of superconducting islands will spontaneously break the global phase rotation symmetry (at zero temperature).

## 7.3 The Anderson–Higgs mechanism

The superconductor spontaneously breaks a continuous symmetry, and you might therefore reasonably expect it to host Nambu–Goldstone modes on top of its ordered state. The gauge fields that feature so prominently in the theory of superconductivity, however, mediate long-ranged Coulomb interactions between the Cooper pairs. Since Goldstone's theorem does not apply in the presence of long-ranged interactions, there is then no guarantee that any gapless modes will exist in the ordered state. In fact, coupling the gapless NG mode of a neutral superfluid to the gapless photon of Maxwell electromagnetism makes both excitations massive within the superconductor.

The quickest way to see how this happens, is to rewrite the Ginzburg–Landau effective free energy of Eq. (7.1) as:

$$
\begin{aligned}
\mathcal{F}_{\mathrm{GL}} &= \frac{1}{2\mu_0}(\nabla \times \mathbf{A})^2 + \frac{\hbar^2}{2m^*}|\psi|^2\left(\nabla\varphi - \frac{e^*}{\hbar}\mathbf{A}\right)^2 + \frac{\hbar^2}{2m^*}(\nabla|\psi|)^2 + \frac{1}{2}r|\psi|^2 + \frac{1}{4}u|\psi|^4 \\
&= \frac{1}{2\mu_0}(\nabla \times \tilde{\mathbf{A}})^2 + \frac{e^{*2}}{2m^*}|\psi|^2\tilde{\mathbf{A}}^2 + \frac{\hbar^2}{2m^*}(\nabla|\psi|)^2 + \frac{1}{2}r|\psi|^2 + \frac{1}{4}u|\psi|^4.
\end{aligned}
\tag{7.7}
$$

In the first line we explicitly wrote the field in terms of its amplitude and phase, $\psi = |\psi|e^{i\varphi}$, while in the second line we *defined* $\tilde{\mathbf{A}} \equiv \mathbf{A} - \frac{\hbar}{e^*}\nabla\varphi$. Notice that the newly defined field $\tilde{\mathbf{A}}(x)$ is invariant under the gauge transformations of Eq. (7.4), which means that, like the scalar amplitude field $|\psi|$, it is a physical and observable degree of freedom. Furthermore, this new field is proportional to the Noether current of Eq. (7.3).

In the superconducting phase, the parameter $r$ is negative and the free energy has a minimum for non-zero values of the field amplitude, $\langle|\psi|\rangle \neq 0$. The second term in Eq. (7.4) can then be interpreted to be a mass term for the field $\tilde{\mathbf{A}}$ (you can check this by explicitly minimizing $\mathcal{F}_{\mathrm{GL}}$ for a fixed value of $|\psi|$). In the ordered phase then, both of the fields appearing in the free energy, the vector field $\tilde{\mathbf{A}}$ as well as the amplitude field $|\psi|$, are massive in the sense of having a gapped dispersion.

In this manifestly gauge-invariant description, the would-be NG mode $\varphi(x)$ seems to have disappeared completely. This is described in various places in the literature by noting that $\varphi$ cannot be a physical degree of freedom because it can be 'removed' from the free energy by imposing the unitary gauge fix $\varphi \equiv 0$. It is also sometimes said 'to be in an unphysical part of Hilbert space', because when using the Coulomb gauge fix $\nabla \cdot \mathbf{A} \equiv 0$, the field $\varphi$ does not couple to any observables. In a neutral superfluid, however, $\varphi(x)$ is real a propagating mode, and physical degrees of freedom cannot simply disappear when including interactions with additional fields in a theory.

In fact, the $\varphi$ excitation has not disappeared. Rather, it is included in the newly defined vector field $\tilde{\mathbf{A}}$. Before coupling to the field $\psi$, we were free to choose the Coulomb gauge fix $\nabla \cdot \mathbf{A} = 0$, which eliminates the longitudinal component as a degree of freedom. However, the new field $\tilde{\mathbf{A}}$ is invariant under gauge transformations, so none of its components can be removed or fixed by employing gauge freedom. In a way, the degree of freedom carried by $\varphi$ can be said to be transferred to or represented by the longitudinal component of $\tilde{\mathbf{A}}$.

A field-theory formulation of the same argument would be to observe that a massless vector field in three dimensions carries two degrees of freedom: the two transverse polarisations. A massive vector field like $\tilde{\mathbf{A}}$ in Eq. (7.7), however, carries three degrees of freedom, which include the longitudinal component. An alternative and more detailed derivation can be done within the formalism of Hamiltonian constraints mentioned in Section 1.5.3, for which an accessible treatment can be found in Ref. [24].

This transformation from the fields $\mathbf{A}$ and $\varphi$ to the new vector field $\tilde{\mathbf{A}}$ is sometimes described as "the vector field has eaten the Goldstone boson and obtained its degree of freedom". The

vector field is also said to "have gotten fat by becoming a massive field". This mechanism for the emergence of a massive vector field is called the *Anderson–Higgs mechanism* [97, 98]. It also goes by several other names, depending on the subfields of physics in which it is discussed.

**Anderson–Higgs mechanism**

### 7.3.1 The Meissner effect

To see some of the consequences of the vector potential becoming massive inside a superconductor, consider the equation of motion obtained by varying the free energy $\mathcal{F}_{GL}$ with respect to the electromagnetic vector potential **A**:

$$\frac{1}{\mu_0}\nabla \times (\nabla \times \mathbf{A}) = \frac{e^*\hbar}{m^*}|\psi|^2(\nabla\varphi - \frac{e^*}{\hbar}\mathbf{A}). \tag{7.8}$$

In deriving these equations of motion, we considered only static field configurations. Including dynamic terms, Eq. (7.8) becomes a wave equation for **A** (and hence **E** and **B**), which yields the dispersion relation for photons. Notice that the right-hand side of the equation vanishes outside of the superconductor, where the field $|\psi|$ is zero. Within the superconductor, the right-hand side may be non-zero and is in fact equal to the supercurrent $\mathbf{j}_e$ defined in Eq. (7.3).

Recall that the supercurrent is proportional to the *vector potential* **A**, rather than to the electric field **E**, as a normal current would be. To see the physical implication of this unusual proportionality to the vector potential, we take the curl of both sides of Eq. (7.8), use the fact that the curl of a gradient vanishes, and substitute $\nabla \cdot \mathbf{B} = 0$, to find the expression:

$$\left(\nabla^2 - \frac{1}{\lambda_L^2}\right)\mathbf{B} = 0. \tag{7.9}$$

Here $\lambda_L = \sqrt{\frac{m^*}{\mu_0 e^{*2}|\psi|^2}}$ defines the *London penetration depth*. The solution to this equation for the field **B** is an exponentially decaying function with length scale $\lambda_L$. That is, magnetic fields can only penetrate the superconductor up to a distance $\lambda_L$ from the surface, and magnetic fields, even static ones, are expelled from the inside of the superconductor. This is known as the *Meissner effect*.

**penetration depth**

**Meissner effect**

Since the supercurrent $\mathbf{j}_e$ is proportional to the vector potential, it obeys a similar equation $(\nabla^2 - 1/\lambda_L^2)\mathbf{j}_e = 0$. This does not mean that superconductors do not carry supercurrents. On the contrary, the dissipationless persistent current is the primary hallmark of superconductivity. However, a stationary flow of Cooper pairs can only exist within a region of width $\lambda_L$ from the edge, while wave-like excitations are gapped and decay exponentially in time.

### 7.3.2 The Higgs boson

Besides the phase variable or would-be Goldstone mode and the electromagnetic vector potential, which combine to form the massive vector field $\tilde{\mathbf{A}}$, the free energy of Eq. (7.7) also contains the amplitude field $|\psi(x)|$. Excitations of this propagating, massive degree of freedom are called the *amplitude mode* of the superconducting state. In the Mexican-hat potential of Fig. 5.4, it corresponds to oscillations perpendicular to the flat direction. Although the existence and physical properties of this mode are straightforward to understand theoretically, its experimental detection in condensed matter systems is complicated and has only recently been achieved [99].

**amplitude mode**

In the context of elementary particle physics, the amplitude mode is known as the *Higgs boson*. The Higgs boson has no *a priori* connection to gauge freedom. In fact neutral superfluids, charge density waves, and various other phases of matter have amplitude modes, at least in principle. However, if gauge fields couple to the Higgs field $\psi$, and the Higgs field has a vacuum expectation value $\langle\psi\rangle \neq 0$, then the Anderson–Higgs mechanism ensures the

**Higgs boson**

gauge fields to be massive. In the Standard Model of elementary particles, the $W$- and $Z$-gauge bosons of the electroweak interaction become massive in this way (see Exercise 7.4).

To be clear about terminology: the field $\psi$ is called the Higgs field. Excitations of the amplitude $|\psi|$ on top of its vacuum expectation value are called Higgs bosons. The gauge fields becoming massive by being coupled to the Higgs fields, and the simultaneous conversion of the NG mode into an additional massive component of the gauge field, is called the Anderson–Higgs mechanism.

---

**Exercise 7.4 (Non-Abelian gauge fields)** Recall the Higgs Lagrangian Eq. (3.11), being expressed in terms of a $U(2)$-field $\check{\Phi}$. This Lagrangian is invariant under the global symmetry $\check{\Phi} \to L\check{\Phi}$, $L \in SU(2)$, but not under the local transformation where $L(x)$ can depend on space. Similar to the $U(1)$-transformation, we can try to fix this by introducing a gauge field.

Define $\mathsf{A}_\mu(x) = \sum_{a=1}^3 A_\mu^a(x) T_a$, where $T_a$ are the Lie-algebra generators of $SU(2)$ as $2 \times 2$-matrices, see Exercise 1.7, and the $A_\mu^a$ are real-valued vector fields, one for each $a$. Define the $SU(2)$-gauge-covariant derivative $D_\mu \equiv \partial_\mu \mathbb{I} - ig\mathsf{A}_\mu$.

**a.** Show that $D_\mu\check{\Phi}(x)$ transforms as $D_\mu\check{\Phi}(x) \to L(x)\big(D_\mu\check{\Phi}(x)\big)$ under the combined transformations:

$$\check{\Phi}(x) \to L(x)\check{\Phi}(x),$$

$$\mathsf{A}_\mu(x) \to L(x)\mathsf{A}_\mu(x)L^\dagger(x) - \frac{i}{g}\big(\partial_\mu L(x)\big)L^\dagger(x). \tag{7.10}$$

The field strength of a non-Abelian gauge field (or *Yang–Mills field*) is

$$\mathsf{F}_{\mu\nu} = \partial_\mu\mathsf{A}_\nu - \partial_\nu\mathsf{A}_\mu - ig[\mathsf{A}_\mu, \mathsf{A}_\nu]. \tag{7.11}$$

Here $\mathsf{F}_{\mu\nu}(x) = \sum_{a=1}^3 F_{\mu\nu}^a(x) T_a$, and $F_{\mu\nu}^a(x)$ are real-valued fields.

**b.** Show that $\mathsf{F}_{\mu\nu}$ transforms under the local transformation Eq. (7.10) as $L\mathsf{F}_{\mu\nu}L^{-1}$.

Therefore the Lagrangian

$$\mathcal{L} = \text{Tr}\Big[\frac{1}{2}(D_\mu\check{\Phi})^\dagger(D^\mu\check{\Phi}) - \frac{1}{2}r\check{\Phi}^\dagger\check{\Phi} - \frac{1}{4}u(\check{\Phi}^\dagger\check{\Phi})^2 - \frac{1}{4}\mathsf{F}_{\mu\nu}\mathsf{F}^{\mu\nu}\Big] \tag{7.12}$$

is invariant under the transformation Eq. (7.10). The $k > 0$-components of these transformations are gauge freedoms, not symmetries. They denote the superfluous degrees of freedom contained in the $A_\mu^a$. Note that now the global symmetry $L(x) = L$ also transforms the gauge field, in contrast to the Abelian case Eq. (7.4).

The Lagrangian is also still invariant under the *global* transformation $\check{\Phi} \to \check{\Phi}R, R \in SU(2)$, recall Exercise 3.2. When $r < 0$, the global part of $L$ together with $R$ is broken down to the diagonal subgroup where $R = L^\dagger$ [100].

**c.** The potential then has a minimum at $\det\langle\check{\Phi}\rangle \neq 0$. Show that one can always perform a gauge transformation $L(x)$ such that $\langle\check{\Phi}\rangle = \text{diag}(v, v)$, $v \in R$. This is the $SU(2)$-equivalent of the unitary gauge fix. Hint: it is easier to use the vector representation for $\check{\Phi}$, see Exercise 3.2.

**d.** If we further assume that $\det\langle\breve{\Phi}\rangle$ is constant (no Higgs boson), then in the unitary gauge fix we have $\partial_\mu\breve{\Phi} = 0$. Show that there is a mass term $\frac{1}{2}M^2 A_\mu^a A^{a\mu}$ for the gauge field in the Lagrangian Eq. (7.12) and determine the mass $M$. Hint: substitute the expectation value $\langle\breve{\Phi}\rangle = \mathrm{diag}(\nu, \nu)$, and use the $SU(2)$-anticommutator $\{T_a, T_b\} = \frac{1}{2}\delta_{ab}\mathbb{I}$.

This is the Anderson–Higgs mechanism for $SU(2)$ gauge-Higgs theory, where the gauge fields become massive while massless NG bosons are absent. In the Standard Model, the unified $SU(2) \times U(1)$ electroweak force couples to the Higgs field, breaking down to the residual $U(1)$ electromagnetic subgroup so that the photon remains massless. (In this *chiral gauge theory* where only the group $L$ is made local, there is also an issue with the so-called *chiral anomaly*, but that is beyond the scope of these notes. See for instance Refs. [71, 101] for details.)

Another example of the non-Abelian Anderson–Higgs mechanism is *colour superconductivity* in quantum chromodynamics (QCD), which is suggested to occur in quark matter at very high density, such as in a neutron star. Here quarks form Cooper pairs and can condense. The $SU(3)$ gauge fields ("gluons") become massive in the same way.

**colour superconductivity**

## 7.4 Vortices

In neutral superfluids, the broken phase-rotation symmetry allows for topological defects in the form of vortex excitations. As discussed in Section 6, these may enter the superfluid when external torque is applied, and they have a topological charge equal to the winding number of the phase. Since the presence of a vortex affects the orientation of the order parameter throughout the superfluid, vortices exert long-range forces on each other, and the energy of a single vortex grows with the system size.

In a charged superfluid, or superconductor, the combination of the phase degree of freedom with the electromagnetic vector potential into a single massive vector field changes the nature of the vortex excitations. First of all, if a vortex is present, the amplitude of the order parameter vanishes at its core. The Meissner effect, which normally expels magnetic fields from the bulk of the superconductor, is therefore not operative in the core region, and the magnetic field can penetrate there. The resulting field profile for the vortex excitation of a superconductor is shown in Fig. 7.1.

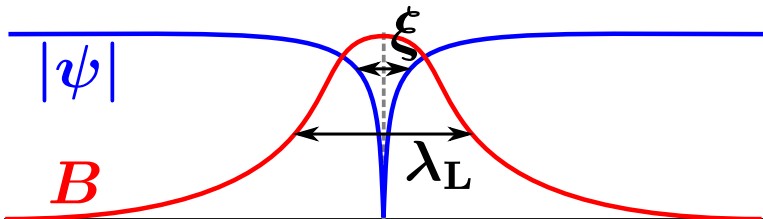

Figure 7.1: Cross section of a vortex in a superconductor. The amplitude of the order parameter field $|\psi|$ falls to zero at the core with the coherence length $\xi = \sqrt{\hbar^2/m^*|r|}$, while the magnetic field **B** can penetrate the superconductor up to the London penetration depth $\lambda_L$ from the core.

To find out how much magnetic flux can penetrate the superconductor through a single vortex core, consider once more the relation between supercurrent and vector potential of Eq. (7.8). We can integrate both sides of the equation along a closed contour $\mathcal{C}$ encircling the vortex core. If the contour is taken sufficiently far from the vortex core, the magnetic field will be completely screened by the Meissner effect, as shown in Fig. 7.1, and the left-hand side is

zero. We are then left with the equation:

$$\frac{\hbar}{e^*} \oint_{\mathcal{C}} d\mathbf{x} \cdot \nabla \varphi = \oint_{\mathcal{C}} d\mathbf{x} \cdot \mathbf{A} = \int_{\mathcal{S}} d\mathbf{S} \cdot \mathbf{B}. \qquad (7.13)$$

Here we used Stokes' theorem in the last equality, and $\mathcal{S}$ is the area enclosed by $\mathcal{C}$. The right-hand side is just the total flux through $\mathcal{S}$. On the left-hand side, we see that $\oint \nabla \varphi = 2\pi n$ is precisely the winding number of the vortex. Therefore, the magnetic flux penetrating the superconductor through vortices is *quantised* in units of $\Phi_0 = h/e^*$, called the flux quantum. **flux quanti-sation**

The decay of the magnetic field radially outward from the vortex core can be understood intuitively by realising that the winding of $\varphi$ does not depend on the size of the contour along which it is calculated. Close to the vortex core, where the field amplitude $|\psi|$ is very low, the phase winding leads to a circular flow of electric supercurrent, according to Eq. (7.3). This supercurrent opposes the externally applied magnetic field, leading to its decay. Of course this is precisely the same physics as what we discussed before for the surface of a superconductor, and the length scale over which the magnetic field decays is just the London penetration depth, $\lambda_{\mathrm{L}}$. Notice however, that the generated magnetic field in turn cancels the supercurrent, which also decays within a London penetration depth. In stark contrast to the situation in a neutral superfluid, the order parameter field outside of an area of radius $\lambda_{\mathrm{L}}$ around the vortex core is entirely unaffected by the topological defect. Well-separated vortices in a superconductor therefore have no interaction, and the energy of a single vortex does not depend on the system size. In fact, for a three-dimensional superconductor with $\lambda_{\mathrm{L}} \gg \xi$, the energy of a straight vortex line of length $l_z$ can be shown to be $E_{\mathrm{vortex}} \approx l_z \left(\frac{n\Phi_0}{\lambda_{\mathrm{L}}}\right)^2 \ln \frac{\lambda_{\mathrm{L}}}{\sqrt{2}\xi} = l_z \frac{h^2}{m^*} \pi n^2 |\psi|^2 \ln \frac{\lambda_{\mathrm{L}}}{\sqrt{2}\xi}$, where $n$ is the winding number of the vortex[16] [74]. As expected, the energy of the vortex scales as $\ln \lambda_{\mathrm{L}}$, rather than $\ln L$, because the order parameter is affected only up to distances smaller than the London penetration depth.

The total vortex free energy contains both the field-independent energy cost $E_{\mathrm{vortex}}$ associated with the local suppression of the superconducting order, and the magnetic energy gained by allowing the electromagnetic field to pass straight through the superconductor rather than expelling it with the Meissner effect. This latter effect provides an energy gain $-\mathbf{H} \cdot \mathbf{B}$, with $\mathbf{H}$ the externally applied magnetic field and $\mathbf{B} = \Phi_0 l_z$ the induced magnetic field within the vortex core. Above some critical value of field, $H_{c1} = E_{\mathrm{vortex}}/\Phi_0 l_z$, it becomes energetically favourable to let the magnetic field in through vortex lines rather than to expel it completely. The resulting state is called an Abrikosov vortex lattice, and it is the distinctive feature of so-called type-II superconductors with $\lambda_{\mathrm{L}} > \sqrt{2}\xi$.

Increasing the applied field beyond $H_{c1}$, the vortices will get more and more closely packed, until the point where their cores start to overlap, and superconductivity is destroyed throughout the material. Since the total flux must be distributed over vortices that each carry a single flux quantum, the cores cover the entire superconductor when $\Phi_0/H \propto \xi^2$, so that superconductivity breaks down for fields higher than $H_{c2} \propto \Phi_0/\xi^2 > H_{c1}$. The two critical magnetic field strengths in type-II superconductors are known as their lower and upper critical fields.

If on the other hand $\lambda_{\mathrm{L}} < \sqrt{2}\xi$, the estimate for the vortex energy, $E_{\mathrm{vortex}}$, breaks down. A more careful analysis will show that in that case, it becomes energetically favourable to create defects with the highest possible vorticity, rather than separating them into isolated flux quanta. An Abrikosov vortex lattice can then not be formed, and the bulk remains in a Meissner state, expelling any magnetic fields towards the edges of the sample. The externally applied magnetic field can be increased in strength until it is large enough to destroy the superconducting order altogether at a single critical field strength $H_c$. This type of behavior is known as type-I superconductivity.

---

[16] In this section, the definition of the coherence length $\xi$ differs by a factor of $\sqrt{2}$ from that of Eq. 5.19, in order to have the expressions here match those of Ref. [74]

## 7.5 Charged BKT phase transition

The coupling of the phase degree of freedom to the electromagnetic field in a superconductor alters the characteristics of its phase transition as compared to a neutral superfluid. The situation in two dimensions is particularly interesting, bringing together several topics discussed in these lecture notes. The main physics can be described in terms of heuristic arguments, by comparing the properties of vortices in superconductors with their superfluid counterparts.

Recall that neutral systems with $U(1)$-symmetry, like superfluids, undergo a BKT phase transition in two dimensions, as described in Section 6.3. This comes about from the combinations of two ingredients. First, the Mermin–Wagner–Hohenberg–Coleman theorem of Section 4.2 precludes the formation of long-range order in two dimensions due to the divergence of thermal corrections to the order parameter by gapless NG modes. However, algebraic long-range order is possible, due to bound vortex–antivortex pairs. Secondly, the energy cost of these pairs is balanced by their entropy, with both scaling logarithmically in system size. This leads to a critical temperature and phase transition at the temperature where vortex pairs can unbind.

In the case of the charged superfluid, both of these arguments need to be adjusted. First of all, the Anderson–Higgs mechanism of Section 7.3 renders all excitations in the superconductor massive, and in particular does not allow for gapless NG modes. This means that there is no divergence as $k \to 0$ in the order parameter corrections of Eq. (4.22). Thus, at first sight, there is no obstruction to having truly long-range ordered two-dimensional superconductors even at non-zero temperatures.

However, we should also take into account the fact that the vortex excitations of the neutral superfluid are altered by the electromagnetic field in the superconducting state. In Section 7.4 we have seen that the winding of the order parameter is counteracted by the penetrating magnetic field, so that the energy of one vortex is finite and does not scale with the system size, being instead of the order of $\ln \lambda_L/\xi$.

The BKT transition in a superfluid comes about from the balance between the entropy and energy of vortices. The entropy of a vortex just counts the number the possible locations for its core, so the form $S \propto \ln L^2/\xi^2$ is still valid for vortices in superconductors as well. On the other hand, the energy of superconducting vortices no longer scales with system size. This implies that for large systems with $L \gg \lambda_L$, the entropic gain of having vortices outweighs their energetic cost even at the lowest temperatures. In other words, the BKT transition temperature at which vortex–antivortex pairs unbind is pushed all the way to zero, and should destroy superconducting order at any temperature.

In reality, both considerations are a bit beside the point. Since we live in a three-dimensional world, even the thinnest superconductors still couple to three-dimensional electromagnetic fields. Vortices can thus interact over large distances through the electromagnetic fields that they create in the vacuum surrounding the superconducting film. This can be described by introducing an *effective penetration depth* of magnetically mediated interactions within the two-dimensional superconductor, which turns out to be [102]:

$$\lambda_{L,2D} = \frac{\lambda_L^2}{w} = \lambda_L \frac{\lambda_L}{w}. \tag{7.14}$$

Here $w$ is the thickness of the superconducting film. If it is small compared to $\lambda_L$, the effective penetration depth becomes very large. From the definition of $\lambda_L$, we see that it is inversely proportional to $e^*$, so that a large penetration depth is equivalent to having a very weak coupling between electromagnetic fields and the superconducting condensate. In other words, a truly two-dimensional superconductor embedded in a three-dimensional vacuum behaves just like a neutral superfluid. The gap of the would-be NG modes becomes very small, and thermal fluctuations can prevent the formation of true long-range order. Simultaneously, the energy

cost of vortices grows and starts to depend on the system size again, pushing the BKT transition temperature up from zero. This explains the experimental observation of BKT transitions in both thin type-II superconductors and in Josephson junction arrays [102].

## 7.6 Order of the superconducting phase transition

Even in three dimensions, where true long-range order certainly exists, the superconducting phase transition is affected by the coupling between the phase and electromagnetic vector potential. Both of these fields can fluctuate, and both types of fluctuations can alter the critical behaviour from the mean-field expectation, in the style of Sections 5.4 and 5.5.

One way to investigate the consequences of the fluctuations in the vector potential is to *integrate out* the gauge field. Recall that the partition function is a sum over all possible configurations. It is then possible, at least formally, to perform a partial sum that includes all possible configurations of the gauge fields while keeping the order parameter field fixed. This turns out to be natural thing to do in the path integral formulation of quantum field theory. The result will be a new, 'effective' theory for the order parameter field in which the gauge field does not explicitly appear, but which may include different terms and changed values of coefficients as compared to the original theory.

As a shortcut to this formal procedure, we can follow Ref. [103], and start from the free energy of the Ginzburg–Landau model in three dimensions in Eq. (7.1). We then expand the minimal coupling term $|(\nabla - ie^*\mathbf{A}/\hbar)\psi|^2$, and replace both the vector potential and its powers by their expectation values, calculated with fixed order-parameter field $\psi$. There are no static electromagnetic fields in the superconducting state, so the terms linear in $\mathbf{A}$ must vanish, but the fluctuations $\langle \mathbf{A} \cdot \mathbf{A} \rangle$ do not. In the free energy of Eq. (7.1) we thus make the replacement:

$$\int d^3x \left( \frac{e^{*2}}{2m^*}|\psi|^2\mathbf{A}^2 \right) \rightarrow \frac{e^{*2}}{2m^*}|\psi|^2 \int d^3x \, \langle \mathbf{A}(\mathbf{x})^2 \rangle$$
$$= \frac{e^{*2}}{2m^*}|\psi|^2 \int \frac{d^3k}{(2\pi)^3} \, \langle \mathbf{A}(\mathbf{k}) \cdot \mathbf{A}(-\mathbf{k}) \rangle. \tag{7.15}$$

The expectation value for the fluctuations can be computed for fixed $|\psi|$ from the London equation (7.9), giving the two-point correlation function for the vector potential in the Coulomb gauge, in momentum space:

$$\langle A_i(\mathbf{k})A_j(-\mathbf{k}) \rangle = \mu_0 \frac{\delta_{ij} - \frac{k_i k_j}{k^2}}{k^2 + \frac{1}{\lambda_L^2}}. \tag{7.16}$$

Using this expression to carry out the integral in the final line of Eq. (7.15) results in:

$$\frac{e^{*2}\mu_0}{4\pi m^*}|\psi|^2\Lambda - \frac{e^{*3}\mu_0^{3/2}}{8m^{*3/2}}|\psi|^3. \tag{7.17}$$

Here, we introduced a cut-off momentum scale $\Lambda$, which we take to be much larger than $1/\lambda_L$. This final expression now replaces the coupling term proportional to $|\psi|^2\mathbf{A}^2$ in the Ginzburg–Landau theory. The result is a modified, effective, theory formulated entirely in terms of the order parameter field. The first term in Eq. (7.17) is proportional to $|\psi|^2$ and slightly renormalises the value of $r$ in the original theory. The second term however, introduces a new term in the effective theory proportional to $|\psi|^3$. This will cause the minimum of the free energy to develop at non-zero $|\psi|$, similar to what is shown in Figure 5.3. The phase transition in the presence of fluctuations must therefore be first-order (discontinuous), rather than second-order (continuous), as would have been expected from mean-field theory. This is

called a *fluctuation-induced first-order* phase transition. In field theory, this way of arriving at a discontinuous phase transition is known as the *Coleman–Weinberg mechanism*.

**Coleman–Weinberg mechanism**

In practice, it turns out that experiments on many superconductors show critical properties that are very close to the mean-field predictions, including for instance the value of the critical exponent $\beta$ being 1/2, as predicted by Eq. (5.4). One reason for this, is that the interval where fluctuations are important, as quantified by the Ginzburg temperature $t_G$ of Eq. (5.25), is very small, so that much of the experimentally accessible parameter regime are not in the 'true' critical region. But this is not the whole story. As mentioned in Section 5.4, the Ginzburg temperature for strongly type-II superconductors (with $\lambda_L \gg \xi$) can be sizeable. These same conditions also allow for the emergence of stable vortices, as discussed in Section 7.4. Using a duality mapping of the kind introduced in Section 6.5.1, it was found that presence of topological defects, which is not taken into account in the calculations above, again alters the effective theory in the critical region and causes the superconducting phase transition in three dimensions to stay second-order as long as $\lambda_L/\xi \gtrsim 1$ [104]. This prediction has since been confirmed in numerical simulations.

# A   Other aspects of spontaneous symmetry breaking

Symmetry and symmetry-breaking affect almost all areas and aspects of physics. These lecture notes have focussed on an important, but nevertheless limited, scope within the realm of spontaneous symmetry breaking, and necessarily leave out many interesting effects and observations. Although this is unavoidable in general, it is particularly regrettable for a few topics that are especially closely related to the main content of these notes, or that are the focus of especially intense current interest in the literature. We therefore include brief introductions to the role of symmetry in some selected topics as short appendices.

## A.1   Glasses

The rigidity of collective states of matter is governed by the spontaneous breaking of symmetry. This is most apparent in crystalline solids, where the framework of gapless Nambu-Goldstone modes (in this case, phonons) assures crystals to really be *rigid solids*. However, most solid states that surround us in everyday life, are not crystalline: one can collectively call such states *glasses*.

 **glass**

 A glass is a solid when it comes to short-time response functions, and indeed glasses are in many macroscopic ways indistinguishable from crystalline solids: they feel rigid, carry phonons, can break, and, most importantly, they break the translational symmetry of space. On the microscopic level, however, glasses are disordered. For example, the $SiO_2$ molecules in an ordinary windowpane sit, immobile, at seemingly random positions, and there is no long-range order in any conventional correlation function. Notice that, in this respect, glasses differ from polycrystalline systems, in which crystalline order exists at some intermediate length scale.

 A detailed theoretical description of the glass phase remains elusive to this day. In fact, one of the major open question is whether the glass phase is even a genuine phase of matter, or if it should perhaps be understood as an extremely slow, viscous liquid. A closely related question is whether there exists a true thermodynamic *glass transition*, separating a high-temperature liquid from a low-temperature stable glass phase. Both of these questions are difficult to answer because most glasses are in practice made by supercooling a liquid to avoid crystallisation, and noticing that upon further cooling the viscosity increases exponentially (see also the related discussion in Section A.3).

 In theoretical simulations, the word 'glasses' is often used for systems with so-called *quenched disorder*. These are described by Hamiltonians in which the values of parameters are chosen at random from some probability distribution. A famous example is the Sherrington-Kirkpatrick model, where each Ising spin has a random interaction with every other spin. This model has a low-temperature rigid phase without long-range order, akin to a glass. However, the states observed in these types of simulations do not spontaneously break translational symmetry, because the quenched disorder already explicitly broke it to begin with. Nonetheless, within an ensemble of Hamiltonians with different realisations of the randomly chosen parameter values, quenched disorder glasses do break the symmetry between different disorder realisations, which is known as *replica symmetry breaking*. Some good books on such quenched disorder glasses are Refs. [105–107]. A detailed introduction of glasses without quenched disorder, which can for example be made using the supercooling of liquids, can be found in Refs. [108, 109].

 **replica symmetry breaking**

 Finally, a second class of solid materials without long-range order is made by *quasicrystals*. In a quasicrystal, there is no periodicity in the atomic positions. The Fourier transform of the atomic positions, however, does consist of perfectly sharp peaks. In fact there are infinitely many of them, arranged in a fractal pattern. A quasicrystal can therefore be discovered by having an unusual but sharp electron diffraction pattern. From the perspective of symmetry breaking, it is interesting to note that a quasicrystal *completely* breaks translational symmetry

 **quasicrystal**

(unlike normal crystals, which break it into a discrete subgroup), but that they can break rotational symmetry down to just a discrete subgroup. The remaining rotational symmetries in quasicrystals can be types that are not allowed in crystalline matter, such as five-fold or ten-fold, which gives another window for discovering them.

## A.2 Many-body entanglement

A recent development in quantum physics is the study of *many-body entanglement*. This topic can be introduced most easily by first reviewing some properties of two-particle entanglement, of which many good discussions can also be found in elementary textbooks on quantum mechanics, such as Refs. [1, 110, 111].

First then, consider a system with two spin-$\frac{1}{2}$ degrees of freedom. The full Hilbert space is spanned by four states, $\mathcal{H} = \{|\uparrow\rangle_1 \otimes |\uparrow\rangle_2, |\downarrow\rangle_1 \otimes |\uparrow\rangle_2, |\uparrow\rangle_1 \otimes |\downarrow\rangle_2, |\downarrow\rangle_1 \otimes |\downarrow\rangle_2\}$. Any state can be written as a linear superposition of these four states. A particularly interesting one to consider, is the singlet state:

$$|0\rangle = \frac{1}{\sqrt{2}} \left( |\downarrow\rangle_1 \otimes |\uparrow\rangle_2 - |\uparrow\rangle_1 \otimes |\downarrow\rangle_2 \right). \tag{A.1}$$

This state is certainly entangled, but just from its definition, it is not easy to quantify *how* entangled it is.

One way to approach this question, is to consider the properties of the *reduced density matrix*. To begin with, we (arbitrarily) write the full Hilbert space as a tensor product of two parts: $\mathcal{H} = \{|\uparrow\rangle_1, |\downarrow\rangle_1\} \otimes \{|\uparrow\rangle_2, |\downarrow\rangle_2\}$. The reduced density matrix on spin 1 is then obtained by 'tracing' the full density matrix over the degrees of freedom of spin 2. In the case of the pure state of Eq. (A.1), the full density matrix is just the projection operator $\rho = |0\rangle\langle 0|$, and the trace operation can be carried out explicitly:

$$
\begin{aligned}
\rho_1 &= \text{Tr}_2 \, \rho \\
&= \langle \uparrow_2 |\rho| \uparrow_2 \rangle + \langle \downarrow_2 |\rho| \downarrow_2 \rangle \\
&= \frac{1}{2} \left( |\uparrow_1\rangle\langle\uparrow_1| + |\downarrow_1\rangle\langle\downarrow_1| \right).
\end{aligned} \tag{A.2}
$$

The fact that spins 1 and 2 were entangled is reflected in the fact that the reduced density matrix $\rho_1$ no longer represents a pure state. To quantify this, Von Neumann proposed to compute the *entanglement entropy* of the reduced density matrix, defined as:

$$S_{\text{vN}} = -\text{Tr}\rho_1 \log \rho_1. \tag{A.3}$$

In the case of the singlet state, the entanglement entropy equals $S_{\text{vN}} = \log 2$. In recent years, entanglement has also been studied in systems with many degrees of freedom, like a many-spin system. In such cases, one can still talk about *bipartite entanglement* in the same way as we showed for two spins. One needs to (again arbitrarily) split the system into two parts (commonly called *A* and *B*), compute the reduced density matrix on one subsystem, and use that to compute the entanglement entropy defined by Eq. (A.3).

Typically, it is then interesting to explore how the entanglement entropy scales with the size of subsystem. There are a few special cases to consider. The first is a so-called 'product state', like the Néel antiferromagnet, in which there is literally no entanglement at all between any two spins. In this case the entanglement entropy is identically zero. The second case is when spins are only entangled with a few spins that are close by. This is then reflected in the entanglement entropy by the fact that it scales with the size of the boundary of region *A*, in what is known as 'area law' entanglement. All ground states of locally interacting systems obey the area law. A final, extreme case is when each spin is entangled with every other spin, in

which case the entanglement entropy scales with the volume of subsystem *A*. Such a 'volume law' typically applies to sufficiently highly excited eigenstates of any interacting Hamiltonian. In Section A.5, we will also introduce topologically ordered states, which are considered to be the most entangled states possible, though a thorough discussion of their entanglement properties goes beyond the scope of these lecture notes (see for example Ref. [112]). The possible connection between entanglement entropy and the classical, statistical entropy at any given temperature, is currently an active field of research.

To see how many-body entanglement is related to spontaneous symmetry breaking, recall the discussion of Section 2.7, in which we showed that exact ground states of symmetric Hamiltonians displaying long-range correlations, are an indication that the symmetry in these models may be spontaneously broken. This implies that unstable ground states are always highly entangled, while stable symmetry-breaking states are unentangled product states that satisfy cluster decomposition. One must be careful, however, before equating the absence of bipartite entanglement with stability too quickly. Take for example the unstable ground state of the Ising model, defined in Eq. (2.50). In quantum information theory, this state is also known as the Greenberger–Horne–Zeilinger (GHZ) state, and is considered a maximally entangled state. The entanglement entropy, however, is exactly $S_{vN} = \log 2$, regardless of how the system is divided in two. Clearly, entanglement entropy by itself is not a sufficient tool for distinguishing between stable (unentangled) and unstable (entangled) ground states, and the search for alternative measures of entanglement is ongoing.

Nevertheless the entanglement entropy does capture some subtle properties related to spontaneous symmetry breaking. In 'type A' symmetry-breaking systems with a symmetric finite size ground state, the entanglement entropy has logarithmic corrections to its generic area law. The coefficient in front of the logarithm counts the number of Nambu–Goldstone modes $N_{NG}$, [113, 114]:

$$S_{vN}(L^D) = aL^{D-1} + \frac{N_{NG}(D-1)}{2} \log L + \dots. \tag{A.4}$$

Furthermore, the so-called *entanglement Hamiltonian* $H \equiv -\log \rho_A$, with $\rho_A$ is the reduced density matrix, has the same structure as the tower of states of Section 2.6. These are just two examples of how information about the symmetry-broken state and its excitations is hidden inside the entanglement structure of the symmetric ground state.

### A.3 Dynamics of spontaneous symmetry breaking

Throughout most of these lecture notes, we focussed on equilibrium properties of phases of matter, and neglected any discussion of how the transition between different phases comes about or evolves as a function of time. We did mention the Kibble–Zurek mechanism in Section 6.5.2, which describes the formation of topological defects during a second-order phase transition. There are several other, closely related, processes connected to dynamical phase transitions.

For instance, as described in our discussion of the Kibble–Zurek mechanism, going through the (second-order) paramagnetic-to-ferromagnetic phase transition, will typically result in a ferromagnetic state consisting of multiple magnetic domains, with more-or-less random orientations with respect to one another. Even if these happen not to lock in any topological defects, and in spite of torque exerted on each other by the domains, the system tends to avoid the formation of a large single domain because of its sizeable magnetostatic energy cost.

In first-order phase transitions, a related phenomenon arises because of the presence of an energy barrier between the symmetric and asymmetric phases. This may prevent the system from relaxing to a symmetry-broken configuration from a symmetric metastable state. For example, very slowly cooling a very pure liquid within a very smooth container, it can remain

domain

liquid up to several tens of degrees below the critical temperature, in an effect known as *supercooling*. As soon as the liquid is perturbed by some motion, or a temperature difference, or an imperfection on the container, however, the local potential energy may cause it to cross the energy barrier to the solid phase. At that point the liquid will crystallise locally, and this enables neighbouring regions to crystallise as well. Thus, the solidification spreads (typically very quickly) throughout the medium, from a single nucleation event.

> **supercooling**

> **nucleation**

Finally, the scale invariance associated with the critical point, mentioned in Section 5.5, pertains not only to equilibrium properties such as specific heat, but extends to dynamical phenomena as well. It leads to the emergence of *dynamical critical exponents* for quantities such as damping and relaxation. This is reviewed in Ref. [115,116].

> **critical exponent – dynamical**

## A.4 Quantum phase transitions

Our discussion of phase transitions in Chapter 5 covered ordinary, thermal phase transitions. Here the change from one phase of matter to another occurs as a result of a change in temperature. The transition from the ordered to the disordered phase (low to high temperature, lower to higher symmetry) can then be said to be due to *thermal fluctuations*.

It is also possible to study the change of a system at fixed temperature as some other parameter is varied. The density, external magnetic field, pressure, or applied current for example are readily accessible tuning parameters whose variation can cause a material to change its phase matter. These transitions, however, can always be viewed as thermal transitions by tuning to the critical value of the alternative parameter and then changing temperature. This is particularly obvious when considering the phase diagram with temperature on one axis and the alternative parameter on the other. Whichever way you cross a transition line in such a diagram, the crossing point, and hence the critical behaviour, can also be reached in a purely thermal transition.

This is not true for transitions that occur at precisely zero temperature, in which the system undergoes a *quantum phase transition* between two distinct stable phases as some parameter $p$ is tuned through a *quantum critical point* $p = p_c$. At zero temperature, the system should be in its ground state at any value of the tuning parameter (but recall the discussion in Section 2.6), and thermal excitations or fluctuations do not play any role. It is sometimes said that, in analogy to thermal transitions, the quantum phase transition is due to *quantum fluctuations*, but as explained in Section 4 that is a misnomer. The system is in a ground state of its Hamiltonian throughout the transition, and nothing fluctuates or changes in time. The actual situation is that, far away from the critical point at $p_c$, the quantum system is in a state that is close to a classical state in the sense of Section 2.5.3. With respect to this classical state, quantum *corrections* become more and more important as $p$ approaches $p_c$, and they completely overwhelm the classical correlations at the quantum critical point. Far across on the other side of the transition, the system (often) approaches a different classical state.

> **quantum phase transition**

> **quantum critical point**

Just like in the thermal phase transitions discussed in Section 5.5, the quantum critical point is characterised by scale invariance and universality, dictated by the symmetry. It turns out that the universality class of a quantum critical point is linked to that of a thermal critical point with the same symmetry breaking pattern, but in one dimension higher. This can be understood by recalling the calculation of the thermal and quantum corrections in Section 4.2. At zero temperature all Matsubara frequencies must be taken into account and these effectively become an additional dimension. At non-zero temperature on the other hand, a finite number of Matsubara frequencies can always be interpreted as thermal corrections to the static equilibrium state.

Exactly zero temperature is beyond the reach of any experiment in the real world, so the discussion of quantum phase transitions may seem purely academic. However, it turns out that the existence of the quantum critical point has large influence on the physics near the

critical value $p_c$ even at non-zero temperatures, sometimes up to hundreds of Kelvins. This is referred to as *quantum criticality*, and is one of the strongest manifestations of non-classical physics in condensed matter—much more so than say superconductors, which after all are just as classical as rocks and chairs. A good resource on this topic is Sachdev's textbook [117]. **quantum criticality**

Notice that many phase transitions in (particle) field theory are at zero density, and hence zero temperature, making them a sort of quantum phase transition. Since these theories are Lorentz invariant, it is immediately clear that the time dimension should be treated in these transitions on equal footing with the spatial dimensions. There is no quantum critical region in the phase diagram, since finite temperatures are not accessible by construction.

## A.5  Topological order

The discovery of the integer quantum Hall effect made clear that symmetry and symmetry breaking alone do not suffice to exhaustively identify all interesting phases of matter. The quantum Hall effect occurs in two-dimensional electron gases—states which are well described by non-interacting electrons—in a perpendicular magnetic field. The field forces the electrons into orbital motion, leading to the formation of Landau levels. The longitudinal conductivity is then completely suppressed for values of the magnetic field at which the electrons completely fill some of the available Landau levels. Going from a field strength with a single filled level to one with two filled levels corresponds to a transition between different quantum Hall phases, characterised by different (quantised) values of the transverse or Hall conductivity. None of the quantum Hall states have any non-trivial symmetry breaking whatsoever though, and from a symmetry perspective none of them can be distinguished from an ordinary (electron) gas or liquid. In particular, there is no local order parameter in any of them. **quantum Hall effect**

Instead of an order parameter, the different quantum Hall states may be labeled by a quantised *topological invariant,* which in this case is just the value of the (quantised) Hall conductivity. While not directly related to the topological defects of Section 6, the invariants are topological in the sense that they are calculated by taking an integral over the whole system, and that they are unaffected by local deformations of the state or Hamiltonian. The physics of the quantum Hall states can be generalised to include *topological insulators* and *topological superconductors*. These too, are characterised by topological invariants, and always remain disordered in the sense of any spontaneously broken symmetry. They do often occur in crystalline materials however, and the lattice symmetries play an intricate role in determining the number and types of available topological invariants [118]. Again all these invariants are topological in the sense that they are insensitive to any local perturbations. Even adding weak interactions will typically not destroy the topology, as long as the interactions do not break any symmetries. A special feature of topological materials is the necessary existence of a gapless mode at the interface between two materials with different topological invariants. These are known as symmetry-protected edge modes, and they cannot be avoided as long as the interface does not break any symmetries necessary for the definition of the associated invariant. **topological insulator** **symmetry-protected topological order**

Returning to the quantum Hall effect, it was found that states with zero longitudinal and quantised Hall conductance may also emerge at magnetic field values corresponding to certain rational filling fractions for the Landau levels, in what was soon dubbed the *fractional quantum Hall effect*. In this case interactions between the excitations are actually strong and essential in establishing the state. To distinguish between distinct fractional quantum Hall states as well as other interacting quantum liquids, Xiao-Gang Wen pioneered the notion of *topological order* [119]. Unlike the integer quantum Hall systems and topological insulators, topologically ordered materials have a non-zero ground state degeneracy, with distinct ground states labelled by a topological invariant. However, they occur in strongly interacting rather than non-interacting systems, and this qualitatively changes many of its other features. Topologically ordered systems are for example long-range entangled (as defined in Section A.2), as **topological order**

opposed to the short-range entanglement of topological insulators and the like. They also typically have fractionalised excitations, such as the quasiparticles in the fractional quantum Hall states, whose electric charge is a rational fraction of the elementary electron charge. Reviews of this still new and hotly debated notion of order are, in order of increasing sophistication, Refs. [119], [120], and [121]. Even superconductors where the condensed bosons are composite (i.e. the charge of the constituent particles is a fraction of the charge of the bosons), like the BCS superconductor discussed in Section 7.1, may be considered to be topologically ordered [122].

## A.6 Time crystals

A crystalline solid is a medium in which translational symmetry is broken down to a discrete but infinite subgroup. One can wonder if there exist systems that break *time* translation symmetry to a discrete infinite subgroup, which can then be said to be a *time crystal*. Soon after **time crystal** Wilczek proposed this idea [123], however, it was pointed out that spontaneous breaking of time translation symmetry is fundamentally impossible in any equilibrium state of matter [124, 125]. These no-go theorems did not exclude the possibility that time crystals can exist out of equilibrium. More specifically, in an oscillating state with period $T$, the time translation symmetry is discrete from the outset, and can be described for example using Floquet theory. But the periodic evolution can have further *spontaneous* breaking of time translations, leading a recurring dynamics with longer period $nT$, where $n$ can be any integer. Such systems are called *discrete time crystals*. **discrete time crystals**

Notice that in order to avoid trivialities, the emergence of a longer time period in a discrete time crystal should be accompanied by a notion of *rigidity*, in the sense that even when the driving is not at precisely the preferred frequency, the time-ordered state still emerges. Systems displaying this type of discrete time-crystalline order were reported in experimental setups of trapped ions driven by periodic laser pulses [126], as well as in diamond spin impurities driven by microwave radiation [127]. A recent review can be found in Ref. [128].

## A.7 Higher-form symmetry

The Noether charge was defined in Eq. (1.13) as $Q = \int_V d^D x \, j^0(x)$, in terms of an integral of the Noether charge density over all of space. This operator is a global symmetry generator for ordinary local fields, describing point particles. One can say the point particles are charged under this symmetry generator. The notion of *generalised global symmetry*, or *higher-form symmetry* generalises this concept [129]. It considers spatially extended charged objects, like **higher-form symmetry** lines or surfaces. The symmetry generators then have to defined as integrals over a lower-dimensional space that the extended object intersects.

This can be illustrated using ordinary Maxwell electromagnetism in empty 3+1D spacetime. We already know there is a $U(1)$ gauge freedom, defined for example in Eq. (1.38). In addition to this, we can define the line operators along some contour $\mathcal{C}$:

$$W_{\mathcal{C}} = e^{i \oint dx^\mu A_\mu}, \qquad \text{Wilson loop,} \qquad (A.5)$$

$$H_{\mathcal{C}} = e^{i \oint dx^\mu \tilde{A}_\mu}, \qquad \text{'t Hooft loop.} \qquad (A.6)$$

Here the Maxwell field strength tensor $F_{\mu\nu} = \partial_\mu A_\nu - \partial_\nu A_\mu$ can be used to define the so-called dual photon field $\tilde{A}_\mu$ by writing $F_{\mu\nu} = \partial_\kappa \epsilon_{\kappa\mu\nu\lambda} \tilde{A}_\lambda$. The electric and magnetic symmetry generators, which act on these line operators, can be defined as:

$$Q_E = \int_\Sigma dS_{tm} \, F_{tm} = \int_\Sigma dS_{tm} \, E_m, \qquad Q_M = \int_\Sigma dS_{mn} \, F_{mn} = \int_\Sigma dS_{mn} \, \epsilon_{mnk} B_k. \qquad (A.7)$$

Here $\Sigma$ is a surface perpendicular to $\mathcal{C}$, taking the place of the volume integral of ordinary symmetry generators. In the language of differential forms, these symmetry operators are said to be *1-form symmetries* because the symmetry transformation acting on the line operators results in a 1-form (vector) valued phase $\sim e^{i\oint_\mathcal{C} dx^\mu \Lambda_\mu}$. Similarly, $p$-form symmetries can written as operators defined on a $(D-p)$-dimensional subspace, which act on $p$-dimensional objects with $p$-form phase factors.

The higher-form symmetries can be broken, and the line operators then act as (non-local) order parameters or generalised correlation functions. Clearly, the expectation value of a line operator depends on its integration contour $\mathcal{C}$. If it satisfies an *area law*, that is $W_\mathcal{C} \propto e^{-|\mathcal{S}|}$, with $\mathcal{S}$ the area enclosed by $\mathcal{C}$, and $|\mathcal{S}|$ its areal size, then the magnitude of the line operator expectation value falls off quickly with increasing contour size, and the symmetry is said to be unbroken. If, on the other hand, it satisfies a *perimeter law*, $W_\mathcal{C} \propto e^{-|\mathcal{C}|}$, with $|\mathcal{C}|$ the length of $\mathcal{C}$, its magnitude falls off slowly with contour size, and the symmetry is said to be broken. This also generalises to higher forms with the appropriate integration domains.

Interestingly, the magnetic symmetry $Q_\mathrm{M}$ is spontaneously broken in the Maxwell vacuum, as the 't Hooft line $H_\mathcal{C}$ satisfies a perimeter law. The associated Nambu–Goldstone mode is the photon $A_\mu$ itself. This higher-form symmetry is restored in a superconductor, where the photon becomes gapped by the Anderson–Higgs mechanism, as described in Section 7.3. This is, in a sense, a dual description of the superconducting phase transition. One way to understand how it comes about, is to realise that $Q_\mathrm{M}$ counts magnetic flux enclosed in its contour. In the vacuum, magnetic fields are free, or unconfined, and cost little energy to create. They are analogous for example to bosons in a Bose–Einstein condensate. On the other hand, magnetic flux in a superconductor is confined into vortex lines and is expensive to create, analogous to ordinary gapped bosonic excitations. The $p$-form generalisation of the Goldstone theorem is given in Refs. [130, 131].

There is an interesting connection to the topology of Sec. A.5: when a higher form discrete symmetry is spontaneously broken, the system will exhibit topological order [132].

## A.8 Decoherence and the measurement problem

One of the hallmarks of quantum physics is the ability to create superpositions of any two states of a system. If the system is completely isolated, such superpositions never decay (although a superposition of energy eigenstates with different energies will show Rabi oscillations). In reality, however, no system is completely isolated, and in practice it is hard to maintain coherent superpositions of macroscopically distinct states for any extended period of time.

The detailed understanding of this observation can be called *the decoherence program*, which was pioneered by Zeh in the 1970s [133–138]. One of the important conceptual successes of decoherence, is the identification of a set of *stable* states. Since quantum states can in principle be described using any basis of Hilbert space, one may wonder why we do see everyday objects in eigenstates of one operator (such as position), but hardly ever in eigenstates of others (like momentum). The resolution of this paradox lies in the fact that the coupling of any observable system to the environment selects a preferred basis (*pointer basis* in decoherence jargon), which consists of the eigenstates of the full Hamiltonian describing the system, its environment, and the interaction between them [135]. In particular, if the energy scale related to the interaction is larger than the typical energy scale of the systems (and the environment), the pointer basis is entirely set by the interaction Hamiltonian.

A second important observation in the decoherence program, is that of the process of decoherence itself. Starting from any pure state $|\psi\rangle$, we may write its density matrix, $\rho = |\psi\rangle\langle\psi|$, in the pointer basis. Interaction with the environment (a large number of uncontrollable degrees of freedom, also known as a heat bath) then typically causes the off-diagonal elements of the *reduced* density matrix (traced over all uncontrollable environmental degrees of freedom,

decoher-
ence

compare with Section A.2) to decay exponentially quickly, leaving a mixed ensemble of pointer states. The understanding of this decoherence is of paramount importance in experiments aiming to exploit quantum superpositions, such as quantum computations or simulations. A good exposition of the decoherence program can be found in the textbook by Schlosshauer [139].

The evolution of a pure state to a mixed state is suggestive of what happens during a quantum measurement. Following unitary time evolution, the interaction between a microscopic quantum object and a measurement machine should typically lead to a superposition of pointer states. Yet, in any single experiment only a single outcome is ever observed. The decoherence program addresses some aspects of this *measurement problem*. In particular, it explains which outcomes may be observed in a macroscopic device interacting with an uncontrollable environment, by defining the pointer basis. Within the reduced density matrix description, it also explains the suppression of any signs of quantum interference between pointer states, through the disappearance of the off-diagonal matrix elements. Decoherence, however, does not solve the measurement problem, because it cannot explain which of the available pointer states will be observed as the single outcome of any particular single measurement [140]. This is not just a case of ignorance, as in classical measurement, because the decoherence program can say nothing about the outcome of a single measurement even if you are given perfect knowledge of the initial state, perfect control over the interactions, and unlimited computational power.

Quantum measurement is a more fundamental problem. Under unitary time evolution, quantum dynamics always evolves a single initial state to a unique final state. Given an initial superposition of pointer states in a typical quantum measurement, however, we sometimes observe one outcome, and sometimes another. The description of the measurement process must therefore be non-unitary [140, 141].

Several authors have noticed peculiar parallels between the measurement problem and spontaneous symmetry breaking [24, 142–146]. Given a symmetric Hamiltonian, any superposition of states in the broken-symmetry manifold is in principle equally likely to be realised. Yet, some states turn out to be stable while others are not. Similarly, starting from a symmetric state at high temperatures, cooling an object through a phase transition will result in one of the many equivalent stable symmetry-breaking states being 'spontaneously' chosen. Unlike the equilibrium case, imperfections that break the symmetry do not give rise to a singular limit in the dynamics of phase transitions, and the evolution of a single symmetric state to a single ordered state must again be non-unitary [145]. Based on this observation, it has been suggested that even the unitarity of quantum mechanical time evolution itself is a symmetry that may be spontaneously broken in an extension of quantum theory encompassing both quantum measurement and the dynamics of phase transitions [146].

# B  Further reading

We can recommend several textbooks and review articles to continue your study of spontaneous symmetry breaking.

**Symmetry and Noether's theorem**

- H. Goldstein, C. Poole and J. Safko. *Classical Mechanics.* Addison Wesley (2002) — a solid standard reference.

- H. Kleinert. *Multivalued fields.* World Scientific 2008, Chapter 3 — two side-by-side derivations of the theorem, both in mechanics and field theory.

- J. Butterfield *On Symmetry and Conserved Quantities in Classical Mechanics.* in W. Demopoulos and I. Pitowsky (ed.) *Physical Theory and its Interpretation.* Springer (2006). — in-depth review of the theorem from Hamiltonian and Lagrangian perspectives.

- K.A. Brading. *Which symmetry? Noether, Weyl, and conservation of electric charge.* Stud. Hist. Phil. Sci. B33(1), 3 (2002) — lucid treatment of Noether's second theorem.

- M. Banados and I. Reyes. *A short review on Noether's theorems, gauge symmetries and boundary terms.* Int. J. Mod. Phys. D 25(10), 1630021 (2016) — very similar to our treatment.

**Symmetry breaking**

- S. Weinberg. *Quantum Theory of Fields, Volume II.* Cambridge University Press, 1996, Chapter 19 "Spontaneously Broken Global Symmetries" — an excellent introduction of SSB from the high energy perspective, including a discussion of pions and quark mass terms.

- G. Guralnik, C. Hagen and T. Kibble. *Broken symmetries and the Goldstone theorem.* In *Advances in Particle Physics*, vol 2. pp 567–708, Interscience, New York (1967) — one of the earliest reviews on symmetry breaking and the Anderson–Higgs mechanism.

- P.W. Anderson. *Basic Notions of Condensed Matter Physics.* Benjaming/Cummings, 1984, Chapter 1 — perspective of condensed matter physics, discussing Bose condensation, crystals, magnets, and so forth.

- K. Landsman. *Foundations of Quantum Theory.* Springer, 2017, Chapter 10 — spontaneous symmetry breaking from a mathematical physics perspective.

**Goldstone theorem**

- Guralnik, Hagen & Kibble. *opus citatum* — contains several proofs of the theorem.

- C. Burgess. *Goldstone and pseudo-Goldstone bosons in nuclear, particle and condensed-matter physics.* Phys. Rep. 330(4), 193 (2000) — a modern review.

- T. Brauner. *Spontaneous symmetry breaking and Nambu–Goldstone bosons in quantum many-body systems.* Symmetry 2(2), 609 (2010) — first review to consider type-B NG modes.

- H. Watanabe. *Formula for the number of Nambu-Goldstone modes.* arXiv:1904.00569 — short review of the effective Lagrangian method for counting of NG modes.

**Mermin–Wagner–Hohenberg–Coleman theorem**

- A. Auerbach. *Interacting electrons and quantum magnetism.* Springer–Verlag, New York (1994) — good treatment of spins waves, and Bogoliubov-inequality proof of the Mermin–Wagner theorem.

- N. Nagaosa. *Quantum field theory in condensed matter physics.* Texts and Monographs in Physics. Springer, Berlin (1999) — our derivation in Section 4.2 is based on this one.

**Phase transitions**

- P. Chaikin and T. Lubensky. *Principles of Condensed Matter Physics.* Cambridge University Press (2000) — good review of phase transition physics, also information on topological defects.

- J. Yeomans. *Statistical Mechanics of Phase Transitions.* Clarendon Press (1992);
  N. Goldenfeld. *Lectures on phase transitions and the renormalization group* No. 85 in Frontiers in physics, Perseus Books (1992) — two standard textbooks on phase transitions and criticality.

- I. Herbut. *A Modern Approach to Critical Phenomena.* Cambridge University Press (2007) — another solid textbook with many explicit calculations.

**Superconductivity**

- P. G. De Gennes. *Superconductivity of Metals and Alloys.* Advanced book classics, Perseus, Cambridge, MA (1999) — contains detailed derivations of the ground state wavefunction and the Ginzburg-Landau functional.

- M. Tinkham. *Introduction to Superconductivity.* McGraw Hill (1996) — standard work on many theoretical and practical aspects of superconductors.

- J. Annett. *Superconductivity, Superfluids and Condensates.* Oxford Master Series in Physics, Oxford, (2004) — modern work that includes superfluids as well.

**Topological defects**

- N. Mermin. *The topological theory of defects in ordered media.* Rev. Mod. Phys. 51, 591 (1979) — essential and accessible review.

**Mathematics**

- H. Jones. *Groups, Representations, and Physics.* Insitute of Physics Publishing (1998);
  H. M. Georgi. *Lie algebras in particle physics.* 2nd ed., Frontiers in Physics. Perseus, Cambridge (1999) — two solid introductions to group theory and Lie algebras in physics.

- M. Nakahara. *Geometry, Topology and Physics.* Second Edition, Graduate student series in physics. Taylor & Francis (2003) — advanced physics textbook on topology, including homotopy theory.

# C   Solutions to selected exercises

**Full list of exercises**



SciPost Phys. Lect. Notes 11 (2019)

### Exercise 1.3: Noether's trick

We will show how Noether's trick applies to the action of the Schrödinger field, Eq. (1.16). The essence of the 'trick' is to make the $U(1)$ phase transformation space-time dependent:

$$\psi(x) \;\;\rightarrow\;\; \psi(x) - i\alpha(x)\psi(x), \tag{C.1}$$

$$\psi^*(x) \;\;\rightarrow\;\; \psi^*(x) + i\alpha(x)\psi^*(x), \tag{C.2}$$

where $\alpha(x)$ is real and continuous. The action is manifestly not invariant under this transformation. Specifically, the first term becomes:

$$
\begin{aligned}
\psi^*(x)\partial_t\psi(x) \;\;\rightarrow\;\; & \psi^*(x)\partial_t\psi(x) + i\alpha(x)\psi^*(x)\partial_t\psi(x) - i\psi^*(x)\partial_t(\alpha(x)\psi(x)) + \mathcal{O}(\alpha^2) \\
= \;\; & \psi^*(x)\partial_t\psi(x) - i\psi^*(x)\psi(x)\partial_t\alpha(x),
\end{aligned}
\tag{C.3}
$$

where we used the product rule for differentiation. Applying the same trick for the term with a spatial derivative, we find that under the transformation Eqs. (C.1)-(C.2) the action changes as:

$$\delta S = \int \mathrm{d}t\,\mathrm{d}^D x \left( \hbar\psi^*(x)\psi(x)\partial_t\alpha(x) + i\frac{\hbar^2}{2m}(\partial_n\alpha(x))((\partial_n\psi^*(x))\psi(x) - \psi^*(x)(\partial_n\psi(x))) \right). \tag{C.4}$$

Using partial integration we find that this is indeed of the form:

$$\delta S = \int \mathrm{d}t\,\mathrm{d}^D x \; \alpha(x)\partial_\nu j^\nu, \tag{C.5}$$

with the 4-current $j^\nu$ equal to the Noether current of Eqs. (1.20)-(1.21).

    The solution for the relativistic field theory can be found by simply looking at how the spatial derivative term in the Schrödinger field theory transforms. The index $n$ can be generalised to a relativistic index $\nu$ to find the full solution.

**Exercise 2.1: Classical magnet**

**a.** For every state with a magnetisation $M$, we can make another state by reversing the direction of all spins. This new state has opposite magnetisation $-M$ but the same energy, and hence the same probability in the thermal ensemble. Since the thermal expectation value is just the sum over all possible states, every $M$ is summed with a $-M$ to yield a net zero magnetisation, regardless of temperature.

**b.** In absence of the external field, the lowest energy is obtained by ferromagnetically aligning all the spins. For every possible direction of the magnetisation we find the same energy. Now adding a symmetry-breaking term, the energy of each state changes by $\Delta E = -hNs\cos\theta$, where $\theta$ is the angle between the external field $\mathbf{h}$ and the direction of magnetisation $\hat{\mathbf{n}}$. The state with the lowest energy has $\theta = 0$, so when $\hat{\mathbf{n}}$ is aligned with $\mathbf{h}$. The ground state (that is, the state with lowest energy) has a magnetisation of $\langle M\rangle_{T=0} = Ns\hat{n}$.

**c.** The first limit means: if you have a symmetric spin system, perfectly shielded from any possible external field, you will have zero magnetisation even in the thermodynamic limit. If, however, you take the thermodynamic limit but the system is not perfectly isolated (the second limit), you will *always* break the symmetry and end up with a nonzero magnetisation. This is the essence of spontaneous symmetry breaking.

**Exercise 2.3: Elitzur's theorem for a free gas**

**a.** Remember that in quantum mechanics, the translation operator is obtained by acting with the momentum operator. The operator $U = e^{i\sum_j \mathbf{a}_j \cdot \mathbf{P}_j/\hbar}$ shifts the $j$-th particle by an amount $\mathbf{a}_j$. Because the Hamiltonian is only dependent on $\mathbf{P}_j$ the Hamiltonian is invariant under $U$.

**b.** The ground state wave function is just a product of the wave functions of all individual particles. For all particles except $j = 2$, the ground state is just a plane wave with $\langle\mathbf{P}_j\rangle = \mathbf{0}$. For $j = 2$, the ground state is that of the harmonic oscillator, so

$$\psi_{j=2}(\mathbf{x}) = \left(\frac{\sqrt{m\kappa}}{\pi\hbar}\right)^{1/4}\exp\left(-\frac{\sqrt{m\kappa}}{2\hbar}\mathbf{x}^2\right). \tag{C.6}$$

**c.** The ground state energy of the harmonic oscillator is

$$\bar{E} = \frac{1}{2}\hbar\sqrt{\frac{\kappa}{m}}. \tag{C.7}$$

The expectation value of $\mathbf{X}_{j=2}$ is zero, and its variance is

$$\Delta\mathbf{X}_{j=2}^2 = \frac{\hbar}{2\sqrt{m\kappa}}. \tag{C.8}$$

**d.** In the limit $\kappa \to 0$, we find $\bar{E} = 0$ and $\Delta\mathbf{X}_{j=2}^2 \to \infty$, which you would expect for a delocalised particle.

**e.** The state of the second particle, that we subjected to our symmetry breaking potential, is unaffected by the thermodynamic limit $N \to \infty$. Therefore $\kappa \to 0$ and $N \to \infty$ commute, and there are no singular limits that could give rise to spontaneous symmetry breaking.

**Exercise 2.4: Noether current of the Heisenberg magnet**

**a.** Rewrite the Hamiltonian as $H = \frac{J}{2}\sum_{j\delta}S_j^b S_{j+\delta}^b$ where $\delta$ runs over all neighbouring sites of each $j$. The factor of $\frac{1}{2}$ is to compensate for double counting. The spin commutation relations

are $[S_i^a, S_j^b] = i\hbar \delta_{ij} \epsilon^{abc} S^c$. Therefore, the equation of motion is:

$$\partial_t S_i^a = \frac{i}{\hbar} \frac{J}{2} \sum_{j\delta} \left( S_j^b [S_{j+\delta}^b, S_i^a] + [S_j^b, S_i^a] S_{j+\delta}^b \right)$$

$$= \frac{J}{2} \sum_{j\delta} \left( S_j^b \epsilon^{abc} S_i^c \delta_{i,j+\delta} + \epsilon^{abc} S_i^c \delta_{i,j} S_{j+\delta}^b \right)$$

$$= \frac{J}{2} \epsilon^{abc} \sum_{\delta} \left( S_{i-\delta}^b S_i^c + S_i^c S_{i+\delta}^b \right)$$

$$= -J \epsilon^{abc} \sum_{\delta} S_i^b S_{i+\delta}^c. \tag{C.9}$$

In going to the last line we used the fact that spin-operators on different sites commute, and we relabelled upper indices while exploiting the antisymmetric property of the Levi-Civita symbol.
**b.** For a given $a$, the right-hand side of the equations of motion have the form $-\sum_\delta j_{i,i+\delta}$ with:

$$j_{i,i+\delta} = J \epsilon^{abc} S_i^b S_{i+\delta}^c, \tag{C.10}$$

which is now the locally conserved spin current on the bond from $i$ to $i + \delta$. (Notice that on a lattice, currents are defined on bonds and not on sites!)

### Exercise 2.6: Josephson effect

**a.** The contribution to the equation of motion of the field density on the left due to $H_K$ is given by:

$$-i\hbar \partial_t (\psi_L^* \psi_L) = -i\hbar \left( (\partial_t \psi_L^*) \psi_L + \psi_L^* (\partial_t \psi_L) \right) = K \left( \psi_R^* \psi_L - \psi_L^* \psi_R \right), \tag{C.11}$$

which introduces the Josephson current operator

$$I_J \equiv iK \left( \psi_R^* \psi_L - \psi_L^* \psi_R \right). \tag{C.12}$$

In order to compare with Eq. (2.41), we write the complex order parameter as $\psi_{L/R} = |\psi_{L/R}| e^{i\phi_{L/R}}$. The Josephson current operator now becomes

$$I_J = 2K |\psi_L| |\psi_R| \sin(\phi_R - \phi_L), \tag{C.13}$$

which shows that just like in Eq. (2.41), the current is given by the sine of the phase difference.
**b.** As in the first part, the spin Josephson current is obtained by computing the commutator between $H_K$ and the magnetisation on the left side,

$$\partial_t M_L^a = i \left[ M_L^a, H_K \right] \tag{C.14}$$

$$= iK M_R^b \left[ M_L^a, M_L^b \right] \tag{C.15}$$

$$= -K \epsilon_{abc} M_R^b M_L^c \tag{C.16}$$

$$\equiv I_J^a. \tag{C.17}$$

In vector notation, this reads

$$\mathbf{I}_J = K \mathbf{M}_L \times \mathbf{M}_R. \tag{C.18}$$

Notice that there is no Josephson current if the two magnets $\mathbf{M}_L$ and $\mathbf{M}_R$ are aligned.

### Exercise 2.7: Heisenberg Ferromagnet

**a.** We assume the ground state has all spins polarised in the $z$-direction, so $\langle\psi|S^z|\psi\rangle \neq 0$. Now for the symmetry $Q = S^x$, choosing the interpolating field to be $S^y$ yields $\langle\psi|[S^x, S^y]|\psi\rangle \propto \langle\psi|S^z|\psi\rangle \neq 0$. Similarly, for the symmetry $Q = S^y$, the interpolating field $S^x$ shows that also $S^y$ is spontaneously broken.

**b.** In the state with all spins aligned, acting with $S_i^+ S_j^-$ for sites $i \neq j$ yields zero, because $S_i^+$ annihilates the maximally polarised state. Therefore, we only need to consider the $S_i^z S_j^z$ term. Because the spins are aligned in the $z$-direction the ferromagnetic state is an eigenstate of Eq. (2.18).

### Exercise 3.1: Number of type-B NG modes

**a.** Let $A = -A^T$ be an antisymmetric matrix and $U$ a unitary operator such that $UAU^\dagger = D$ is a diagonal matrix. Then

$$D^* = D^\dagger = UA^\dagger U^\dagger = UA^T U^\dagger = -UAU^\dagger = -D. \tag{C.19}$$

In the third equality we used that $A$ is real. For each eigenvalue $\lambda$ we therefore have $\lambda^* = -\lambda$ so $\lambda$ is purely imaginary.

Alternatively, let $\mathbf{v}$ be an eigenvector of $A$. Then $\lambda|\mathbf{v}|^2 = \mathbf{v}^\dagger A\mathbf{v} = -\mathbf{v}^\dagger A^T \mathbf{v} = -(\mathbf{v}^\dagger A\mathbf{v})^\dagger = -\lambda^*|\mathbf{v}|^2$.

**b.** Let $\mathbf{v}$ be an eigenvector of $A$ with eigenvalue $\lambda$: $A\mathbf{v} = \lambda\mathbf{v}$. Taking the complex conjugate of the whole equation shows $A\mathbf{v}^* = \lambda^*\mathbf{v}^*$, where we used that $A$ is real. So $\lambda^*$ is also an eigenvalue of $A$.

**c.** Let $e_i = \begin{pmatrix} i\lambda_i & 0 \\ 0 & -i\lambda_i \end{pmatrix}$. Then we have $w_i e_i w_i^\dagger = \begin{pmatrix} 0 & \lambda_i \\ -\lambda_i & 0 \end{pmatrix}$ with $w_i = \frac{1}{\sqrt{2}}\begin{pmatrix} 1 & i \\ i & 1 \end{pmatrix}$.

### Exercise 3.3: Heisenberg Ferrimagnet

**a.** The magnetisation in one unit cell is $\frac{2}{N}\langle S_{\text{tot}}^z\rangle = m_A + m_B = S_A - S_B$.

**b.** The Watanabe–Brauner matrix is given by $M_{ab} = -i\langle\psi|[Q_a, j_b^t(x)]|\psi\rangle$. In our case, we have two broken symmetry generators, $S_{\text{tot}}^x$ and $S_{\text{tot}}^y$. Because $[S^x, S^y] = iS^z$, we find

$$M_{xy} = \begin{pmatrix} 0 & i\langle S^z\rangle \\ -i\langle S^z\rangle & 0 \end{pmatrix} = \begin{pmatrix} 0 & i(S_A - S_B) \\ -i(S_A - S_B) & 0 \end{pmatrix}. \tag{C.20}$$

Other matrix elements vanish.

**c.** The staggered magnetisation per unit cell is equal to $\langle N_i^z + N_{i+1}^z\rangle = m_A - m_B = S_A + S_B$.

### Exercise 4.1: $XY$-model quantum corrections

**a.** We use $S^x = \frac{1}{2}(S^+ + S^-)$ to write:

$$\tilde{H}_{XY} = J\sum_{\langle ij\rangle}\left(\frac{1}{4}\left(S_i^+ S_j^+ + S_i^+ S_j^- + S_i^- S_j^+ + S_i^- S_j^-\right) + S_i^z S_j^z\right). \tag{C.21}$$

**b.** In terms of Holstein-Primakoff bosons we get

$$\tilde{H}_{XY} = -\frac{1}{2}|J|S\sum_{\langle ij\rangle}\left(a_i a_j + a_j^\dagger a_i + a_i^\dagger a_j + a_i^\dagger a_j^\dagger\right) - \frac{1}{2}|J|zNS^2 + |J|zS\sum_i n_i, \tag{C.22}$$

where we neglect the quartic term proportional to $n_i n_j$, and we use $J = -|J|$. Here $N$ is the number of sites and $z$ is the number of nearest neighbours on our lattice (the coordination number).

**c.** Following Eq. (4.8)-(4.10), the Fourier transformation yields

$$\tilde{H}_{XY} = -\frac{1}{2}|J|zNS^2 + |J|zS\sum_k \left((1 - \tfrac{1}{2}\gamma_k)a_k^\dagger a_k - \tfrac{1}{4}\gamma_k\left(a_k a_{-k} + a_k^\dagger a_{-k}^\dagger\right)\right). \tag{C.23}$$

After the Bogoliubov transformation, Eq. (4.12),

$$\begin{aligned}
\tilde{H}_{XY} = &-\frac{1}{2}|J|zNS^2 + |J|zS\sum_k\Big[(1 - \tfrac{1}{2}\gamma_k)\sinh^2 u_k - \tfrac{1}{4}\gamma_k\sinh 2u_k \\
&+ ((1 - \tfrac{1}{2}\gamma_k)\cosh 2u_k - \tfrac{1}{2}\gamma_k\sinh 2u_k)b_k^\dagger b_k \\
&+ \tfrac{1}{2}(-\tfrac{1}{2}\gamma_k\cosh 2u_k + (1 - \tfrac{1}{2}\gamma_k)\sinh 2u_k)(b_k^\dagger b_{-k}^\dagger + b_k b_{-k})\Big].
\end{aligned} \tag{C.24}$$

**c.** Choosing

$$\tanh 2u_k = \frac{\gamma_k}{2 - \gamma_k} \tag{C.25}$$

diagonalises the Hamiltonian, which then becomes

$$\tilde{H}_{XY} = -\frac{1}{2}|J|zNS^2 + |J|zS\sum_k\left[\tfrac{1}{2}\left(-1 + \sqrt{1 - \gamma_k}\right) + \sqrt{1 - \gamma_k}\,b_k^\dagger b_k\right]. \tag{C.26}$$

**d.** The ground state energy density is obtained when no spin waves are occupied, hence

$$E/N = -\frac{1}{2}|J|zNS^2 + |J|zS\sum_k\tfrac{1}{2}\left(-1 + \sqrt{1 - \gamma_k}\right), \tag{C.27}$$

which is in $D = 2$ on a square lattice equal to

$$\frac{E_{D=2}}{N} = -2|J|S^2 - 0.0838172|J|S \tag{C.28}$$

and in $D = 3$ on a cubic lattice

$$\frac{E_{D=3}}{N} = -3|J|S^2 - 0.0757964|J|S. \tag{C.29}$$

The magnetisation can be computed using $\langle S^z\rangle/N = S - \frac{1}{N}\sum_k\langle a_k^\dagger a_k\rangle = S - \frac{1}{N}\sum_k\sinh^2 u_k$. Filling in $u_k$ gives

$$\langle S^z\rangle/N = S - \frac{1}{2N}\sum_k\left(-1 + \frac{1}{\sqrt{1 - (\frac{\gamma_k}{2-\gamma_k})^2}}\right). \tag{C.30}$$

In $D = 2$ this gives $S - 0.060964$ and in $D = 3$ $S - 0.0225238$.

### Exercise 5.1: Mean-field order parameter

**a.** The mean-field Hamiltonian Eq. (5.11) has no correlations between neighbouring spins. The expectation value is thus given by the expectation value on a single site,

$$\begin{aligned}
|\mathbf{m}| = \langle(-1)^i\mathbf{S}_i\rangle &= Z^{-1}\mathrm{Tr}\,(-1)^i\mathbf{S}_i e^{-\beta H[m]} \tag{C.31} \\
&= \frac{\sum_{S_i = \pm\frac{1}{2}} S_i e^{2Jz\beta|\mathbf{m}|S_i}}{\sum_{S_i = \pm\frac{1}{2}} e^{2Jz\beta|\mathbf{m}|S_i}} \tag{C.32} \\
&= \frac{1}{2}\frac{e^{Jz\beta|\mathbf{m}|} - e^{-Jz\beta|\mathbf{m}|}}{e^{Jz\beta|\mathbf{m}|} + e^{-Jz\beta|\mathbf{m}|}} \tag{C.33} \\
&= \frac{1}{2}\tanh Jz\beta|\mathbf{m}|. \tag{C.34}
\end{aligned}$$

**b.** The derivative of the free energy Eq. (5.13) with respect to $|\mathbf{m}|$ is

$$
\frac{\partial \mathcal{F}/N}{\partial |\mathbf{m}|} = 2Jz|\mathbf{m}| - \frac{1}{\beta}\frac{\frac{\partial}{\partial|\mathbf{m}|}2\cosh Jz\beta|\mathbf{m}|}{2\cosh Jz\beta|\mathbf{m}|} \tag{C.35}
$$

$$
= 2Jz|\mathbf{m}| - Jz\frac{\sinh Jz\beta|\mathbf{m}|}{\cosh Jz\beta|\mathbf{m}|} \tag{C.36}
$$

$$
= 2Jz|\mathbf{m}| - Jz\tanh Jz\beta|\mathbf{m}| \tag{C.37}
$$

$$
= 0. \tag{C.38}
$$

Setting this derivative to zero yields the condition $|\mathbf{m}| = \frac{1}{2}\tanh Jz\beta|\mathbf{m}|$.

**Exercise 7.3: Josephson junction array**

**a.** Following Eq. (7.4), we perform a gauge transformation with the function $\alpha(x)$, or $\alpha_j$ at the island with index $j$. The phase variable changes according to

$$
\theta_j \to \theta_j - \alpha_j \tag{C.39}
$$

and the parameter $\psi_j^{j+1}$ transforms as

$$
\psi_j^{j+1} \to \psi_j^{j+1} - (\alpha_{j+1} - \alpha_j). \tag{C.40}
$$

With these transformation rules, it is clear that $\theta_j - \theta_{j+1} + \psi_j^{j+1}$ is gauge invariant. Of course, $n_j$ does not change under the gauge transformation either.

**b.** In terms of the new variables $\phi_j$, the cosine term becomes $-J\cos(\phi_j - \phi_{j+1})$. We now make the assumption that $\theta_j \approx \theta_{j+1}$, and expand for a small difference. This yields

$$
H \approx \sum_j \left[\frac{1}{2}Cn_j^2 + \frac{1}{2}J(\phi_j - \phi_{j+1})^2\right]. \tag{C.41}
$$

Here a constant term has been dropped.

**c.** We first perform a Fourier transform, which yields for $k \neq 0$,

$$
H = \sum_{k\neq 0}\left[\frac{1}{2}C|n_k|^2 + \frac{1}{2}J|\phi_k|^2\left|1 - e^{ika}\right|^2\right], \tag{C.42}
$$

where $a$ is the distance between neighbouring islands. Because $n$ and $\phi$ are conjugate variables, for each $k$-mode we found a harmonic oscillator system. The frequency is set by

$$
\hbar\omega(k) = \sqrt{CJ}\left|1 - e^{ika}\right| = 2\sqrt{CJ}\left|\sin\frac{ka}{2}\right|. \tag{C.43}
$$

**d.** At $k = 0$, the phase-dependent part vanishes and we have $H_{k=0} = \frac{1}{2N}Cn_{\text{tot}}^2$ only dependent on $n_{\text{tot}}$. Naturally, this Hamiltonian commutes with $U = e^{i\alpha n_{\text{tot}}}$.

**e.** By introducing the symmetry breaking field $J'$, the $k = 0$ Hamiltonian also looks like a harmonic oscillator. Its frequency is set by $\omega_0 = \sqrt{J'C/N}$. For any nonzero $J'$, the ground state (hence the state at zero temperature) is a Gaussian wavepacket with an uncertainty in the phase of $(\Delta\phi_{\text{ave}})^2 = \frac{\hbar}{2}\sqrt{\frac{C}{J'N}}$. If $J' > 0$ and $N \to \infty$, the uncertainty in the phase variable vanishes and we have spontaneously broken the global phase rotation symmetry. Notice that the other order of limits, namely taking $J' \to 0$ before sending $N \to \infty$, yields a diverging phase uncertainty and therefore a preserved global phase rotation symmetry.

**Exercise 7.4: Non-Abelian gauge fields**

**a.** Write out all terms to find:

$$
\begin{aligned}
D_\mu \check{\Phi} &= (\partial_\mu - i g A_\mu)\check{\Phi} \\
&\rightarrow \left(\partial_\mu - i g L A_\mu L^\dagger - (\partial_\mu L)L^\dagger\right)\left(L\check{\Phi}\right) \\
&= L(\partial_\mu \check{\Phi}) + (\partial_\mu L)\check{\Phi} - i g L A_\mu \check{\Phi} - (\partial_\mu L)\check{\Phi} \\
&= L\left((\partial_\mu - i g A_\mu)\check{\Phi}\right) = L\left(D_\mu \check{\Phi}\right).
\end{aligned}
\tag{C.44}
$$

**b.** First note that $(\partial_\mu L)L^\dagger = -L(\partial_\mu L^\dagger)$ by partial integration. Write out the four terms in $\mathsf{F}_{\mu\nu}$ after gauge transformation:

$$
\begin{aligned}
\mathsf{F}_{\mu\nu} &= \partial_\mu \mathsf{A}_\nu - \partial_\nu \mathsf{A}_\mu - i g \mathsf{A}_\mu \mathsf{A}_\nu + i g \mathsf{A}_\nu \mathsf{A}_\mu \\
&\rightarrow (\partial_\mu L)\mathsf{A}_\nu L^\dagger + L(\partial_\mu \mathsf{A}_\nu)L^\dagger + L\mathsf{A}_\nu(\partial_\mu L^\dagger) - \frac{i}{g}(\partial_\mu \partial_\nu L)L^\dagger - \frac{i}{g}(\partial_\nu L)(\partial_\mu L^\dagger) \\
&\quad - (\partial_\nu L)\mathsf{A}_\mu L^\dagger - L(\partial_\nu \mathsf{A}_\mu)L^\dagger - L\mathsf{A}_\mu(\partial_\nu L^\dagger) + \frac{i}{g}(\partial_\nu \partial_\mu L)L^\dagger + \frac{i}{g}(\partial_\mu L)(\partial_\nu L^\dagger) \\
&\quad - i g L\mathsf{A}_\mu \mathsf{A}_\nu L^\dagger - (\partial_\mu L)\mathsf{A}_\nu L^\dagger + L\mathsf{A}_\mu(\partial_\nu L^\dagger) - \frac{i}{g}(\partial_\mu L)(\partial_\nu L^\dagger) \\
&\quad + i g L\mathsf{A}_\nu \mathsf{A}_\mu L^\dagger + (\partial_\nu L)\mathsf{A}_\mu L^\dagger - L\mathsf{A}_\nu(\partial_\mu L^\dagger) + \frac{i}{g}(\partial_\nu L)(\partial_\mu L^\dagger) \\
&= L(\partial_\mu \mathsf{A}_\nu)L^\dagger - L(\partial_\nu \mathsf{A}_\mu)L^\dagger - i g L\mathsf{A}_\mu \mathsf{A}_\nu L^\dagger + i g L\mathsf{A}_\nu \mathsf{A}_\mu L^\dagger \\
&= L\mathsf{F}_{\mu\nu}L^\dagger.
\end{aligned}
\tag{C.45}
$$

**c.** The matrix field $\check{\Phi}$ corresponds to the vector field $\Phi = \begin{pmatrix} \phi_1 & \phi_2 \end{pmatrix}^{\mathrm{T}}$, which transforms as $\Phi(x) \rightarrow L(x)\Phi(x)$. This field obtains a non-zero expectation value $\langle |\Phi|^2 \rangle = \langle \phi_1^* \phi_1 + \phi_2^* \phi_2 \rangle = v^2 \neq 0$.

A gauge transformation of our desired field configuration is

$$
L(x)\begin{pmatrix} 0 \\ v \end{pmatrix} = \begin{pmatrix} l_2^*(x) & l_1(x) \\ -l_1^*(x) & l_2(x) \end{pmatrix}\begin{pmatrix} 0 \\ v \end{pmatrix} = \begin{pmatrix} l_1(x)v \\ l_2(x)v \end{pmatrix}.
\tag{C.46}
$$

Therefore, for a given field configuration $\begin{pmatrix} \phi_1(x) & \phi_2(x) \end{pmatrix}^{\mathrm{T}}$, we can choose $\phi_1(x) = l_1(x)v$ and $\phi_2(x) = l_2(x)v$, and act with $L^{-1}(x) = L^\dagger(x)$ to obtain $\begin{pmatrix} 0 & v \end{pmatrix}^{\mathrm{T}}$ at any point $x$ in space.

**d.** We can assume $\partial_\mu \check{\Phi} = 0$. Expand first term in the Lagrangian

$$
\begin{aligned}
\mathrm{Tr}\left[(D_\mu \check{\Phi})^\dagger (D^\mu \check{\Phi})\right] &= \mathrm{Tr}\left[\left((\partial_\mu - i g A_\mu)\check{\Phi}\right)^\dagger \left((\partial^\mu - i g A^\mu)\check{\Phi}\right)\right] \\
&\xrightarrow{\partial_\mu \check{\Phi} = 0} \mathrm{Tr}\left[\left(i g A_\mu \check{\Phi}\right)^\dagger \left(-i g A^\mu \check{\Phi}\right)\right] = g^2 \mathrm{Tr}\left[\check{\Phi}^\dagger A_\mu^\dagger A^\mu \check{\Phi}\right].
\end{aligned}
\tag{C.47}
$$

Now use $A_\mu(x) = \sum_{a=1}^3 A_\mu^a(x) T_a$ and $T_a^\dagger = T_a$:

$$
\begin{aligned}
A_\mu^\dagger A^\mu &= \sum_{ab} A_\mu^a T_a T_b A^{b\mu} = \frac{1}{2}\sum_{ab} A_\mu^a \{T_a, T_b\} A^{b\mu} = \frac{1}{4}\sum_{ab} A_\mu^a \delta_{ab} \mathbb{I} A^{b\mu} \\
&= \frac{1}{4} A_\mu^a A^{a\mu} \mathbb{I}. \qquad \text{(summation over } a \text{ implied)}
\end{aligned}
\tag{C.48}
$$

Now substitute $\check{\Phi}$ by its expectation value $\langle \check{\Phi} \rangle = v\mathbb{I}$ to find

$$
\frac{1}{4} g^2 v^2 A_\mu^a A^{a\mu} \mathrm{Tr}\,\mathbb{I} = \frac{1}{2} g^2 v^2 A_\mu^a A^{a\mu}.
\tag{C.49}
$$

The mass is given by $M = g v$.

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

# Index