# Peer review of "An Introduction to Spontaneous Symmetry Breaking"

_SciPost Physics Lecture Notes, doi:SciPost Phys. Lect. Notes 11 (2019)_

## Round 1 · Referee Report · Anonymous · 2019-10-16

Report

Overall I think these lecture notes are of excellent quality. They contain many useful concepts related to spontaneous symmetry breaking, brought together in a coherent and in most parts very pedagogical way. Even though I was familiar with most of the physics treated here, I enjoyed immensely reading the manuscript and I learned some new things, besides rethinking some others which I already knew but which are presented in an original and often thought-provoking way. The real question is of course not whether experts in the field enjoy reading the notes, but whether students and young researchers benefit from it. Overall the answer is a clear yes, and thus I strongly recommend publication and do not have any fundamental criticism. I rather have a few general comments the authors might consider.

(1) For lecture notes intended for undergraduate and graduate students, the ratio of text/interpretation vs equations/formalism is very large. In most parts I find the explanations and illustrations excellent, but I think for somebody new to the topic they might be sometimes hard to follow without having an actual calculation at hand. One example are pages 27-29, but there are many more of this kind. Also, the extensive and detailed interpretation in the long text passages sometimes tends to suppress the students coming up with their own understanding, gained "the hard way", rather than having every detail of the interpretation already digested for them. This point is somewhat subjective, and I leave it up to the authors whether to use it as an inspiration for some improvements.

(2) Given the ambition formulated in the preface of highlighting the importance of spontaneous symmetry breaking in condensed matter physics as well as in high-energy physics, I was expecting some more examples/insights from the high-energy context. For instance the list of examples for NG modes on page 61 contains not a single example from high-energy physics, and color superconductivity, which would be a nice example of a non-abelian superconductor to illustrate the Anderson-Higgs mechanism, is never mentioned. I understand that the focus is on the condensed matter applications (as stated in the preface), but given the very general scope, I would have expected the notes to be a bit more attractive for students in the field of high-energy physics. Again, this is not a suggestion for extensive corrections, but the authors might consider to add a few more examples that demonstrate the applicability of the ideas to systems in high-energy physics.

(3) There are two instances of ambiguous notation which might be confusing for students. Firstly, operators are not distinguished in notation from numbers or fields. This confused me for instance in Sec 1.2.4 where phi_a suddenly turns into an operator while before is was a field (and presumably meant to be the eigenfunction of this operator). To the very least, I would think this notation deserves a comment at the very beginning to warn the reader. Secondly, three-vectors are sometimes denoted in bold, sometimes not. For instance in eq (7.5) both notations seem to be used in a single equation (I don't see a reason why x in that equation should not also be a three-vector). I suggest to use a consistent notation, which should be easy to implement by making all three-vectors bold.

---

## Round 1 · Referee Report · Anonymous · 2019-10-27

Report

These lecture notes provide a comprehensive introduction to symmetry and its breaking phenomena from the modern point of view, which also cover the recent progress of the classification of Nambu-Goldstone modes. I believe that they are helpful to not only graduate students but also working physicists; especially, for high energy physics, who work in the relativistic and infinite volume system because the lecture notes cover finite volume and/or nonrelativistic systems. They are well organized and the quality is high; thus, I recommend the publication of the present lecture notes.
I have several minor comments and questions as follows:

In page 116, the authors discussed the relation between the symmetry breaking and the coefficient in front of the logarithm of the entanglement entropy. I have a naive question. Is the entanglement entropy independent of the type of NG modes? I expect that it counts the number of type-A NG modes. What happens to the entanglement entropy, when type-B NG modes exist?

In page 119, the authors wrote "It has been suggested that even an ordinary, fermionic superconductor like the BCS superconductor (discussed in Section 7.1), may be considered to be topologically ordered [117], although bosonic superconductors certainly are not."
The Cooper pair in the BCS superconductor is charge two, whose low-energy effective theory is described by $\mathbb{Z}_2$ gauge theory (or equivalently, BF theory with level two, which is also an effective theory of topological order. The level-two BF theory has emergent one and two form symmetries, and they are spontaneously broken). The origin of the topological order is related to the charge of the condensate. If the charge is equal or larger than two, it exhibits the topological order. In my opinion, bosonic or fermionic is not important.

In page 120, the authors discussed higher form symmetries, and their breaking. I would like to point out when a higher form discrete symmetry is spontaneously broken, it exhibits a topological order (See, Xiao-Gang Wen, Phys. Rev. B 99, 205139 (2019)).

$D$ is used as the spatial dimensions in Sec. 1 and 2, while $d$ is used in Sec. 3, 4.
It is better to use the same notation of spatial dimensions.

$dx$ and $\mathrm{d}x$ coexist. It is better to use the same symbol.

In page 37, since the normalization of the Pauli matrix is not the standard one (I think the standard normalization is $\mathrm{tr}\sigma^a\sigma^a=2$ with no sum), it is better to refer the representation of Pauli matrices shown in Exercise 1.7.

In page 76-77, the authors wrote "Even so, the momentum integral still diverges in dimensions lower than two due to the factor $k^{d-2}$."
This statement is, of course, true because the spatial dimension in our world is integer. However, $d$ is often treated as a real number and the improper integral converges for $d> 1$ when $d$ is real, so that it is better to write ".. in dimensions lower than or equal to one ...."

In page 80, the endpoint of liquid gas transition is usually called the "critical point."
In my opinion, it is better to use the word "critical point" instead of "critical endpoint,"
although the QCD community often uses "critical endpoint" as the endpoint of the first order line. The critical endpoint is employed as the endpoint of second-order line terminating a first order line (See textbook by Chaikin and Lubensky, for example).

In page 119, The authors wrote, "Like the integer quantum Hall systems and topological insulators, topologically ordered materials have a non-zero ground state degeneracy, ...."
Since the integer quantum Hall systems and topological insulators have the unique ground state, should the sentence be "Unlike the integer ....."?

Requested changes

In page 11, $\psi(x,t)$ should be $\psi({\bf x},t)$, or simply $\psi(x)$.
In page 12, $\delta_s^\nu$ in Eq. (1.27) should be $\Delta_s^\nu$.
In page 26, $\sigma^a$ should be $\sigma_a$ in Eq. (1.50).
In page 56, above Eq. (3.1), $Q=\int dx j^t$ should be $Q=\int \mathrm{d}^dxj^t$.
In page 62, "Nambu-goldstone" should be "Nambu-Goldtone."

---

## Editorial Decision

published